# Space-Time Variability of UTLS Chemical Distribution in the Asian Summer Monsoon Viewed by Limb and Nadir Satellite Sensors

Jiali Luo[1,2], Laura L. Pan[2], Shawn B. Honomichl[2], John W. Bergman[2,3], William J. Randel[2], Gene Francis[2], Cathy Clerbaux[4], Maya George[4], Xiong Liu[5], and Wenshou Tian[1]

[1]Key Laboratory of Semi-Arid Climate Change and College of Atmospheric Sciences, Lanzhou University, Lanzhou, China

[2]National Center for Atmospheric Research, Boulder, Colorado, USA

[3]Bay Area Environmental Research Institute, Sonoma, California, USA

[4]LATMOS/IPSL,UPMCUniversité Paris 06 Sorbonne Universités, UVSQ, CNRS, Paris, France

[5]Harvard-Smithsonian Center for Astrophysics, Cambridge, Massachusetts, USA

*Correspondence to:* Laura Pan (liwen@ucar.edu)

**Abstract.** The Asian Summer Monsoon (ASM) creates a hemispheric scale signature in trace gas distributions in the upper troposphere and lower stratosphere (UTLS). Data from satellite retrievals are the best source of information for characterizing these large-scale signatures. Measurements from the Microwave Limb Sounder (MLS), a limb viewing satellite sensor, have been the most widely used retrieval products for these types of studies. This work explores the information for the ASM influence on UTLS chemical distribution from two nadir-viewing sensors, the Infrared Atmospheric Sounding Interferometer (IASI) and the Ozone Monitoring Instrument (OMI), together with the MLS. Day-to-day changes in carbon monoxide (CO) and ozone ($O_3$) tracer distributions in response to dynamical variability are examined, to assess how well the data from different sensors provide useful information for studying the impact of sub-seasonal scale dynamics on chemical fields. Our result, using June-July-August of 2008 data, shows that although the MLS provides relatively sparse horizontal sampling on daily timescales, interpolated daily CO distributions show a high degree of dynamical consistency with the synoptic scale structure and variability of the anticyclone. Our analysis also shows that the IASI CO retrieval has sufficient sensitivity to produce upper tropospheric (UT) CO with variabilities independent from the lower to middle tropospheric CO. The consistency of IASI CO field with the synoptic scale anticyclone dynamical variability demonstrates that the IASI UT CO product is a physically meaningful dataset. Furthermore, IASI CO vertical cross-sections combined with the daily maps provide the first observational evidence for a model analyses-based hypothesis on the preferred ASM vertical transport location and the subsequent horizontal redistribution via east-west eddy shedding. Similarly, the OMI $O_3$ profile product is shown to be capable of distinguishing the tropospheric dominated air mass in the anticyclone from the stratospheric dominated background on a daily time scale, providing consistent and complementary information to the MLS. These results not only highlight

the complementary information between nadir and limb sensors, but also demonstrate the value of "process-based" retrieval evaluation for characterizing satellite data information content.

## 1 Introduction

As a prominent atmospheric circulation feature in the upper troposphere and lower stratosphere (UTLS) during boreal summer, the Asian Summer Monsoon (ASM) anticyclone's large scale dynamical behaviour has been investigated widely in recent years (e.g., Hoskins and Rodwell, 1995; Highwood and Hoskins, 1998; Zhang et al., 2002; Liu et al., 2007; Wu et al., 2015). The ASM anticyclone is bounded by the westerly jet to the north, easterly jet to the south, and this circulation is linked to enhanced air confinement (e.g., Dunkerton, 1995; Randel and Park, 2006; Garny and Randel, 2016; Fan et al., 2017). Due to the influences of deep convection and air confinement, seasonal-mean chemical composition within the anticyclone near the tropopause displays distinctly surface-like characteristics; boundary layer and tropospheric tracers, such as CO, $H_2O$, HCN, and a large set of hydrocarbons, are significantly enhanced, while $O_3$ as a stratospheric tracer is significantly decreased (Park et al., 2004; Li et al., 2005; Randel and Park, 2006; Park et al., 2007; Randel et al., 2010; Vernier et al., 2011; Garny and Randel, 2013).

Although the ASM anticyclone is a strong and steady feature of seasonal scale circulations in UTLS, it undergoes variations on sub-seasonal timescales. These include 10-20 day east-west migrations and the associated eddy shedding (Hsu and Plumb, 2000; Popovic and Plumb, 2001; Zhang et al, 2002; Garny and Randel, 2013). Previous studies have shown that the monsoon circulation has active/break cycles that are linked to oscillations of deep convection with timescales of 10-20 and 30-60 days (e.g., Krishnamurti and Bhalme, 1976; Krishnamurti and Ardanuy, 1980; Annamalai and Slingo, 2001; Randel and Park, 2006). Zhang et al. (2002) found that the center of the anticyclone shows bimodality in its longitude location that they classify in terms of the Tibetan mode (centered at about 90° E) and the Iranian mode (centered at about 60° E), although the degree of bimodality appears dependent on the meteorological dataset (Nützel et al., 2016). A number of recent studies have shown that sub-seasonal scale dynamical processes in the ASM region may play a significant role in UTLS transport of trace gases (Yan et al., 2011; Garny and Randel, 2013; Pan et al., 2016; Vogel et al., 2016). It is evident that diagnosing intra-seasonal variability of chemical tracers in the UTLS and their interactions with dynamical fields is important for a more complete understanding of the ASM anticyclone's chemical impact.

Satellite observations provide an essential source of information in ASM UTLS-related studies. Data from limb viewing sensors, the Aura Microwave Limb Sounder (MLS), and the Atmospheric Chemistry Experiment Fourier Transform Spectrometer (ACE-FTS), in particular, are two widely used datasets for this purpose (e.g., Park et al., 2007; Randel et al., 2010). These limb sounders offer relatively high vertical resolution but have limited horizontal sampling on daily timescales. Nadir-viewing instruments, on the other hand, offer better horizontal sampling and daily coverage but have limited vertical resolution and are primarily used to study column abundances.

This study aims to examine the representation of sub-seasonal chemical variability in the ASM UTLS from limb and nadir-viewing sensors. Two specific nadir datasets we explore are CO from the Infrared Atmospheric Sounding Interferometer (IASI) and $O_3$ from the Ozone Monitoring Instrument (OMI). These two data sets will be examined

together with MLS CO and $O_3$ data, respectively. CO is a pollution tracer and is also an effective tracer of transport in the troposphere and lower stratosphere (e.g., Bowman, 2006), with a photochemical lifetime of ~2 months in troposphere (Xiao et al., 2007). $O_3$ is an effective transport tracer in the UTLS because of the large gradient in its mixing ratio across the tropopause and its long lifetime relative to transport time scales in the UTLS region. In the UTLS, $O_3$ mostly serves as a stratospheric tracer although it also has tropospheric pollution sources. Short-term variations of $O_3$ in the UTLS are largely linked to synoptic scale disturbances in the tropopause region (e.g., Shapiro, 1980). These satellite datasets are examined with meteorological analyses from the Global Forecasting System (GFS) to address the following questions: 1) Do these nadir viewing instruments, designed primarily for retrieving trace gas column abundance, have sufficient information to show the ASM dynamically-driven trace gas distributions and variability at UTLS levels? 2) Are the data from nadir sensors consistent with the limb-viewing data on sub-seasonal scales with respect to dynamical variability of tracers in the ASM region? 3) What can we learn from the complementary information from limb and nadir viewing instruments?

Although the IASI CO and OMI $O_3$ are compared with MLS CO and $O_3$, respectively, it is not the goal of this work to evaluate the quantitative agreement between the nadir and limb data. The difference in viewing geometries makes limb-viewing and nadir-viewing datasets fundamentally different quantities since the air masses they are sensing represent very different volume and spatial extent (described in section 2). The goal of these comparisons, therefore, is to evaluate whether data from the two types of sensors provide a consistent picture of the ASM dynamical impact on the UTLS tracer distributions and variability. We characterize this type of analysis as *process-based retrieval evaluation*. This analysis provides a perspective of whether the high density horizontal sampling from the nadir sensors supplements information from the limb viewing sensors, despite the relatively coarse vertical resolution, in the region of strong synoptic scale horizontal dynamical variability. This work is therefore not a validation study, but rather, aims to complement previous validation studies of the MLS CO and $O_3$, the IASI CO and the OMI $O_3$ profile product (Livesey et al., 2008; George et al., 2009; Liu et al., 2010a, 2010b; Kroon et al., 2011; De Wachter et al., 2012; Bak et al., 2013; Safieddine et al., 2016; Barret et al., 2016; Huang et al., 2017).

Although both CO and $O_3$ are examined in this work, the focus is on their relationship with the ASM dynamical structure. No attempt is included here to use tracer-tracer relationship as an additional diagnostic for UTLS transport. In the CO analysis, we focus on the upper troposphere (UT) variability associated with convective pumping and the horizontal redistribution by the dynamics of the anticyclone. In the $O_3$ analysis, we focus on the tropopause level and the sensitivity of data to the tropopause structure. Overall, we give more focus on the CO analysis.

## 2 Data Description

### 2.1 Satellite data

For limb-viewing observations, we use MLS Version 4, level 2, 215 hPa CO, 147 hPa CO and 100 hPa $O_3$ data. MLS is a forward-looking sounder on board the Aura satellite launched in July 2004 (Waters et al., 2006). Accurate data

descriptions of CO and $O_3$, including uncertainties, are given in Livesey et al, (2017). Briefly, the vertical resolution of CO retrievals at 147 hPa (215 hPa) is 5.1 km (5.4 km) and the single profile precision is ~16 ppbv (19 ppbv). The systematic uncertainty for 147 hPa CO is the root-sum-square (RSS) of 26 ppbv and 30%. For 215 hPa CO it is 30 ppbv and 30%. The $O_3$ retrieval has a vertical resolution of 3 km at 100 hPa. The single profile precision is estimated to be ~30 ppbv. The systematic uncertainty for 100 hPa $O_3$ is estimated to be 5 ppbv + 7% (Livesey et al., 2017). As a limb sounder, MLS's field of view produces a horizontal resolution of ~6 km across the track and ~300 km, 570 km, and 590 km along the track for 100 hPa $O_3$, 147 hPa, and 215 hPa CO, respectively, MLS has a relatively low daily sampling density (~240 limb scans per orbit with ~3500 profiles during both day and night). In order to make the daily output easier to interpret, daily maps are made by interpolating the output onto a regular grid.

Nadir-viewing observations of CO are obtained from IASI (level 2 data) aboard EUMETSAT's Metop satellite. IASI measures the 'thermal infrared' (TIR) spectrum emitted by the Earth-atmosphere system with twice daily near-global coverage (with 4 simultaneous pixels of 12 km diameter every 50 km), but limited vertical resolution (Clerbaux et al., 2009). The tropospheric CO product is derived from the spectra using the FORLI retrieval algorithm, which uses a single *a priori* profile and covariance matrix (Hurtmans et al., 2012; George et al., 2015). The IASI CO level 2 retrieval product is provided as mixing ratios in 19 1–km layers from surface to 19 km altitude. The retrieval information content analysis, however, shows 0.8 to 2.4 (1.5 to 2.0 at mid-latitudes) 'independent pieces of information' (or degrees of freedom for signal (DOFS); George et al. 2009). How well this information content allows IASI CO retrieval to capture upper tropospheric variability at mid-latitude and tropical latitudes is one of the foci of this study. Our analysis will complement previous validation studies, including in situ measurements from the Measurements of OZone, water vapor, carbon monoxide and nitrogen oxides by Airbus In-service airCraft (MOZAIC) project (correlations ~ 0.7; De Wachter et al., 2012), and satellite observations from the Measurements Of Pollution In The Troposphere (MOPITT) instrument (George et al., 2015). This work also aims to complement previous IASI data analyses which shows the data reproduce monthly mean large-scale features in the UTLS over the ASM region comparable to model results from GEOS-Chem (a chemical transport model coupled to meteorological analysis from the Goddard Earth Observing System GEOS-5; Barret et al., 2016).

Nadir-viewing observations of $O_3$ are obtained from OMI, an $O_3$ sounder aboard the Aura satellite that provides daily global coverage at 13 km x 24 km footprint (Levelt et al., 2006). OMI $O_3$ products include retrievals of both total $O_3$ columns and vertical profiles. In this study, we use the $O_3$ profile product by Liu et al. (2010b) and Huang et al. (2017). $O_3$ profiles are retrieved at 24 vertical layers covering the surface to ~60 km using the optimal estimation technique constrained by a monthly and zonal mean $O_3$ profile climatology (McPeters et al., 2007). The OMI profile retrievals have 6.0-7.0 degrees of freedom (5.0-6.7 in the stratosphere; Liu et al., 2005; Liu et al., 2010b; Liu et al., 2010a). The distribution of the information content is sufficient to resolve the UTLS transition region in part owing to the large $O_3$ gradient across the tropopause, as demonstrated by a number of previous works (Pittman et al., 2009; Liu et al., 2010a; Liu et al., 2010b; Bak et al., 2013). For vertical distribution of the averaging kernels and information content, see Liu et al. (2010b). In this work, we use a level-3 product gridded to 1° longitude x 1° latitude horizontal resolution. Only

the layer 18 product is used, which is a layer centered approximately at the 100 hPa level, with retrieval information from a broad layer of approximately 10 km (Liu et al., 2010b). In the ASM anticyclone region, this layer is contributed more from the UT inside the anticyclone and from the LS outside. OMI has known cross-track dependent biases (Liu et al., 2010a; Liu et al., 2010b). Thus the data points from view zenith angles (VZA) greater than 58° are not used in
the mapping process.

To highlight the horizontal sampling density and vertical sensitivity differences between the limb viewing and nadir viewing sensors, Figure 1 shows the geolocations of all IASI and MLS profiles and the relevant averaging kernels for the study domain (0-180° E, 10° S- 60°N) in a single day (August 1, 2008). Both daytime and night-time samplings are
included. It is apparent from Fig. 1a that IASI has a much denser horizontal coverage than the MLS. Note that both datasets have data gaps: while the MLS orbit tracks are separated by ~ 20 degree longitudes, the IASI retrieval also has significant data gap each day due to the cloud coverage (no IASI products are available if the cloud fraction in the pixel exceeds 25%). The comparison of $O_3$ data sampling densities between OMI and MLS is not shown but it is conceptually similar to that is shown in Fig. 1a. This disparity of sampling density and its implications for representing
synoptic scale variability motivates this work of exploring the utility of nadir viewing data in characterizing chemical distributions in the UTLS on daily to sub-seasonal timescales.

Figure 1b shows the vertical information distribution for both IASI and MLS UT CO retrievals. The IASI averaging kernels for the CO product in 19 layers are shown, which are the average of all individual profiles included in Fig.
1a. Note that the altitude labels for these layers are referring to the centers of the 1-km layer as part of the product identification and they are not intended for representing independent information from each layer. The physical information for these layer products is contributed from a broad layer, as indicated by the averaging kernels (Fig. 1b). To provide a perspective of retrieval information content in the study region, we show the distribution of DOFS for IASI profiles in Fig. 1c. The distribution shows that the majority of the profiles are estimated to have DOFS
close to 2, which supports that IASI CO retrievals should have sufficient information for independent variability in the upper and the middle troposphere. In this study, we aim to evaluate the upper tropospheric CO variability using the IASI CO product. The most relevant retrieval product layers are 12-16 km. Averaging kernels for these layers are highlighted in Fig 1b.

MLS vertical sensitivity is shown by the standard CO averaging kernels for the 215 hPa, 147 hPa and 100 hPa product (Livesey et al., 2017). Although we focus on the 147 hPa product in this analysis, 215 hPa and 100 hPa averaging kernels are included in the figure to contrast the sensitivity distributions in the two instruments. The figure provides a perspective that IASI retrieval information content is optimized for the middle troposphere. The UT information is much weaker, maximized over a range of UT layers, and is not sharply peaked at a particular retrieval
layer. In contrast, MLS information for 147 hPa CO shows a strong maximum near 14 km (~ 150 hPa). The figure also indicates that both the nadir and the limb viewing sensors are expected to have "smoothing errors" in the retrieval.

A similar figure for OMI is not shown, since for ozone analysis we are not focusing on independent information between the upper and lower-to-middle troposphere, rather we focus on stratospheric versus tropospheric influence in ozone distribution near the tropopause level (~100 hPa pressure level) and expect the contrast between the air mass inside and outside the anticyclone to be dominated by the tropopause structure of the region. For more complete averaging kernel discussions, see the MLS data quality document (Livesey et al., 2017), the work of George et al. (2009) for IASI data, and Liu et al. (2010b) for OMI data.

**2.2 Meteorological analysis data**

We use wind fields, geopotential height (GPH), tropopause height, and potential temperature (derived from temperature and pressure) from the GFS operational analysis (a product of the National Centers for Environmental Prediction; NCEP) to diagnose the dynamical variability of the ASM anticyclone. These 6-hourly data have a horizontal resolution of 1° on 26 pressure levels (from 1000 hPa to 10 hPa) (National Centers for Environmental Prediction/National Weather Service/NOAA/U.S. Department of Commerce, 2000). Having pressure and height for the tropopause in the product, and the determination of these levels using the native GFS grid is a major strength of the product that motivated our choice (Pan and Munchak, 2011).

**3 Processing daily maps**

Figure 1a highlights the sampling gaps from both MLS and IASI for mapping daily CO distributions. Careful data interpolation and smoothing to fill data gaps are essential steps for producing daily maps from the available retrievals in each given day. In general, the daily representation from MLS data requires interpolation to increase the density in coverage, while the IASI (and OMI) data densities are reduced by binned averages. We have explored three interpolation algorithms, cosine smoothing, natural neighbour, and inverse distance, for mapping data. All three methods are similar, conceptually, in filling an empty cell with weighted mean of nearby observations, but the weightings are determined differently. After experimenting with various grid sizes and mapping methods, we choose to use 5° x 5° longitudes and latitudes for mapping MLS data and 3° x2° for the IASI data. The results shown in this paper are mapped using the natural neighbour method (Watson, 1992) and followed by a Gaussian smoothing. We find that these steps produce daily maps with a good balance between representing the synoptic scale variability and the information from the data in localized structures.

Figure 2 provides an example using 1 August 2008 data, where maps of retrieved MLS CO (Fig. 2a), interpolated MLS 147 hPa CO (Fig. 2b), and IASI CO in the UT layer (Fig. 2c) are shown. The IASI UT layer CO mixing ratio is produced by interpolating the layer product to 150 hPa. Although the interpolation aimed to find the CO mixing ratio approximation for 150 hPa, it is clear from the averaging kernel that the retrieval represents a layer. The retrieval information for this layer is represented by the averaging kernels for 12.5-15.5 km layers as highlighted in Fig. 1b. To emphasize this limitation in vertical resolution, we refer to this layer as the UT layer in the rest of the paper. Note we have used different color-scales for MLS and IASI CO and the rationale is given in next section. The dynamical fields of 150 hPa GPH and the horizontal wind are superimposed for identifying the location and structure of the ASM

anticyclone. Comparison of the MLS data on the orbital tracks (Fig 2a) and the interpolated map (Fig 2b) provides a useful perspective that the mapping procedure we choose highlights the large scale dynamical consistency of the CO and the flow pattern instead of the fine scale structure. Comparison of MLS (Fig. 2b) and IASI (Fig. 2c) CO maps provide additional perspective that although both datasets show CO enhancement in the region of the ASM anticyclone, the appearance and detail of the enhancement are quite different. These differences are contributed by several factors. For example, the missing data in IASI (due to cloud contamination) and the larger grid size in MLS may both contribute to the difference in the spatial pattern of the enhancement between 90°–120° E and 20°–30° N. Similarly, the filamentary structure in IASI CO near 150° E and 30° N, although hinted in the MLS orbital data, are represented differently in the MLS CO map. Additionally, IASI signal-to-noise ratio is likely degraded over the region of elevated terrain, which will be discussed in later examples. The two datasets are also obtained in slightly different sampling times, as the MLS has a 1:30 equator crossing time and the IASI orbit has a 9:30 equator crossing time. These factors need to be kept in mind when interpreting the details.

**4 Comparisons of MLS and IASI CO**

Although the focus of this study is to characterize chemical tracers' space-time variability and dynamical consistency, we make quantitative comparisons between the MLS and IASI CO data in this section. The comparisons focus on the consistency between the two datasets in representing CO UT variability in the study domain and their representation of the well-demonstrated large-scale spatial pattern associated with the ASM anticyclone on the seasonal scale. Vertical ranges of the data were chosen to optimize the overlap of information from nadir and limb viewing instruments with the vertical extent of the anticyclone. Based on the analyses of the dynamical fields and trajectory calculations, the maximum chemical confinement in the anticyclone in the vertical range is between 200-100 hPa or 12 -16 km (Randel and Park, 2006), although elevated levels of tropospheric tracers in the ASM anticyclone are evident up to 68 hPa in the MLS data (Park et al., 2007). Moreover, the strongest closed circulation of the anticyclone occurs at ~14-15 km, above the main convective outflow level (~ 12 km) (Park et al., 2008). Based on this structure and the vertical information content of IASI CO retrieval (George et al., 2009), we choose to use the IASI UT layer CO mixing ratio and the 147 hPa MLS CO retrieval product. The June-July-August (JJA) season of a single year of 2008 is examined.

Figure 3 shows a scatterplot of IASI CO in the UT layer versus MLS 147 hPa CO level-2 product. Each point represents a co-located daily average in a 10 x 6 degrees longitude-latitude bin in the study domain for all days in the JJA 2008 period. The scatterplot shows that variations of CO in the two datasets are generally consistent and correlated (r = 0.8), although the IASI CO shows a smaller range of variability than MLS (indicated by the slope of the linear fit, 0.55). The smaller variability in IASI CO is likely contributed by a weaker detection sensitivity in the upper troposphere and the use of a single a priori profile in CO retrieval (George et al., 2015).

Figure 4 shows JJA seasonal averages for (a) MLS 147 hPa CO, (b) MLS CO average of 147 and 215 hPa products, and (c) IASI UT layer CO for 2008. Note that different ranges are used in MLS and IASI color bar to adjust for the smaller range of variability in IASI CO as indicated in Fig. 3. Selected GPH contours and wind vectors at 150 hPa for

the same period are shown on all three maps to indicate the seasonal mean location of the anticyclone. The chemical signature of the ASM anticyclone is evident for all three seasonal averages. The clear chemical signature indicates that, despite the relatively weak UT sensitivity, the IASI data are capable of showing the impact of the ASM circulation on UT CO. Spatially, the MLS 147 hPa and IASI UT layer CO mixing ratio fields show noticeable differences in their

horizontal locations. The IASI enhancement pattern shows an overall eastward shift relative to the MLS. There is a pattern of strong enhancement between 120° and 150° E in IASI CO seasonal average that is not clearly present in the MLS 147 hPa seasonal average. In view of the broad vertical structure in IASI averaging kernels for the UT layer, we also constructed a seasonal average layer using both MLS 147 hPa and 215 hPa products (Fig. 4b), which has a region of CO enhancement very comparable to the IASI UT CO pattern. This result shows that IASI UT layer CO is

consistent with MLS CO from the combined 147 hPa and 215 hPa product on seasonal time scales. The comparison also suggests that the region's UT CO enhancement has an east-west tilted vertical structure. The dynamical factors that contribute to the tilted chemical structure likely involve the vertical range of the anticyclone confinement and the altitudinal distributions of the easterly and westerly jets.

In addition to the weaker enhancement and the location offset, CO enhancements over the Tibetan and Iranian plateaus are largely missing in the IASI average. This is likely a result of weakened signal-to-noise ratio in the nadir sensor retrieval due to the higher surface elevation. We re-visit this issue in later sections using daily examples.

Overall, these comparisons provide quantitative and qualitative characterizations on the seasonal and the ASM

regional scale variabilities represented in the IASI CO product relative to the MLS data, which has been widely used to investigate chemical tracer distributions and transport in this region (e.g., Park et al., 2007; Santee et al., 2017). We now proceed to analyse the sub-seasonal scale variability.

**5 Sub-seasonal variability of ASM UT CO from MLS and IASI data**

**5.1 UT CO variability from MLS data**

We begin our examination of daily maps in Fig. 5, which shows mixing ratios of MLS CO at 147 hPa and the dynamical fields (winds and GPH) at 150 hPa for selected days. During the time period, the dynamical evolution of the anticyclone, as indicated by the selected GPH contours, shows different phases of the east-west oscillation (Pan et al., 2016). In this sequence, the anticyclone was initially in the Tibetan mode (July 16, when the maximum of the anticyclone as represented by the GPH was located near the southern edge of the Tibetan plateau). In subsequent days

the anticyclone elongates, and the center migrates westward toward the Iranian mode (July 18). As the anticyclone further elongates, the center eventually splits, and the anticyclone forms a double center (July 22), with the two maxima located around 30°E and 80°E and a hinted 3[rd] center near 135°E as indicated by the wind field. During this time period, the center of maximum CO enhancement also migrated westward from south of Tibetan plateau (16 July), to around 60°E (18 July) and 30°E (22 July). Additional effect of the anticyclone elongation is the appearance of an

additional CO maximum east of the Tibetan plateau, which eventually migrates eastward to western Pacific near

southern Japan (see Fig. 7a, 26 July), similar to the configuration of 16 July (Fig. 5a). The July 16–26 time period therefore provided an example of ASM dynamical and chemical variations in a cycle of 10-20 day east-west oscillation. We refer this additional CO maximum and associated anticyclonic circulation over the region of western Pacific near Japan as the western Pacific mode, which is likely related to a system locally referred to as Bonin High (Enomoto et al., 2003).

To quantify the correlation of CO enhancement with the dynamics of the anticyclone east-west oscillation, we compare the anomaly fields of the GPH and CO through the season in Hovmöller diagrams (Figure 6). The Hovmöller diagrams are constructed by first calculating daily mean GPH and CO in the latitude band of 10°-40° N and 0°-220° E. Note we have extended the longitude range further east in this calculation to include a larger back ground. This is because the study region is dominated by three highs. Including the region outside the highs is necessary for identifying the highs as positive anomalies. The anomaly is derived for each 5° longitude bins by subtracting the daily mean. The mean correlation of the spatial (longitudinal) variability between the CO and GPH anomalies for the three-month period is 0.92. Note that the Hovmöller diagram shown in Fig. 6 is constructed using the interpolated CO field. If using the retrieved CO data only, this correlation is significantly weaker due to the sparse sampling of MLS data. As a comparison, a similar analysis using a global model shows correlation of ~ 0.7 (Pan et al., 2016).

Note that this analysis is very similar to a previous work of ASM dynamical and chemical variability in the context of eddy shedding. Using low PV air as the dynamical tracer, the correlation analyses between daily PV and MLS CO data at 370 K during one season (May-September) resulted in a spatial correlation of ~0.5 (Garny and Randel, 2013).

Figure 6 demonstrates that the UT CO distribution is closely linked to the upper tropospheric dynamical variability of the anticyclone. The dynamics of this east-west oscillation phenomenon is the focus of a number of works (e.g., Hsu and Plumb, 2000; Popovic and Plumb, 2001; Liu et al., 2007) where convective pumping of low PV air to the upper troposphere, followed by eddy shedding creates the transient behaviour of the anticyclone. The persistent low PV at the tropopause level occurs around 90°E (Popovic and Plumb, 2001; Garny and Randel, 2013), which is considered the center of the Tibetan plateau mode (Zhang et al., 2002). The low PV air propagates both westwards and eastwards. A model analysis using CO as a tracer further concludes that the vertical transport of boundary layer air predominantly occurs near the southern flank of the Tibetan plateau, and the enhanced CO over the entire anticyclone is a result of transient mixing and anticyclone confinement (Pan et al., 2016). In Fig. 6 both GPH and MLS CO shows stronger westward propagation in 10-20 periods and relatively smaller eastward propagation.

Overall, Figs. 5 and 6 show that, despite the limited horizontal sampling, MLS data provide enough information to successfully capture the day to day co-variability of CO with the dynamical fields with the help of careful mapping procedures.

**5.2 CO variability associated with ASM dynamics from IASI data analyses**

We begin the IASI data discussion by evaluating the information content in the IASI UT CO retrieval. Although in the literature the term "retrieval information content" almost always refers to the DOFS calculated using forward and retrieval models, we propose an alternative way of demonstrating the information content in this work through the analyses of dynamical consistency. This type of evaluation may bring new insight into the retrieval, since it evaluates the result of the retrieval, which may vary depending on how the sensitivity represented by the DOFS is used.

Figures 7b-c show an example of daily IASI CO maps in two layers: the UT layer and the middle troposphere (MT) layer. Th MT layer is derived from interpolating the IASI layer product to 500 hPa. The two layers are chosen to examine the dependency of the retrieval between the upper tropospheric and mid-tropospheric CO. Dynamically, these two layers are associated with distinct flow patterns, which should have clear signatures in the CO distribution. Comparing the CO fields between these two layers and with the flow patterns, shown by the 150 hPa and 500 hPa GPH and winds, provide an effective test whether the retrieval sensitivity is sufficient to resolve independent upper and lower/middle tropospheric CO variability. This result complements the retrieval information content calculated from DOFS, as shown in Fig. 1c. As a reference, we have also included the MLS CO map for the same day using the 147 hPa product (Fig. 7a).

This chosen day (26 July 2008) follows the sequence of days from Fig. 5 for MLS and Fig. 8 for IASI. Dynamically, the upper tropospheric anticyclone is in a "tri-center" phase of the east-west oscillation, following the elongation shown in Fig. 5. There are three anticyclonic centers: the strongest one over the Tibetan plateau (~ 90°E) and the second near the border of Iranian and Iraq (~ 50°E), both indicated by the maxima of GPH; The third center is over the western Pacific near 140°E, with the closed circulation indicated by the wind arrows. The IASI UT CO mixing ratio map shows a high degree of consistency with the flow pattern at the 150 hPa level, and the distribution in this layer does not appear to be correlated with the MT CO mixing ratio map. This example demonstrates the capability of IASI retrieval to produce CO distribution in the ASM upper troposphere independent from the lower to middle troposphere CO.

Figures 7a and 7b provide another case comparison between the maps based on MLS and IASI (note the different color-scales), adding to the case in Fig. 2. Although the two maps visually show different areas of "hot spots", the overall patterns of CO enhancement are very comparable if using the area greater than ~ 85 ppbv in the MLS map and that of greater than ~ 65 ppbv in the IASI map. Over the Tibetan plateau, the IASI CO map shows decreased enhancement in the high GPH center, consistent with degraded signal-to-noise ratio due to the high terrain (marked by grey shading in Fig. 7c), while over the western Pacific, the IASI CO enhancement is more intense. This comparison provides a single day example and complements the information in Fig. 4 and the associated discussions.

Figure 8 shows the IASI UT CO mixing ratio maps during the same period as MLS maps in Fig. 5. The overall CO enhancement patterns are very comparable to the MLS data if comparing the area of 65 ppbv or greater with MLS

values 85 ppbv or greater. The IASI maps, however, shows additional finer scale structures, consistent to the flow pattern. Similar to the previous example, all three maps show the weakening of CO enhancement over the region of high elevation both over the Tibetan and Iranian plateaus. In all three cases, the IASI maps show much stronger CO enhancement over or around the western Pacific High.

Note that physically there is no reason to expect a perfect correlation between the CO maximum and the GPH maximum, since the dynamical field and the CO mixing ratios are controlled by different processes (Garny and Randel, 2013). A significant correlation in the UT reflects the strong influence of the anticyclone dynamics on the air mass and persistent boundary layer emission and convective pumping. The interesting differences between the MLS and IASI

UT CO enhancement over the western Pacific, again, suggest that the IASI UT retrievals have a broad vertical sensitivity, as shown by the averaging kernels (Fig. 1b).

In addition to UT horizontal variability, IASI data provide opportunities to investigate vertical structure of CO in the monsoon region. One of the significant conclusions from a model study (Pan et al., 2016) is that the upper tropospheric CO enhancement over the Iranian Plateau is not formed by the vertical transport from the local boundary layer. Rather,

it is produced by the westward shedding from the upper troposphere over the region associated with the Tibetan mode. Similar hypothesis can be made for the western Pacific enhancement. We examine the IASI CO cross-sections to search for observational evidence for verifying these hypotheses. Four examples are shown in Fig. 9. These four pressure-latitude cross-sections are selected to examine the vertical structure in the centers of the Tibetan, Iranian and Western Pacific mode. The locations of the cross-sections are marked on the maps in Figs. 7 and 8.

The cross-section in Fig. 9a is at the center of the Tibetan mode (see Fig. 8a for map). The CO enhancement in this case extends from the surface to near 14 km, with a vertical structure consistent with the flow field, i.e., the vertical structure of the enhancement is collocated with the region of strong vertical winds over northern India and the southern flank of the Tibetan plateau. Dynamically, this is identified as the ascending branch of the monsoon Hadley cell (Wang, 2006). For more discussion on the climatological flow structure in the meridional plane, see analyses in

Zhang et al. (2002). This example also shows that in this region, the plateau is taking away approximately half of the atmosphere, consequently degrading the nadir sensor's signal-to-noise ratio for retrieval, leading to a weakened CO enhancement over the plateau at higher altitude. This factor likely contributed to the difference between MLS 147 hPa and IASI UT CO data based maps over the plateau (see Figs. 5a and 8a).

The cross-sections in Figs. 9b and 9d are two examples of the CO enhancement over the western Pacific High. In both

cases, the enhanced layers are shown in the upper troposphere. Similarly, Fig. 9c shows an example of an enhanced UT CO layer near the southern edge of the Iranian plateau. In all three cases (Figs. 9b-d) the wind fields indicate a change of circulation from strong vertical motion in the lower-mid troposphere to the horizontal flow dominated upper troposphere. Overall, the cross-sections support the hypothesis that the UT CO enhancement over the middle east and the western Pacific are not a result of local vertical transport but are produced by UT redistribution via westward and

eastward eddy shedding.

Figure 9 not only provides observational evidence supporting the model-based hypothesis on transport structure, it also provides evidence supporting the ability of the IASI retrieval to resolve independent variability in the upper

tropospheric CO. Note that in each cross-section, we have also included the retrieval *a priori* profile as the left-most column. Since the IASI retrieval uses a single *a priori* profile, the left-most column on each of the four panels are identical. The UT variability shown in each cross-section is not only dynamically consistent but also independent from the lower-to middle troposphere and the *a priori* profile. The effective use of information content in the IASI retrieval is powerfully demonstrated in these cross-sections, complementing and much more enlightening than the averaging kernels shown in Fig. 1b.

Similar to Fig. 6, we show Hovmöller diagrams of daily anomaly fields for 150 hPa GPH and IASI UT CO mixing ratio for JJA 2008 (Fig. 10) to quantify the correlation in sub-seasonal variability. As expected, the weakened retrieval signal over the plateaus produced non-physical structure around 100°E longitude segment. On both the eastern and western edges, the CO anomaly shows a tendency of eastward shift relative to the GPH anomaly, a feature that is consistent with the discussion on Fig. 4. The overall correlation is 0.69.

**6 UTLS O$_3$ analysis using MLS and OMI data**

We now turn our attention to the UTLS O$_3$ from MLS and OMI. While CO is a boundary layer pollution tracer, O$_3$ in the UTLS region is foremost a transport tracer highlighting the influence of the stratosphere, although its distribution can also be affected by photochemical production. Here, the influence of monsoon convection on the UT O$_3$ distribution is somewhat complicated since the polluted air masses tend to have enhanced precursors for ozone production. For these reasons, we focus on analysing ozone variability at the UTLS level using 100 hPa MLS data and the OMI layer 18 product. The large scale O$_3$ distribution at the 100 hPa level over the ASM region reflects the tropospheric influence on the air mass inside the anticyclone in contrast to the stratospheric influence outside. The structure of the bulging tropopause in the monsoon region (Bian et al., 2012; Pan et al., 2016) has a significant influence of the O$_3$ distribution at the 100 hPa level. Lower O$_3$ mixing ratios are expected inside the anticyclone in the layer near 100 hPa since the tropopause is at a lower pressure inside the anticyclone than it is outside in this region. Previous work analysing MLS 100 hPa CO and O$_3$ led to a similar conclusion (Park et al., 2007, Fig. 9). We aim to examine how well the data from MLS, which has relatively sparse horizontal sampling but better vertical resolution, and OMI, which has high density coverage horizontally but with coarse vertical resolution, represent the correlation between the ozone field and the sub-seasonal scale dynamical variability of the tropopause in the ASM region.

**6.1 Comparison of 100 hPa MLS and OMI O3 data on seasonal scale variability**

Similar to the CO analysis, we first compare the two O$_3$ datasets on seasonal time scales. Figure 11 shows 100 hPa MLS and OMI average O$_3$ for JJA 2008. Also included in the figure are seasonal averages of a few selected dynamical fields for the same time period. The 100 hPa wind field is included to show the anticyclonic flow associated with the ASM. The location of the anticyclone is marked by the 16.7 km GPH contour and the contours of tropopause intersection with the 100 hPa and 105 hPa pressure surfaces. The contours of the tropopause pressure and the GPH show a small south-north offset. The 100 hPa O$_3$ gradient change is well aligned with the tropopause contours, supporting the concept of ASM creating a tropospheric "bubble" in the otherwise stratospheric background at this level. Both the MLS and OMI based seasonal mean show low O$_3$ in the area of higher tropopause as expected. MLS O$_3$

shows a band of high $O_3$ near the southern edge of the anticyclone. This is a well-known dynamical structure associated with the mixing of high latitude stratospheric air driven by the anticyclonic flow (e.g., Konopka et al., 2010). This band of high $O_3$ appears weaker on the OMI map. The average of the finer structure with spatial variability and the limitation of the coarse vertical resolution in detecting a shallow layer may both contribute to the weaker seasonal appearance.

To evaluate the consistency in representing variability in daily data, Figure 12 shows a scatterplot of OMI versus MLS daily grid point average $O_3$ near the 100 hPa in the study region over the JJA 2008 period. The grid point average is done daily in each co-located 10 x 6 degree longitude-latitude box through the study domain. This figure is similar to the CO scatterplot in Fig. 3, but the correlation between the OMI and MLS $O_3$ is much better with both the slope (0.94) and the correlation coefficient (0.96) near unity.

Figures 11 and 12 characterize the good overall agreement between OMI and MLS $O_3$ on seasonal and ASM regional scales. We now proceed to examine the daily and sub-seasonal variability represented by the two datasets.

**6.2 Representation of sub-seasonal scale variability from MLS and OMI $O_3$**

Figure 13 shows maps of MLS 100 hPa $O_3$, OMI layer 18 $O_3$ and the tropopause pressure for two selected days in July 2008. Dynamical fields of the GPH and horizontal wind are superimposed on the $O_3$ maps. The 105 hPa tropopause contour is included in all maps. Both sets of $O_3$ maps exhibit the characteristic low $O_3$ mixing ratios inside the anticyclone. Here the 105 hPa tropopause contour appears to correlate well with the $O_3$ and wind field gradients. Note that the tropopause pressure here is from the GFS final analysis product, which is based on the WMO thermal tropopause definition. Since this quantity is derived from the vertical gradient and is not analysed on the pressure surface, it's intersection with the pressure surface can appear noisy. Gaussian smoothing is applied to the 1 x 1 degree tropopause data on all maps.

In the two selected days, the dynamical structures of the anticyclone are in two different phases as discussed in relation to Figs. 5 and 8. The ASM influence at the tropopause level shows a wider longitudinal range on the 18[th] (approximately 20°-130° E), and it is westward migrated on the 22[nd] (approximately 10°–110° E) and with a double-centered structure. The OMI $O_3$ map on 18[th] shows a close correspondence with the longitudinal range of the tropopause pressure, while the MLS map shows a westward shift of the low $O_3$ area. The difference in horizontal sampling density is likely a contributor. On 22[nd], both MLS and OMI $O_3$ gradients are well co-located with the anticyclone boundary as indicated by the 105 hPa tropopause contour. The MLS $O_3$ structure shows a more well-defined double-centered structure. OMI map shows a smaller $O_3$ depression over the Tibetan plateau. We speculate that surface elevation may have contributed to the structure in OMI $O_3$, similar to the IASI CO discussion. The high ozone band on the southern side of the anticyclone shows a large difference between MLS and OMI, with MLS having a much wider structure. Both the coarser horizontal sampling of MLS and the coarser vertical resolution of OMI for resolving this shallow layer may contribute to this difference.

The Hovmöller diagrams in Fig. 14 examine sub-seasonal variations and the relationship between the tropopause pressure and 100 hPa $O_3$ field during JJA season of 2008. All three fields in the figure are dominated by the persistent location of the anticyclone as indicated by the lower tropopause pressure and of $O_3$ mixing ratios between 30°E and

100°E. All three Hovmöller diagrams exhibit westward propagation in 10-20 day timescales. The correlation in the variability along the longitudinal dimension is 0.90 between the tropopause pressure and MLS $O_3$, and 0.76 between the tropopause pressure and OMI $O_3$. In both cases, the interpolated fields are used to calculate the correlations. The strong correlation between the tropopause structure and $O_3$ supports the conceptual model that the higher tropopause over the ASM forms a region of tropospheric "bubble" above the mean level of tropical tropopause for the season. This structure enables a unique transport pathway for air masses in the "bubble" to enter the lower stratosphere via horizontal eddy shedding, bypassing the equatorial tropical tropopause (e.g., Garny and Randel, 2016; Ploeger et al., 2017).

While the two $O_3$ datasets provide generally consistent large scale ozone structure, there are visible differences between MLS and OMI in small-scale structures. Potential impacts of clouds on retrievals at 100 hPa is discussed in a recent OMI validation study (Huang et al., 2017). The weaker $O_3$ depression near 90°E is likely contributed by the impact of surface elevation on the OMI retrieval. A better understanding of the small-scale structures can benefit from validation studies using airborne measurements targeting the ASM UTLS structure.

## 7 Conclusions and discussions

We have examined space-time variability of chemical tracers in the UTLS associated with the ASM represented by nadir viewing (IASI and OMI) satellite instruments in comparison with a widely used limb viewing (MLS) dataset. Using CO (a boundary layer pollution tracer) and $O_3$ (a stratospheric tracer), we focus on the strengths and limitations of these data for representing the distribution and variability of UTLS chemical tracers in the region of the dynamically variable ASM anticyclone. We explore whether the much denser horizontal samplings of the nadir sensors provide information complementary to the higher vertical resolution limb data for the tracer daily distribution in response to synoptic scale variability.

Our CO analysis shows that, despite a relatively coarse horizontal sampling on daily timescales, interpolated MLS 147 hPa daily CO field exhibits a high degree of correlation with the dynamical variability on synoptic scales (Figs. 5 and 6). The spatial correlation between the CO anomaly and the GPH anomaly at the 150 hPa for the ASM region is 0.92 for the 2008 JJA season studied. The same correlation for IASI CO is much weaker (r = 0.69) (Fig. 10), largely due to the missing UT enhancement over the elevated surface of the Tibetan plateau. There is also an eastward shift in CO positive anomaly pattern relative to the GPH. A comparison between IASI and the MLS CO seasonal averages leads to an insight that IASI UT CO includes contributions from a broad layer, comparable to the range of the combined 147 hPa and 215 hPa MLS product, which is consistent with the broad vertical structure shown in IASI averaging kernels.

Quantitatively, IASI UT CO shows a consistent variability with the MLS 147 hPa product over the ASM season and region, although IASI CO has a smaller range of variability and misses the enhancement over the plateaus, likely due to the regions' elevated surface which reduces the nadir viewing sensor's signal (Figs 3 and 4). On daily to weekly time scales, IASI's data resolve finer structures in CO distribution owing to its higher horizontal sampling density. The most important complementary information is provided by IASI vertical cross-sections (Fig. 9), which provide

information identifying the region of upward transport. Selected examples provided first observation evidence supporting the model-based hypothesis that the large-scale UT enhancement over ASM is a combined result of vertical pumping and horizontal re-distribution at UTLS level via eddy shedding (Pan et al., 2016).

In the $O_3$ analysis, nadir sensor data from OMI shows a good agreement with MLS $O_3$ near the 100 hPa level when averaged seasonally and when compared using 10 x 6 longitude-latitude grid point daily average (Figs. 11 and 12). The dynamical consistency of OMI $O_3$ mixing ratios in the layer 18 product (centered near 100 hPa) on seasonal and sub-seasonal timescales demonstrates the sufficient information for the nadir viewing datasets to contribute to the ASM dynamically-driven UTLS $O_3$ variability. Both MLS and OMI $O_3$ variability in the region exhibit good

correlations with the tropopause pressure, supporting the conceptual model that ASM creates a tropospheric "bubble" above the season's average tropopause in the tropics (Pan et al., 2016).

The CO maps from different layers (Fig.7) and selected cross-sections (Fig.9) both provide strong evidence that IASI has sufficient information content to discriminate upper tropospheric CO variability from that in the lower to middle troposphere. This result is consistent with and complementary to the model estimates of retrieval information content,

which shows that the DOFS for the interested region is approximately 2 (Fig. 1c). The overall dynamical consistency found in IASI CO maps and cross-section demonstrates the value of IASI CO data for ASM transport studies. OMI $O_3$ product in the layer near 100 hPa also shows a high degree of correlation with the MLS product, and dynamical consistency with the variability of the tropopause. Results of this study therefore demonstrate the approach of "process-based" retrieval information content evaluation. This type of evaluation is different from traditional

validation studies, where the goals are focused on retrieval accuracies and precisions, and often involve quantitative comparisons with independent and better trusted data. This type of evaluation also complements the traditional information content analyses based on forward and inverse model calculations, and gives additional physical meaning to information content from data application in process studies.

Overall, our analysis demonstrates the value of high horizontal sampling density from the nadir viewing sensors in

capturing the dynamical variability of UTLS tracer distributions. Although the retrieval has fewer degrees of freedom for each profile, the large number of profiles retrieved daily at finer footprints produces valuable information regarding horizontal dynamical variability. The result of this analysis, not only demonstrated the significant role of ASM sub-seasonal scale dynamics in UTLS chemical distributions, but also bring new insight on the dynamics of the ASM through the differences of these two types of sensors.

**Statement.** The authors declare that they have no conflict of interest.

**Acknowledgments.** This work is in part J Luo's PhD research, funded by the National Science Foundation of China (41705021, 41630421 and 41575038). The work is in part conducted at the National Center for Atmospheric Research,

operated by the University Corporation for Atmospheric Research under sponsorship of the United State National Science Foundation. The IASI mission is a joint mission of EUMETSAT and the Centre National d'Etudes Spatiales (CNES, France). We thank the ULB team (Daniel Hurtmans, Pierre Coheur) for the development of the FORLI-CO

retrieval algorithm, and Mijeong Park for helpful discussions. We also thank three anonymous reviewers for their helpful comments and suggestions.

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

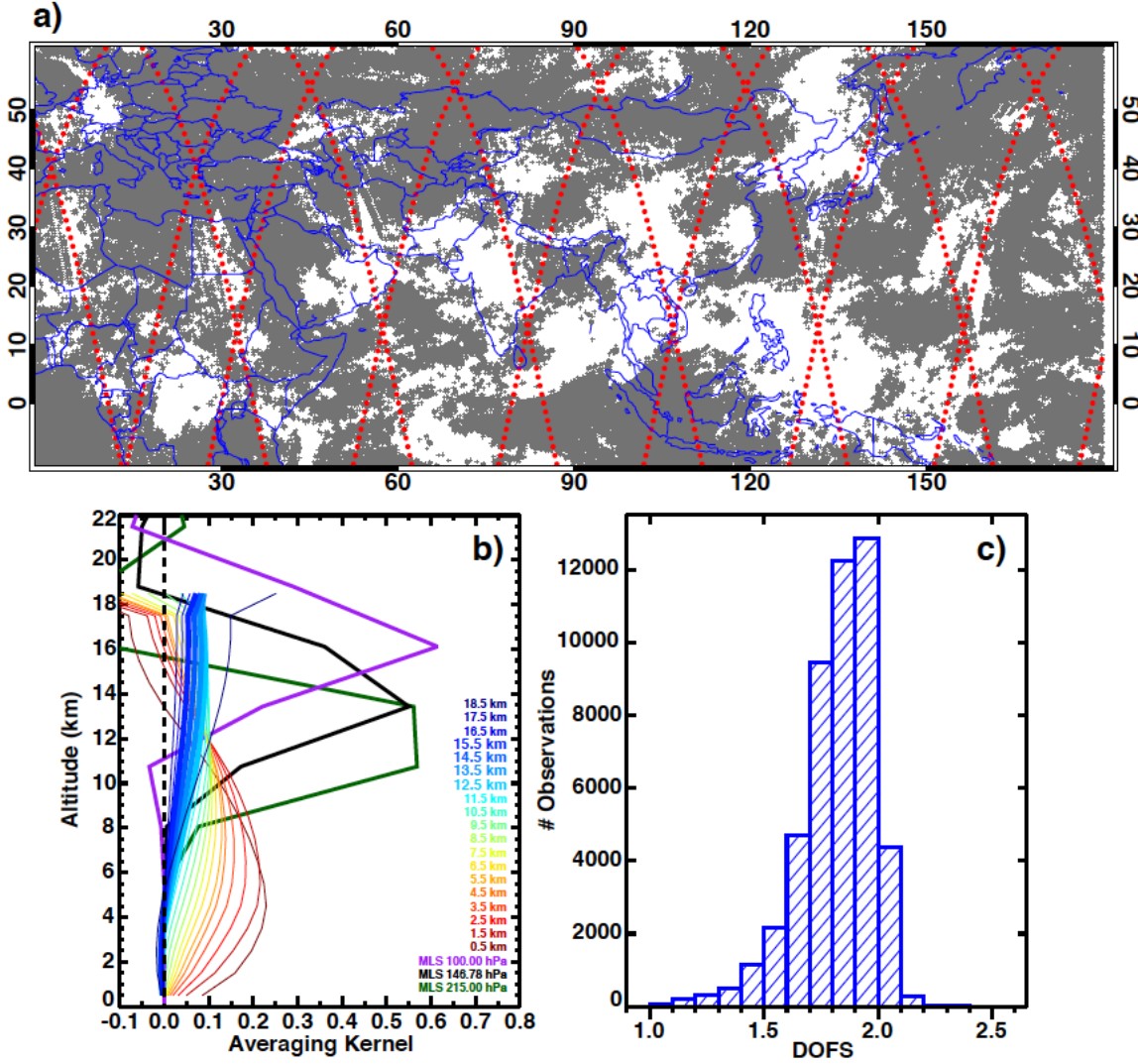

**Figure 1: (a)** Retrieval geolocations for IASI CO (gray crosses) and MLS CO (red dots) on August 1, 2008 for the study domain (0°–180° E, 10°S– 60°N). Both day and night observations are included. **(b)** IASI averaging kernels for 19 retrieval layers from surface to 19 km, labelled by the layer-center altitudes, and the standard MLS averaging kernels for the UTLS products (215, 147, and 100 hPa). The IASI curves are the averages of all profiles from the study domain on August 1st 2008. **(c)** Distribution of degrees of freedom for signal (DOFS) for all IASI profiles on August 1st 2008 within [0, 40N] latitude and [40 E-150 E] longitude.

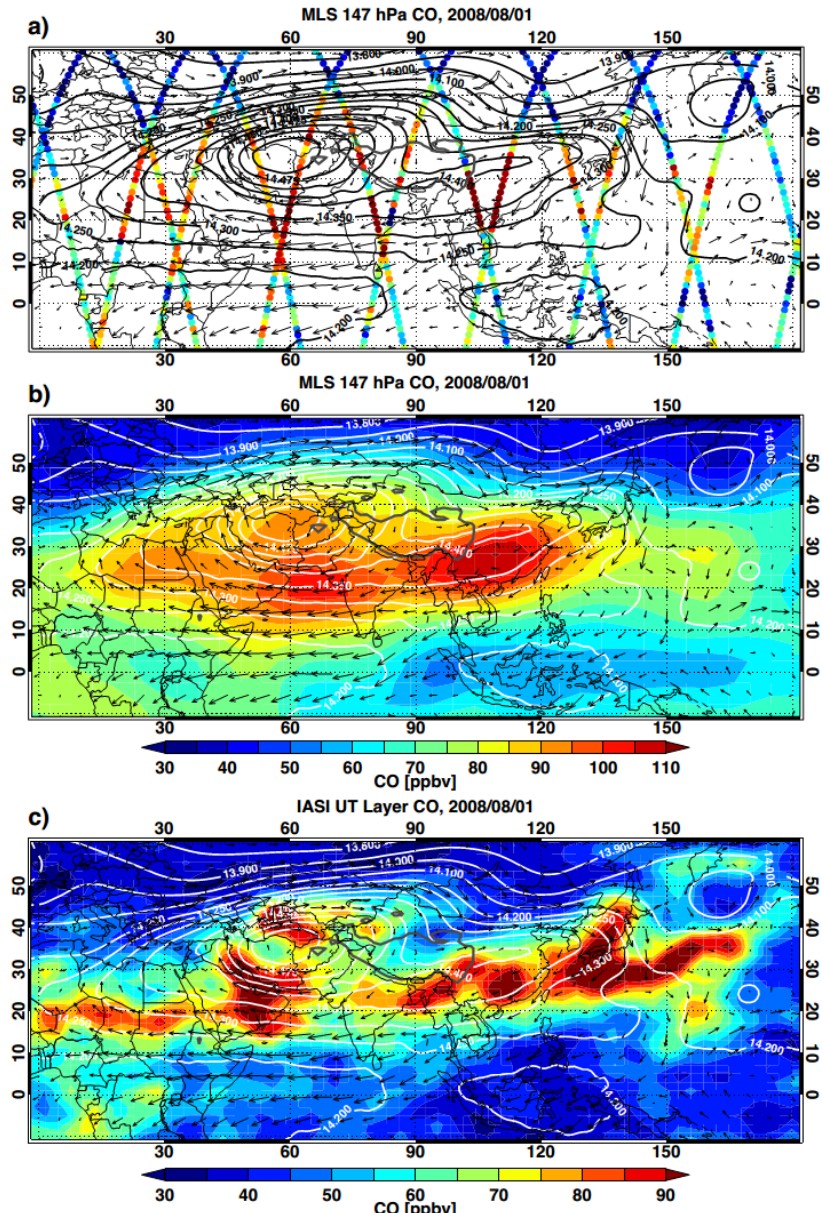

**Figure 2: (a) MLS 147 hPa CO mixing ratio at retrieval geolocations on August 1, 2008, (b) the interpolated map of MLS CO and (c) the map of IASI CO in the upper troposphere (UT) layer. The selected Geopotential Height (GPH) contours (white) and horizontal winds (black arrows) at 150 hPa are superimposed. The MLS CO map is made with 5°x5° longitude and latitude grids. The IASI CO map is made using 3°x2° grids. Both are interpolated using the natural neighbor algorithm (Watson, 1992).**

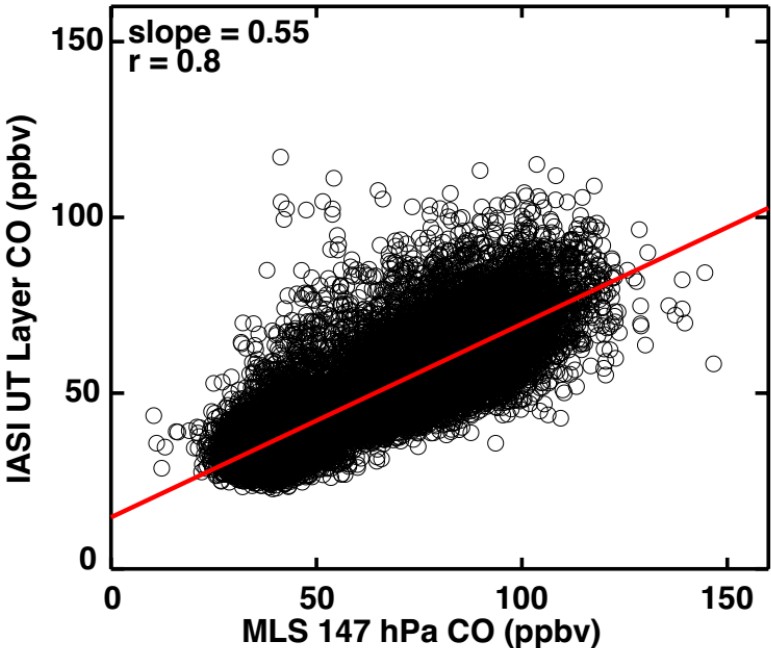

**Figure 3: Scatterplot of IASI UT layer CO versus MLS 147 hPa CO for June, July, and August (JJA), 2008. Each data point represents a daily average of CO level-2 data from IASI and MLS in the same 10 x 6 degree longitude-latitude box in the study domain. The red line shows a linear fit. Correlation and slope for the linear fit are given in the upper left corner of the panel.**

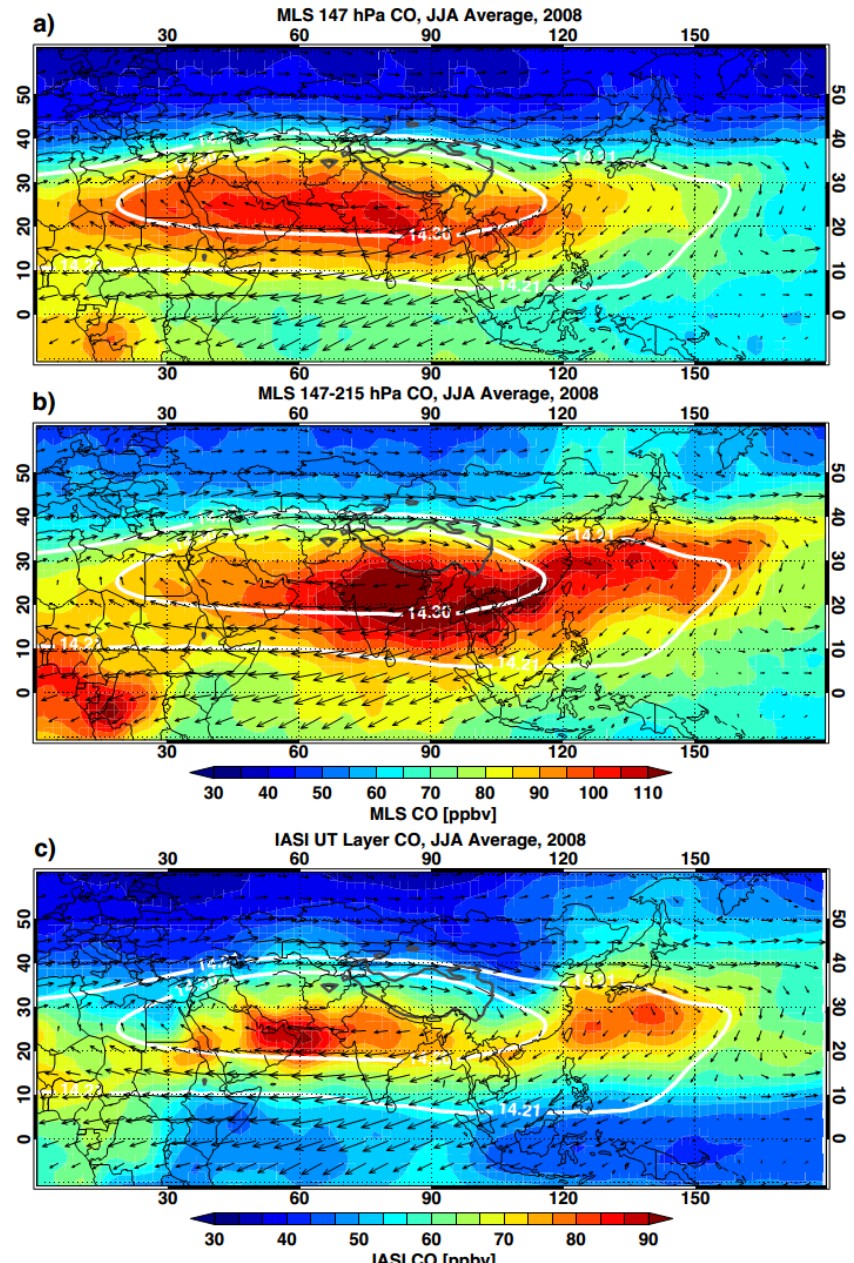

**Figure 4: JJA 2008 seasonal average CO mixing ratio for (a) MLS 147 hPa, (b) MLS 147 and 215 hPa average, and (c) IASI UT layer. Superimposed white contours are the14.3 km and 14.2 km GPH (from GFS analysis) at 150 hPa. Note that the color scales for IASI and MLS CO are different. Both MLS and IASI are 2° × 2° longitude-latitude binned averages.**

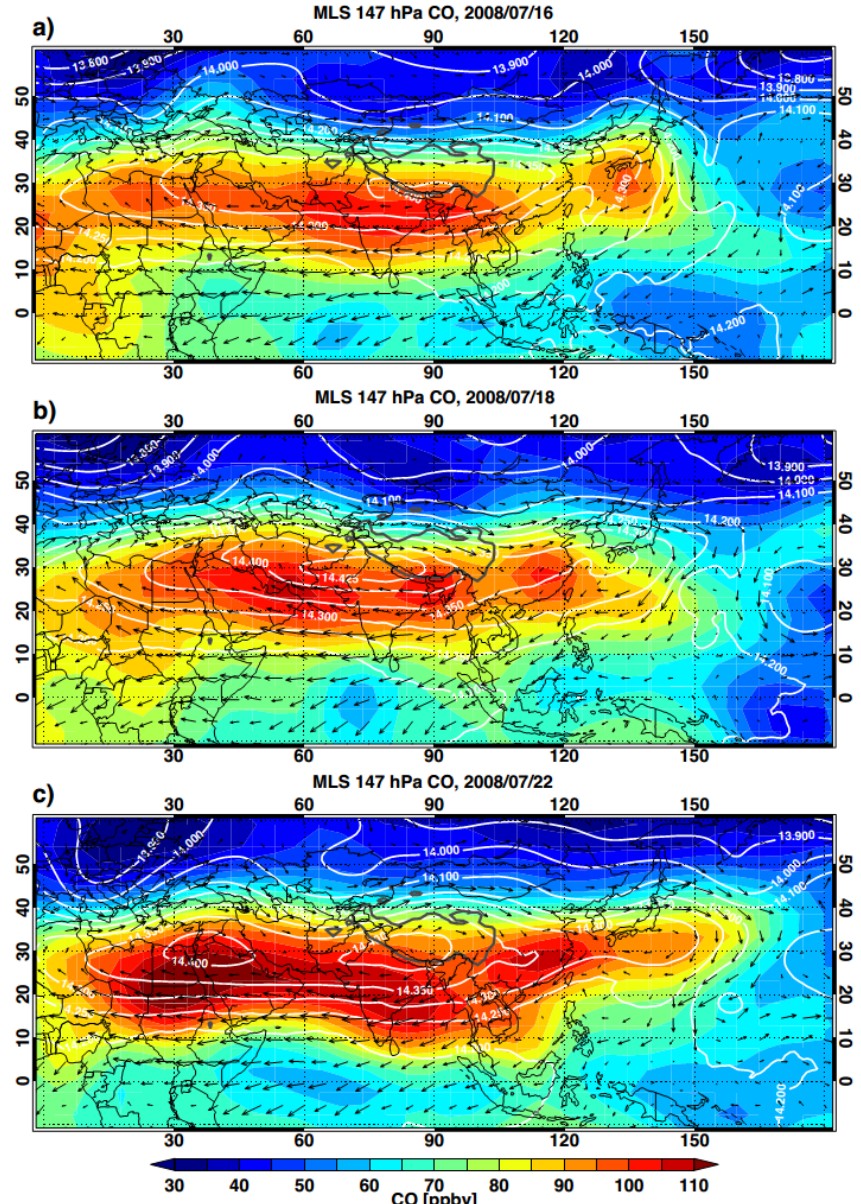

**Figure 5: Daily maps of MLS CO mixing ratio at 147 hPa (color shading) on (a) July 16, (b) July 18, (c) July 22 2008. Dynamical fields of GPH (white contours) and horizontal winds (black arrows) are superimposed. Maps are interpolated using natural neighbor algorithm (Watson, 1992) to 5°x5° longitude-latitude grids. The location of the Tibetan plateau (using 3 km elevation) is also shown in the maps (thick gray).**

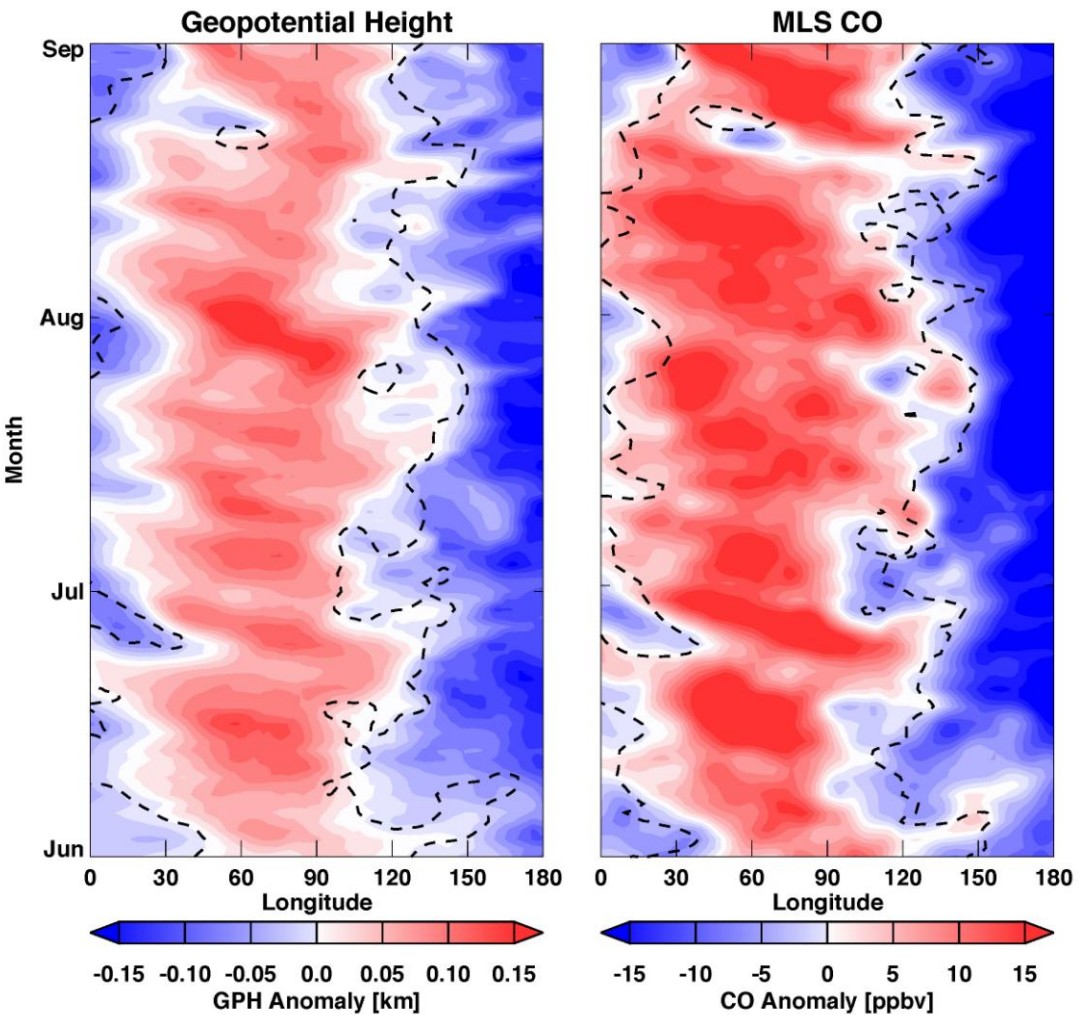

**Figure 6: Hovmöller diagrams of the 150 hPa GPH and 147 hPa MLS CO anomaly for JJA 2008. The anomalies are calculated with respect to daily means over the latitude band 10°–40° N and longitude range 0°– 220°E, in 5° longitude bins. The dashed line in each panel indicates the location of the mean (zero anomaly) of the opposite field. The Pearson's correlation of the two fields for the 3-month period is 0.92.**

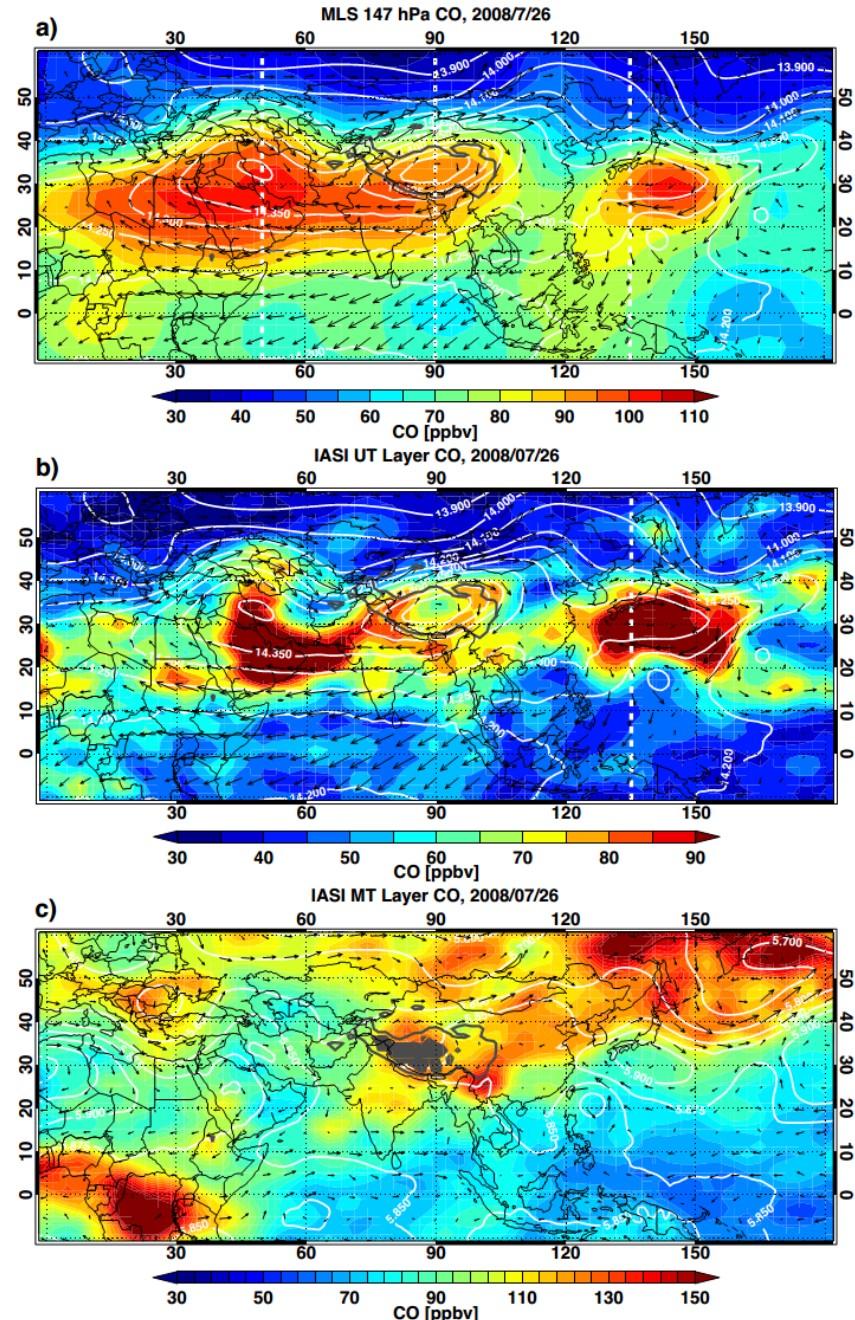

**Figure 7: (a) MLS 147 hPa CO (b) IASI UT layer CO, and (c) IASI middle troposphere (MT) layer CO for a selected day (26 July 2008). Dynamical fields of GPH (white contours) and horizontal winds (black arrows) for the corresponding levels are superimposed. Elevated terrain is indicated by gray shadings for the 500 hPa map in (c). The location of the Tibetan plateau (using 3 km elevation) is also shown in the maps (thick gray line). The dashed white line in (b) marks the location of the cross-section shown in Fig. 9d.**

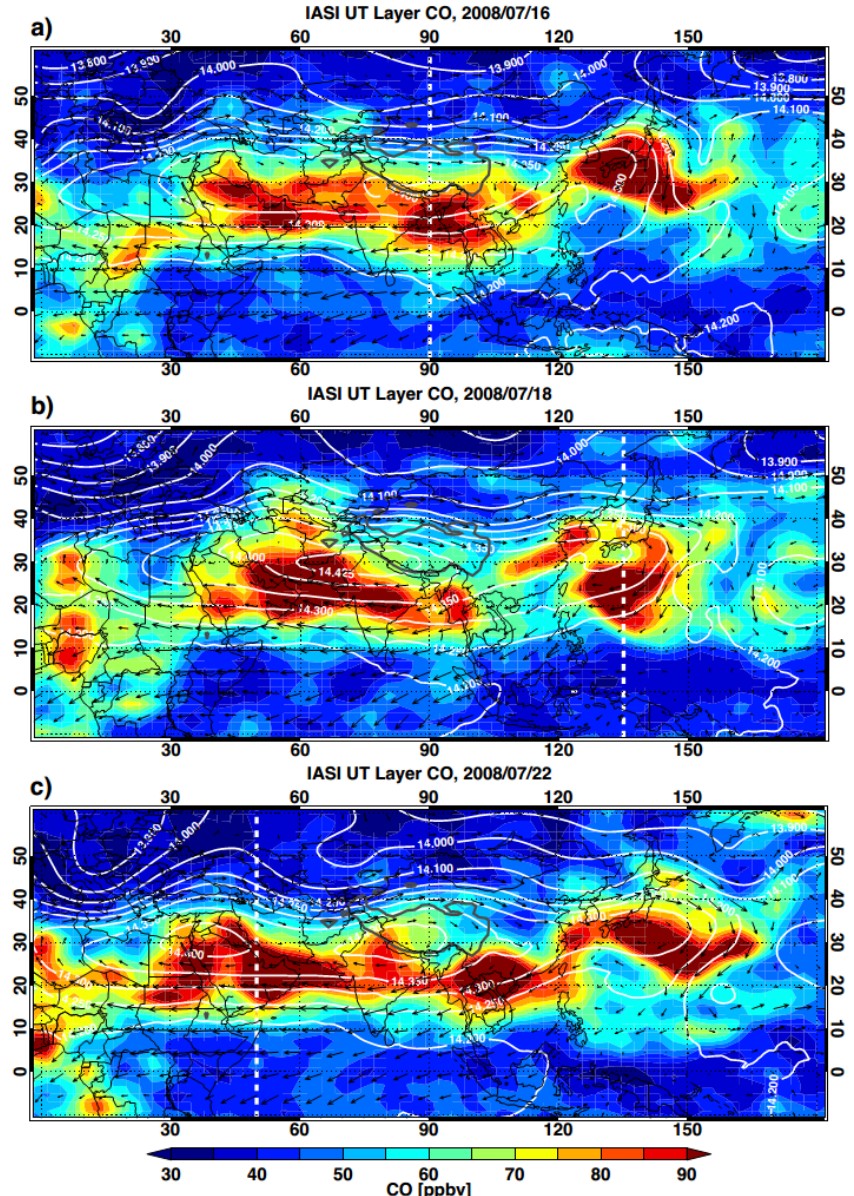

**Figure 8: Same as Figure 5, but for IASI UT layer CO mixing ratio. The maps are interpolated to 3°x2° longitude-latitude grids. The dashed white lines mark the location of the cross-sections shown in Fig. 9.**

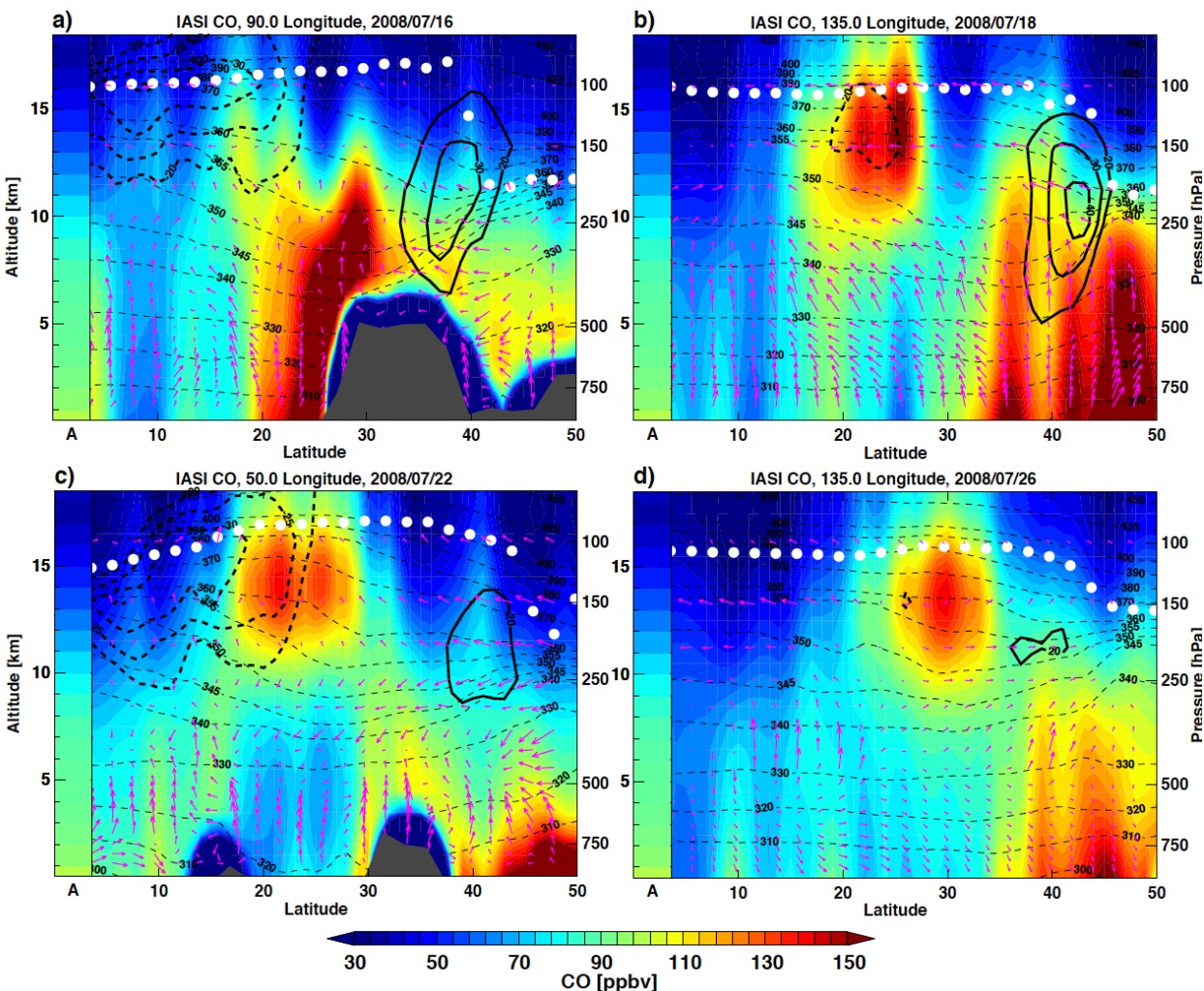

**Figure 9: Selected latitude-height cross-sections of the IASI CO retrieval. The retrieval a priori profile is shown as the left-most column in each panel (marked as "A" on x-axis). The days and the location of the cross sections are selected to highlight the different vertical structures of the three modes of the anticyclone: a) 90°E on July 16 (Tibetan mode), b) 135°E on July 18 (Western Pacific mode), c) 50°E on July 22 (Iranian mode), and d) 135°E on July 27 (Western Pacific mode). The corresponding maps are given in Figures 7 and 8. A number of dynamical fields are overlaid, including zonal winds (black contours, solid (dashed) for Westley (Eastley)), meridional wind (pink arrows), potential temperatures (thin black dashed lines), and the tropopause height (white dots).**

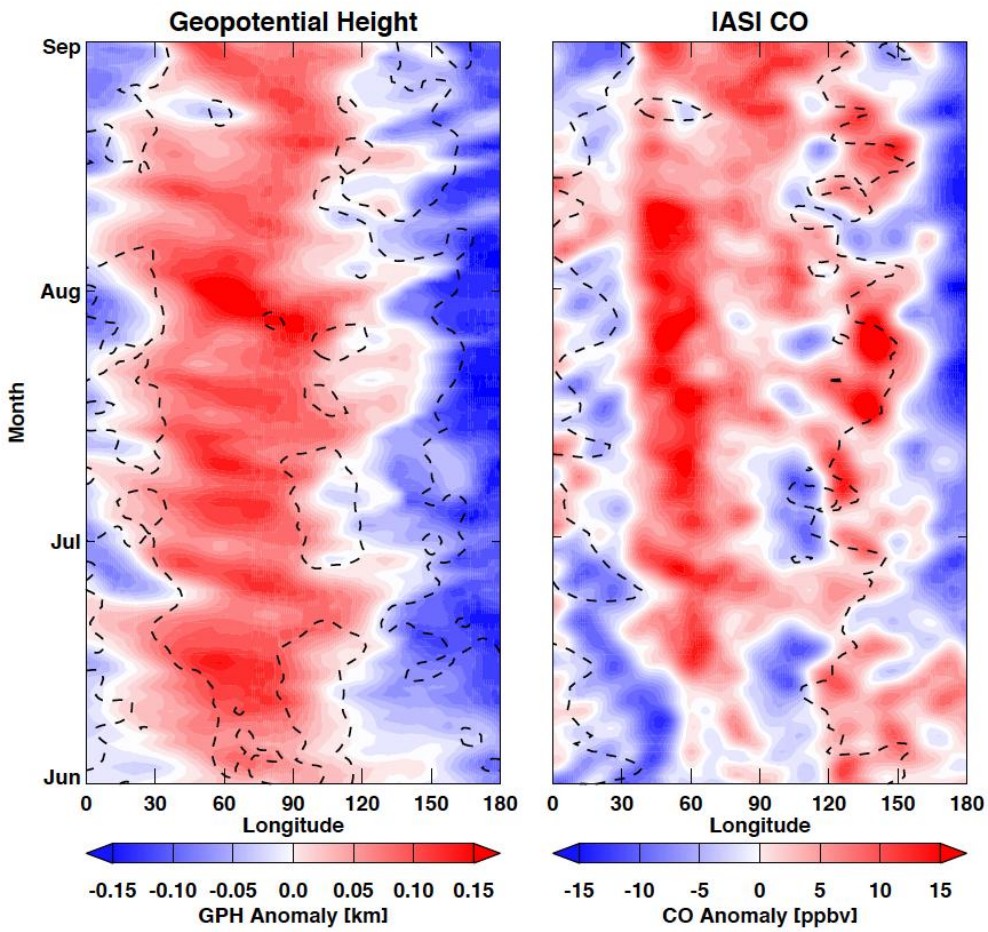

**Figure 10: Same as Figure 6 but for IASI UT layer CO mixing ratio anomaly. The Pearson's correlation of the two fields for the 3-month period is 0.69.**

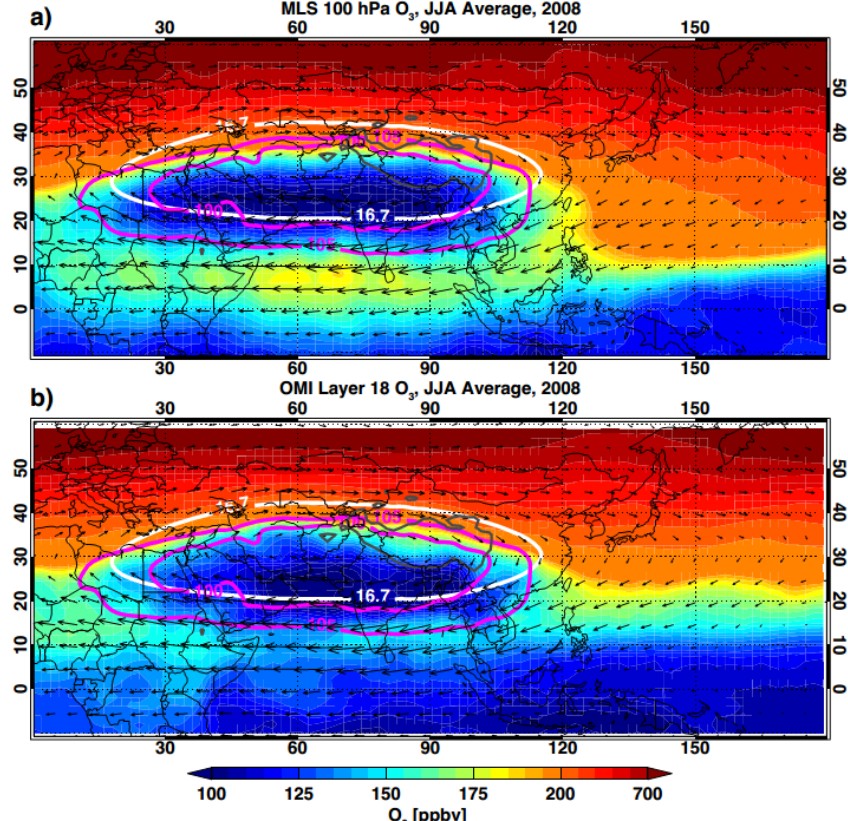

**Figure 11: JJA seasonal average O₃ mixing ratio for (a) MLS 100 hPa product and (b) OMI layer 18 product for 2008. Superimposed white contours are the 16.7 km GPH at 100 hPa and magenta contours are the 100 and 105 hPa tropopause pressure, i.e., the intersection of the tropopause with the 100 and 105 hPa pressure surfaces. Both MLS and OMI are 2° × 2° longitude-latitude binned averages.**

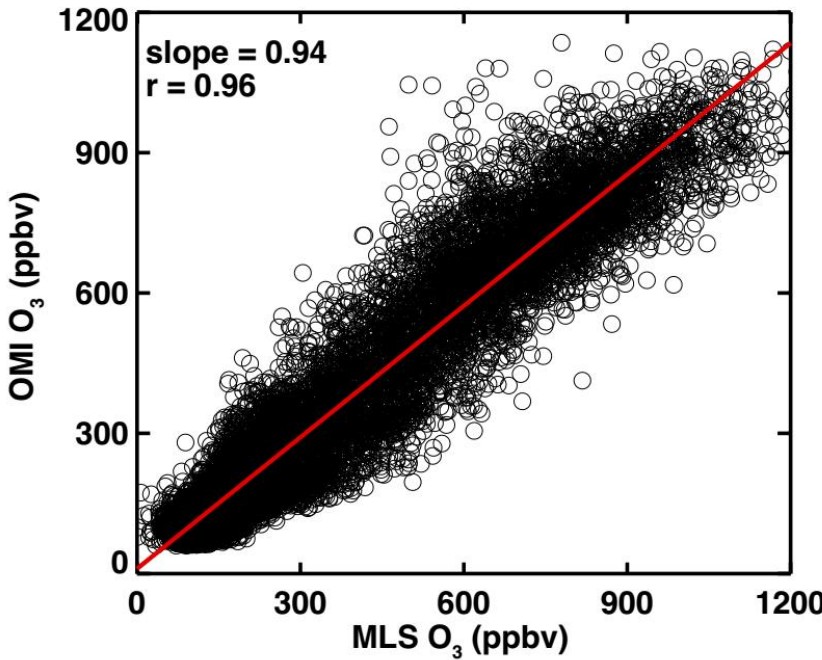

**Figure 12: Same as Fig.3 but for OMI layer 18 O$_3$ mixing ratio versus MLS O$_3$ mixing ratio at 100 hPa for JJA 2008. The red line shows a linear fit. Correlation and slope for the linear fit are given in the upper left corner of the panel.**

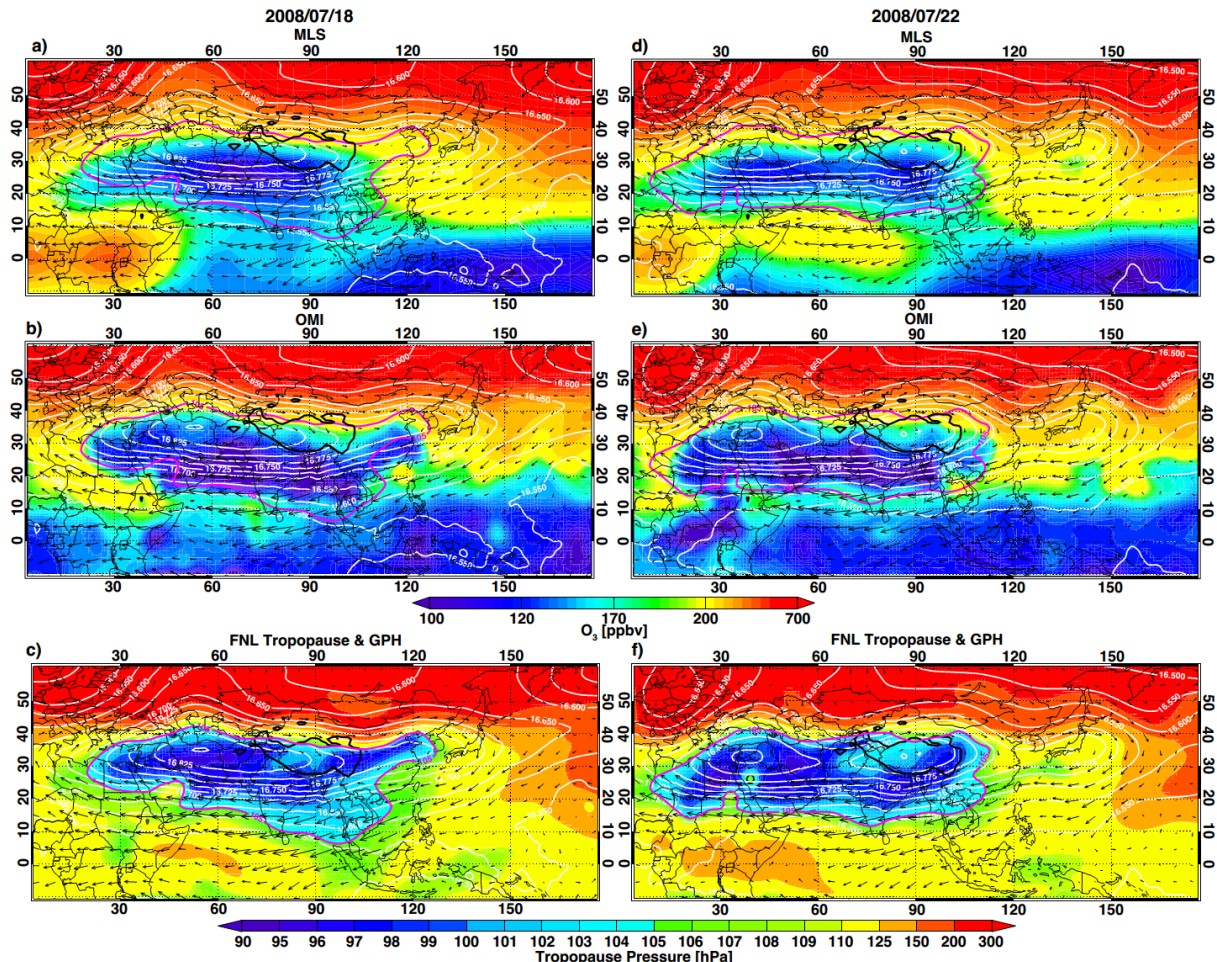

**Figure 13: Daily maps of MLS O$_3$ mixing ratio at 100 hPa and OMI O$_3$ mixing ratio in layer 18 (color shading) for 18 July (a, b) and 22 July (d, e) 2008. Tropopause pressure maps for the same selected two days are in (c, f). Dynamical fields of GPH (white contours), horizontal winds (gray arrows), and 105 hPa tropopause pressure contour (pink) are superimposed. MLS maps are interpolated using natural neighbor method on 5°x5° longitude-latitude grids while OMI maps are interpolated on 1°x1° longitude-latitude grids. The tropopause pressure is from the GFS product. A Gaussian smoothing is applied to all maps. The location of the Tibetan plateau (using 3 km elevation) is also shown in the maps.**

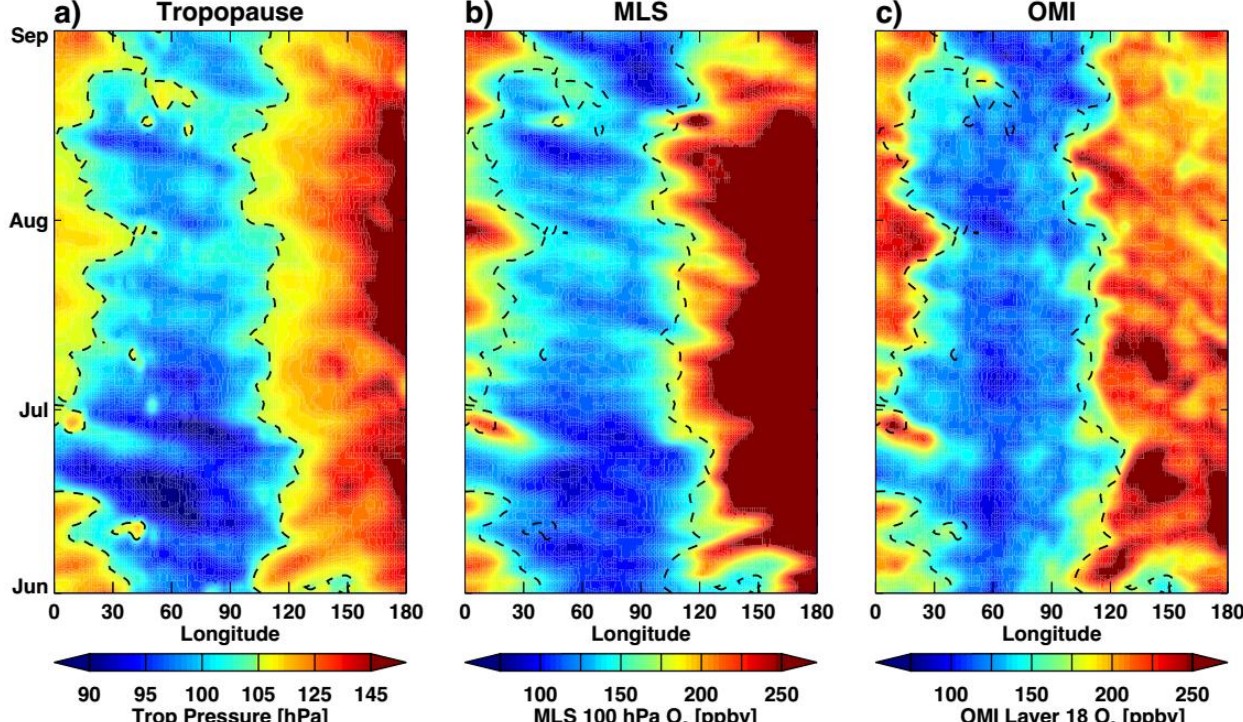

**Figure 14: Longitudinal-time (Hovmöller) diagrams for (a) tropopause pressure, (b) MLS 100 hPa O$_3$ mixing ratio , and (c) OMI layer 18 O$_3$ mixing ratio for JJA season 2008. The Hovmöller diagram is constructed using daily average over 15°–35° N. MLS data has been averaged over 5° longitude bins, and OMI data averaged in 1° longitude bins. The 105 hPa zonal average tropopause pressure is shown by the dashed line on all three fields. Gaussian smoothing is applied to all three datasets.**

