# Peer review of "Space-Time Variability of UTLS Chemical Distribution in the Asian Summer Monsoon Viewed by Limb and Nadir Satellite Sensors"

_Atmospheric Chemistry and Physics, 2017_

## Referee Comment (RC1) · Anonymous Referee #1 · 26 Jul 2017

This manuscript investigates whether coarse vertical but high horizontal resolution measurements from two nadir-viewing sensors – IASI and OMI – can be used to characterize changes in UTLS composition in response to the Asian summer monsoon (ASM). Comparisons with Aura MLS limb sounding data are made to explore how well the nadir measurements represent enhancements in CO and depressions in O3 within the ASM anticyclone as well as the sub-seasonal variations therein. The manuscript is generally well organized and well written (although further copy-editing, beyond the minor points listed below, is needed before final publication) and could make a valuable contribution to the extensive ASM literature. However, in my opinion the manuscript has a significant shortcoming that needs to be addressed before I can recommend it

for publication: The authors have not done enough to demonstrate that IASI CO retrievals actually do resolve CO enhancements in the UTLS associated with the ASM. Certainly the data have some sensitivity in the upper troposphere. But the analysis presented in this manuscript has not gone far enough to paint a truly convincing picture that IASI has sufficient sensitivity in the upper troposphere to discriminate variations there from the equally large (or possibly even larger) variations in the middle and lower troposphere to which it has much greater sensitivity. I also have a number of specific comments and points of clarification (detailed below) that I would like to see addressed.

General comments:

* Figure 1b shows that nearly as much information is contributed to the IASI "12-16 km" product from 10 and 18 km as from 12-16 km. Moreover, there is a not insignificant degree of overlap with the averaging kernel for the so-called "0-12 km" partial column; in fact, the averaging kernel for the "0-12 km" partial column shows considerably more sensitivity in the upper troposphere than does the averaging kernel for the "12-16 km" partial column. In addition, non-negligible contributions to the "12-16 km" layer derive from 4-10 km. Thus it seems somewhat misleading to refer to the volume of air being sensed by IASI, which is at least 8-10 km thick, as the "12-16 km" CO value throughout the paper.

* The study by Barret et al. [2016] is cited to support the statement that the IASI CO retrieval captures UTLS variability at middle and tropical latitudes (P4, L2-4). But that study focused on examining large-scale monthly mean features, not sub-seasonal localized variations such as eddy shedding. Furthermore, Barret et al. note that the smoothing of GEOS-Chem model fields imposed by the IASI averaging kernels mixes high CO concentrations from near the surface into the middle and upper troposphere. Thus I feel that the authors need to do more to conclusively demonstrate the power of IASI CO measurements to distinguish upper tropospheric variations from those at lower levels. One approach might be to perform sensitivity tests with model output (e.g., from GEOS-Chem as in Barret et al., or from WACCM4 as in Pan et al. [2016], etc.) to

investigate the influence of the CO distributions in the lower and middle troposphere on the partial columns measured by IASI. Results from the "raw" model fields multiplied by the IASI averaging kernels could be compared to those derived from the same model output but with mid-tropospheric CO abundances reduced or enhanced by some fraction (say, 20%). Differences between the inferred "12-16 km" layer averages from such tests would help to gauge how much independent information IASI provides in the upper troposphere.

* Why was a comparison of the vertical resolution similar to that in Figure 1b not shown for MLS and OMI O3?

Specific substantive comments:

* P2, L16: I don't believe that Garny & Randel [2015] is the correct reference for this point. First, the citation for that paper is incorrect: it should be ACP 16, 2703-2718, 2016. Second, that transport pathways paper is not appropriate here: I'm guessing that the authors meant to cite Garny & Randel [2013] instead.

* P3, L2-3: Three main questions are articulated that this study is aimed at. The first two are clearly addressed in the manuscript. However, I am not sure that the third question – What can we learn from the complementary information from limb and nadir viewing instruments? – has actually been explicitly touched on. It would be good if the authors could add a sentence or two, probably in the Conclusions section, that circle back to this question to offer guidance on how the limb and nadir data sets could be applied synergistically to investigate specific science questions.

* P3, L22-23: According to the most recent MLS Data Quality Document [Livesey et al., 2017] (not [2015] as cited in the manuscript), the accuracy of MLS v4 O3 at 100 hPa is +0.005 + 7% ppmv, or ~20 ppbv for 200 ppbv of O3, not 50 ppbv. Although the along-track resolution is ~300 km for the O3 product at 100 hPa as stated, it is ~550-600 km for CO at 147 hPa.

\* P4, L28-P5, L4: The formulation of Figure 1b is slightly confusing. Why are different x-axes used for the IASI and MLS averaging kernels? They could be compared more readily if plotted on the same scale. At the least, the zero lines for the two axes should be aligned (and perhaps a vertical line drawn through zero to guide the eye).

\* P5, L1-4: First, according to Livesey et al. [2017], the full width at half maximum of the MLS CO averaging kernel (by which the vertical resolution is defined) is 5 km, as stated on P3, L21, not 6 km as given here. Second, it would be better to refer to this as "vertical resolution" rather than "vertical distribution".

\* P5, L5-10: It is interesting that the authors chose to use the GFS operational analysis for this study, rather than one of the more commonly used analyses/reanalyses such as GEOS-5, MERRA or ERA-I. A sentence or two motivating that choice would be appropriate. This is especially true in light of the work of Nuetzel et al. [2016] showing that substantial bimodality in the ASM anticyclone, as asserted in previous studies, is apparent only in the NCEP-NCAR reanalysis and not in data from other modern assimilation systems.

\* P5, L19: Park et al. [2007] note that confined tropospheric air masses are present in the ASM anticyclone up to 68 hPa.

\* P5, L29: I understand the desire to employ different color bars for MLS and IASI CO. However, Figure 2 could give a false impression of the degree of agreement between the two fields to readers not paying close attention to the figure or its caption. Therefore the fact that the color scales are not the same should be mentioned in the main text as well as the figure caption.

\* P6, L3-8: I agree that OMI data appear to represent UTLS O3 in the ASM region fairly well. However, the OMI JJA O3 field does not reflect the signature of tongues of extratropical stratospheric air being transported equatorward and westward around the edge of the anticyclone, which results in a ring of higher ozone surrounding the low values in the anticyclone center. This structure, which has been discussed in several

previous papers based on MLS and also MIPAS data, is readily apparent in Fig. 2c but is barely visible in Fig. 2d. I find this surprising given OMI's higher horizontal sampling/resolution and the fact that other smaller-scale features are seen in the map in Fig. 2d. On the other hand, such a signature of monsoon-induced stratosphere-to-troposphere transport is prominent in the daily maps of Fig. 11. Is its absence in Fig. 2d just a color scale issue (since Figs. 2 and 10/11 have different color bar increments)? Or does the disparity in the JJA means in Fig. 2 imply that MLS and OMI observe different seasonal evolution of this transport feature, some of which lies equatorward of 15N and thus may not be captured in Fig. 13?

* P6, L11: In these lines the "monsoon region" is defined as 0-50N, 0-150E, which is a rather broad area to label as being directly influenced by the ASM. I presume such an extensive region was used in order to encompass a range of values of both the tropospheric and the stratospheric tracer. If so, perhaps that should be explicitly noted, especially given that the ASM region is defined differently for different plots: on P4, L31 it is defined as 15-35N, 30-140E, and on P8, L12 as 15-35N, 0-150E.

* P8, L4-10: Again, it is not obvious to me that IASI has sufficient sensitivity in the upper troposphere to distinguish a "plume" of CO up to (and above, according to the figure) the tropopause that is clearly separable from the large abundances of CO in the lower atmosphere. The color bar in Figure 7 is strongly saturated at the high end. It would be better to adjust the color scale to allow some of the structure in enhanced CO abundances to become visible. That might reveal instances of localized enhancements in the upper troposphere that are not connected to the generally higher mixing ratios in the lower troposphere, providing more confidence that they are not simply a manifestation of contamination from below.

* P8, L33 – P9, L2: (1) The sentence "IASI CO data have a higher horizontal resolution and are able to detect the impacts of vertical transport in the troposphere" is somewhat confusing, because as written it seems to imply that the higher horizontal resolution enables the detection of the impacts of vertical transport. I think it would be better to more

clearly separate these two points. Perhaps something along these lines would work: "IASI CO data have higher horizontal resolution than MLS measurements. Despite its coarse vertical resolution, IASI is able to detect the impacts of vertical transport in the troposphere". (2) In my opinion the conclusion in the last sentence of this section that some of the finer scale structure evident in IASI CO data is attributable to eastward eddy shedding over the western Pacific is stated too definitively. That may be true to some extent, but as Figure 1b shows and the discussion on P7 makes clear, the IASI "12-16 km" layer average is substantially influenced by the mid-tropospheric CO distribution. In fact, the authors have basically said as much in the lines just above (P8, L19-21). Thus, unless the sensitivity analysis suggested earlier conclusively shows that such a statement is fully justified, I would like to see the wording in the last sentence of this section softened. The same comment applies to a similar sentence in the Abstract (P1, L23-25). (3) Although this work demonstrates the utility of IASI data for studying the evolution of CO over the ASM region, I think the essential bottom line is summed up on P8, L32-33: "MLS and IASI data have different advantages. MLS data are better for examining features with a shallow vertical extent, such as the ASM anticyclone, provided those features have a large enough horizontal scale." I feel that it would be appropriate to repeat this sentiment in the Conclusions section, and possibly in the Abstract as well.

* P9, L5-7: It might be good to remind readers here that, although UTLS O3 is mainly a tracer of stratospheric air as noted, its distribution can also be affected by photochemical production, as alluded to on P2, L30 of the Introduction. So the interpretation of O3 fields in the ASM region is not necessarily straightforward.

* P9, L19-21: "OMI data show a sharper transition of O3 field across the edge of the anticyclone (as indicted by the 105 hPa tropopause contour) ... differences between MLS and OMI are more pronounced at low latitudes". (1) Why is the tropopause being used to define the edge of the anticyclone here? Elsewhere GPH is used to denote the anticyclone boundary, not the tropopause. (2) How is the tropopause shown in

these figures being defined? I realize that it is taken from the GFS analysis, but is a thermal or dynamical (PV-based) definition being used? That should be clarified. (3) Differences between MLS and OMI may well be more pronounced at low latitudes, but it is difficult to tell from Figs. 10 and 11 since the color scale saturates at high latitudes.

* P10, L16: Although Garny & Randel [2013] did show that spatial variations in CO are well correlated with variations in the region of low PV defining the anticyclone, they did not find evidence of the kind of bimodality in the location of the anticyclone that Yan et al. did. So although it is certainly appropriate to cite Garny and Randel (among many others not listed) for highlighting the significant role of ASM dynamical variability in controlling UTLS tracer distributions, I don't think it is quite fair to include that reference for different "modes" of anticyclone behavior.

* P10, L24-26: In general, the Conclusions section overlooks the potentially significant contamination in the IASI CO "12-16 km" layer average from lower altitudes. Such influence from below is likely to be another factor explaining the apparent lack of consistency with MLS 147 hPa CO and should be acknowledged here.

* P11, L7: I'm intrigued by the notion that the results from this study might be used to refine the IASI or OMI retrievals. Could the authors say more about that, perhaps provide an illustrative example of how these findings could inform IASI or OMI retrieval algorithm development? Also, please clarify which "differences" are being referred to – differences from MLS?

Minor points of clarification, wording / figure suggestions, and grammar / typo corrections:

* P1, L15: The IASI and OMI acronyms should be spelled out here

* P 1, L17: "changes . . . is" –> "changes . . . are"

* P1, L19: "result shows" –> "results show"

* P1, L23: I suggest "captures" rather than "show[s]"

* P1, L31: As the first sentence of the paper notes, the ASM anticyclone has been investigated widely in recent years, but the small subset of references cited for this point seems somewhat arbitrary. Many more equally relevant papers could have been included, so it would be appropriate to add "e.g." at the front of the list.

* P2, L10: "i.e." –> "e.g."

* P2, L12: "in terms of"

* P2, L31: "Short time" –> "Short-term"

* P2, L33: "nadir view" –> "nadir viewing"

* P3, L5: "make" –> "makes"

* P3, L7-8: delete the second instance of "quantitative comparisons"

* P3, L13: "much" –> "more"

* P3, L13: "UTLS chemical tracers variability" –> "variability of UTLS chemical tracers"

* P3, L15: "aim" –> "aims" and "supplement" –> "supplements"

* P3, L16: "inform" is not quite the right word. Perhaps "examine", or something similar

* P3, L23: "and has" –> "with"

* P3, L31: "degrees of freedom signal" –> "degrees of freedom for signal"

* P4, L1-2: The MOZAIC and MOPITT acronyms should be spelled out; also add "and" before "satellite"

* P4, L7: Does Huang et al. [2016] really discuss the OMI O3 profile product? I believe this reference is incorrect.

* P4, L9: delete "and" before "zonal mean". Also, "NCEP" has not yet been defined.

* P4, L11: It seems odd to use a tilde with such precise numbers for the DOFs ("~6.0-

7.0")

* P4, L17: "which" –> "for which" and "is" –> "are"

* P4, L20-21: add "data" after "O3", "began" before "in January", and "has" before "impacted". Again, I don't think that Huang et al. [2016] is the correct reference for OMI O3 profiles.

* P4, L29: "analyses" –> "analysis"

* P4, L33: I think it is potentially confusing for readers to refer to the MLS data quality document in this manner here but as Livesey et al. on P3. Please be consistent. Also, the web site information should be provided in the reference list as part of the Livesey et al. citation.

* P5, L1: add "thick" after "8 km"

* P5, L16: "dataset" –> "datasets"

* P6, L5: "tied" –> "tied to"

* P6, L6: delete "in" after "within"

* P6, L17-18: "the UTLS chemical impact by ASM anticyclone" –> "the impact of the ASM anticyclone on UTLS chemical composition". Also, I think it would be more accurate to say "a picture *largely* consistent with that from MLS".

* P6, L32: "the empty" –> "an empty"

* P7, L11: "splits" –> "split"

* P7, L18: "enhancement" –> "enhancements"

* P7, L21: "are much more extended in longitudinal range compared to the MLS, co-located and mimic the east-west extent of the" –> "is much more extended in longitudinal range compared to that from MLS, co-located with and reflecting the east-west extent of the"

* P7, L22: "distributions" –> "distributions on" and "that are not" –> "that is not"

* P7, L25: "on the south" –> "to the south"

* P8, L20-21: "is contributed by the retrieval information in the level lower than that represented by 150 hPa dynamical field" –> "reflect the influence of retrieval information from a level lower than that represented by the 150 hPa dynamical field"

* P8, L30: It is not clear what is meant by "the two sensors are influenced by different over vertical columns". I suggest instead something along the lines of "the two sensors sample quite different volumes of air".

* P9, L7-9: I suggest rearranging /rewording this sentence and replacing "interception", which is not correct here: "The structure of the bulging tropopause in the monsoon region (indicated by the intersection of the tropopause with the 105 hPa pressure level in Figs. 10 & 11) (Bian et al., 2012; Pan et al., 2016) has a significant influence on the O3 distribution."

* P9, L11-12: It is not entirely clear that the Fig. 9 being referred to here is from Park et al., not the current manuscript.

* P9, L18-19: "OMI also shows similar distribution" –> "OMI (Fig. 11) also shows similar morphology". Also, "Quantitatively" –> "Qualitatively" and "indicted" –> "indicated"

* P9, L31: "is" –> "are"

* P10, L1-2: "sectional anomalies" –> "regional anomaly". Also, "vertical and horizontal samplings of two satellites" –> "vertical resolution and horizontal sampling of the two satellite instruments"

* P10, L9: I think that "reduced" would be better than "weakened" here

* P10, L12-13: To be perfectly clear, please change "the weaker UT sensitivity" to "IASI's weaker UT sensitivity" and add "its" in front of "retrieval". Also, "product" could be deleted.

* P10, L14: "dynamic" –> "dynamical"

* P10, L16: "Garney" –> "Garny"

* P10, L29: "convective-driven" –> "convectively driven"

* P11, L6: "dynamic" –> "dynamical"

* P11-15, references: Several references (e.g., Pan, Park 2007, Randel 2006, Vernier, etc) are incomplete (e.g., pages and/or doi missing)

* P12, L15: I am not familiar with the 2015 paper by George et al., but I am quite sure that "Bmc Medical Genetics, 8, 4095-4135" is not the correct citation for it

* P16, Fig 1a caption: please clarify whether the statement about the symbols being enlarged in Figure 1a applies to both data sets or only to MLS.

* P17, Fig 2 caption: "mean of CO" –> "mean CO" and "Note the" –> "Note that the". Also it probably would be a good idea to specify in the caption that the GPH values are also taken from the GFS analysis.

* P17, Fig 4 caption: "geolocation" –> "geolocations" and "selected GPH of" –> "selected GPH values at". Also, a few MLS data points at various spots in the map in Fig 4a are plotted in black – are these points off the color scale (on both ends)? That shouldn't be the case given the way the color bar is constructed.

* P20, Fig 6 caption: note here also that the color scale in this figure differs from that used in Fig 5.

* P21, Fig 7 caption: "white dash" –> "white dashed lines"

* P22, Fig 8 caption: "5 deg bins" –> "5 deg longitude bins" and "period" –> "periods"

* P23, Fig 9 caption: "dash lines" –> "dashed lines"

* P24, Fig 10 caption: "white" –> "white contours" and "interception" –> "intersection"

* P25, Fig 11 caption: "mapped in" –> "mapped onto a" and "grids" –> "grid"

* P26, Fig 12 caption: add "bins" after "longitude" for OMI data

* P27, Fig 13 caption: "dash lines" –> "dashed lines"
* * *

---

## Referee Comment (RC2) · Anonymous Referee #2 · 29 Jul 2017

This paper investigates the sensitivity of nadir looking satellite sensors to changes in the chemical composition of the UTLS region within the Asian summer monsoon (ASM) as compared to typically adopted limb sounders. It explores both seasonal and day-to-day variability attempting to exploit the high horizontal resolution of nadir sounders to better depict horizontal structures in the ASM distribution of tracers of pollution and stratospheric air (CO and O3). This study could give a valuable contribution both to the broad issue of exploiting nadir sensors at UTLS altitude, and to the ability of observe and understand this region at fine spatial and temporal scale. However, attempting to tackle both challenges in the same study, the authors have failed, in my opinion, to achieve enough robustness and the study needs substantial improvement before I can
recommend it for publication on ACP.

GENERAL COMMENTS

The manuscript is generally well written and structured, although some editing may improve it. Despite appreciating this effort of promoting a synergetic use of nadir and limb data, I find there is a certain lack of overall clarity on the aims of the paper. The study is introduced as a qualitative comparison of nadir and limb data (see introduction), but with the ambition that their results would support the use of this approach in future research. It is not clear to me whether the study aims therefore at validating the use of the adopted nadir observations under ASM UTLS or trust support from the literature and aims at producing novel results for the ASM UTLS region. They focus on 3 questions: the first two point to a validating exercise which cannot be kept at a qualitative level, the third one to a more general interpretation of the results for the ASM UTLS region, although leaving it with no clear answer.

The authors use the observed ASM atmospheric composition (in comparison with MLS) to support the use of the adopted nadir data and then use the same nadir data to investigate the details of the ASM atmospheric composition at finer scale (going beyond MLS). The observed details and fine scale variability has to be proved to be a real measure of the natural variability and not a combination of perturbed conditions and a low sensitivity retrieval. The authors appreciably introduce averaging kernels and mention vertical resolution/degrees of freedom for signal (DOFS) but do not use them quantitatively. E.g., the IASI mean averaging kernel for the 12-16 km observation (Fig 1) is so broad that contributions from layers outside the range are expected and the high CO cannot be assigned to a shallow layer, contrary to MLS data. The same applies to O3. A vertical resolution of 10-14 km: the apriori profile is likely to be simply scaled to match the average sensed value of the whole tropospheric column. Indeed, the horizontal details presented by the authors are very interesting, but should they be read more like an on/off effect of a perturbation at an unknown tropospheric layer? You need to quantify how much of the results are coming from the measurement and

from what layer. In order to support the authors' claims, there is a need for performing sensitivity studies to investigate the response of the retrieved profile to perturbations at varying altitude that mimic the observed ASM behavior, contributions from different layers and possible contaminations (clouds?). To this end, both retrieval simulations and atmospheric model simulations (e.g. the referenced Pan et al., 2016) would be of help. Furthermore, comparison between sounders could be performed more quantitatively, e.g. introducing convolution with the averaging kernels.

I find very unusual the choice of using two different years for the analysis of the two targets. I see no reason not to compare results for nadir CO and O3 for the exact same days and regions (and also O3 from the two nadir sensors) and verify that the small-scale structures you attribute to natural variability are indeed consistent among the two targets. Even then, you may still be seeing a retrieval artefact but in response to sensitivity to different layers of the atmosphere, so giving further support to your approach.

I encourage the authors to consider these main issues and the specific comments below in order to improve the paper to make it a valuable reference for future studies.

SPECIFIC COMMENTS (P=page, L=line)

Abstract:

P1 L14: the use of "information content" is misleading since it is not analyzed in the manuscript. You only quote mean estimates of DOFS from the literature. On the other hand, adding an analysis of the information content from the retrievals would be of great help in understating the sensitivity of the nadir observations under these conditions. How much information for each point of your profiles is coming from the measurement? How much from the apriori? Adding maps with this information could give convincing support. P1 L16: same.

P1 L15: possibly due to a typo, the sentence starting with "Day-to-day behavior" should

be rephrased discussing first the seasonal scale analysis, then moving to the day-to-day and finer scales in the following sentence.

P1L 20: I would tune this sentence down depending on further support to the analysis of the layers that actually produce the signature in IASI data. P1 L21: The same issue applies to OMI data, for which you actually show no profiles in your work. P1 L24: same.

1. Introduction P3 L4-9: this is a key statement for justifying the lack of robust analysis during the comparison. I feel the authors failed to extract enough support from the listed literature in order to prove the claims of their paper. See comments to the listed references here below. If the ambition is to show how the synergetic use of nadir and limb sounding can improve the understanding of the ASM UTLS, then you need to quantify the agreement, more carefully analyze the limitations of the two and what parts of the nadir observations can be trusted. You could then propose a strategy for how to merge the horizontally coarser and less frequent vertical information from limb sounders to drive the finer and more frequent picture coming from nadir observations.

P3 L9-14: how can a qualitative comparison help to assess the information of these nadir viewing datasets for ASM UTLS studies? Please explain. P3 L14-16: I am doubtful on what part of the study investigates the information content and how your analysis can help further studies if the horizontal information cannot be located at a correct altitude range.

2. Data Description

P3 L19-P4 L17: I think the available support for your study from the literature you list has been overestimated. In general, the profiles of CO and O3 retrieved from IASI and OMI have such a low vertical resolution and DOFS that have almost no sensitivity to differentiate layers in the troposphere, allowing for 1 or maximum 2 independent partial columns (or points in your profiles) throughout the whole troposphere (see e.g. George et al., 2015). Liu et al (2010b) show OMI O3 vertical profiles have 0-1.5 DOFS

in the troposphere, or 14/11 km vertical resolution at 12.5/17.5 km altitude. Kroon et al (2011) show (their Fig. 1 and discussion) that in the troposphere it is almost impossible to distinguish different layers in OMI O3 (certainly true around the 100 hPa level). Wachter et al (2012) validation reach only 225 hPa for IASI CO, i.e. below the layer adopted in this study. Bak et al. (2013) state that a large smoothing error is introduced in OMI profiles by the retrieva. Safieddine et al (2016) only use 0-6 km columns. Barret et al (2016) indeed compared IASI vertical profiles and results from a transport model, although working on monthly means only, so leaving the validity of small scale day-to-day data in the UTLS unsupported. Averaging over long time periods or large regions, the resulting profiles are very smooth and agreement with low information content retrievals can be very good. But is this residing on the apriori climatologies? Instead, you intend to use the nadir data under variable conditions and unknown vertical gradients. Conditions of strong vertical variability may be completely missed by nadir retrievals: see e.g. Gazeaux et al. (2013) AMT where IASI O3 profiles completely miss to reproduce the plumes at various tropospheric altitude observed by ozonesondes over Antarctica. How do these limitations affect your analysis?

P3 L19-P4 L17: are there detection deficiencies expected for the three sensors in the UTLS region? E.g., how is IASI affected by clouds? Could you also help the reader understanding how the numbers you give as degrees of freedom translate in how much information can be extracted from the observations? To what extent are 0.8-2.4 DOFS in IASI CO sufficient "to capture upper tropospheric variability" as you write? If you can retrieve 1 to 2 independent partial columns, help the reader to understand that you expect only 1 or 2 independent points in your profiles which will then be used to scale the climatological profile.

P3 L19 –P4 L17: At what local time are the observations taken? Are the fixed local times of the measurements affecting your analysis?

P3 L23: Could you use the horizontal resolution to predict at what scale you expect MLS data to lose sensitivity to finer scales as compared to nadir observations? Is this

reflected in your results (e.g. the 100 km features seen in IASI CO daily maps but missed by MLS)?

P3 L25: how are these and following interpolations affecting your study? E.g., averages and interpolation tend to add further smoothing errors in the profiles therefore reducing the vertical resolution. How are the vertical profiles changing depending on the horizontal interpolation you adopt? One may expect the agreement between nadir and limb data to improve reducing resolution (and therefore variability).

P3 L28: please carefully discuss the vertical resolution for IASI CO (and OMI O3) as this is a key element in your analysis.

P3 L29: the use of a single a priori profile could lead to a bias in the average you perform and could be removed from the final result to reduce its influence.

P4 L1-4: see comments on supporting references above.

P4 L5-17: see comments on supporting references above. L11: you should clearly state that the DOFS in the troposphere are 0-1.5 and that the troposphere is then observed as one single column (Liu et al. 2010b). L14: the fact that there is useful information does not mean you can control where that is coming from (e.g., what about contamination by photochemical production?) Note that for example the averaging kernels in Liu et al. 2010b under tropospheric perturbed conditions peak at 5 km altitude in their case study in the tropics (their Fig. 5): so as long as the signal is coming from the upper troposphere, what you show is sound. But if the signal is coming from different layers then you give a wrong picture, with no ability to distinguish among the two cases. Please find support for the kind of conditions you analyze and provide sensitivity tests. Can you support via e.g. MLS that the whole signal in the region is coming from the upper troposphere and then constrain the nadir observations?

P4 L18-23: your use of different years is quite unusual when studying simultaneously different targets since further insight can be achieved by inter-comparison. Without

studying at least one year with both targets before showing what happens in a different year, the robustness of your analysis is largely reduced. CO data from IASI on MetOP-A in 2008 were used from George et al. (2015). And why was O3 from IASI not used instead of/in comparison to OMI O3? If the observed small-scale variability is not an artefact of the poor retrieval (i.e., the smoothed response to local composition of the 1 or 2 independent partial tropospheric columns you can retrieve), then the very same structures should be present both in CO and O3. Even then, you may be seeing the same retrieval artefact but it would give further support to your approach since the retrieval of the two targets would be most sensitive to different layers of the atmosphere.

P4 L25: Figure 1a. It would be good to focus only on the ASM region under investigation, plotting the data with actual size of the nadir footprint and an indicative footprint of the smoothed horizontal area defining each MLS measurement. To complete the information, I would also show a companion plot with a vertical cross section passing through the ASM region showing the position of MLS tangent points and the 1 or 2 independent points of the nadir profiles (maybe with error bars indicating their vertical resolution).

P4 L29: I would rephrase "vertical informration distribution". Figure 1b. It would be useful to add the same figure for OMI O3 AKs (and possibly for IASI O3). Or simply discuss whether they are similar.

P4 L30: explain whether IASI 12-16 km partial columns are a standard IASI product or you calculate them to match the anticyclone vertical range. Could you compare them to partial columns from MLS too?

3. Comparison

P5 L12-14: could you show the vertical extent of the anticyclone in one of your plots (e.g. in the additional panel of Figure 1a I suggested, or in Figure 7)?

P5 L23: it would be helpful to add a conversion to km or hPa for the various levels you

adopt (e.g. is 147 hPa for MLS CO about 14 km and layer 18 about 16-17 km?). Have you tried repeating the analysis at different levels? Does it remain consistent?

P5 L24: see comments above, why not choosing the same year for both targets and prevent a self-consistency comparison?

P5 L28 – P6 L9: the analysis on JJA averages show quite large differences in CO and a much better agreement in O3. Even for O3, the wave pattern at 0/30N is not visible in OMI data. Why? I would expect finer scales to be resolved by nadir? Is that at a layer outside the nadir sensitivity? Can you give a more in-depth interpretation of how to read these results based on the low DOFS nadir data have? P6 L10 ...: The scatter plots of Fig 3 are very useful: here clearly the correlation of O3 is very good and the sensitivity to CO is weak with likely contamination from other layers. I recall these are 3-month averages: have you performed a similar comparison on shorter time scales? Can you compare the agreement/standard deviation you find to e.g. what is an accepted comparison in validation studies from the literature? Since these are comparisons of 3-month averages and not coincident profiles, it seems to me that the sentence P5 L17 states more confidence than what is shown by the data.

4. Sub-seasonal variability

P7 L19-20, L22-23, L28-32, Fig 5/6: IASI CO is showing enhanced values within the ASM region but also scattered features outside. Why would MLS not pick for example the feature of enhanced CO at 90E and 10S/0 if it were around the 147 hPa layer? How can you distinguish cases with high CO in the upper troposphere from cases with high CO in the middle troposphere (see the high levels of CO in e.g. Fig 7)? The different evolution of MLS CO and IASI CO over the days points toward signals coming from different layers. P8 L1: I think you are underestimating the shortages of this analysis and should be more careful with this statement. There is some sensitivity to UT CO but you cannot distinguish it from sensitivity to the rest of the troposphere. Fig 7 would be more useful if compared to MLS data (in the upper range) and to a model simulation to

understand the vertical structures and compare it to that reproduced by the 2 DOFS of IASI CO.

P8 L11-L21, Fig 8: I find it difficult to read the figure as reported in the text. Could you over plot the zero-anomaly contour of one on the other? Or the GPH shape? Could you mark the trajectory of the moving local maxima to highlight the propagation of anomalies? I do not see a convincing agreement. I see it clearly in the model by Pan et al (2016). Again, with sensitivity studies you would have more support not to speculate in L19-21. Please quantify what is coming from the UT.

P8 L22..., Fig 9: the regional timeseries look more robust, which seems to me supporting the fact that when you average more data together you remove variability and can find a better agreement. But this is not supporting your attempt to use nadir data to investigate finer scales. Can you help extracting more information from the timeseries? What parts are in robust agreement and why? What happens if you over plot timeseries for the Tibetan and Iranian lobes? What if you over plot timeseries at different altitude?

P8 L32-34: it would be more convincing if the agreement with Pan et al. (2016) was shown adding their data to the figures. Please rephrase "able to detect the impacts of vertical transport".

P9 L13-L21, Fig 10/11: there are large differences that need to be investigated with sensitivity tests. Are they due to different layers being involved? What is coming from measurement of the UT and what is contamination? You need to quantify this to identify what you can extract from the data you show.

P9 L22-L29, Fig 12: see comments for Fig 8. How can you get different frequencies for the migrating anomalies among the three plots?

P9 L30: I am not sure how you attribute the differences to the sampling densities rather than (lack of) sensitivity to different layers.

P10 L1-2: Can you investigate more these timeseries and identify when and why they

correlate and when not? The reason of the weak correlation needs to be better expressed since to me these timeseries should give the strongest support to your analysis: if they fail to correlate, can you trust the nadir data in the way you adopted them?

5. Conclusions

P10 L6- P11 L9: I think the conclusions should better reflect the limitations of the adopted nadir data and the caution needed to deal with them under these circumstances.

FIGURES

General: for some reason, the figures at actual size on screen (especially labels) show poor quality, whereas they have very good quality when enlarged. Could you tune this?

Fig 1a: I would limit the map to the ASM region. Please add a panel with vertical snapshot – see text. Blue/red is very hard to read. Could you have the symbols with the actual size of the footprints (horizontal resolution for MLS) when limiting the map to the ASM region?

Fig 1b: increase legend font size. Possibly place the zero aligned. Why are the numbers so different? I would like to see also the same plots for O3.

Fig 2: it is a bit confusing to have a different color scale for the two. Could you add a remark in the text? Remove "(GPH)" or "geopotential height" from caption as it was already explained. Please mention first "a,b", then "c,d" in the caption.

Fig 3: could you provide some significance test? Could you add also a comparison of the vertical profiles with their standard deviations too?

Fig 5 and 6: please join the two figures to allow easier comparison. Fonts at actual size are not readable. Why do you fill missing data? Please let them visible.

Fig 7: could you add an approximate vertical scale in km too since you use both in your text? Could you add the vertical and horizontal extent of the anticyclone? You

appreciably pay attention to the horizontal interpolation for MLS data, but to my understanding this plot is produced with only 2 independent points on the vertical. Is this correct? If so, could you add a comment/a sign for this?

Fig 8: could you help the reader over plotting contour of one zero anomaly line on the other? Or of the GPH? In a similar fashion as in Pan et al. (2016).

Fig 10-11: please join the two figures and see comments above.

Fig 12: consider over plotting reference contours as in Fig 8.

TECHNICAL CORRECTIONS

General: please note that the use of the article "the" is recurrently not consistent throughout the manuscript. E.g., P1 L17: "the ASM UTLS trace gas", P1 L21 "of ASM anticyclone", P1 L25 "of the ASM anticyclone", etc.

P1L13-14: since you first introduce MLS, I would link to it in the following sentence where you state you work on IASI and OMI and avoid the reader to wait to know why MLS was introduced: e.g., P1 L15 ". . . IASI and OMI, IN COMPARISON TO THE MLS LIMB SOUNDER."

P1 L14: "these type" – > "this type"

P1 L15: "IASI" and "OMI" were not introduced before and needs to be explained

P1 L16: remove "in the UTLS" after "(O3)" as it is repeated at the end of the sentence.

P1 L17: "variability is" – > "variability are"

P1 L18: remove comma after "explored"

P1 L19: – > "results show"

P1 L26: remove "(GPH)"

P2 L4: ";" – > ":"

P2 L12: – > "of the Tibetan mode"

P2 L15: "Asian summer monsoon" – > "ASM"

P2 L18: – > "chemical impact."

P2 L21: – > "are widely used for this purpose"?

P2 L25: this is slightly confusing since in the title you mention "Limb and Nadir…" (limb first), and here you introduce your work stating you work only on nadir data

P2 L26: "Two specific dataset we explore" seems as the reader should expect further nadir data to be used in the study

P2 L28: 2-3 months at what altitude/which layer?

P2 L29: "long lifetime" at what altitude/which layer?

P2 L30: "pollution sources" with negligible or not negligible impact?

P2 L31: "will be examined" – > "were examined"

P2 L32: remove "the" in front of "MLS", this is the first time you introduce them

P3 L1: remove "the" in front of "UTLS levels"

P3 L15: – > "also aims to"

P4 L17: "which" – > "whose", "is greater" – > "are greater"

P5 L19: "pressure" – > "vertical"

P6 L24: "10-20 day" – > "a 10-20 day"

P7 L9: – > "(Aug 18, when the center of the . . ."

P7 L19: – > "Compared to MLS…"

P8 L30: – > "by differenCES over…"

P12 L15: George et al reference is wrong

P13 L14: Liu et al misses page/volume number. P13 L 30: McPeters et al. misses page/volume number. Please check other references have the same problem.

---

## Referee Comment (RC3) · Anonymous Referee #3 · 29 Aug 2017

General commentÂă:

The main objective of the paper is to demonstrate the ability of nadir viewing sensors to document the sub-seasonal variability of CO (IASI) and O3 (OMI) in the UTLS during the Asian Summer Monsoon (ASM) and their relationship with the dynamics of the Asian Monsoon Anticyclone (AMA). Since more than a decade this subject has been widely studied and documented with various satellite sensors and models. The goal of the present paper is not to bring new insight about the dynamical processes that control CO and O3 in the UTLS during the ASM. It mostly aims at demonstrating the

capability of IASI to document UTLS CO. As detailed below, this demonstration is not fully convincing. Finally, the complementarity of CO and O3 data to document the AMA dynamics is not put forward because data of different years are used for both sensors. Three questions are proposed in the introduction (P2L33-34 P3L1-3) but no clear, positive and thorough answers are given in the paper as discussed in this review. For these reasons, I think this paper is not publishable in ACP and that major revisions and improvements are needed before the paper is re-submitted. As the paper focuses on satellite data capability in the UTLS, the authors should strengthen their demonstration and validation of IASI UTLS CO and the paper would rather be submitted to AMT than to ACP. Once or if IASI is proven to be able to document UTLS CO on a sub-seasonal scale in the AMA, they should take advantage of the complementarity of CO and O3 data from nadir sensors and of the long records to bring new insights in the ASM and AMA science.

Detailed comments ă:

IASI UTLS COÂ ă: The largest part of the paper is dedicated to demonstrate that IASI is able to document the day to day variability of CO in the UTLS during the ASM over Asia.

-1/ The first part of the demonstration is based on literature. According to George et al. (2009) the IASI retrievals contain 0.8 to 2.4 independent element of information depending probably on the location. More details about the theoretical independent layers (when DFS > 1.5) and a focus on the region of interest (AMA) should be provided. Are there about 2 elements of information consistently above the AMA region or just about one ? The paper of De Wachter et al. (2012) is cited as a validation of IASI UTLS CO. Nevertheless, this paper compares IASI and MOZAIC columns for the 470-250 hPa range which is not the UTLS. George et al. (2009) compares MOPITT and IASI CO tropospheric columns but not specifically UTLS columns. Barret et al. (2016) comparisons of IASI and model distributions concerns another IASI retrieval product.

-2/ The second part is based on the retrieval characterization with the averaging kernels presented for the AMA region in Figure 1b. The 0-12 km kernel shows the largest sensitivity (HWHM) with values exceeding 0.7 between 2 and 13 km. IASI is therefore clearly sensitive in the mid-upper troposphere. The 12-16 km kernel displays a very low sensitivity over the whole troposphere and UTLS with a weak maximum with low values (below 0.25!) above 10 km. Furthermore, the values of the 12-16 km AvK are much lower than the values of the 0-12 km over the UTLS altitude range. This means that (i) the sensitivity to the UTLS is very low (ii) the two kernels are not independent with clear resolved maxima (iii) more information about the 12-16 km is contained in the 0-12 km retrieved column than in the 12-16 km column. From this characterization it is therefore not clearly proven that IASI is able to provide independent information about UTLS CO. The AvK is averaged over a large region with very different surfacesÂă: desert, high mountains, low lands, oceans... As the AvKs depend on surface properties, there is probably no homogeneity in the vertical sensitivity over the whole region and more details should be provided rather than refering to the general "averaging kernels discussion" from George et al. (2009).

- 3/ The third and largest part of the demonstration is based on comparisons/validation with MLS UTLS CO. - The MLS daily data are interpolated to provide a global distribution. This methodology is questionable with so sparse data (space of about 15° longitude or 1600 km between observations in the tropicsÂă!). This can be seen on Figure 4 with plumes of high CO over the Bay of Bengal or the north western Pacific that are not related to real observations. A better methodology should be to average over 5 to 7 days as is normally done with MLS data and to compare with IASI averages or to use data assimilation. - Concerning the averaged seasonal distributions, there is a relative agreement between IASI and MLS CO. Nevertheless looking in more details, there are large discrepancies which are not so clear because of the large domain displayed. The inside of the AMA is characterized by homogeneous high CO with MLS while lower CO appears to the north (north of 30°) and to the east of the AMA with IASI. Furthermore, the latitudinal gradient of CO between the tropical UT (high CO) and

extratropical LS (lower CO) detected by MLS is not detected by IASI. It is especially clear for the southern tropical Indian Ocean and Pacific around 30S. These discrepancies are probably due to the fact that IASI detects the highest UTLS enhancements of CO that coincide with the deepest convection in the monsoon region from the Arabian sea to south-east Asia. It seems that IASI also detects UTLS enhancements over the African monsoon because of strong convective uplift of BB plumes. It could be interesting to show the whole African monsoon region to confirm this ability and strengthen the demonstration as it is the aim of the paper. The border of the AMA is shown with a very thick white line that partly hide the CO distribution. - - - An important proof of the ability of IASI to detect high CO in the UTLS independently from the lower-mid troposphere would be to show the tropospheric columns. In the eastern and central part of the AMA (ASM region) high tropospheric CO due to large emissions should be present contrarily to the western and northern part. The latitude-pressure section displayed in Fig. 7 rather shows that the information is mixed over the whole troposphere and UTLS (see comment below). - The correlation plot between IASI and MLS CO also shows some consistency between the two datasets but the correlation coefficient is rather low ($r2 = 0.52$ « 1). As mentioned, this low $r2$ and the reduced slope indicates a rather low sensitivity of IASI to UTLS CO. Whether it is due to the use of a single a priori profile is not obvious and has not been demonstrated. Without specific sensitivity tests using other a priori profiles, the statement p6l14-15 should be removed or modified. - Finally, daily MLS and IASI CO distributions corresponding to different dynamical situations of the AMA are compared. It is mentioned that with IASI "the spatial distributions show many finer scale structures" than with MLS. Given the low UTLS sensitivity and possible mixing of UTLS and tropospheric information as discussed above, it is difficult to know whether these finer scale structures are really in the UTLS. FurthermoreĂ: (i)- we see no holes in the IASI daily distributions in the ASM region impacted by very large and deep clouds. This means that the IASI observations have been interpolated and that part of the finer scale structures could be interpolation artifacts such as those discussed for MLS. It would be much better to see raw IASI data without interpolation.

Otherwise, averages over a few days would be much better than absolutely trying to show daily distributions with spurious features. (ii)- many of those finer scale structures are not detected by MLS such a CO bubbles in the south eastern Indian Ocean on 18 and 20 August, or over the north western Pacific on 20 and 24 August... over equatorial Africa, some high CO spots are detected by IASI and not by MLS on 18, 20, 24 August. They may correspond to BB plumes in the middle troposphere but this has to be discussed thorough-fully. (iii)- the westward elongation of the AMA highlighted by the GPH field (west of 30°E) is very clear and consistent with MLS but not detected by IASI on 24 and 26 of August. This is again a probable evidence that IASI is not sensitive enough to detect UTLS CO enhancements that correspond to low integrated amounts outside of the convective and polluted regions.

The answer to point 2 and 3 found in the paper is given P8L19-21, starting with "we speculate...". This is probably the answer of the discrepancies but (i) if true it proves that IASI is not really able to detect CO in the UTLS independently from the mid troposphere and therefore that the product is not so valid to characterize the AMA composition and dynamics at the sub-seasonal scale (ii) as this is somehow the central point of the paper, the demonstration should be more thorough and based on comparisons between 0-12 km and 12-16 km distributions (as mentioned above) and a latitude pressure sections at a location west of 90E and/or a longitude-pressure section at the center of the AMA to show wether the east-west lower tropospheric and UTLS gradients are different.

The vertical structure of IASI CO over the ASM region is finally displayed on Figure 7. As discussed in the paper, the uplift of CO within the monsoon region up to 100 hPa is "clear" around 20°N. Nevertheless this latitude-pressure section needs some explanations aboutÂǎ: (i) the CO concentrations is homogeneously high over the whole troposphere and UTLS in the convective region. This could be explained by the color scale which saturates at 100 ppbv (in that case the authors should change this color scale to show the detailed vertical structure) or by the fact that the information provided

by IASI is not sufficient to separate the lower tropospheric pollution from the convective outflow which seem to be mixed together. Indeed, convection detrains preferentially at high altitude (above 200 hPa) and convective enhancement should not be connected to lower tropospheric enhanced CO. (ii) We also see large CO vmr north of the Himalaya range from the surface to 200 hPa. This region (Tibetan plateau) is not impacted by important emissions nor by deep convection. How can such elevated CO reaching such high altitudes be explainedÂă? (iii) The same is true between 10S and 10N below 300 hPa. Which emissions and transport processes are responsible for these high CO concentrations over the Indian Ocean up to 300 hPa ?

The Hovmoller diagram of Fig. 8 corroborates the low sensitivity of IASI to UTLS CO enhancements in the western part of the AMA and the detection of features of high CO on the eastern part that are not detected by MLS and not consistent with GPH anomalies. This is rather problematic to state that IASI is able to detect CO in the UTLS.

OMI UTLS O3Âă: As already mentioned, the choice of 2008 for OMI and 2012 for IASI raises the usefulness of presenting/using both datasets. The explanation of that choice is not fully convincing. IASI 2008 data are available and it is not clear that the results for a single monsoon season may be impacted by jumps induced by changing L2 data (when does this change occurs?). It is mentioned that "O3 and CO data are examined separately" which is a real weakness of the paper because (i) if the aim was to characterize the relationship between UTLS composition and AMA dynamics, the complementarity of both gases as pollution and stratospheric tracers would have been a strength (ii) the OMI part is much shorter than the IASI and comes to the conclusion that OMI O3 documents correctly the UTLS which has been shown elsewhere. Therefore, the OMI O3 part could be removed from the paper which should concentrate in proving the IASI CO UTLS capability. A future paper could focus on the dynamics-composition relationship using the O3/CO complementarity.

IASI is also providing information about the O3 vertical distribution and should logically

have been used preferentially for obvious coincidence criterion with IASI CO. Why is IASI O3 not used or even mentioned in the paperĂă?

---

## Author Comment (AC1) · 4 Oct 2017

**Response to reviewer 1**

Laura Pan on behalf of all co-authors

We thank the reviewer for the constructive comments and many helpful suggestions. Due to the significant amount of revisions (i.e., most of the figures are changed), we would like to provide an overview of revisions in addition to the point-by-point responses. This overview is included as a supplement.

**Point-by-point responses**

This manuscript investigates whether coarse vertical but high horizontal resolution measurements from two nadir-viewing sensors – IASI and OMI – can be used to characterize changes in UTLS composition in response to the Asian summer monsoon (ASM). Comparisons with Aura MLS limb sounding data are made to explore how well the nadir measurements represent enhancements in CO and depressions in $O_3$ within the ASM anticyclone as well as the sub-seasonal variations therein. The manuscript is generally well organized and well written (although further copy-editing, beyond the minor points listed below, is needed before final publication) and could make a valuable contribution to the extensive ASM literature. However, in my opinion the manuscript has a significant shortcoming that needs to be addressed before I can recommend it for publication: The authors have not done enough to demonstrate that IASI CO retrievals actually do resolve CO enhancements in the UTLS associated with the ASM. Certainly the data have some sensitivity in the upper troposphere. But the analysis presented in this manuscript has not gone far enough to paint a truly convincing picture that IASI has sufficient sensitivity in the upper troposphere to discriminate variations there from the equally large (or possibly even larger) variations in the middle and lower troposphere to which it has much greater sensitivity. I also have a number of specific comments and points of clarification (detailed below) that I would like to see addressed.

We have addressed this main concern of IASI UT sensitivity issue. See Overview of Revision point 1).

General comments:
* Figure 1b shows that nearly as much information is contributed to the IASI "12-16 km" product from 10 and 18 km as from 12-16 km. Moreover, there is a not insignificant degree of overlap with the averaging kernel for the so-called "0-12 km" partial column; in fact, the averaging kernel for the "0-12 km" partial column shows considerably more sensitivity in the upper troposphere than does the averaging kernel for the "12-16 km" partial column. In addition, non-negligible contributions to the "12-16 km" layer derive from 4-10 km. Thus it seems somewhat misleading to refer to the volume of air being

sensed by IASI, which is at least 8-10 km thick, as the "12-16 km" CO value throughout the paper.

We have revised the Fig 1b to show individual retrieval layer averaging kernels and focus on the 150 hPa level. Although the IASI sensitivity at this level is broad, the peak contribution is from the UT.

\* The study by Barret et al. [2016] is cited to support the statement that the IASI CO retrieval captures UTLS variability at middle and tropical latitudes (P4, L2-4). But that study focused on examining large-scale monthly mean features, not sub-seasonal localized variations such as eddy shedding. Furthermore, Barret et al. note that the smoothing of GEOS-Chem model fields imposed by the IASI averaging kernels mixes high CO concentrations from near the surface into the middle and upper troposphere. Thus I feel that the authors need to do more to conclusively demonstrate the power of IASI CO measurements to distinguish upper tropospheric variations from those at lower levels. One approach might be to perform sensitivity tests with model output (e.g., from GEOS-Chem as in Barret et al., or from WACCM4 as in Pan et al. [2016], etc.) to investigate the influence of the CO distributions in the lower and middle troposphere on the partial columns measured by IASI. Results from the "raw" model fields multiplied by the IASI averaging kernels could be compared to those derived from the same model output but with mid-tropospheric CO abundances reduced or enhanced by some fraction (say, 20%). Differences between the inferred "12-16 km" layer averages from such tests would help to gauge how much independent information IASI provides in the upper troposphere.

This comment is addressed in point 1) of the overview and demonstrated by Figs 7r & 9r, which are included in the overview file.

\* Why was a comparison of the vertical resolution similar to that in Figure 1b not shown for MLS and OMI O3?

We examine the AK for CO analyses because the identification of UT signature is important for the CO analysis. The focus is different for the ozone analysis, because 1) ozone is not a boundary layer tracer so we do not examine ozone for vertical transport, and 2) we expect ozone ASM signature at the 100 hPa to be associated with the tropopause structure (see revision overview point 2b). When analyzing the OMI ozone at 100 hPa, the message is that OMI retrieval at 100 hPa can identify tropospheric (inside the anticyclone) or stratospheric (outside the anticyclone) dominated region, i.e., separates UT from LS. We do not feel it is necessary to show already published averaging kernels in this case and we simply direct the interested readers to Liu et al. (2010b) fig 5. for the OMI averaging kernels.

Specific substantive comments:

* P2, L16: I don't believe that Garny & Randel [2015] is the correct reference for this point. First, the citation for that paper is incorrect: it should be ACP 16, 2703-2718,2016. Second, that transport pathways paper is not appropriate here: I'm guessing that the authors meant to cite Garny & Randel [2013] instead.

Revised

* P3 L2-3: Three main questions are articulated that this study is aimed at. The first two are clearly addressed in the manuscript. However, I am not sure that the third question – What can we learn from the complementary information from limb and nadir viewing instruments? – has actually been explicitly touched on. It would be good if the authors could add a sentence or two, probably in the Conclusions section, that circle back to this question to offer guidance on how the limb and nadir data sets could be applied synergistically to investigate specific science questions.

See revision overview point 3)

* P3, L22-23: According to the most recent MLS Data Quality Document [Livesey et al., 2017] (not [2015] as cited in the manuscript), the accuracy of MLS v4 O3 at 100 hPa is +0.005 + 7% ppmv, or ~20 ppbv for 200 ppbv of O3, not 50 ppbv. Although the along-track resolution is ~300 km for the O3 product at 100 hPa as stated, it is ~550-600 km for CO at 147 hPa.

Revised

* P4, L28-P5, L4: The formulation of Figure 1b is slightly confusing. Why are different x-axes used for the IASI and MLS averaging kernels? They could be compared more readily if plotted on the same scale. At the least, the zero lines for the two axes should be aligned (and perhaps a vertical line drawn through zero to guide the eye).

This Fig has been revised to have same axes.

* P5, L1-4: First, according to Livesey et al. [2017], the full width at half maximum of the MLS CO averaging kernel (by which the vertical resolution is defined) is 5 km, as stated on P3, L21, not 6 km as given here. Second, it would be better to refer to this as "vertical resolution" rather than "vertical distribution".

Revised

\* P5, L5-10: It is interesting that the authors chose to use the GFS operational analysis for this study, rather than one of the more commonly used analyses/reanalyses such as GEOS-5, MERRA or ERA-I. A sentence or two motivating that choice would be appropriate. This is especially true in light of the work of Nuetzel et al. [2016] showing that substantial bimodality in the ASM anticyclone, as asserted in previous studies, is apparent only in the NCEP-NCAR reanalysis and not in data from other modern assimilation systems.

The use of GFS final analyses data is largely motivated by the high resolution and good quality tropopause information. We included a reference of the validation analyses (Pan and Munchak, 2011) in the revised version. We have look at the comparable days of GOES-5 data and do not find it to have significant difference for the daily analyses in this work.

\* P5, L19: Park et al. [2007] note that confined tropospheric air masses are present in the ASM anticyclone up to 68 hPa.

Noted in the revision.

\* P5, L29: I understand the desire to employ different color bars for MLS and IASI CO. However, Figure 2 could give a false impression of the degree of agreement between the two fields to readers not paying close attention to the figure or its caption. Therefore the fact that the color scales are not the same should be mentioned in the main text as well as the figure caption.

We have re-evaluated the choice of colorbar ranges and . In this revised version, we have noted in the text when the different colorbar is used.

\* P6, L3-8: I agree that OMI data appear to represent UTLS O3 in the ASM region fairly well. However, the OMI JJA O3 field does not reflect the signature of tongues of extratropical stratospheric air being transported equatorward and westward around the edge of the anticyclone, which results in a ring of higher ozone surrounding the low values in the anticyclone center. This structure, which has been discussed in several

previous papers based on MLS and also MIPAS data, is readily apparent in Fig. 2c but is barely visible in Fig. 2d. I find this surprising given OMI's higher horizontal sampling/resolution and the fact that other smaller-scale features are seen in the map in Fig. 2d. On the other hand, such a signature of monsoon-induced stratosphere- totroposphere transport is prominent in the daily maps of Fig. 11. Is its absence in Fig. 2d just a color scale issue (since Figs. 2 and 10/11 have different color bar increments)? Or does the disparity in the JJA means in Fig. 2 imply that MLS and OMI observe different seasonal evolution of this transport feature, some of which lies equatorward of 15N and thus may not be captured in Fig. 13?

We think the lack of clear stratospheric "tongue" in the seasonal OMI map is possibly a result of averaging a small scale feature that has variable daily locations. This is consistent with the narrower latitude range of the stratospheric ozone signature south of the anticyclone. In general, this particular "wrapping" around the anticyclone is a relatively shallow layer. OMI profile product likely has more sensitivity to the "deeper" layer and has weaker response to this shallow layer.

* P6, L11: In these lines the "monsoon region" is defined as 0-50N, 0-150E, which is a rather broad area to label as being directly influenced by the ASM. I presume such an extensive region was used in order to encompass a range of values of both the tropospheric and the stratospheric tracer. If so, perhaps that should be explicitly noted, especially given that the ASM region is defined differently for different plots: on P4, L31 it is defined as 15-35N, 30-140E, and on P8, L12 as 15-35N, 0-150E.

Wordings are modified to be more consistent. The larger domain is now referred to as "the study domain".

* P8, L4-10: Again, it is not obvious to me that IASI has sufficient sensitivity in the upper troposphere to distinguish a "plume" of CO up to (and above, according to the figure) the tropopause that is clearly separable from the large abundances of CO in the lower atmosphere. The color bar in Figure 7 is strongly saturated at the high end. It would be better to adjust the color scale to allow some of the structure in enhanced CO abundances to become visible. That might reveal instances of localized enhancements in the upper troposphere that are not connected to the generally higher mixing ratios in the lower troposphere, providing more confidence that they are not simply a manifestation of contamination from below.

See Revision Overview point 1) and the new figure 9r. We have much more discussions on this point in the revised manuscript.

* P8, L33 – P9, L2: (1) The sentence "IASI CO data have a higher horizontal resolution and are able to detect the impacts of vertical transport in the troposphere" is somewhat confusing, because as written it seems to imply that the higher horizontal resolution enables the detection of the impacts of vertical transport. I think it would be better to

more clearly separate these two points. Perhaps something along these lines would work: "IASI CO data have higher horizontal resolution than MLS measurements. Despite its coarse vertical resolution, IASI is able to detect the impacts of vertical transport in the troposphere". (2) In my opinion the conclusion in the last sentence of this section that some of the finer scale structure evident in IASI CO data is attributable to eastward eddy shedding over the western Pacific is stated too definitively. That may be true to some extent, but as Figure 1b shows and the discussion on P7 makes clear, the IASI "12-16 km" layer average is substantially influenced by the mid-tropospheric CO distribution. In fact, the authors have basically said as much in the lines just above (P8, L19-21). Thus, unless the sensitivity analysis suggested earlier conclusively shows that such a statement is fully justified, I would like to see the wording in the last sentence of this section softened. The same comment applies to a similar sentence in the Abstract (P1, L23-25). (3) Although this work demonstrates the utility of IASI data for studying the evolution of CO over the ASM region, I think the essential bottom line is summed up on P8, L32-33: "MLS and IASI data have different advantages. MLS data are better for examining features with a shallow vertical extent, such as the ASM anticyclone, provided those features have a large enough horizontal scale." I feel that it would be appropriate to repeat this sentiment in the Conclusions section, and possibly in the Abstract as well.

These discussions are significantly revised. Also part of the ambiguity was due to the lack of clear evidence that IASI has independent UT sensitivity. With the new analysis, some of the conclusion statement now can be made stronger.

* P9, L5-7: It might be good to remind readers here that, although UTLS O3 is mainly a tracer of stratospheric air as noted, its distribution can also be affected by photochemical production, as alluded to on P2, L30 of the Introduction. So the interpretation of O3 fields in the ASM region is not necessarily straightforward.

Thank you. This point is much more emphasized in the revision.

* P9, L19-21: "OMI data show a sharper transition of O3 field across the edge of the anticyclone (as indicted by the 105 hPa tropopause contour)... differences between MLS and OMI are more pronounced at low latitudes". (1) Why is the tropopause being used to define the edge of the anticyclone here? Elsewhere GPH is used to denote the anticyclone boundary, not the tropopause. (2) How is the tropopause shown in these figures being defined? I realize that it is taken from the GFS analysis, but is a thermal or dynamical (PV-based) definition being used? That should be clarified. (3) Differences between MLS and OMI may well be more pronounced at low latitudes, but it is difficult to tell from Figs. 10 and 11 since the color scale saturates at high latitudes.

We will make it clear in the revision

* P10, L16: Although Garny & Randel [2013] did show that spatial variations in CO are well correlated with variations in the region of low PV defining the anticyclone, they did not find evidence of the kind of bimodality in the location of the anticyclone that Yan et al. did. So although it is certainly appropriate to cite Garny and Randel (among many others not listed) for highlighting the significant role of ASM dynamical variability in controlling UTLS tracer distributions, I don't think it is quite fair to include that reference for different "modes" of anticyclone behavior.

Revised.

* P10, L24-26: In general, the Conclusions section overlooks the potentially significant contamination in the IASI CO "12-16 km" layer average from lower altitudes. Such influence from below is likely to be another factor explaining the apparent lack of consistency with MLS 147 hPa CO and should be acknowledged here.

These issues are resolved in the updated analyses.

* P11, L7: I'm intrigued by the notion that the results from this study might be used to refine the IASI or OMI retrievals. Could the authors say more about that, perhaps provide an illustrative example of how these findings could inform IASI or OMI retrieval algorithm development? Also, please clarify which "differences" are being referred to – differences from MLS?

Specific comments will be made in the revision.

Following minor comments are all considered in the revision. Thank you for your hard work to help.

Minor points of clarification, wording/figure suggestions, and grammar / typo corrections:
* P1, L15: The IASI and OMI acronyms should be spelled out here
* P 1, L17: "changes . . . is" –> "changes . . . are"
* P1, L19: "result shows" –> "results show"
* P1, L23: I suggest "captures" rather than "show[s]"

* P1, L31: As the first sentence of the paper notes, the ASM anticyclone has been investigated widely in recent years, but the small subset of references cited for this

point seems somewhat arbitrary. Many more equally relevant papers could have been included, so it would be appropriate to add "e.g." at the front of the list.

* P2, L10: "i.e." –> "e.g."

* P2, L12: "in terms of"

* P2, L31: "Short time" –> "Short-term"

* P2, L33: "nadir view" –> "nadir viewing"

* P3, L5: "make" –> "makes"

* P3, L7-8: delete the second instance of "quantitative comparisons"

* P3, L13: "much" –> "more"

* P3, L13: "UTLS chemical tracers variability" –> "variability of UTLS chemical tracers"

* P3, L15: "aim" –> "aims" and "supplement" –> "supplements"

* P3, L16: "inform" is not quite the right word. Perhaps "examine", or something similar

* P3, L23: "and has" –> "with"

* P3, L31: "degrees of freedom signal" –> "degrees of freedom for signal"

* P4, L1-2: The MOZAIC and MOPITT acronyms should be spelled out; also add "and"
before "satellite"

* P4, L7: Does Huang et al. [2016] really discuss the OMI O3 profile product? I believe this reference is incorrect.

* P4, L9: delete "and" before "zonal mean". Also, "NCEP" has not yet been defined.

* P4, L11: It seems odd to use a tilde with such precise numbers for the DOFs ("~6.0-7.0")

* P4, L17: "which" –> "for which" and "is" –> "are"

* P4, L20-21: add "data" after "O3", "began" before "in January", and "has" before "impacted". Again, I don't think that Huang et al. [2016] is the correct reference for OMI O3 profiles.

* P4, L29: "analyses" –> "analysis"

* P4, L33: I think it is potentially confusing for readers to refer to the MLS data quality document in this manner here but as Livesey et al. on P3. Please be consistent. Also, the web site information should be provided in the reference list as part of the Livesey et al. citation.

* P5, L1: add "thick" after "8 km"

* P5, L16: "dataset" –> "datasets"

* P6, L5: "tied" –> "tied to"

* P6, L6: delete "in" after "within"

* P6, L17-18: "the UTLS chemical impact by ASM anticyclone" –> "the impact of the ASM anticyclone on UTLS chemical composition". Also, I think it would be more accu- rate to say "a picture *largely* consistent with that from MLS".

* P6, L32: "the empty" –> "an empty"

* P7, L11: "splits" –> "split"

* P7, L18: "enhancement" –> "enhancements"

* P7, L21: "are much more extended in longitudinal range compared to the MLS, co- located and mimic the east-west extent of the" –> "is much more extended in longitu- dinal range compared to that from MLS, co-located with and reflecting the east-west extent of the"

* P7, L22: "distributions" –> "distributions on" and "that are not" –> "that is not"

* P7, L25: "on the south" –> "to the south"

* P8, L20-21: "is contributed by the retrieval information in the level lower than that represented by 150 hPa dynamical field" –> "reflect the influence of retrieval information from a level lower than that represented by the 150 hPa dynamical field"

* P8, L30: It is not clear what is meant by "the two sensors are influenced by different over vertical columns". I suggest instead something along the lines of "the two sensors sample quite different volumes of air".

* P9, L7-9: I suggest rearranging /rewording this sentence and replacing "interception", which is not correct here: "The structure of the bulging tropopause in the monsoon region (indicated by the intersection of the tropopause with the 105 hPa pressure level in Figs. 10 & 11) (Bian et al., 2012; Pan et al., 2016) has a significant influence on the O3 distribution."

* P9, L11-12: It is not entirely clear that the Fig. 9 being referred to here is from Park et al., not the current manuscript.

* P9, L18-19: "OMI also shows similar distribution" –> "OMI (Fig. 11) also shows similar morphology". Also, "Quantitatively" –> "Qualitatively" and "indicted" –> "indicated"

* P9, L31: "is" –> "are"

* P10, L1-2: "sectional anomalies" –> "regional anomaly". Also, "vertical and horizontal samplings of two satellites" –> "vertical resolution and horizontal sampling of the two satellite instruments"

* P10, L9: I think that "reduced" would be better than "weakened" here

* P10, L12-13: To be perfectly clear, please change "the weaker UT sensitivity" to "IASI's weaker UT sensitivity" and add "its" in front of "retrieval". Also, "product" could be deleted.

* P10, L14: "dynamic" –> "dynamical"

* P10, L16: "Garney" –> "Garny"

* P10, L29: "convective-driven" –> "convectively driven"

* P11, L6: "dynamic" –> "dynamical"

* P11-15, references: Several references (e.g., Pan, Park 2007, Randel 2006, Vernier, etc) are incomplete (e.g., pages and/or doi missing)

* P12, L15: I am not familiar with the 2015 paper by George et al., but I am quite sure that "Bmc Medical Genetics, 8, 4095-4135" is not the correct citation for it

* P16, Fig 1a caption: please clarify whether the statement about the symbols being enlarged in Figure 1a applies to both data sets or only to MLS.

* P17, Fig 2 caption: "mean of CO" –> "mean CO" and "Note the" –> "Note that the". Also it probably would be a good idea to specify in the caption that the GPH values are also taken from the GFS analysis.

* P17, Fig 4 caption: "geolocation" –> "geolocations" and "selected GPH of" –> "selected GPH values at". Also, a few MLS data points at various spots in the map in Fig 4a are plotted in black − are these points off the color scale (on both ends)? That shouldn't be the case given the way the color bar is constructed.

* P20, Fig 6 caption: note here also that the color scale in this figure differs from that used in Fig 5.

* P21, Fig 7 caption: "white dash" –> "white dashed lines"

* P22, Fig 8 caption: "5 deg bins" –> "5 deg longitude bins" and "period" –> "periods"

* P23, Fig 9 caption: "dash lines" –> "dashed lines"

* P24, Fig 10 caption: "white" –> "white contours" and "interception" –> "intersection"

* P25, Fig 11 caption: "mapped in" –> "mapped onto a" and "grids" –> "grid"

* P26, Fig 12 caption: add "bins" after "longitude" for OMI data

* P27, Fig 13 caption: "dash lines" –> "dashed lines"

---

## Author Comment (AC2) · 4 Oct 2017

**Overview of the Revisions**

Laura Pan on behalf of all co-authors

We thank all three reviewers for the constructive comments and many helpful suggestions. Due to the significant amount of revision (i.e., most of the figures are changed), we would like to provide this Overview of Revisions in addition to the point-by-point responses. The changes summarized in this overview will be referred to in the point-by-point response to each reviewer.

The major criticisms from the three reviewers can be summarized as the following four aspects: 1) IASI CO retrieval upper tropospheric (UT) sensitivity is not adequately demonstrated, 2) The CO and  $O_3$  data are not examined using data from the same year and are not combined as a complementary analysis, 3) Potential complementary information from the nadir and limb data was posed as a key question but not clearly answered, and 4) the objective of examining representations of the sub-seasonal scale variability in the limb and the nadir data is not sufficiently met. In the following, we outline how each of the four aspects are addressed in the revision.

- 1) IASI CO retrieval UT sensitivity
  - a) To verify and demonstrate the IASI retrieval UT sensitivity, we have introduced two new figures, Figs. 7r and 9r (here we us "r" to label the revised version). These two figures are included at the end of this overview.
  - b) Fig. 7r shows IASI CO at 150 and 500 hPa levels for a selected day. The consistency of the CO distribution with the flow pattern at the UT level and the distinct differences between the upper and middle troposphere distributions are effective indications of the independent UT information in the IASI CO retrieval.
  - c) Fig. 9r shows four selected cross-sections and the a priori profile for the retrieval. The cross-sections are selected with correspondence maps in Figs. 7r and 8r (not included). Here again the dynamical consistency of the locations of the UT enhancement, the independent variabilities of the UT and the lower to middle troposphere are all good indications of effective UT information content. In addition, the comparison between the vertical structure of the cross-section and the a priori profile is also a powerful demonstration of the UT retrieval information content.
  - d) We revised the averaging kernel plot (Fig. 1r-b) to show the averaging kernels for individual IASI retrieval layers and also MLS UT level averaging kernels for comparison.
- 2)  $O_3$  and CO data from the same year
  - a) In this revision, we re-worked our analysis to use IASI 2008 CO data, which was not available to us at the time of the previous analysis.
  - b) Note, however, the CO and O3 are analyzed independently in this work. These two tracers serve to examine two different aspects of the ASM dynamics:
    - The CO analysis is the main focus of this work, because the UT (150 hPa) CO variability is associated with both the convective uplifting of the boundary layer air and the horizontal transport driven by the anticyclone dynamics.

- For the O3 analysis we focus on the 100 hPa level where the depression of ozone mixing ratio is primarily a result of the tropopause structure in this region and reflecting the troposphere (stratosphere) dominated air mass inside (outside) the anticyclone at this level. We choose to examine the 100 hPa O3 to examine how MLS and OMI O3 data seeing this structure.
- As a tracer at UT, O3 is a more complicated as also pointed out by the reviewer 1. O3 can be positively correlated with CO if the air is convective uplifted from a heavily polluted boundary layer. This type of CO-O3 relationship analysis should be pursued by a dedicated study.
- 3) Nadir viewing sensors' contributions and how the two types of sensor complement each other
  - a) We have changed the approach to IASI CO analysis from using 12-16 km layer average to data near the 150 hPa level. This allowed a much better coincidence between the CO field and the dynamical field. With some additional technical improvements, we now see a consistent picture of sub-seasonal east-west variability of the UT CO from both MLS and IASI.
  - b) Although consistent, the IASI data show a stronger enhancement over the western Pacific mode of the anticyclone and a weaker enhancement over the Tibetan Plateau region compared to the MLS data. The former is likely due to the factor that the enhancement over the western pacific is potentially a layer with contribution from ~200 hPa. See Fig. 4r for seasonal average comparisons that supports this "tilted" structure. The latter is likely contributed by the reduced signal-to-noise ratio due to the reduced atmospheric column in the region of high surface elevation. The monsoon season's frequent cloudy conditions also cause wide range of missing data in this region. See Fig. 1r for cloud contribution to missing data in IASI.
  - c) The nadir data (both IASI and OMI) provide much more details in horizontal structure and variability due to the denser daily coverage. The quantitative evaluation of these structure should be targeted in future validation studies.
  - d) Furthermore, the vertical cross-section from IASI demonstrated the capability of identifying the source region in vertical transport. The 3D CO distribution also show that the CO UT enhancements in the Iranian mode (and the western Pacific mode) are not vertically transported from the local boundary layer, which supports model interpretations of eddy shedding. This is the first observational confirmation of the model study on the topic (Pan et al., 2016).
- 4) Satellite data representation of sub-seasonal scale variability in the ASM UT composition
  - a) We have focused more on the sub-seasonal UT CO variability seen by both the MLS and IASI data. We show that both datasets provide good representation of synoptic scale variability, but careful interpolations and gap fillings are required. An excellent correlation between the MLS CO and the Geopotential Height anomalies is demonstrated in a Hovmöller diagram (Fig 6r)
  - b) Most interestingly, both datasets show a regular enhancement over the western Pacific near Japan, associated with a local anticyclone system, known as the Bonin High. A good example is given by Fig 7r.

In additional to these key changes, we have explored a number of different technics to improve the methods of interpolations and filling data gaps. The improved analysis leads to a much stronger set of conclusions in this paper. Below we included the figures specifically referenced in this overview.

---

## Author Comment (AC3) · 5 Oct 2017

**Response to Reviewer 2**

Laura Pan on behalf of all co-authors

**Point-by-point responses**

This paper investigates the sensitivity of nadir looking satellite sensors to changes in the chemical composition of the UTLS region within the Asian summer monsoon (ASM) as compared to typically adopted limb sounders. It explores both seasonal and day- to-day variability attempting to exploit the high horizontal resolution of nadir sounders to better depict horizontal structures in the ASM distribution of tracers of pollution and stratospheric air (CO and O3). This study could give a valuable contribution both to the broad issue of exploiting nadir sensors at UTLS altitude, and to the ability of observe and understand this region at fine spatial and temporal scale. However, attempting to tackle both challenges in the same study, the authors have failed, in my opinion, to achieve enough robustness and the study needs substantial improvement before I can recommend it for publication on ACP.

We appreciate the reviewer's criticisms and have made major revisions to address these general concerns. The overview of the revision is given in a separate file.

GENERAL COMMENTS

The manuscript is generally well written and structured, although some editing may improve it. Despite appreciating this effort of promoting a synergetic use of nadir and limb data, I find there is a certain lack of overall clarity on the aims of the paper. The study is introduced as a qualitative comparison of nadir and limb data (see introduction), but with the ambition that their results would support the use of this approach in future research. It is not clear to me whether the study aims therefore at validating the use of the adopted nadir observations under ASM UTLS or trust support from the literature and aims at producing novel results for the ASM UTLS region. They focus on 3 questions: the first two point to a validating exercise which cannot be kept at a qualitative level, the third one to a more general interpretation of the results for the ASM UTLS region, although leaving it with no clear answer.

We agree with the reviewer that the previous version of the paper has significant weakness in addressing the objectives. We have made significant revisions to improve four aspects of the paper as detailed in the overview.

This work, however, was not designed to be a quantitative comparison, as stated in the last paragraph of the introduction. In the revision, we further emphasize that this work is NOT a validation study in its traditional sense, i.e., to compare the measurements to trusted source and quantify the accuracy and precision of the retrievals. Instead, this is an exercise of "process-based retrieval EVALUATION". Here the goal is not to substantiate the specific values produced in the retrieval, but rather, the representation of a process, i.e. the Asian monsoon dynamics and transport, in the UTLS CO and $O_3$ field, which is the overarching idea of the three questions. In the revision, we made effort to clearly answer all three questions posed in the introduction.

The authors use the observed ASM atmospheric composition (in comparison with MLS) to support the use of the adopted nadir data and then use the same nadir data to investigate the details of the ASM atmospheric composition at finer scale (going beyond MLS). The observed details and fine scale variability has to be proved to be a real measure of the natural variability

and not a combination of perturbed conditions and a low sensitivity retrieval. The authors appreciably introduce averaging kernels and mention vertical resolution/degrees of freedom for signal (DOFS) but do not use them quantitatively. E.g., the IASI mean averaging kernel for the 12-16 km observation (Fig1) is so broad that contributions from layers outside the range are expected and the high CO cannot be assigned to a shallow layer, contrary to MLS data. The same ap- plies to O3. A vertical resolution of 10-14 km: the a priori profile is likely to be simply scaled to match the average sensed value of the whole tropospheric column. Indeed, the horizontal details presented by the authors are very interesting, but should they be read more like an on/off effect of a perturbation at an unknown tropospheric layer? You need to quantify how much of the results are coming from the measurement and from what layer. In order to support the authors' claims, there is a need for performing sensitivity studies to investigate the response of the retrieved profile to perturbations at varying altitude that mimic the observed ASM behavior, contributions from different layers and possible contaminations (clouds?). To this end, both retrieval simulations and atmospheric model simulations (e.g. the referenced Pan et al., 2016) would be of help. Furthermore, comparison between sounders could be performed more quantitatively, e.g. introducing convolution with the averaging kernels.

We have included new figures to demonstrate the IASI UT sensitivity. Please see point 1) in the overview.

Again, we do not aim to make quantitative comparison. As stated in the paper, the two sensors (MLS and IASI) are observing very different air masses.

I find very unusual the choice of using two different years for the analysis of the two targets. I see no reason not to compare results for nadir CO and O3 for the exact same days and regions (and also O3 from the two nadir sensors) and verify that the small-scale structures you attribute to natural variability are indeed consistent among the two targets. Even then, you may still be seeing a retrieval artefact but in response to sensitivity to different layers of the atmosphere, so giving further support to your approach.

We have re-worked the CO analysis using IASI 2008 data. See point 2) in the overview.

I encourage the authors to consider these main issues and the specific comments below in order to improve the paper to make it a valuable reference for future studies.

SPECIFIC COMMENTS (P=page, L=line)

Abstract:

P1 L14: the use of "information content" is misleading since it is not analyzed in the manuscript. You only quote mean estimates of DOFS from the literature. On the other hand, adding an analysis of the information content from the retrievals would be of great help in understating the sensitivity of the nadir observations under these conditions. How much information for each point of your profiles is coming from the measurement? How much from the apriori? Adding maps with this information could give convincing support. P1 L16: same.

In the revision, we aim to make this point clear that conventionally the phrase "information content" has a specific meaning in retrieval, and it is often quantified by the "degree-of-freedom in signals (DOFS)". In this work, we advocate a "process-based" retrieval evaluation where the information is evaluated for how effective the DOFS is in retrieval results, in this case, to represent the Asian monsoon dynamics and transport. The dynamical consistency we demonstrated with the new figures is a clear indication that although IASI has relatively weak

UT information, the retrieval used the information well to represent the dynamical variability effectively. Figs 7r, 8r, and 9r in particular demonstrate that the effectiveness of the information content is shown by the geophysical consistency of the CO maps and cross sections.

P1 L15: possibly due to a typo, the sentence starting with "Day-to-day behavior" should be rephrased discussing first the seasonal scale analysis, then moving to the day-to- day and finer scales in the following sentence.

Revised.

P1L 20: I would tune this sentence down depending on further support to the analysis of the layers that actually produce the signature in IASI data. P1 L21: The same issue applies to OMI data, for which you actually show no profiles in your work.

For OMI $O_3$, see point 2) in overview of revision

P1 L24: same.

Introduction P3 L4-9: this is a key statement for justifying the lack of robust analysis during the comparison. I feel the authors failed to extract enough support from the listed literature in order to prove the claims of their paper. See comments to the listed references here below. If the ambition is to show how the synergetic use of nadir and limb sounding can improve the understanding of the ASM UTLS, then you need to quantify the agreement, more carefully analyze the limitations of the two and what parts of the nadir observations can be trusted. You could then propose a strategy for how to merge the horizontally coarser and less frequent vertical information from limb sounders to drive the finer and more frequent picture coming from nadir observations.

Again, it is not the goal of this work to "validate" the nadir sensors' retrieval through comparisons with MLS data. The goal is to examine the consistency of both limb and nadir data with the dynamical field over the Asian monsoon region. See point 3) and 4) in the overview for the nadir data's contribution beyond what we can learn from MLS.

P3 L9-14: how can a qualitative comparison help to assess the information of these nadir viewing datasets for ASM UTLS studies? Please explain. P3 L14-16: I am doubtful on what part of the study investigates the information content and how your analysis can help further studies if the horizontal information cannot be located at a correct altitude range.

Vertical range of the IASI CO information is demonstrated in the revision. See Figs 7r and 9r included in the overview of revision.

Data Description

P3 L19-P4 L17: I think the available support for your study from the literature you list has been overestimated. In general, the profiles of CO and O3 retrieved from IASI and OMI have such a low vertical resolution and DOFS that have almost no sensitivity to differentiate layers in the troposphere, allowing for 1 or maximum 2 independent partial columns (or points in your profiles) throughout the whole troposphere (see e.g. George et al., 2015). Liu et al (2010b) show OMI O3 vertical profiles have 0-1.5 DOFS in the troposphere, or 14/11 km vertical resolution at 12.5/17.5 km altitude. Kroon et al (2011) show (their Fig. 1 and discussion) that in the troposphere it is almost impossible to distinguish different layers in OMI O3 (certainly true around the 100 hPa level). Wachter et al (2012) validation reach only 225 hPa for IASI CO, i.e. below the layer adopted in this study. Bak et al. (2013) state that a large smoothing error is introduced in OMI profiles by the retrieval. Safieddine et al (2016) only use 0-6 km columns. Barret et al (2016) indeed compared IASI vertical profiles and

results from a transport model, although working on monthly means only, so leaving the validity of small scale day-to-day data in the UTLS unsupported. Averaging over long time periods or large regions, the resulting profiles are very smooth and agreement with low information content retrievals can be very good. But is this residing on the apriori climatologies? Instead, you intend to use the nadir data under variable conditions and unknown vertical gradients. Conditions of strong vertical variability may be completely missed by nadir retrievals: see e.g. Gazeaux et al. (2013) AMT where IASI O3 profiles completely miss to reproduce the plumes at various tropospheric altitude observed by ozonesondes over Antarctica. How do these limitations affect your analysis?

The references cited in the data description meant to give the background for this work. The current study does not depend on the previous validation studies, but rather, is motivated by the limited information from validating though direct comparisons. The approach of the process-based evaluation of retrieval information is to complement the direct comparisons between the datasets.

For the ozone analyses, we are not expecting the OMI data to show multiple tropospheric layers, rather, we expect the data to recognize the stratosphere and the troposphere, which is fully supported from the averaging kernels. The result is positive, as shown in the figures in the previous submission. As a result, we have demonstrated the effectiveness of process-based evaluation.

P3 L19-P4 L17: are there detection deficiencies expected for the three sensors in the UTLS region? E.g., how is IASI affected by clouds? Could you also help the reader understanding how the numbers you give as degrees of freedom translate in how much information can be extracted from the observations? To what extent are 0.8-2.4 DOFS in IASI CO sufficient "to capture upper tropospheric variability" as you write? If you can retrieve 1 to 2 independent partial columns, help the reader to understand that you expect only 1 or 2 independent points in your profiles which will then be used to scale the climatological profile.

In the revision, we discussed the effect of cloud. In IASI retrieval, cloud coverage greater than 25% are not retrieved, which leave significant data gap in the monsoon region.

See point 1) in Overview of Revision for the question of "to what extent the IASI CO capture the UT variability".

P3 L19 –P4 L17: At what local time are the observations taken? Are the fixed local times of the measurements affecting your analysis?

The dynamical variability we are targeting are synoptic scale features and are not affected by the observation time per se.

P3 L23: Could you use the horizontal resolution to predict at what scale you expect MLS data to lose sensitivity to finer scales as compared to nadir observations? Is this reflected in your results (e.g. the 100 km features seen in IASI CO daily maps but missed by MLS)?

Both instruments have their challenges with regarding the data gap. In the revised manuscript, we will provide more discussion on the importance of carefully interpolating the data.

P3 L25: how are these and following interpolations affecting your study? E.g., aver- ages and interpolation tend to add further smoothing errors in the profiles therefore reducing the vertical resolution. How are the vertical profiles changing depending on the horizontal interpolation you adopt? One may expect the agreement between nadir and limb data to improve reducing resolution (and therefore variability).

Specific discussions on interpolation and smoothing are discussed in the revision. In the updated figure, both MLS and IASI are interpolated using the same method (Natural Neighbor method) and followed by 1-sigma Gaussian smoothing. For the 3D structure we again focus on the variability signature not the absolution value of enhancement.

P3 L28: please carefully discuss the vertical resolution for IASI CO (and OMI O3) as this is a key element in your analysis.

We agree. See point 1) of Overview of revision.

P3 L29: the use of a single a priori profile could lead to a bias in the average you perform and could be removed from the final result to reduce its influence.

This is an interesting subject to discuss with the retrieval team.

P4 L1-4: see comments on supporting references above.

P4 L5-17: see comments on supporting references above. L11: you should clearly state that the DOFS in the troposphere are 0-1.5 and that the troposphere is then observed as one single column (Liu et al. 2010b). L14: the fact that there is useful information does not mean you can control where that is coming from (e.g., what about contamination by photochemical production?) Note that for example the averaging kernels in Liu et al. 2010b under tropospheric perturbed conditions peak at 5 km altitude in their case study in the tropics (their Fig. 5): so as long as the signal is coming from the upper troposphere, what you show is sound. But if the signal is coming from different layers then you give a wrong picture, with no ability to distinguish among the two cases. Please find support for the kind of conditions you analyze and provide sensitivity tests. Can you support via e.g. MLS that the whole signal in the region is coming from the upper troposphere and then constrain the nadir observations?

Again, for ozone study, we focus on the separate between stratosphere and troposphere, which OMI data demonstrated excellent consistency with the tropopause data.

P4 L18-23: your use of different years is quite unusual when studying simultaneously different targets since further insight can be achieved by inter-comparison. Without studying at least one year with both targets before showing what happens in a different year, the robustness of your analysis is largely reduced. CO data from IASI on MetOP- A in 2008 were used from George et al. (2015). And why was O3 from IASI not used instead of/in comparison to OMI O3? If the observed small-scale variability is not an artefact of the poor retrieval (i.e., the smoothed response to local composition of the 1 or 2 independent partial tropospheric columns you can retrieve), then the very same structures should be present both in CO and O3. Even then, you may be seeing the same retrieval artefact but it would give further support to your approach since the retrieval of the two targets would be most sensitive to different layers of the atmosphere.

Addressed in Overview of Revision (point 2)

P4 L25: Figure 1a. It would be good to focus only on the ASM region under investiga- tion, plotting the data with actual size of the nadir footprint and an indicative footprint of the smoothed horizontal area defining each MLS measurement. To complete the information, I would also show a companion plot with a vertical cross section passing through the ASM region showing the position of MLS tangent points and the 1 or 2 independent points of the nadir profiles (maybe with error bars indicating their vertical resolution).

Fig 1a is revised to show only the study region. Vertical cross section is in Fig 9r.

P4 L29: I would rephrase "vertical information distribution". Figure 1b. It would be useful to add the same figure for OMI O3 AKs (and possibly for IASI O3). Or simply discuss whether they are similar.

See Overview of Revision point 1) and 2)

P4 L30: explain whether IASI 12-16 km partial columns are a standard IASI product or you calculate them to match the anticyclone vertical range. Could you compare them to partial columns from MLS too?

Revised to analyze IASI CO data at 150 hPa level. See Overview.

Comparison

P5 L12-14: could you show the vertical extent of the anticyclone in one of your plots (e.g. in the additional panel of Figure 1a I suggested, or in Figure 7)?

Shown in Fig. 9r

P5 L23: it would be helpful to add a conversion to km or hPa for the various levels you adopt (e.g. is 147 hPa for MLS CO about 14 km and layer 18 about 16-17 km?). Have you tried repeating the analysis at different levels? Does it remain consistent?

147 hPa is approximately 14 km and layer 18 is approximately 100 hPa are stated in the text.

P5 L24: see comments above, why not choosing the same year for both targets and prevent a self-consistency comparison?

P5 L28 – P6 L9: the analysis on JJA averages show quite large differences in CO and a much better agreement in O3. Even for O3, the wave pattern at 0/30N is not visible in OMI data. Why? I would expect finer scales to be resolved by nadir? Is that at a layer outside the nadir sensitivity? Can you give a more in-depth interpretation of how to read these results based on the low DOFS nadir data have? P6 L10 . . .:

We will enhance the discussions of the comparison, but the goal of the quantitative comparison in this work is mainly to provide a baseline to look at the variabilities together.

The scatter plots of Fig 3 are very useful: here clearly the correlation of O3 is very good and the sensitivity to CO is weak with likely contamination from other layers. I recall these are 3-month averages: have you performed a similar comparison on shorter time scales? Can you compare the agreement/standard deviation you find to e.g. what is an accepted comparison in validation studies from the literature? Since these are comparisons of 3-month averages and not coincident profiles, it seems

to me that the sentence P5 L17 states more confidence than what is shown by the data.

Again, the quantitative comparisons are made to characterize the differences between the two types of data and to have a baseline to look at the variability together. The scatter plots are daily binned points, i.e. the average of the data that represent the same region and same day, to the extent of different viewing angles allow. These are all points for the three months but it is not a three month average.

Sub-seasonal variability

P7 L19-20, L22-23, L28-32, Fig 5/6: IASI CO is showing enhanced values within the ASM region but also scattered features outside. Why would MLS not pick for example the feature of enhanced CO at 90E and 10S/0 if it were around the 147 hPa layer? How can you distinguish cases with high CO in the upper troposphere from cases with high CO in the middle troposphere (see the high levels of CO in e.g. Fig 7)? The different evolution of MLS CO and IASI CO over the days points toward signals coming from different layers. P8 L1: I think you are underestimating the shortages of this analysis and should be more careful with this statement. There is some sensitivity to UT CO but you cannot distinguish it from sensitivity to the rest of the troposphere. Fig 7 would be more useful if compared to MLS data (in the upper range) and to a model simulation to understand the vertical structures and compare it to that reproduced by the 2 DOFS of IASI CO.

These figures are all updated in the revision

P8 L11-L21, Fig 8: I find it difficult to read the figure as reported in the text. Could you over plot the zero-anomaly contour of one on the other? Or the GPH shape? Could you mark the trajectory of the moving local maxima to highlight the propagation of anomalies? I do not see a convincing agreement. I see it clearly in the model by Pan et al (2016). Again, with sensitivity studies you would have more support not to speculate in L19-21. Please quantify what is coming from the UT.

Revised to include zero anomaly. See Fig. 6r in the Overview for the flavor of new figures.

P8 L22. . ., Fig 9: the regional timeseries look more robust, which seems to me sup- porting the fact that when you average more data together you remove variability and can find a better agreement. But this is not supporting your attempt to use nadir data to investigate finer scales. Can you help extracting more information from the timeseries? What parts are in robust agreement and why? What happens if you over plot timeseries for the Tibetan and Iranian lobes? What if you over plot timeseries at different altitude?

These figures are changed in the revision.

P8 L32-34: it would be more convincing if the agreement with Pan et al. (2016) was shown adding their data to the figures. Please rephrase "able to detect the impacts of vertical transport".

See Fig 6r in Overview – the dynamical consistency in satellite data based Hovmoller is comparable but better than the model result.

P9 L13-L21, Fig 10/11: there are large differences that need to be investigated with sensitivity tests. Are they due to different layers being involved? What is coming from measurement of the UT and what is contamination? You need to quantify this to identify what you can extract from the data you show.

The main difference in the structure is from sampling differences and smoothing.

P9 L22-L29, Fig 12: see comments for Fig 8. How can you get different frequencies for the migrating anomalies among the three plots?

Sampling density differences will contribute to the differences in apparent frequency. The figure is revised.

P9 L30: I am not sure how you attribute the differences to the sampling densities rather than (lack of) sensitivity to different layers.

Comment on the vertical sensitivity will be added in revision.

P10 L1-2: Can you investigate more these timeseries and identify when and why they correlate and when not? The reason of the weak correlation needs to be better ex- pressed since to me these timeseries should give the strongest support to your analy- sis: if they fail to correlate, can you trust the nadir data in the way you adopted them?

The time series figures are no long in the revised paper.

Conclusions

P10 L6- P11 L9: I think the conclusions should better reflect the limitations of the adopted nadir data and the caution needed to deal with them under these circum- stances.

See overview of revision for discussion.

FIGURES

General: for some reason, the figures at actual size on screen (especially labels) show poor quality, whereas they have very good quality when enlarged. Could you tune this?

Revised

Fig 1a: I would limit the map to the ASM region. Please add a panel with vertical snapshot – see text. Blue/red is very hard to read. Could you have the symbols with the actual size of the footprints (horizontal resolution for MLS) when limiting the map to the ASM region?

Revised

Fig 1b: increase legend font size. Possibly place the zero aligned. Why are the num- bers so different? I would like to see also the same plots for O3.

Revised.  See point 2) in overview

Fig 2: it is a bit confusing to have a different color scale for the two.  Could you add  a remark in the text? Remove "(GPH)" or "geopotential height" from caption as it was already explained. Please mention first "a,b", then "c,d" in the caption.

Fig 3: could you provide some significance test? Could you add also a comparison of the vertical profiles with their standard deviations too?

**Following figures are all revised.**

Fig 5 and 6: please join the two figures to allow easier comparison. Fonts at actual size are not readable. Why do you fill missing data? Please let them visible.

Fig 7: could you add an approximate vertical scale in km too since you use both in your text? Could you add the vertical and horizontal extent of the anticyclone? You appreciably pay attention to the horizontal interpolation for MLS data, but to my under- standing this plot is produced with only 2 independent points on the vertical. Is this correct? If so, could you add a comment/a sign for this?

Fig 8: could you help the reader over plotting contour of one zero anomaly line on the other? Or of the GPH? In a similar fashion as in Pan et al. (2016).

Fig 10-11: please join the two figures and see comments above. Fig 12: consider over plotting reference contours as in Fig 8.

**Relevant technical corrections are made in revised version.**

TECHNICAL CORRECTIONS

General: please note that the use of the article "the" is recurrently not consistent throughout the manuscript. E.g., P1 L17: "the ASM UTLS trace gas", P1 L21 "of ASM anticyclone", P1 L25 "of the ASM anticyclone", etc.

P1L13-14: since you first introduce MLS, I would link to it in the following sentence where you state you work on IASI and OMI and avoid the reader to wait to know why MLS was introduced: e.g., P1 L15 ". . . IASI and OMI, IN COMPARISON TO THE MLS LIMB SOUNDER."

P1 L14: "these type" – > "this type"

P1 L15: "IASI" and "OMI" were not introduced before and needs to be explained

P1 L16: remove "in the UTLS" after "(O3)" as it is repeated at the end of the sentence. P1 L17: "variability is" – >
"variability are"

P1 L18: remove comma after "explored" P1 L19: – > "results show"

P1 L26: remove "(GPH)" P2 L4: ";" – > ":"

P2 L12: – > "of the Tibetan mode"

P2 L15: "Asian summer monsoon" – > "ASM" P2 L18: – > "chemical impact."

P2 L21: – > "are widely used for this purpose"?

P2 L25: this is slightly confusing since in the title you mention "Limb and Nadir…" (limb first), and here you introduce your work stating you work only on nadir data

P2 L26: "Two specific dataset we explore" seems as the reader should expect further nadir data to be used in the study

P2 L28: 2-3 months at what altitude/which layer? P2 L29: "long lifetime" at what altitude/which layer?

P2 L30: "pollution sources" with negligible or not negligible impact? P2 L31: "will be examined" – > "were examined"

P2 L32: remove "the" in front of "MLS", this is the first time you introduce them P3 L1: remove "the" in front of "UTLS levels"

P3 L15: – > "also aims to"

P4 L17: "which" – > "whose", "is greater" – > "are greater" P5 L19: "pressure" – > "vertical"

P6 L24: "10-20 day" – > "a 10-20 day"

P7 L9: – > "(Aug 18, when the center of the …" P7 L19: – > "Compared to MLS…"

P8 L30: – > "by differenCES over. . ."

P12 L15: George et al reference is wrong

P13 L14: Liu et al misses page/volume number. P13 L 30: McPeters et al. misses page/volume number. Please check other references have the same problem.

---

## Author Comment (AC4) · 5 Oct 2017

**Response to reviewer 3**

Laura Pan on behalf of all co-authors

We thank the reviewer for many suggestions. Please see the overall response to reviewers, summarized in the Overview of Revisions.

**Point-by-point responses**

General comment:

The main objective of the paper is to demonstrate the ability of nadir viewing sensors to document the sub-seasonal variability of CO (IASI) and O3 (OMI) in the UTLS dur- ing the Asian Summer Monsoon (ASM) and their relationship with the dynamics of the Asian Monsoon Anticyclone (AMA). Since more than a decade this subject has been widely studied and documented with various satellite sensors and models. The goal of the present paper is not to bring new insight about the dynamical processes that control CO and O3 in the UTLS during the ASM. It mostly aims at demonstrating the capability of IASI to document UTLS CO. As detailed below, this demonstration is not fully convincing. Finally, the complementarity of CO and O3 data to document the AMA dynamics is not put forward because data of different years are used for both sensors. Three questions are proposed in the introduction (P2L33-34 P3L1-3) but no clear, positive and thorough answers are given in the paper as discussed in this review. For these reasons, I think this paper is not publishable in ACP and that major revisions and improvements are needed before the paper is re-submitted. As the paper focuses on satellite data capability in the UTLS, the authors should strengthen their demonstration and validation of IASI UTLS CO and the paper would rather be submitted to AMT than to ACP. Once or if IASI is proven to be able to document UTLS CO on a sub-seasonal scale in the AMA, they should take advantage of the complementarity of CO and O3 data from nadir sensors and of the long records to bring new insights in the ASM and AMA science.

These general concerns are addressed in the Overview of Revisions.

Detailed comments:

IASI UTLS COĂă : The largest part of the paper is dedicated to demonstrate that IASI is able to document the day to day variability of CO in the UTLS during the ASM over Asia.

-1/ The first part of the demonstration is based on literature. According to George et al. (2009) the IASI retrievals contain 0.8 to 2.4 independent element of information depending probably on the location. More details about the theoretical independent layers (when DFS > 1.5) and a focus on the region of interest (AMA) should be provided. Are there about 2 elements of information consistently above the AMA region or just about one ? The paper of De Wachter et al. (2012) is cited as a validation of IASI UTLS CO. Nevertheless, this paper compares IASI and MOZAIC columns for the 470- 250 hPa range which is not the UTLS. George et al. (2009) compares MOPITT and IASI CO tropospheric columns but not specifically UTLS columns. Barret et al. (2016) comparisons of IASI and model distributions concerns another IASI retrieval product.

These references are not cited to demonstrate IASI's capability for ASM study but to provide the context and to acknowledge the prior studies using IASI CO data. Our goal is to demonstrate the information by the dynamical consistency, which complements the theoretical study of information content, i.e. DOFS. See Point 1) in the Overview.

-2/ The second part is based on the retrieval characterization with the averaging kernels presented for the AMA region in Figure 1b. The 0-12 km kernel shows the largest sensitivity (HWHM) with values exceeding 0.7 between 2 and 13 km. IASI is therefore clearly sensitive in the mid-upper troposphere. The 12-16 km kernel displays a very low sensitivity over the whole troposphere and UTLS with a weak maximum with low values (below 0.25!) above 10 km. Furthermore, the values of the 12-16 km AvK are much lower than the values of the 0-12 km over the UTLS altitude range. This means that (i) the sensitivity to the UTLS is very low (ii) the two kernels are not independent with clear resolved maxima (iii) more information about the 12-16 km is contained in the 0-12 km retrieved column than in the 12-16 km column. From this characterization it is therefore not clearly proven that IASI is able to provide independent information about UTLS CO. The AvK is averaged over a large region with very different surfaces : desert, high mountains, low lands, oceans... As the AvKs depend on surface proper- ties, there is probably no homogeneity in the vertical sensitivity over the whole region and more details should be provided rather than refering to the general "averaging kernels discussion" from George et al. (2009).

This section, as part of data description, is also provided to set the background for the analyses. In the revision, we have provided the averaging kernels for the study domain. The goal of this work, however, is not a more detailed theoretical analysis of the information content, but rather to demonstrate the effect of the information content by looking at the performance of the data, i.e., a "process-based retrieval evaluation". This approach was not well demonstrated in the previous submission, but we are meeting the objective in this revision. See figs. 7r and 9r in the Overview of Revisions.

- 3/ The third and largest part of the demonstration is based on comparisons/validation with MLS UTLS CO. - The MLS daily data are interpolated to provide a global distribution. This methodology is questionable with so sparse data (space of about $15°$ longitude or 1600 km between observations in the tropics!). This can be seen on Figure 4 with plumes of high CO over the Bay of Bengal or the north western Pacific that are not related to real observations. A better methodology should be to average over 5 to 7 days as is normally done with MLS data and to compare with IASI aver- ages or to use data assimilation. - Concerning the averaged seasonal distributions, there is a relative agreement between IASI and MLS CO. Nevertheless looking in more details, there are large discrepancies which are not so clear because of the large do- main displayed. The inside of the AMA is characterized by homogeneous high CO with MLS while lower CO appears to the north (north of $30°$) and to the east of the AMA with IASI. Furthermore, the latitudinal gradient of CO between the tropical UT (high CO) and extratropical LS (lower CO) detected by MLS is not detected by IASI. It is especially clear for the southern tropical Indian Ocean and Pacific around 30S. These discrepan- cies are probably due to the fact that IASI detects the highest UTLS enhancements of CO that coincide

with the deepest convection in the monsoon region from the Arabian sea to south-east Asia. It seems that IASI also detects UTLS enhancements over the African monsoon because of strong convective uplift of BB plumes. It could be inter esting to show the whole African monsoon region to confirm this ability and strengthen the demonstration as it is the aim of the paper. The border of the AMA is shown with a very thick white line that partly hide the CO distribution. - - - An important proof of the ability of IASI to detect high CO in the UTLS independently from the lower-mid tropo- sphere would be to show the tropospheric columns. In the eastern and central part of the AMA (ASM region) high tropospheric CO due to large emissions should be present contrarily to the western and northern part. The latitude-pressure section displayed in Fig. 7 rather shows that the information is mixed over the whole troposphere and UTLS (see comment below). - The correlation plot between IASI and MLS CO also shows some consistency between the two datasets but the correlation coefficient is rather low ($r2 = 0.52 \ll 1$). As mentioned, this low r2 and the reduced slope indicates a rather low sensitivity of IASI to UTLS CO. Whether it is due to the use of a single a priori profile is not obvious and has not been demonstrated. Without specific sensitivity tests using other a priori profiles, the statement p6l14-15 should be removed or modified. - Finally, daily MLS and IASI CO distributions corresponding to different dynamical situations of the AMA are compared. It is mentioned that with IASI "the spatial distributions show many finer scale structures" than with MLS. Given the low UTLS sensitivity and possible mixing of UTLS and tropospheric information as discussed above, it is difficult to know whether these finer scale structures are really in the UTLS. FurthermoreÂă: (i)- we see no holes in the IASI daily distributions in the ASM region impacted by very large and deep clouds. This means that the IASI observations have been interpolated and that part of the finer scale structures could be interpolation artifacts such as those discussed for MLS. It would be much better to see raw IASI data without interpolation.

Otherwise, averages over a few days would be much better than absolutely trying to show daily distributions with spurious features. (ii)- many of those finer scale structures are not detected by MLS such a CO bubbles in the south eastern Indian Ocean on 18 and 20 August, or over the north western Pacific on 20 and 24 August... over equa- torial Africa, some high CO spots are detected by IASI and not by MLS on 18, 20, 24 August. They may correspond to BB plumes in the middle troposphere but this has to be discussed thorough-fully. (iii)- the westward elongation of the AMA highlighted by the GPH field (west of 30˚E) is very clear and consistent with MLS but not detected by IASI on 24 and 26 of August. This is again a probable evidence that IASI is not sensitive enough to detect UTLS CO enhancements that correspond to low integrated amounts outside of the convective and polluted regions.

We agree with the reviewer that the issue of interpolation to fill the orbital gaps and missing data in single-day maps was not carefully discussed in the previous submission. In the revision, we have dedicated a figure and a paragraph of specific discussions on the topic. We have tried various multi-day running mean etc. Eventually we found that a careful interpolation of data from each day can be an effective use of the data for analyzing the day-to-day dynamical consistency.

Fig. 6r in the Overview of Revisions provides the Hovmoller diagram for the JJA 2008, where MLS 147 hPa CO and GPH showed excellent correlation.

The answer to point 2 and 3 found in the paper is given P8L19-21, starting with "we speculate...". This is probably the answer of the discrepancies but (i) if true it proves that IASI is not really able to detect CO in the UTLS independently from the mid troposphere and therefore that the product is not so valid to characterize the AMA com- position and dynamics at the sub-seasonal scale (ii) as this is somehow the central point of the paper, the demonstration should be more thorough and based on comparisons between 0-12 km and 12-16 km distributions (as mentioned above) and a latitude pressure sections at a location west of 90E and/or a longitude-pressure section at the center of the AMA to show wether the east-west lower tropospheric and UTLS gradients are different.

This is addressed in the revision. See Overview point 1).

The vertical structure of IASI CO over the ASM region is finally displayed on Figure7. As discussed in the paper, the uplift of CO within the monsoon region up to 100 hPa is "clear" around 20°N. Nevertheless this latitude-pressure section needs some explanations aboutÂă : (i) the CO concentrations is homogeneously high over the whole troposphere and UTLS in the convective region. This could be explained by the color scale which saturates at 100 ppbv (in that case the authors should change this color scale to show the detailed vertical structure) or by the fact that the information provided by IASI is not sufficient to separate the lower tropospheric pollution from the convective outflow which seem to be mixed together. Indeed, convection detrains preferentially at high altitude (above 200 hPa) and convective enhancement should not be connected to lower tropospheric enhanced CO. (ii) We also see large CO vmr north of the Himalaya range from the surface to 200 hPa. This region (Tibetan plateau) is not impacted  by important emissions nor by deep convection. How can such elevated CO reaching such high altitudes be explainedÂă ? (iii) The same is true between 10S and 10N below 300 hPa. Which emissions and transport processes are responsible for these high CO concentrations over the Indian Ocean up to 300 hPa ?

We have provided 4 vertical cross sections and their corresponding maps in revised figures. See Figs 7r and 9r in the Overview.

The Hovmoller diagram of Fig. 8 corroborates the low sensitivity of IASI to UTLS CO enhancements in the western part of the AMA and the detection of features of high CO on the eastern part that are not detected by MLS and not consistent with GPH anomalies. This is rather problematic to state that IASI is able to detect CO in the UTLS.

We have revised this figure. In the revised analysis, the two sensors' dynamical consistency are demonstrated separately. We also make a point that the dynamical consistency is much better demonstrated using carefully interpolated data, instead of coarsely binned and smoothed data. One of the new Hovmoller diagram is included in the Overview.

OMI UTLS O3:  As already mentioned, the choice of 2008 for OMI and 2012 for IASI raises the usefulness of presenting/using both datasets. The explanation of that choice is not fully convincing. IASI 2008 data are available and it is not clear that the results for a single monsoon season may be impacted by jumps induced by changing L2 data (when does this change occurs?). It is mentioned that "O3 and CO data are examined separately" which is a real weakness of the paper because (i) if the

aim was to characterize the relationship between UTLS composition and AMA dynamics, the complementarity of both gases as pollution and stratospheric tracers would have been a strength (ii) the OMI part is much shorter than the IASI and comes to the conclusion that OMI O3 documents correctly the UTLS which has been shown elsewhere. There- fore, the OMI O3 part could be removed from the paper which should concentrate in proving the IASI CO UTLS capability. A future paper could focus on the dynamics- composition relationship using the O3/CO complementarity.

See Point 2) in the Overview of Revisions. It is true that the OMI part is shorter, but the data demonstrated nicely the ozone at 100 hPa is strongly correlated with the ASM tropopause structure.

IASI is also providing information about the O3 vertical distribution and should logically ave been used preferentially for obvious coincidence criterion with IASI CO. Why is IASI O3 not used or even mentioned in the paper?

At the time when this work was performed there was still a bias issue with the IASI product (eg see discussion in Boynard et al. 2016), and hence we used the OMI data that we had available. The bias is now corrected in the current version of the IASI dataset and we will look into using the IASI ozone in future work.

---

## Author Response (AR1)

Note to reviewers and editors:

1) The point-by-point response to all three reviews have been uploaded on 23 July 2017. We will not duplicate them here.

2) Due to the large amount of changes in the revision, the marked change file is not very meaningful or helpful, especially in PDF. We nevertheless provide the PDF here for the requirement.

**Style Definition:** Normal: Font:(Asian) +Theme Body Asian (宋体), 10 pt, (Asian) Chinese (PRC), Widow/Orphan control, Adjust space between Latin and Asian text, Adjust space between Asian text and numbers

**Style Definition:** Heading 1: Font:(Asian) Times New Roman, 10 pt, Bold, Font color: Black, English (UK), Kern at 16 pt, Justified, Space Before: 24 pt, After: 12 pt, Outline numbered + Level: 1 + Numbering Style: 1, 2, 3, ... + Start at: 1 + Alignment: Left + Aligned at: 0" + Indent at: 0.3", Keep with next

**Style Definition:** Heading 2: Font:(Asian) Times New Roman, English (UK), Space Before: 12 pt, After: 12 pt, Outline numbered + Level: 2 + Numbering Style: 1, 2, 3, ... + Start at: 1 + Alignment: Left + Aligned at: 0" + Indent at: 0.4", Keep with next

**Style Definition:** Heading 3: Outline numbered + Level: 3 + Numbering Style: 1, 2, 3, ... + Start at: 1 + Alignment: Left + Aligned at: 0" + Indent at: 0.5", No widow/orphan control

**Style Definition:** Heading 4: Outline numbered + Level: 4 + Numbering Style: 1, 2, 3, ... + Start at: 1 + Alignment: Left + Aligned at: 0" + Indent at: 0.6"

**Style Definition:** Heading 5: Outline numbered + Level: 5 + Numbering Style: 1, 2, 3, ... + Start at: 1 + Alignment: Left + Aligned at: 0" + Indent at: 0.7"

**Style Definition:** Heading 6: Outline numbered + Level: 6 + Numbering Style: 1, 2, 3, ... + Start at: 1 + Alignment: Left + Aligned at: 0" + Indent at: 0.8"

**Style Definition:** Heading 7: Outline numbered + Level: 7 + Numbering Style: 1, 2, 3, ... + Start at: 1 + Alignment: Left + Aligned at: 0" + Indent at: 0.9"

**Style Definition:** Heading 8: Outline numbered + Level: 8 + Numbering Style: 1, 2, 3, ... + Start at: 1 + Alignment: Left + Aligned at: 0" + Indent at: 1"

**Style Definition:** Heading 9: Outline numbered + Level: 9 + Numbering Style: 1, 2, 3, ... + Start at: 1 + Alignment: Left + Aligned at: 0" + Indent at: 1.1"

**Style Definition:** List Paragraph: Font:(Asian) SimSun, 10.5 pt, Indent: Left: 0.5", First line: 0", Don't add space between paragraphs of the same style, No widow/orphan control

**Style Definition:** Footer

**Style Definition:** Caption: Font:(Asian) Times New Roman

**Style Definition:** Balloon Text: Font:(Asian) +Theme Body Asian (宋体)

**Style Definition:** Header: Font:(Asian) +Theme Body Asian (宋体)

**Style Definition:** Authors: Font:(Asian) Times New Roman

**Style Definition:** Affiliation: Font:(Asian) Times New Roman

**Style Definition:** Comment Text: No widow/orphan control

**Style Definition:** Revision

[Figure]

**Space-Time Variability of UTLS Chemical Distribution in the Asian Summer Monsoon Viewed by Limb and Nadir Satellite Sensors**

Jiali Luo[1,2], Laura L. Pan[2], Shawn B. Honomichl[2], John W. Bergman[2,3], William J. Randel[2], Gene Francis[2], Cathy Clerbaux[4], Maya George[4], Xiong Liu[5] and Wenshou Tian[1]

[1]Key Laboratory of Semi-Arid Climate Change and College of Atmospheric Sciences, Lanzhou University, Lanzhou, China

[2]National Center for Atmospheric Research, Boulder, Colorado, USA

[3]Bay Area Environmental Research Institute, Sonoma, California, USA

[4]LATMOS/IPSL, UPMC Université Paris 06 Sorbonne Universités, UVSQ, CNRS, Paris, France

[5]Harvard-Smithsonian Center for Astrophysics, Cambridge, Massachusetts, USA

*Correspondence to:* Laura Pan (liwen@ucar.edu)

**Abstract.** The Asian Summer Monsoon (ASM) creates a hemispheric scale signature in trace gas distributions in the upper troposphere and lower stratosphere (UTLS). Data from satellite retrievals are the best source of information for characterizing these large-scale signatures. Measurements from the Microwave Limb Sounder (MLS), a limb viewing satellite sensor, have been the most widely used retrieval products for these types of studies. This work explores the information for the ASM influence on UTLS chemical distribution from two nadir-viewing sensors, the Infrared Atmospheric Sounding Interferometer (IASI) and the Ozone Monitoring Instrument (OMI), together with the MLS. Day-to-day changes in carbon monoxide (CO) and ozone ($O_3$) tracer distributions in response to dynamical variability are examined, to assess how well the data from different sensors provide useful information for studying the impact of sub-seasonal scale dynamics on chemical fields. Our result, using June-July-August of 2008 data, shows that although the MLS provides relatively sparse horizontal sampling on daily timescales, interpolated daily CO distributions show a high degree of dynamical consistency with the synoptic scale structure and variability of the anticyclone. Our analysis also shows that the IASI CO retrieval has sufficient sensitivity to produce upper tropospheric (UT) CO with variabilities independent from the lower to middle tropospheric CO. The consistency of IASI CO field with the synoptic scale anticyclone dynamical variability demonstrates that the IASI UT CO product is a physically meaningful dataset. Furthermore, IASI CO vertical cross-sections combined with the daily maps provide the first observation-based evidence for a model analyses-based hypothesis on the preferred ASM vertical transport location and the subsequent horizontal re-distribution via east-west eddy shedding. Similarly, the OMI $O_3$ profile product is shown to be capable of distinguishing the tropospheric dominated air mass in the anticyclone from the stratospheric dominated background on a daily time scale, providing consistent and complementary information to the MLS. These results not only highlight the complementary information between

[revised manuscript text omitted]

**Formatted** ... [299]

**Formatted** ... [301]

**Formatted** ... [302]

**Formatted** ... [303]

**Formatted** ... [305]

**Formatted** ... [306]

**Formatted** ... [308]

**Formatted** ... [309]

**Formatted** ... [310]

**Formatted** ... [311]

**Formatted** ... [312]

**Formatted** ... [313]

**Formatted** ... [314]

**Formatted** ... [315]

**Formatted** ... [316]

**Formatted** ... [317]

**Formatted** ... [318]

**Formatted** ... [319]

**Formatted** ... [320]

**Formatted** ... [321]

**Formatted** ... [322]

**Formatted** ... [323]

**Formatted** ... [324]

**Formatted** ... [325]

**Formatted** ... [326]

**Formatted** ... [327]

**Formatted** ... [328]

... [329]

to the $CO_2$ scatterplot in Fig. 3, but the correlation between the OMI and MLS $O_3$ is much better with both the slope (0.94) and the correlation coefficient (0.96) near unity.

Figures 11 and 12 characterize the good overall agreement between OMI and MLS $O_3$ on seasonal and ASM regional scales. We now proceed to examine the daily and sub-seasonal variability represented by the two datasets.

**6.2 Representation of sub-seasonal scale variability from MLS and OMI $O_3$**

Figure 13 shows maps of MLS and OMI $O_3$ mixing ratios at 100 hPa and the tropopause pressure for two selected days in July 2008. Dynamical fields of the GPH and horizontal wind are superimposed on the $O_3$ maps. The 105 hPa tropopause contour is included in all maps. Both sets of $O_3$ maps exhibit the characteristic low $O_3$ mixing ratios inside the anticyclone. Here the 105 hPa tropopause contour appears to correlate well with the $O_3$ and wind field gradients. Note that the tropopause pressure here is from the GFS final analysis product, which is based on the WMO thermal tropopause definition. Since this quantity is derived from the vertical gradient and is not analysed on the pressure surface, it's interception with the pressure surface can appear noisy. Gaussian smoothing is applied to the 1 x 1 degree maps.

In the two selected days, the dynamical structures of the anticyclone are in two different phases as discussed in relation to Figs. 5 and 8. The ASM influence at the tropopause level shows a wider longitudinal range on the 18[th] (approximately 20°–130° E), and it is westward migrated on the 22[nd] (approximately 10°–110° E) and with a double-centered structure. The OMI $O_3$ map on 18[th] shows a close correspondence with the longitudinal range of the tropopause pressure, while the MLS map shows a westward shift of the low $O_3$ area. The difference in horizontal sampling density is likely a contributor. On 22[nd], both MLS and OMI $O_3$ gradients are well co-located with the anticyclone boundary as indicated by the 105 hPa tropopause contour. The MLS $O_3$ structure shows a more well-defined double-centered structure. OMI map shows a smaller $O_3$ depression over the Tibetan plateau. We speculate that surface elevation may have contributed to the structure, similar to the IASI CO discussion. The high ozone band on the southern side of the anticyclone shows a large difference between MLS and OMI, with MLS having a much wider structure. Both the coarser horizontal sampling of MLS and the weaker vertical resolution for this potentially shallow layer in OMI may contribute to this difference.

The Hovmöller diagrams in Fig. 14 examine sub-seasonal variations and the relationship between the tropopause pressure and 100 hPa $O_3$ field during JJA season of 2008. All three fields in the figure are dominated by the persistent location of the anticyclone as indicated by the lower tropopause pressure and of $O_3$ mixing ratios between 30°E and 100°E. All three Hovmöller diagrams exhibit westward propagation in 10-20 day timescales. The correlation in the variability along the longitudinal dimension is 0.90 between the tropopause pressure and MLS $O_3$, and 0.76 between the tropopause pressure and OMI $O_3$. In both cases, the interpolated fields are used to calculate the correlations. The strong correlation between the tropopause structure and $O_3$ supports the conceptual model that the higher tropopause over the ASM forms a region of tropospheric "bubble" above the mean level of tropical tropopause for the season. This structure enables a unique transport pathway of horizontal eddy shedding of air mass in the "bubble" to the lower stratosphere, bypassing the equatorial tropical tropopause (e.g., Garny and Randel,

2016; Ploeger et al., 2017). The air masses in the "bubble" is expected to contain convective lofted air masses from the polluted boundary layer and with high water vapour content.

While the two $O_3$ datasets provide generally consistent large scale ozone structure, there are visible differences between MLS and OMI in small-scale structures. Potential impacts of clouds on retrievals at 100 hPa is discussed in a recent OMI validation study (Huang et al., 2017). The weaker $O_3$ depression near 90°E is likely contributed by the impact of surface elevation on retrieval. A better understanding of the small-scale structures can benefit from validation studies using airborne measurements targeting the ASM UTLS structure.

**7 Conclusions and discussions**

We have examined space-time variability of chemical tracers in the UTLS associated with the ASM represented by nadir viewing (IASI and OMI) satellite instruments in comparison with a widely used limb viewing (MLS) dataset. Using CO (a boundary layer pollution tracer) and $O_3$ (a stratospheric tracer), we focus on the strengths and limitations of these data for representing the distribution and variability of UTLS chemical tracers in the region of the dynamically variable ASM anticyclone. We explore whether the much denser horizontal samplings of the nadir sensors provide information complementary to the limb data for the tracer daily distribution in response to synoptic scale variability.

[revised manuscript text omitted]

) at 100 hPa from June 1 to August 31 2008, averaged over 60-90° E and 15-35° N (i.e., the 60-90° average of values in Fig. 12). Dash lines represent mean values of each variable in the period. The data are smoothed using a 3 day running mean.

) at 100 hPa from June 1 to August 31 2008, averaged over 60-90° E and 15-35° N (i.e., the 60-90° average of values in Fig. 12). Dash lines represent mean valu... [605]

| age 2: [1] Deleted | Microsoft Office User | 11/29/17 11:26:00 AM |

| age 2: [2] Formatted | Microsoft Office User | 11/29/17 11:26:00 AM |

ndent: Left:  0", Space Before:  18 pt, Don't add space between paragraphs of the same style, Line spacing:  1.5 lines

| age 2: [3] Formatted | Microsoft Office User | 11/29/17 11:26:00 AM |

ont color: Text 1, English (UK)

| age 2: [4] Formatted | Microsoft Office User | 11/29/17 11:26:00 AM |

ont:10 pt, Font color: Text 1

| age 2: [5] Formatted | Microsoft Office User | 11/29/17 11:26:00 AM |

uthors, Indent: Left:  0", Space Before:  0 pt, Line spacing:  1.5 lines

| age 2: [6] Formatted | Microsoft Office User | 11/29/17 11:26:00 AM |

ont:10 pt, Font color: Text 1, Superscript, Not Raised by / Lowered by

| age 2: [6] Formatted | Microsoft Office User | 11/29/17 11:26:00 AM |

ont:10 pt, Font color: Text 1, Superscript, Not Raised by / Lowered by

| age 2: [6] Formatted | Microsoft Office User | 11/29/17 11:26:00 AM |

ont:10 pt, Font color: Text 1, Superscript, Not Raised by / Lowered by

| age 2: [6] Formatted | Microsoft Office User | 11/29/17 11:26:00 AM |

ont:10 pt, Font color: Text 1, Superscript, Not Raised by / Lowered by

| age 2: [6] Formatted | Microsoft Office User | 11/29/17 11:26:00 AM |

ont:10 pt, Font color: Text 1, Superscript, Not Raised by / Lowered by

| age 2: [6] Formatted | Microsoft Office User | 11/29/17 11:26:00 AM |

ont:10 pt, Font color: Text 1, Superscript, Not Raised by / Lowered by

| age 2: [6] Formatted | Microsoft Office User | 11/29/17 11:26:00 AM |

ont:10 pt, Font color: Text 1, Superscript, Not Raised by / Lowered by

| age 2: [6] Formatted | Microsoft Office User | 11/29/17 11:26:00 AM |

ont:10 pt, Font color: Text 1, Superscript, Not Raised by / Lowered by

| age 2: [6] Formatted | Microsoft Office User | 11/29/17 11:26:00 AM |

ont:10 pt, Font color: Text 1, Superscript, Not Raised by / Lowered by

| age 2: [6] Formatted | Microsoft Office User | 11/29/17 11:26:00 AM |

ont:10 pt, Font color: Text 1, Superscript, Not Raised by / Lowered by

| age 2: [6] Formatted | Microsoft Office User | 11/29/17 11:26:00 AM |

ont:10 pt, Font color: Text 1, Superscript, Not Raised by / Lowered by

| age 2: [6] Formatted | Microsoft Office User | 11/29/17 11:26:00 AM |

ont:10 pt, Font color: Text 1, Superscript, Not Raised by / Lowered by

| age 2: [6] Formatted | Microsoft Office User | 11/29/17 11:26:00 AM |

ont:10 pt, Font color: Text 1, Superscript, Not Raised by / Lowered by

| age 2: [6] Formatted | Microsoft Office User | 11/29/17 11:26:00 AM |
|---|---|---|

ont:10 pt, Font color: Text 1, Superscript, Not Raised by / Lowered by

| age 2: [6] Formatted | Microsoft Office User | 11/29/17 11:26:00 AM |
|---|---|---|

ont:10 pt, Font color: Text 1, Superscript, Not Raised by / Lowered by

| age 2: [6] Formatted | Microsoft Office User | 11/29/17 11:26:00 AM |
|---|---|---|

ont:10 pt, Font color: Text 1, Superscript, Not Raised by / Lowered by

| age 2: [6] Formatted | Microsoft Office User | 11/29/17 11:26:00 AM |
|---|---|---|

ont:10 pt, Font color: Text 1, Superscript, Not Raised by / Lowered by

| age 2: [6] Formatted | Microsoft Office User | 11/29/17 11:26:00 AM |
|---|---|---|

ont:10 pt, Font color: Text 1, Superscript, Not Raised by / Lowered by

| age 2: [6] Formatted | Microsoft Office User | 11/29/17 11:26:00 AM |
|---|---|---|

ont:10 pt, Font color: Text 1, Superscript, Not Raised by / Lowered by

| age 2: [7] Formatted | Microsoft Office User | 11/29/17 11:26:00 AM |
|---|---|---|

ffiliation, Indent: Left:  0", Space Before:  0 pt, Line spacing:  1.5 lines, Tabs:Not at  0.47"

| age 2: [8] Formatted | Microsoft Office User | 11/29/17 11:26:00 AM |
|---|---|---|

ont:10 pt, Font color: Text 1, Superscript, Not Raised by / Lowered by

| age 2: [8] Formatted | Microsoft Office User | 11/29/17 11:26:00 AM |
|---|---|---|

ont:10 pt, Font color: Text 1, Superscript, Not Raised by / Lowered by

| age 2: [8] Formatted | Microsoft Office User | 11/29/17 11:26:00 AM |
|---|---|---|

ont:10 pt, Font color: Text 1, Superscript, Not Raised by / Lowered by

| age 2: [8] Formatted | Microsoft Office User | 11/29/17 11:26:00 AM |
|---|---|---|

ont:10 pt, Font color: Text 1, Superscript, Not Raised by / Lowered by

| age 2: [8] Formatted | Microsoft Office User | 11/29/17 11:26:00 AM |
|---|---|---|

ont:10 pt, Font color: Text 1, Superscript, Not Raised by / Lowered by

| age 2: [8] Formatted | Microsoft Office User | 11/29/17 11:26:00 AM |
|---|---|---|

ont:10 pt, Font color: Text 1, Superscript, Not Raised by / Lowered by

| age 2: [8] Formatted | Microsoft Office User | 11/29/17 11:26:00 AM |
|---|---|---|

ont:10 pt, Font color: Text 1, Superscript, Not Raised by / Lowered by

| age 2: [8] Formatted | Microsoft Office User | 11/29/17 11:26:00 AM |
|---|---|---|

ont:10 pt, Font color: Text 1, Superscript, Not Raised by / Lowered by

| age 2: [8] Formatted | Microsoft Office User | 11/29/17 11:26:00 AM |
|---|---|---|

ont:10 pt, Font color: Text 1, Superscript, Not Raised by / Lowered by

| age 2: [8] Formatted | Microsoft Office User | 11/29/17 11:26:00 AM |
|---|---|---|

ont:10 pt, Font color: Text 1, Superscript, Not Raised by / Lowered by

**age 2: [8] Formatted** **Microsoft Office User** **11/29/17 11:26:00 AM**

ont:10 pt, Font color: Text 1, Superscript, Not Raised by / Lowered by

**age 2: [8] Formatted** **Microsoft Office User** **11/29/17 11:26:00 AM**

ont:10 pt, Font color: Text 1, Superscript, Not Raised by / Lowered by

**age 2: [8] Formatted** **Microsoft Office User** **11/29/17 11:26:00 AM**

ont:10 pt, Font color: Text 1, Superscript, Not Raised by / Lowered by

**age 2: [8] Formatted** **Microsoft Office User** **11/29/17 11:26:00 AM**

ont:10 pt, Font color: Text 1, Superscript, Not Raised by / Lowered by

**age 2: [8] Formatted** **Microsoft Office User** **11/29/17 11:26:00 AM**

ont:10 pt, Font color: Text 1, Superscript, Not Raised by / Lowered by

**age 2: [8] Formatted** **Microsoft Office User** **11/29/17 11:26:00 AM**

ont:10 pt, Font color: Text 1, Superscript, Not Raised by / Lowered by

**age 2: [8] Formatted** **Microsoft Office User** **11/29/17 11:26:00 AM**

ont:10 pt, Font color: Text 1, Superscript, Not Raised by / Lowered by

**age 2: [8] Formatted** **Microsoft Office User** **11/29/17 11:26:00 AM**

ont:10 pt, Font color: Text 1, Superscript, Not Raised by / Lowered by

**age 2: [8] Formatted** **Microsoft Office User** **11/29/17 11:26:00 AM**

ont:10 pt, Font color: Text 1, Superscript, Not Raised by / Lowered by

**age 2: [8] Formatted** **Microsoft Office User** **11/29/17 11:26:00 AM**

ont:10 pt, Font color: Text 1, Superscript, Not Raised by / Lowered by

**age 2: [8] Formatted** **Microsoft Office User** **11/29/17 11:26:00 AM**

ont:10 pt, Font color: Text 1, Superscript, Not Raised by / Lowered by

**age 2: [8] Formatted** **Microsoft Office User** **11/29/17 11:26:00 AM**

ont:10 pt, Font color: Text 1, Superscript, Not Raised by / Lowered by

**age 2: [8] Formatted** **Microsoft Office User** **11/29/17 11:26:00 AM**

ont:10 pt, Font color: Text 1, Superscript, Not Raised by / Lowered by

**age 2: [8] Formatted** **Microsoft Office User** **11/29/17 11:26:00 AM**

ont:10 pt, Font color: Text 1, Superscript, Not Raised by / Lowered by

**age 2: [8] Formatted** **Microsoft Office User** **11/29/17 11:26:00 AM**

ont:10 pt, Font color: Text 1, Superscript, Not Raised by / Lowered by

**age 2: [8] Formatted** **Microsoft Office User** **11/29/17 11:26:00 AM**

ont:10 pt, Font color: Text 1, Superscript, Not Raised by / Lowered by

**age 2: [9] Formatted** **Microsoft Office User** **11/29/17 11:26:00 AM**

ont:10 pt, Font color: Text 1, Superscript, Not Raised by / Lowered by

| Page 2: [10] Formatted | Microsoft Office User | 11/29/17 11:26:00 AM |
|---|---|---|

Affiliation, Line spacing: 1.5 lines

| Page 2: [11] Formatted | Microsoft Office User | 11/29/17 11:26:00 AM |
|---|---|---|

Font color: Text 1

| Page 2: [12] Formatted | Microsoft Office User | 11/29/17 11:26:00 AM |
|---|---|---|

Font:10 pt, Font color: Text 1, Superscript, Not Raised by / Lowered by

| Page 2: [12] Formatted | Microsoft Office User | 11/29/17 11:26:00 AM |
|---|---|---|

Font:10 pt, Font color: Text 1, Superscript, Not Raised by / Lowered by

| Page 2: [13] Formatted | Microsoft Office User | 11/29/17 11:26:00 AM |
|---|---|---|

Font:10 pt, Font color: Text 1, Superscript, Not Raised by / Lowered by

| Page 2: [13] Formatted | Microsoft Office User | 11/29/17 11:26:00 AM |
|---|---|---|

Font:10 pt, Font color: Text 1, Superscript, Not Raised by / Lowered by

| Page 2: [14] Formatted | Microsoft Office User | 11/29/17 11:26:00 AM |
|---|---|---|

Font color: Text 1

| Page 2: [15] Formatted | Microsoft Office User | 11/29/17 11:26:00 AM |
|---|---|---|

Font:10 pt, Font color: Text 1, Superscript, Not Raised by / Lowered by

| Page 2: [15] Formatted | Microsoft Office User | 11/29/17 11:26:00 AM |
|---|---|---|

Font:10 pt, Font color: Text 1, Superscript, Not Raised by / Lowered by

| Page 2: [16] Formatted | Microsoft Office User | 11/29/17 11:26:00 AM |
|---|---|---|

ndent: Left:  0", Space Before:  6 pt, After:  18 pt, Line spacing:  1.5 lines, Tabs:Not at  0.47"

| Page 2: [17] Formatted | Microsoft Office User | 11/29/17 11:26:00 AM |
|---|---|---|

Font color: Text 1, English (UK)

| Page 2: [18] Formatted | Microsoft Office User | 11/29/17 11:26:00 AM |
|---|---|---|

Font color: Text 1, English (UK)

| Page 2: [19] Formatted | Microsoft Office User | 11/29/17 11:26:00 AM |
|---|---|---|

Font color: Text 1, English (UK)

| Page 2: [20] Formatted | Microsoft Office User | 11/29/17 11:26:00 AM |
|---|---|---|

Font color: Text 1, English (UK)

| Page 2: [20] Formatted | Microsoft Office User | 11/29/17 11:26:00 AM |
|---|---|---|

Font color: Text 1, English (UK)

| Page 2: [20] Formatted | Microsoft Office User | 11/29/17 11:26:00 AM |
|---|---|---|

Font color: Text 1, English (UK)

| Page 2: [20] Formatted | Microsoft Office User | 11/29/17 11:26:00 AM |
|---|---|---|

Font color: Text 1, English (UK)

| Page 2: [20] Formatted | Microsoft Office User | 11/29/17 11:26:00 AM |
|---|---|---|

Font color: Text 1, English (UK)

| Page 2: [20] Formatted | Microsoft Office User | 11/29/17 11:26:00 AM |
|---|---|---|

Font color: Text 1, English (UK)

| age 2: [20] Formatted | Microsoft Office User | 11/29/17 11:26:00 AM |
|---|---|---|

ont color: Text 1, English (UK)

| age 2: [20] Formatted | Microsoft Office User | 11/29/17 11:26:00 AM |
|---|---|---|

ont color: Text 1, English (UK)

| age 2: [20] Formatted | Microsoft Office User | 11/29/17 11:26:00 AM |
|---|---|---|

ont color: Text 1, English (UK)

| age 2: [20] Formatted | Microsoft Office User | 11/29/17 11:26:00 AM |
|---|---|---|

ont color: Text 1, English (UK)

| age 2: [20] Formatted | Microsoft Office User | 11/29/17 11:26:00 AM |
|---|---|---|

ont color: Text 1, English (UK)

| age 2: [20] Formatted | Microsoft Office User | 11/29/17 11:26:00 AM |
|---|---|---|

ont color: Text 1, English (UK)

| age 2: [20] Formatted | Microsoft Office User | 11/29/17 11:26:00 AM |
|---|---|---|

ont color: Text 1, English (UK)

| age 2: [20] Formatted | Microsoft Office User | 11/29/17 11:26:00 AM |
|---|---|---|

ont color: Text 1, English (UK)

| age 2: [20] Formatted | Microsoft Office User | 11/29/17 11:26:00 AM |
|---|---|---|

ont color: Text 1, English (UK)

| age 2: [20] Formatted | Microsoft Office User | 11/29/17 11:26:00 AM |
|---|---|---|

ont color: Text 1, English (UK)

| age 2: [20] Formatted | Microsoft Office User | 11/29/17 11:26:00 AM |
|---|---|---|

ont color: Text 1, English (UK)

| age 2: [20] Formatted | Microsoft Office User | 11/29/17 11:26:00 AM |
|---|---|---|

ont color: Text 1, English (UK)

| age 2: [20] Formatted | Microsoft Office User | 11/29/17 11:26:00 AM |
|---|---|---|

ont color: Text 1, English (UK)

| age 2: [20] Formatted | Microsoft Office User | 11/29/17 11:26:00 AM |
|---|---|---|

ont color: Text 1, English (UK)

| age 2: [20] Formatted | Microsoft Office User | 11/29/17 11:26:00 AM |
|---|---|---|

ont color: Text 1, English (UK)

| age 2: [20] Formatted | Microsoft Office User | 11/29/17 11:26:00 AM |
|---|---|---|

ont color: Text 1, English (UK)

| age 2: [20] Formatted | Microsoft Office User | 11/29/17 11:26:00 AM |
|---|---|---|

ont color: Text 1, English (UK)

| age 2: [20] Formatted | Microsoft Office User | 11/29/17 11:26:00 AM |
|---|---|---|

ont color: Text 1, English (UK)

| age 2: [20] Formatted | Microsoft Office User | 11/29/17 11:26:00 AM |
|---|---|---|

ont color: Text 1, English (UK)

| age 2: [20] Formatted | Microsoft Office User | 11/29/17 11:26:00 AM |

ont color: Text 1, English (UK)

| age 2: [20] Formatted | Microsoft Office User | 11/29/17 11:26:00 AM |

ont color: Text 1, English (UK)

| age 2: [20] Formatted | Microsoft Office User | 11/29/17 11:26:00 AM |

ont color: Text 1, English (UK)

| age 2: [20] Formatted | Microsoft Office User | 11/29/17 11:26:00 AM |

ont color: Text 1, English (UK)

| age 2: [20] Formatted | Microsoft Office User | 11/29/17 11:26:00 AM |

ont color: Text 1, English (UK)

| age 2: [20] Formatted | Microsoft Office User | 11/29/17 11:26:00 AM |

ont color: Text 1, English (UK)

| age 2: [20] Formatted | Microsoft Office User | 11/29/17 11:26:00 AM |

ont color: Text 1, English (UK)

| age 2: [20] Formatted | Microsoft Office User | 11/29/17 11:26:00 AM |

ont color: Text 1, English (UK)

| age 2: [20] Formatted | Microsoft Office User | 11/29/17 11:26:00 AM |

ont color: Text 1, English (UK)

| age 2: [20] Formatted | Microsoft Office User | 11/29/17 11:26:00 AM |

ont color: Text 1, English (UK)

| age 2: [20] Formatted | Microsoft Office User | 11/29/17 11:26:00 AM |

ont color: Text 1, English (UK)

| age 2: [20] Formatted | Microsoft Office User | 11/29/17 11:26:00 AM |

ont color: Text 1, English (UK)

| age 2: [20] Formatted | Microsoft Office User | 11/29/17 11:26:00 AM |

ont color: Text 1, English (UK)

| age 2: [20] Formatted | Microsoft Office User | 11/29/17 11:26:00 AM |

ont color: Text 1, English (UK)

| age 2: [20] Formatted | Microsoft Office User | 11/29/17 11:26:00 AM |

ont color: Text 1, English (UK)

| age 2: [20] Formatted | Microsoft Office User | 11/29/17 11:26:00 AM |

ont color: Text 1, English (UK)

| age 2: [20] Formatted | Microsoft Office User | 11/29/17 11:26:00 AM |

ont color: Text 1, English (UK)

| age 2: [20] Formatted | Microsoft Office User | 11/29/17 11:26:00 AM |

ont color: Text 1, English (UK)

| age 2: [20] Formatted | Microsoft Office User | 11/29/17 11:26:00 AM |

ont color: Text 1, English (UK)

| age 2: [20] Formatted | Microsoft Office User | 11/29/17 11:26:00 AM |
|---|---|---|

ont color: Text 1, English (UK)

| age 2: [20] Formatted | Microsoft Office User | 11/29/17 11:26:00 AM |
|---|---|---|

ont color: Text 1, English (UK)

| age 2: [20] Formatted | Microsoft Office User | 11/29/17 11:26:00 AM |
|---|---|---|

ont color: Text 1, English (UK)

| age 2: [20] Formatted | Microsoft Office User | 11/29/17 11:26:00 AM |
|---|---|---|

ont color: Text 1, English (UK)

| age 2: [20] Formatted | Microsoft Office User | 11/29/17 11:26:00 AM |
|---|---|---|

ont color: Text 1, English (UK)

| age 2: [20] Formatted | Microsoft Office User | 11/29/17 11:26:00 AM |
|---|---|---|

ont color: Text 1, English (UK)

| age 2: [20] Formatted | Microsoft Office User | 11/29/17 11:26:00 AM |
|---|---|---|

ont color: Text 1, English (UK)

| age 2: [21] Formatted | Microsoft Office User | 11/29/17 11:26:00 AM |
|---|---|---|

ont color: Text 1, English (UK), Not Expanded by / Condensed by

| age 2: [21] Formatted | Microsoft Office User | 11/29/17 11:26:00 AM |
|---|---|---|

ont color: Text 1, English (UK), Not Expanded by / Condensed by

| age 2: [21] Formatted | Microsoft Office User | 11/29/17 11:26:00 AM |
|---|---|---|

ont color: Text 1, English (UK), Not Expanded by / Condensed by

| age 2: [21] Formatted | Microsoft Office User | 11/29/17 11:26:00 AM |
|---|---|---|

ont color: Text 1, English (UK), Not Expanded by / Condensed by

| age 2: [21] Formatted | Microsoft Office User | 11/29/17 11:26:00 AM |
|---|---|---|

ont color: Text 1, English (UK), Not Expanded by / Condensed by

| age 2: [21] Formatted | Microsoft Office User | 11/29/17 11:26:00 AM |
|---|---|---|

ont color: Text 1, English (UK), Not Expanded by / Condensed by

| age 2: [21] Formatted | Microsoft Office User | 11/29/17 11:26:00 AM |
|---|---|---|

ont color: Text 1, English (UK), Not Expanded by / Condensed by

| age 2: [21] Formatted | Microsoft Office User | 11/29/17 11:26:00 AM |
|---|---|---|

ont color: Text 1, English (UK), Not Expanded by / Condensed by

| age 2: [21] Formatted | Microsoft Office User | 11/29/17 11:26:00 AM |
|---|---|---|

ont color: Text 1, English (UK), Not Expanded by / Condensed by

| age 2: [21] Formatted | Microsoft Office User | 11/29/17 11:26:00 AM |
|---|---|---|

ont color: Text 1, English (UK), Not Expanded by / Condensed by

| age 2: [21] Formatted | Microsoft Office User | 11/29/17 11:26:00 AM |
|---|---|---|

ont color: Text 1, English (UK), Not Expanded by / Condensed by

| age 2: [21] Formatted | Microsoft Office User | 11/29/17 11:26:00 AM |
|---|---|---|

ont color: Text 1, English (UK), Not Expanded by / Condensed by

| age 2: [21] Formatted | Microsoft Office User | 11/29/17 11:26:00 AM |
|---|---|---|

ont color: Text 1, English (UK), Not Expanded by / Condensed by

| age 2: [21] Formatted | Microsoft Office User | 11/29/17 11:26:00 AM |
|---|---|---|

ont color: Text 1, English (UK), Not Expanded by / Condensed by

| age 2: [21] Formatted | Microsoft Office User | 11/29/17 11:26:00 AM |
|---|---|---|

ont color: Text 1, English (UK), Not Expanded by / Condensed by

| age 2: [22] Formatted | Microsoft Office User | 11/29/17 11:26:00 AM |
|---|---|---|

ont color: Text 1, English (UK)

| age 2: [22] Formatted | Microsoft Office User | 11/29/17 11:26:00 AM |
|---|---|---|

ont color: Text 1, English (UK)

| age 2: [22] Formatted | Microsoft Office User | 11/29/17 11:26:00 AM |
|---|---|---|

ont color: Text 1, English (UK)

| age 2: [22] Formatted | Microsoft Office User | 11/29/17 11:26:00 AM |
|---|---|---|

ont color: Text 1, English (UK)

| age 2: [22] Formatted | Microsoft Office User | 11/29/17 11:26:00 AM |
|---|---|---|

ont color: Text 1, English (UK)

| age 2: [23] Deleted | Microsoft Office User | 11/29/17 11:26:00 AM |
|---|---|---|

5 upper troposphere

| age 2: [24] Formatted | Microsoft Office User | 11/29/17 11:26:00 AM |
|---|---|---|

ont color: Text 1, English (UK)

| age 2: [24] Formatted | Microsoft Office User | 11/29/17 11:26:00 AM |
|---|---|---|

ont color: Text 1, English (UK)

| age 2: [24] Formatted | Microsoft Office User | 11/29/17 11:26:00 AM |
|---|---|---|

ont color: Text 1, English (UK)

| age 2: [24] Formatted | Microsoft Office User | 11/29/17 11:26:00 AM |
|---|---|---|

ont color: Text 1, English (UK)

| age 2: [25] Formatted | Microsoft Office User | 11/29/17 11:26:00 AM |
|---|---|---|

ont color: Text 1, English (UK)

| age 2: [26] Formatted | Microsoft Office User | 11/29/17 11:26:00 AM |
|---|---|---|

ont color: Text 1, English (UK)

| age 2: [26] Formatted | Microsoft Office User | 11/29/17 11:26:00 AM |
|---|---|---|

ont color: Text 1, English (UK)

| age 2: [26] Formatted | Microsoft Office User | 11/29/17 11:26:00 AM |
|---|---|---|

ont color: Text 1, English (UK)

| age 2: [26] Formatted | Microsoft Office User | 11/29/17 11:26:00 AM |
|---|---|---|

ont color: Text 1, English (UK)

| age 2: [27] Deleted | Microsoft Office User | 11/29/17 11:26:00 AM |
|---|---|---|

n the UTLS from these two nadir-viewing sensors are analysed in comparison to MLS to examine the information content for the ASM

UTLS trace gas analyses. Day-to-day changes in

| age 2: [28] Formatted | Microsoft Office User | 11/29/17 11:26:00 AM |
|---|---|---|

ont color: Text 1, English (UK)

| age 2: [29] Formatted | Microsoft Office User | 11/29/17 11:26:00 AM |
|---|---|---|

ont color: Text 1, English (UK)

| age 2: [30] Formatted | Microsoft Office User | 11/29/17 11:26:00 AM |
|---|---|---|

ont color: Text 1, English (UK)

| age 2: [31] Formatted | Microsoft Office User | 11/29/17 11:26:00 AM |
|---|---|---|

ont color: Text 1, English (UK)

| age 2: [32] Formatted | Microsoft Office User | 11/29/17 11:26:00 AM |
|---|---|---|

ont color: Text 1, English (UK)

| age 2: [32] Formatted | Microsoft Office User | 11/29/17 11:26:00 AM |
|---|---|---|

ont color: Text 1, English (UK)

| age 2: [32] Formatted | Microsoft Office User | 11/29/17 11:26:00 AM |
|---|---|---|

ont color: Text 1, English (UK)

| age 2: [32] Formatted | Microsoft Office User | 11/29/17 11:26:00 AM |
|---|---|---|

ont color: Text 1, English (UK)

| age 2: [32] Formatted | Microsoft Office User | 11/29/17 11:26:00 AM |
|---|---|---|

ont color: Text 1, English (UK)

| age 2: [32] Formatted | Microsoft Office User | 11/29/17 11:26:00 AM |
|---|---|---|

ont color: Text 1, English (UK)

| age 2: [32] Formatted | Microsoft Office User | 11/29/17 11:26:00 AM |
|---|---|---|

ont color: Text 1, English (UK)

| age 2: [32] Formatted | Microsoft Office User | 11/29/17 11:26:00 AM |
|---|---|---|

ont color: Text 1, English (UK)

| age 2: [33] Deleted | Microsoft Office User | 11/29/17 11:26:00 AM |
|---|---|---|

both nadir-viewing instruments capture the impact of ASM dynamics on spatial distribution of tracers in

0 the UTLS. Despite

| age 2: [34] Formatted | Microsoft Office User | 11/29/17 11:26:00 AM |
|---|---|---|

ont color: Text 1, English (UK)

| age 2: [35] Formatted | Microsoft Office User | 11/29/17 11:26:00 AM |
|---|---|---|

ont color: Text 1, English (UK)

| age 2: [36] Formatted | Microsoft Office User | 11/29/17 11:26:00 AM |
|---|---|---|

ont color: Text 1, English (UK)

| age 2: [37] Formatted | Microsoft Office User | 11/29/17 11:26:00 AM |
|---|---|---|

ont color: Text 1, English (UK)

| Page 2: [38] Formatted | Microsoft Office User | 11/29/17 11:26:00 AM |
|---|---|---|

Font color: Text 1, English (UK)

| Page 2: [39] Formatted | Microsoft Office User | 11/29/17 11:26:00 AM |
|---|---|---|

Font color: Text 1, English (UK)

| Page 2: [39] Formatted | Microsoft Office User | 11/29/17 11:26:00 AM |
|---|---|---|

Font color: Text 1, English (UK)

| Page 2: [39] Formatted | Microsoft Office User | 11/29/17 11:26:00 AM |
|---|---|---|

Font color: Text 1, English (UK)

| Page 2: [39] Formatted | Microsoft Office User | 11/29/17 11:26:00 AM |
|---|---|---|

Font color: Text 1, English (UK)

| Page 1: [40] Deleted | Microsoft Office User | 11/29/17 11:26:00 AM |
|---|---|---|
| Page 3: [41] Deleted | Microsoft Office User | 11/29/17 11:26:00 AM |
|---|---|---|

The high horizontal sampling density of IASI data show finer structures in the horizontal distribution of CO compared to the limb

viewing MLS, including CO enhancement in the upper troposphere over the western Pacific resulting from the eastward eddy

| Page 3: [42] Formatted | Microsoft Office User | 11/29/17 11:26:00 AM |
|---|---|---|

Font color: Text 1, English (UK)

| Page 3: [43] Formatted | Microsoft Office User | 11/29/17 11:26:00 AM |
|---|---|---|

Heading 1, Left, No bullets or numbering, Tabs:Not at 0.61"

| Page 3: [44] Deleted | Microsoft Office User | 11/29/17 11:26:00 AM |
|---|---|---|

25 shedding of the ASM anticyclone. Sub-seasonal variability of tracers is correlated with the dynamical structure of the anticyclone

as represented by the geopotential height (GPH) field, and systematic differences between the nadir and limb sounder results

are discussed.

| Page 3: [45] Formatted | Microsoft Office User | 11/29/17 11:26:00 AM |
|---|---|---|

Font:Not Bold

| Page 3: [46] Deleted | Microsoft Office User | 11/29/17 11:26:00 AM |
|---|---|---|

0

| Page 3: [47] Formatted | Microsoft Office User | 11/29/17 11:26:00 AM |
|---|---|---|

Normal, Indent: Left: 0", Line spacing: 1.5 lines, Tabs:Not at 0.47"

| Page 3: [48] Formatted | Microsoft Office User | 11/29/17 11:26:00 AM |
|---|---|---|

Font color: Text 1, English (UK)

| Page 3: [49] Deleted | Microsoft Office User | 11/29/17 11:26:00 AM |
|---|---|---|

anticyclone has been investigated widely in recent years  (Hoskins and Rodwell,   1995;

—————————————Section Break (Next Page)—————————————

lighwood and Hoskins, 1998; Zhang et al., 2002; Liu et al., 2007; Wu et al., 2015). It is bounded by the westerly jet to the north, asterly jet to the south, and noted for prolonged air confinement (Dunkerton, 1995).

| | | |
|---|---|---|
| age 3: [50] Formatted | Microsoft Office User | 11/29/17 11:26:00 AM |
| ont color: Text 1, English (UK) | | |
| age 3: [50] Formatted | Microsoft Office User | 11/29/17 11:26:00 AM |
| ont color: Text 1, English (UK) | | |
| age 3: [50] Formatted | Microsoft Office User | 11/29/17 11:26:00 AM |
| ont color: Text 1, English (UK) | | |
| age 3: [51] Deleted | Microsoft Office User | 11/29/17 11:26:00 AM |
| | | |
| age 3: [52] Formatted | Microsoft Office User | 11/29/17 11:26:00 AM |
| ont color: Text 1, English (UK) | | |
| age 3: [52] Formatted | Microsoft Office User | 11/29/17 11:26:00 AM |
| ont color: Text 1, English (UK) | | |
| age 3: [52] Formatted | Microsoft Office User | 11/29/17 11:26:00 AM |
| ont color: Text 1, English (UK) | | |
| age 3: [52] Formatted | Microsoft Office User | 11/29/17 11:26:00 AM |
| ont color: Text 1, English (UK) | | |
| age 3: [52] Formatted | Microsoft Office User | 11/29/17 11:26:00 AM |
| ont color: Text 1, English (UK) | | |
| age 3: [52] Formatted | Microsoft Office User | 11/29/17 11:26:00 AM |
| ont color: Text 1, English (UK) | | |
| age 3: [52] Formatted | Microsoft Office User | 11/29/17 11:26:00 AM |
| ont color: Text 1, English (UK) | | |
| age 3: [52] Formatted | Microsoft Office User | 11/29/17 11:26:00 AM |
| ont color: Text 1, English (UK) | | |
| age 3: [52] Formatted | Microsoft Office User | 11/29/17 11:26:00 AM |
| ont color: Text 1, English (UK) | | |
| age 3: [52] Formatted | Microsoft Office User | 11/29/17 11:26:00 AM |
| ont color: Text 1, English (UK) | | |
| age 3: [52] Formatted | Microsoft Office User | 11/29/17 11:26:00 AM |
| ont color: Text 1, English (UK) | | |
| age 3: [52] Formatted | Microsoft Office User | 11/29/17 11:26:00 AM |
| ont color: Text 1, English (UK) | | |
| age 3: [52] Formatted | Microsoft Office User | 11/29/17 11:26:00 AM |

ont color: Text 1, English (UK)

| Page 3: [52] Formatted | Microsoft Office User | 11/29/17 11:26:00 AM |
|---|---|---|

ont color: Text 1, English (UK)

| Page 3: [52] Formatted | Microsoft Office User | 11/29/17 11:26:00 AM |
|---|---|---|

ont color: Text 1, English (UK)

| Page 3: [52] Formatted | Microsoft Office User | 11/29/17 11:26:00 AM |
|---|---|---|

ont color: Text 1, English (UK)

| Page 3: [52] Formatted | Microsoft Office User | 11/29/17 11:26:00 AM |
|---|---|---|

ont color: Text 1, English (UK)

| Page 3: [52] Formatted | Microsoft Office User | 11/29/17 11:26:00 AM |
|---|---|---|

ont color: Text 1, English (UK)

| Page 3: [52] Formatted | Microsoft Office User | 11/29/17 11:26:00 AM |
|---|---|---|

ont color: Text 1, English (UK)

| Page 3: [52] Formatted | Microsoft Office User | 11/29/17 11:26:00 AM |
|---|---|---|

ont color: Text 1, English (UK)

| Page 3: [52] Formatted | Microsoft Office User | 11/29/17 11:26:00 AM |
|---|---|---|

ont color: Text 1, English (UK)

| Page 3: [52] Formatted | Microsoft Office User | 11/29/17 11:26:00 AM |
|---|---|---|

ont color: Text 1, English (UK)

| Page 3: [52] Formatted | Microsoft Office User | 11/29/17 11:26:00 AM |
|---|---|---|

ont color: Text 1, English (UK)

| Page 3: [52] Formatted | Microsoft Office User | 11/29/17 11:26:00 AM |
|---|---|---|

ont color: Text 1, English (UK)

| Page 3: [52] Formatted | Microsoft Office User | 11/29/17 11:26:00 AM |
|---|---|---|

ont color: Text 1, English (UK)

| Page 3: [53] Formatted | Microsoft Office User | 11/29/17 11:26:00 AM |
|---|---|---|

ont color: Text 1, English (UK)

| Page 3: [54] Deleted | Microsoft Office User | 11/29/17 11:26:00 AM |
|---|---|---|

2005; Randel and Park, 2006; Park et al., 2007; Randel et al., 2010; Vernier et al., 2011; Garny and Randel, 2013).

| Page 3: [55] Formatted | Microsoft Office User | 11/29/17 11:26:00 AM |
|---|---|---|

Normal, Left, Indent: First line:  0", Right:  0", Space Before:  0 pt, Line spacing:  1.5 lines

| Page 3: [56] Formatted | Microsoft Office User | 11/29/17 11:26:00 AM |
|---|---|---|

ont color: Text 1, English (UK)

| Page 3: [56] Formatted | Microsoft Office User | 11/29/17 11:26:00 AM |
|---|---|---|

ont color: Text 1, English (UK)

| Page 3: [56] Formatted | Microsoft Office User | 11/29/17 11:26:00 AM |
|---|---|---|

ont color: Text 1, English (UK)

| Page 3: [56] Formatted | Microsoft Office User | 11/29/17 11:26:00 AM |

ont color: Text 1, English (UK)

| Page 3: [56] Formatted | Microsoft Office User | 11/29/17 11:26:00 AM |

ont color: Text 1, English (UK)

| Page 3: [56] Formatted | Microsoft Office User | 11/29/17 11:26:00 AM |

ont color: Text 1, English (UK)

| Page 3: [56] Formatted | Microsoft Office User | 11/29/17 11:26:00 AM |

ont color: Text 1, English (UK)

| Page 3: [56] Formatted | Microsoft Office User | 11/29/17 11:26:00 AM |

ont color: Text 1, English (UK)

| Page 3: [56] Formatted | Microsoft Office User | 11/29/17 11:26:00 AM |

ont color: Text 1, English (UK)

| Page 3: [56] Formatted | Microsoft Office User | 11/29/17 11:26:00 AM |

ont color: Text 1, English (UK)

| Page 3: [56] Formatted | Microsoft Office User | 11/29/17 11:26:00 AM |

ont color: Text 1, English (UK)

| Page 3: [56] Formatted | Microsoft Office User | 11/29/17 11:26:00 AM |

ont color: Text 1, English (UK)

| Page 3: [56] Formatted | Microsoft Office User | 11/29/17 11:26:00 AM |

ont color: Text 1, English (UK)

| Page 3: [56] Formatted | Microsoft Office User | 11/29/17 11:26:00 AM |

ont color: Text 1, English (UK)

| Page 3: [56] Formatted | Microsoft Office User | 11/29/17 11:26:00 AM |

ont color: Text 1, English (UK)

| Page 3: [56] Formatted | Microsoft Office User | 11/29/17 11:26:00 AM |

ont color: Text 1, English (UK)

| Page 3: [56] Formatted | Microsoft Office User | 11/29/17 11:26:00 AM |

ont color: Text 1, English (UK)

| Page 3: [56] Formatted | Microsoft Office User | 11/29/17 11:26:00 AM |

ont color: Text 1, English (UK)

| Page 3: [56] Formatted | Microsoft Office User | 11/29/17 11:26:00 AM |

ont color: Text 1, English (UK)

| Page 3: [56] Formatted | Microsoft Office User | 11/29/17 11:26:00 AM |

ont color: Text 1, English (UK)

| Page 3: [56] Formatted | Microsoft Office User | 11/29/17 11:26:00 AM |
|---|---|---|
| Font color: Text 1, English (UK) | | |
| Page 3: [56] Formatted | Microsoft Office User | 11/29/17 11:26:00 AM |
| Font color: Text 1, English (UK) | | |
| Page 3: [56] Formatted | Microsoft Office User | 11/29/17 11:26:00 AM |
| Font color: Text 1, English (UK) | | |
| Page 3: [56] Formatted | Microsoft Office User | 11/29/17 11:26:00 AM |
| Font color: Text 1, English (UK) | | |
| Page 3: [56] Formatted | Microsoft Office User | 11/29/17 11:26:00 AM |
| Font color: Text 1, English (UK) | | |
| Page 3: [56] Formatted | Microsoft Office User | 11/29/17 11:26:00 AM |
| Font color: Text 1, English (UK) | | |
| Page 3: [56] Formatted | Microsoft Office User | 11/29/17 11:26:00 AM |
| Font color: Text 1, English (UK) | | |
| Page 3: [56] Formatted | Microsoft Office User | 11/29/17 11:26:00 AM |
| Font color: Text 1, English (UK) | | |
| Page 3: [56] Formatted | Microsoft Office User | 11/29/17 11:26:00 AM |
| Font color: Text 1, English (UK) | | |
| Page 3: [56] Formatted | Microsoft Office User | 11/29/17 11:26:00 AM |
| Font color: Text 1, English (UK) | | |
| Page 3: [56] Formatted | Microsoft Office User | 11/29/17 11:26:00 AM |
| Font color: Text 1, English (UK) | | |
| Page 3: [56] Formatted | Microsoft Office User | 11/29/17 11:26:00 AM |
| Font color: Text 1, English (UK) | | |
| Page 3: [56] Formatted | Microsoft Office User | 11/29/17 11:26:00 AM |
| Font color: Text 1, English (UK) | | |
| Page 3: [56] Formatted | Microsoft Office User | 11/29/17 11:26:00 AM |
| Font color: Text 1, English (UK) | | |
| Page 3: [56] Formatted | Microsoft Office User | 11/29/17 11:26:00 AM |
| Font color: Text 1, English (UK) | | |
| Page 3: [56] Formatted | Microsoft Office User | 11/29/17 11:26:00 AM |
| Font color: Text 1, English (UK) | | |
| Page 3: [56] Formatted | Microsoft Office User | 11/29/17 11:26:00 AM |
| Font color: Text 1, English (UK) | | |
| Page 3: [57] Formatted | Microsoft Office User | 11/29/17 11:26:00 AM |
| Font color: Text 1, English (UK) | | |
| Page 3: [58] Deleted | Microsoft Office User | 11/29/17 11:26:00 AM |

Hsu and Plumb, 2000; Popovic and Plumb, 2001).

| Page 3: [59] Formatted | Microsoft Office User | 11/29/17 11:26:00 AM |
|---|---|---|

Font color: Text 1, English (UK)

| Page 3: [60] Formatted | Microsoft Office User | 11/29/17 11:26:00 AM |
|---|---|---|

Font color: Text 1, English (UK)

| Page 3: [61] Deleted | Microsoft Office User | 11/29/17 11:26:00 AM |
|---|---|---|

0

| Page 3: [62] Formatted | Microsoft Office User | 11/29/17 11:26:00 AM |
|---|---|---|

Font:10 pt, Font color: Text 1, English (UK)

| Page 3: [62] Formatted | Microsoft Office User | 11/29/17 11:26:00 AM |
|---|---|---|

Font:10 pt, Font color: Text 1, English (UK)

| Page 3: [63] Deleted | Microsoft Office User | 11/29/17 11:26:00 AM |
|---|---|---|

i.e., Krishnamurti and Bhalme, 1976; Krishnamurti and Ardanuy, 1980; Annamalai and Slingo, 2001; Randel and Park, 2006). Zhang et al. (2002)

| Page 3: [64] Formatted | Microsoft Office User | 11/29/17 11:26:00 AM |
|---|---|---|

Font color: Text 1, English (UK)

| Page 3: [64] Formatted | Microsoft Office User | 11/29/17 11:26:00 AM |
|---|---|---|

Font color: Text 1, English (UK)

| Page 3: [64] Formatted | Microsoft Office User | 11/29/17 11:26:00 AM |
|---|---|---|

Font color: Text 1, English (UK)

| Page 3: [64] Formatted | Microsoft Office User | 11/29/17 11:26:00 AM |
|---|---|---|

Font color: Text 1, English (UK)

| Page 3: [64] Formatted | Microsoft Office User | 11/29/17 11:26:00 AM |
|---|---|---|

Font color: Text 1, English (UK)

| Page 3: [64] Formatted | Microsoft Office User | 11/29/17 11:26:00 AM |
|---|---|---|

Font color: Text 1, English (UK)

| Page 3: [64] Formatted | Microsoft Office User | 11/29/17 11:26:00 AM |
|---|---|---|

Font color: Text 1, English (UK)

| Page 3: [64] Formatted | Microsoft Office User | 11/29/17 11:26:00 AM |
|---|---|---|

Font color: Text 1, English (UK)

| Page 3: [64] Formatted | Microsoft Office User | 11/29/17 11:26:00 AM |
|---|---|---|

Font color: Text 1, English (UK)

| Page 3: [64] Formatted | Microsoft Office User | 11/29/17 11:26:00 AM |
|---|---|---|

Font color: Text 1, English (UK)

| Page 3: [64] Formatted | Microsoft Office User | 11/29/17 11:26:00 AM |
|---|---|---|

Font color: Text 1, English (UK)

| Page 3: [64] Formatted | Microsoft Office User | 11/29/17 11:26:00 AM |
|---|---|---|

ont color: Text 1, English (UK)

| age 3: [64] Formatted | Microsoft Office User | 11/29/17 11:26:00 AM |
|---|---|---|

ont color: Text 1, English (UK)

| age 3: [64] Formatted | Microsoft Office User | 11/29/17 11:26:00 AM |
|---|---|---|

ont color: Text 1, English (UK)

| age 3: [64] Formatted | Microsoft Office User | 11/29/17 11:26:00 AM |
|---|---|---|

ont color: Text 1, English (UK)

| age 3: [64] Formatted | Microsoft Office User | 11/29/17 11:26:00 AM |
|---|---|---|

ont color: Text 1, English (UK)

| age 3: [64] Formatted | Microsoft Office User | 11/29/17 11:26:00 AM |
|---|---|---|

ont color: Text 1, English (UK)

| age 3: [64] Formatted | Microsoft Office User | 11/29/17 11:26:00 AM |
|---|---|---|

ont color: Text 1, English (UK)

| age 3: [64] Formatted | Microsoft Office User | 11/29/17 11:26:00 AM |
|---|---|---|

ont color: Text 1, English (UK)

| age 3: [64] Formatted | Microsoft Office User | 11/29/17 11:26:00 AM |
|---|---|---|

ont color: Text 1, English (UK)

| age 3: [64] Formatted | Microsoft Office User | 11/29/17 11:26:00 AM |
|---|---|---|

ont color: Text 1, English (UK)

| age 3: [64] Formatted | Microsoft Office User | 11/29/17 11:26:00 AM |
|---|---|---|

ont color: Text 1, English (UK)

| age 3: [64] Formatted | Microsoft Office User | 11/29/17 11:26:00 AM |
|---|---|---|

ont color: Text 1, English (UK)

| age 3: [64] Formatted | Microsoft Office User | 11/29/17 11:26:00 AM |
|---|---|---|

ont color: Text 1, English (UK)

| age 3: [64] Formatted | Microsoft Office User | 11/29/17 11:26:00 AM |
|---|---|---|

ont color: Text 1, English (UK)

| age 3: [64] Formatted | Microsoft Office User | 11/29/17 11:26:00 AM |
|---|---|---|

ont color: Text 1, English (UK)

| age 3: [64] Formatted | Microsoft Office User | 11/29/17 11:26:00 AM |
|---|---|---|

ont color: Text 1, English (UK)

| age 3: [64] Formatted | Microsoft Office User | 11/29/17 11:26:00 AM |
|---|---|---|

ont color: Text 1, English (UK)

| age 3: [64] Formatted | Microsoft Office User | 11/29/17 11:26:00 AM |
|---|---|---|

ont color: Text 1, English (UK)

| age 3: [64] Formatted | Microsoft Office User | 11/29/17 11:26:00 AM |

ont color: Text 1, English (UK)

| age 3: [64] Formatted | Microsoft Office User | 11/29/17 11:26:00 AM |

ont color: Text 1, English (UK)

| age 3: [64] Formatted | Microsoft Office User | 11/29/17 11:26:00 AM |

ont color: Text 1, English (UK)

| age 3: [64] Formatted | Microsoft Office User | 11/29/17 11:26:00 AM |

ont color: Text 1, English (UK)

| age 3: [64] Formatted | Microsoft Office User | 11/29/17 11:26:00 AM |

ont color: Text 1, English (UK)

| age 3: [64] Formatted | Microsoft Office User | 11/29/17 11:26:00 AM |

ont color: Text 1, English (UK)

| age 3: [64] Formatted | Microsoft Office User | 11/29/17 11:26:00 AM |

ont color: Text 1, English (UK)

| age 3: [64] Formatted | Microsoft Office User | 11/29/17 11:26:00 AM |

ont color: Text 1, English (UK)

| age 3: [64] Formatted | Microsoft Office User | 11/29/17 11:26:00 AM |

ont color: Text 1, English (UK)

| age 3: [64] Formatted | Microsoft Office User | 11/29/17 11:26:00 AM |

ont color: Text 1, English (UK)

| age 3: [64] Formatted | Microsoft Office User | 11/29/17 11:26:00 AM |

ont color: Text 1, English (UK)

| age 3: [64] Formatted | Microsoft Office User | 11/29/17 11:26:00 AM |

ont color: Text 1, English (UK)

| age 3: [64] Formatted | Microsoft Office User | 11/29/17 11:26:00 AM |

ont color: Text 1, English (UK)

| age 3: [64] Formatted | Microsoft Office User | 11/29/17 11:26:00 AM |

ont color: Text 1, English (UK)

| age 3: [64] Formatted | Microsoft Office User | 11/29/17 11:26:00 AM |

ont color: Text 1, English (UK)

| age 3: [64] Formatted | Microsoft Office User | 11/29/17 11:26:00 AM |

ont color: Text 1, English (UK)

| age 3: [64] Formatted | Microsoft Office User | 11/29/17 11:26:00 AM |

ont color: Text 1, English (UK)

| age 3: [64] Formatted | Microsoft Office User | 11/29/17 11:26:00 AM |

ont color: Text 1, English (UK)

| age 3: [64] Formatted | Microsoft Office User | 11/29/17 11:26:00 AM |
| --- | --- | --- |

ont color: Text 1, English (UK)

| age 3: [64] Formatted | Microsoft Office User | 11/29/17 11:26:00 AM |
| --- | --- | --- |

ont color: Text 1, English (UK)

| age 3: [64] Formatted | Microsoft Office User | 11/29/17 11:26:00 AM |
| --- | --- | --- |

ont color: Text 1, English (UK)

| age 3: [64] Formatted | Microsoft Office User | 11/29/17 11:26:00 AM |
| --- | --- | --- |

ont color: Text 1, English (UK)

| age 3: [64] Formatted | Microsoft Office User | 11/29/17 11:26:00 AM |
| --- | --- | --- |

ont color: Text 1, English (UK)

| age 3: [64] Formatted | Microsoft Office User | 11/29/17 11:26:00 AM |
| --- | --- | --- |

ont color: Text 1, English (UK)

| age 3: [64] Formatted | Microsoft Office User | 11/29/17 11:26:00 AM |
| --- | --- | --- |

ont color: Text 1, English (UK)

| age 3: [64] Formatted | Microsoft Office User | 11/29/17 11:26:00 AM |
| --- | --- | --- |

ont color: Text 1, English (UK)

| age 3: [64] Formatted | Microsoft Office User | 11/29/17 11:26:00 AM |
| --- | --- | --- |

ont color: Text 1, English (UK)

| age 3: [64] Formatted | Microsoft Office User | 11/29/17 11:26:00 AM |
| --- | --- | --- |

ont color: Text 1, English (UK)

| age 3: [64] Formatted | Microsoft Office User | 11/29/17 11:26:00 AM |
| --- | --- | --- |

ont color: Text 1, English (UK)

| age 3: [64] Formatted | Microsoft Office User | 11/29/17 11:26:00 AM |
| --- | --- | --- |

ont color: Text 1, English (UK)

| age 3: [64] Formatted | Microsoft Office User | 11/29/17 11:26:00 AM |
| --- | --- | --- |

ont color: Text 1, English (UK)

| age 3: [64] Formatted | Microsoft Office User | 11/29/17 11:26:00 AM |
| --- | --- | --- |

ont color: Text 1, English (UK)

| age 3: [64] Formatted | Microsoft Office User | 11/29/17 11:26:00 AM |
| --- | --- | --- |

ont color: Text 1, English (UK)

| age 3: [64] Formatted | Microsoft Office User | 11/29/17 11:26:00 AM |
| --- | --- | --- |

ont color: Text 1, English (UK)

| age 3: [64] Formatted | Microsoft Office User | 11/29/17 11:26:00 AM |
| --- | --- | --- |

ont color: Text 1, English (UK)

| age 3: [64] Formatted | Microsoft Office User | 11/29/17 11:26:00 AM |
| --- | --- | --- |

ont color: Text 1, English (UK)

| age 3: [64] Formatted | Microsoft Office User | 11/29/17 11:26:00 AM |
|---|---|---|

ont color: Text 1, English (UK)

| age 3: [64] Formatted | Microsoft Office User | 11/29/17 11:26:00 AM |
|---|---|---|

ont color: Text 1, English (UK)

| age 3: [64] Formatted | Microsoft Office User | 11/29/17 11:26:00 AM |
|---|---|---|

ont color: Text 1, English (UK)

| age 3: [64] Formatted | Microsoft Office User | 11/29/17 11:26:00 AM |
|---|---|---|

ont color: Text 1, English (UK)

| age 3: [64] Formatted | Microsoft Office User | 11/29/17 11:26:00 AM |
|---|---|---|

ont color: Text 1, English (UK)

| age 3: [64] Formatted | Microsoft Office User | 11/29/17 11:26:00 AM |
|---|---|---|

ont color: Text 1, English (UK)

| age 3: [64] Formatted | Microsoft Office User | 11/29/17 11:26:00 AM |
|---|---|---|

ont color: Text 1, English (UK)

| age 3: [64] Formatted | Microsoft Office User | 11/29/17 11:26:00 AM |
|---|---|---|

ont color: Text 1, English (UK)

| age 3: [64] Formatted | Microsoft Office User | 11/29/17 11:26:00 AM |
|---|---|---|

ont color: Text 1, English (UK)

| age 3: [64] Formatted | Microsoft Office User | 11/29/17 11:26:00 AM |
|---|---|---|

ont color: Text 1, English (UK)

| age 3: [64] Formatted | Microsoft Office User | 11/29/17 11:26:00 AM |
|---|---|---|

ont color: Text 1, English (UK)

| age 3: [64] Formatted | Microsoft Office User | 11/29/17 11:26:00 AM |
|---|---|---|

ont color: Text 1, English (UK)

| age 3: [64] Formatted | Microsoft Office User | 11/29/17 11:26:00 AM |
|---|---|---|

ont color: Text 1, English (UK)

| age 3: [64] Formatted | Microsoft Office User | 11/29/17 11:26:00 AM |
|---|---|---|

ont color: Text 1, English (UK)

| age 3: [64] Formatted | Microsoft Office User | 11/29/17 11:26:00 AM |
|---|---|---|

ont color: Text 1, English (UK)

| age 3: [65] Formatted | Microsoft Office User | 11/29/17 11:26:00 AM |
|---|---|---|

ont color: Text 1, English (UK), Not Expanded by / Condensed by

| age 3: [65] Formatted | Microsoft Office User | 11/29/17 11:26:00 AM |
|---|---|---|

ont color: Text 1, English (UK), Not Expanded by / Condensed by

| age 3: [65] Formatted | Microsoft Office User | 11/29/17 11:26:00 AM |
|---|---|---|

ont color: Text 1, English (UK), Not Expanded by / Condensed by

| age 3: [65] Formatted | Microsoft Office User | 11/29/17 11:26:00 AM |
|---|---|---|

ont color: Text 1, English (UK), Not Expanded by / Condensed by

| age 3: [65] Formatted | Microsoft Office User | 11/29/17 11:26:00 AM |
|---|---|---|

ont color: Text 1, English (UK), Not Expanded by / Condensed by

| age 3: [65] Formatted | Microsoft Office User | 11/29/17 11:26:00 AM |
|---|---|---|

ont color: Text 1, English (UK), Not Expanded by / Condensed by

| age 3: [65] Formatted | Microsoft Office User | 11/29/17 11:26:00 AM |
|---|---|---|

ont color: Text 1, English (UK), Not Expanded by / Condensed by

| age 3: [65] Formatted | Microsoft Office User | 11/29/17 11:26:00 AM |
|---|---|---|

ont color: Text 1, English (UK), Not Expanded by / Condensed by

| age 3: [65] Formatted | Microsoft Office User | 11/29/17 11:26:00 AM |
|---|---|---|

ont color: Text 1, English (UK), Not Expanded by / Condensed by

| age 3: [65] Formatted | Microsoft Office User | 11/29/17 11:26:00 AM |
|---|---|---|

ont color: Text 1, English (UK), Not Expanded by / Condensed by

| age 3: [65] Formatted | Microsoft Office User | 11/29/17 11:26:00 AM |
|---|---|---|

ont color: Text 1, English (UK), Not Expanded by / Condensed by

| age 3: [65] Formatted | Microsoft Office User | 11/29/17 11:26:00 AM |
|---|---|---|

ont color: Text 1, English (UK), Not Expanded by / Condensed by

| age 3: [65] Formatted | Microsoft Office User | 11/29/17 11:26:00 AM |
|---|---|---|

ont color: Text 1, English (UK), Not Expanded by / Condensed by

| age 3: [65] Formatted | Microsoft Office User | 11/29/17 11:26:00 AM |
|---|---|---|

ont color: Text 1, English (UK), Not Expanded by / Condensed by

| age 3: [65] Formatted | Microsoft Office User | 11/29/17 11:26:00 AM |
|---|---|---|

ont color: Text 1, English (UK), Not Expanded by / Condensed by

| age 3: [65] Formatted | Microsoft Office User | 11/29/17 11:26:00 AM |
|---|---|---|

ont color: Text 1, English (UK), Not Expanded by / Condensed by

| age 3: [65] Formatted | Microsoft Office User | 11/29/17 11:26:00 AM |
|---|---|---|

ont color: Text 1, English (UK), Not Expanded by / Condensed by

| age 3: [65] Formatted | Microsoft Office User | 11/29/17 11:26:00 AM |
|---|---|---|

ont color: Text 1, English (UK), Not Expanded by / Condensed by

| age 3: [65] Formatted | Microsoft Office User | 11/29/17 11:26:00 AM |
|---|---|---|

ont color: Text 1, English (UK), Not Expanded by / Condensed by

| age 3: [65] Formatted | Microsoft Office User | 11/29/17 11:26:00 AM |
|---|---|---|

ont color: Text 1, English (UK), Not Expanded by / Condensed by

| age 3: [66] Deleted | Microsoft Office User | 11/29/17 11:26:00 AM |
|---|---|---|

| age 3: [67] Formatted | Microsoft Office User | 11/29/17 11:26:00 AM |

ont:10 pt, Font color: Text 1, English (UK)

| age 3: [67] Formatted | Microsoft Office User | 11/29/17 11:26:00 AM |

ont:10 pt, Font color: Text 1, English (UK)

| age 3: [68] Formatted | Microsoft Office User | 11/29/17 11:26:00 AM |

ont color: Text 1, English (UK)

| age 3: [69] Deleted | Microsoft Office User | 11/29/17 11:26:00 AM |

Yan et al., 2011; Garny and Randel, 2015; Pan et al., 2016).

| age 3: [70] Formatted | Microsoft Office User | 11/29/17 11:26:00 AM |

ont color: Text 1, English (UK)

| age 3: [71] Formatted | Microsoft Office User | 11/29/17 11:26:00 AM |

ont color: Text 1, English (UK)

| age 3: [72] Formatted | Microsoft Office User | 11/29/17 11:26:00 AM |

ont color: Text 1, English (UK)

| age 3: [73] Deleted | Microsoft Office User | 11/29/17 11:26:00 AM |

0

| age 3: [74] Formatted | Microsoft Office User | 11/29/17 11:26:00 AM |

ont:10 pt, Font color: Text 1, English (UK)

| age 3: [74] Formatted | Microsoft Office User | 11/29/17 11:26:00 AM |

ont:10 pt, Font color: Text 1, English (UK)

| age 3: [74] Formatted | Microsoft Office User | 11/29/17 11:26:00 AM |

ont:10 pt, Font color: Text 1, English (UK)

| age 3: [74] Formatted | Microsoft Office User | 11/29/17 11:26:00 AM |

ont:10 pt, Font color: Text 1, English (UK)

| age 3: [74] Formatted | Microsoft Office User | 11/29/17 11:26:00 AM |

ont:10 pt, Font color: Text 1, English (UK)

| age 3: [75] Formatted | Microsoft Office User | 11/29/17 11:26:00 AM |

ont color: Text 1, English (UK)

| age 3: [75] Formatted | Microsoft Office User | 11/29/17 11:26:00 AM |

ont color: Text 1, English (UK)

| age 3: [76] Formatted | Microsoft Office User | 11/29/17 11:26:00 AM |

ont color: Text 1, English (UK)

| age 3: [76] Formatted | Microsoft Office User | 11/29/17 11:26:00 AM |

ont color: Text 1, English (UK)

| age 3: [77] Formatted | Microsoft Office User | 11/29/17 11:26:00 AM |

ont color: Text 1, English (UK)

| age 3: [77] Formatted | Microsoft Office User | 11/29/17 11:26:00 AM |

ont color: Text 1, English (UK)

| age 3: [77] Formatted | Microsoft Office User | 11/29/17 11:26:00 AM |

ont color: Text 1, English (UK)

| age 3: [77] Formatted | Microsoft Office User | 11/29/17 11:26:00 AM |

ont color: Text 1, English (UK)

| age 3: [78] Formatted | Microsoft Office User | 11/29/17 11:26:00 AM |

Normal, Left, Right:  0", Space Before:  0 pt

| age 3: [79] Formatted | Microsoft Office User | 11/29/17 11:26:00 AM |

ont color: Text 1, English (UK)

| age 3: [79] Formatted | Microsoft Office User | 11/29/17 11:26:00 AM |

ont color: Text 1, English (UK)

| age 3: [79] Formatted | Microsoft Office User | 11/29/17 11:26:00 AM |

ont color: Text 1, English (UK)

| age 3: [79] Formatted | Microsoft Office User | 11/29/17 11:26:00 AM |

ont color: Text 1, English (UK)

| age 3: [79] Formatted | Microsoft Office User | 11/29/17 11:26:00 AM |

ont color: Text 1, English (UK)

| age 3: [79] Formatted | Microsoft Office User | 11/29/17 11:26:00 AM |

ont color: Text 1, English (UK)

| age 4: [80] Deleted | Microsoft Office User | 11/29/17 11:26:00 AM |

CO is a pollution tracer and an effective tracer of transport in the troposphere and lower stratosphere (e.g., Bowman, 2006) since it has

photochemical lifetime of 2–3 months (Xiao et al., 2007).

| age 4: [81] Formatted | Microsoft Office User | 11/29/17 11:26:00 AM |

ont color: Text 1, English (UK)

| age 4: [81] Formatted | Microsoft Office User | 11/29/17 11:26:00 AM |

ont color: Text 1, English (UK)

| age 4: [81] Formatted | Microsoft Office User | 11/29/17 11:26:00 AM |

ont color: Text 1, English (UK)

| age 4: [81] Formatted | Microsoft Office User | 11/29/17 11:26:00 AM |

ont color: Text 1, English (UK)

| age 4: [81] Formatted | Microsoft Office User | 11/29/17 11:26:00 AM |

ont color: Text 1, English (UK)

| age 4: [81] Formatted | Microsoft Office User | 11/29/17 11:26:00 AM |

ont color: Text 1, English (UK)

| age 4: [81] Formatted | Microsoft Office User | 11/29/17 11:26:00 AM |

ont color: Text 1, English (UK)

| age 4: [81] Formatted | Microsoft Office User | 11/29/17 11:26:00 AM |

ont color: Text 1, English (UK)

| age 4: [81] Formatted | Microsoft Office User | 11/29/17 11:26:00 AM |

ont color: Text 1, English (UK)

| age 4: [81] Formatted | Microsoft Office User | 11/29/17 11:26:00 AM |

ont color: Text 1, English (UK)

| age 4: [81] Formatted | Microsoft Office User | 11/29/17 11:26:00 AM |

ont color: Text 1, English (UK)

| age 4: [81] Formatted | Microsoft Office User | 11/29/17 11:26:00 AM |

ont color: Text 1, English (UK)

| age 4: [81] Formatted | Microsoft Office User | 11/29/17 11:26:00 AM |

ont color: Text 1, English (UK)

| age 4: [81] Formatted | Microsoft Office User | 11/29/17 11:26:00 AM |

ont color: Text 1, English (UK)

| age 4: [81] Formatted | Microsoft Office User | 11/29/17 11:26:00 AM |

ont color: Text 1, English (UK)

| age 4: [81] Formatted | Microsoft Office User | 11/29/17 11:26:00 AM |

ont color: Text 1, English (UK)

| age 4: [81] Formatted | Microsoft Office User | 11/29/17 11:26:00 AM |

ont color: Text 1, English (UK)

| age 4: [81] Formatted | Microsoft Office User | 11/29/17 11:26:00 AM |

ont color: Text 1, English (UK)

| age 4: [81] Formatted | Microsoft Office User | 11/29/17 11:26:00 AM |

ont color: Text 1, English (UK)

| age 4: [81] Formatted | Microsoft Office User | 11/29/17 11:26:00 AM |

ont color: Text 1, English (UK)

| age 4: [81] Formatted | Microsoft Office User | 11/29/17 11:26:00 AM |

ont color: Text 1, English (UK)

| age 4: [81] Formatted | Microsoft Office User | 11/29/17 11:26:00 AM |

ont color: Text 1, English (UK)

| age 4: [81] Formatted | Microsoft Office User | 11/29/17 11:26:00 AM |

ont color: Text 1, English (UK)

| age 4: [81] Formatted | Microsoft Office User | 11/29/17 11:26:00 AM |

ont color: Text 1, English (UK)

| age 4: [81] Formatted | Microsoft Office User | 11/29/17 11:26:00 AM |

ont color: Text 1, English (UK)

| age 4: [81] Formatted | Microsoft Office User | 11/29/17 11:26:00 AM |
|---|---|---|

ont color: Text 1, English (UK)

| age 4: [81] Formatted | Microsoft Office User | 11/29/17 11:26:00 AM |
|---|---|---|

ont color: Text 1, English (UK)

| age 4: [81] Formatted | Microsoft Office User | 11/29/17 11:26:00 AM |
|---|---|---|

ont color: Text 1, English (UK)

| age 4: [81] Formatted | Microsoft Office User | 11/29/17 11:26:00 AM |
|---|---|---|

ont color: Text 1, English (UK)

| age 4: [81] Formatted | Microsoft Office User | 11/29/17 11:26:00 AM |
|---|---|---|

ont color: Text 1, English (UK)

| age 4: [81] Formatted | Microsoft Office User | 11/29/17 11:26:00 AM |
|---|---|---|

ont color: Text 1, English (UK)

| age 4: [81] Formatted | Microsoft Office User | 11/29/17 11:26:00 AM |
|---|---|---|

ont color: Text 1, English (UK)

| age 4: [81] Formatted | Microsoft Office User | 11/29/17 11:26:00 AM |
|---|---|---|

ont color: Text 1, English (UK)

| age 4: [81] Formatted | Microsoft Office User | 11/29/17 11:26:00 AM |
|---|---|---|

ont color: Text 1, English (UK)

| age 4: [81] Formatted | Microsoft Office User | 11/29/17 11:26:00 AM |
|---|---|---|

ont color: Text 1, English (UK)

| age 4: [81] Formatted | Microsoft Office User | 11/29/17 11:26:00 AM |
|---|---|---|

ont color: Text 1, English (UK)

| age 4: [81] Formatted | Microsoft Office User | 11/29/17 11:26:00 AM |
|---|---|---|

ont color: Text 1, English (UK)

| age 4: [81] Formatted | Microsoft Office User | 11/29/17 11:26:00 AM |
|---|---|---|

ont color: Text 1, English (UK)

| age 4: [81] Formatted | Microsoft Office User | 11/29/17 11:26:00 AM |
|---|---|---|

ont color: Text 1, English (UK)

| age 4: [81] Formatted | Microsoft Office User | 11/29/17 11:26:00 AM |
|---|---|---|

ont color: Text 1, English (UK)

| age 4: [81] Formatted | Microsoft Office User | 11/29/17 11:26:00 AM |
|---|---|---|

ont color: Text 1, English (UK)

| age 4: [81] Formatted | Microsoft Office User | 11/29/17 11:26:00 AM |
|---|---|---|

ont color: Text 1, English (UK)

| age 4: [82] Deleted | Microsoft Office User | 11/29/17 11:26:00 AM |
|---|---|---|

0

| age 4: [83] Formatted | Microsoft Office User | 11/29/17 11:26:00 AM |
|---|---|---|
| ont:10 pt, Font color: Text 1, English (UK) | | |

| age 4: [83] Formatted | Microsoft Office User | 11/29/17 11:26:00 AM |
|---|---|---|
| ont:10 pt, Font color: Text 1, English (UK) | | |

| age 4: [83] Formatted | Microsoft Office User | 11/29/17 11:26:00 AM |
|---|---|---|
| ont:10 pt, Font color: Text 1, English (UK) | | |

| age 4: [83] Formatted | Microsoft Office User | 11/29/17 11:26:00 AM |
|---|---|---|
| ont:10 pt, Font color: Text 1, English (UK) | | |

| age 4: [83] Formatted | Microsoft Office User | 11/29/17 11:26:00 AM |
|---|---|---|
| ont:10 pt, Font color: Text 1, English (UK) | | |

| age 4: [83] Formatted | Microsoft Office User | 11/29/17 11:26:00 AM |
|---|---|---|
| ont:10 pt, Font color: Text 1, English (UK) | | |

| age 4: [84] Formatted | Microsoft Office User | 11/29/17 11:26:00 AM |
|---|---|---|
| ont color: Text 1, English (UK) | | |

| age 4: [84] Formatted | Microsoft Office User | 11/29/17 11:26:00 AM |
|---|---|---|
| ont color: Text 1, English (UK) | | |

| age 4: [84] Formatted | Microsoft Office User | 11/29/17 11:26:00 AM |
|---|---|---|
| ont color: Text 1, English (UK) | | |

| age 4: [85] Formatted | Microsoft Office User | 11/29/17 11:26:00 AM |
|---|---|---|
| ont color: Text 1, English (UK) | | |

| age 4: [86] Formatted | Microsoft Office User | 11/29/17 11:26:00 AM |
|---|---|---|
| ont color: Text 1, English (UK) | | |

| age 4: [87] Formatted | Microsoft Office User | 11/29/17 11:26:00 AM |
|---|---|---|
| ont color: Text 1, English (UK) | | |

| age 4: [88] Formatted | Microsoft Office User | 11/29/17 11:26:00 AM |
|---|---|---|
| ont color: Text 1, English (UK) | | |

| age 4: [89] Formatted | Microsoft Office User | 11/29/17 11:26:00 AM |
|---|---|---|
| ont color: Text 1, English (UK) | | |

| age 4: [89] Formatted | Microsoft Office User | 11/29/17 11:26:00 AM |
|---|---|---|
| ont color: Text 1, English (UK) | | |

| age 4: [89] Formatted | Microsoft Office User | 11/29/17 11:26:00 AM |
|---|---|---|
| ont color: Text 1, English (UK) | | |

| age 4: [89] Formatted | Microsoft Office User | 11/29/17 11:26:00 AM |
|---|---|---|
| ont color: Text 1, English (UK) | | |

| age 4: [89] Formatted | Microsoft Office User | 11/29/17 11:26:00 AM |
|---|---|---|
| ont color: Text 1, English (UK) | | |

| age 4: [89] Formatted | Microsoft Office User | 11/29/17 11:26:00 AM |
|---|---|---|

ont color: Text 1, English (UK)

| Page 4: [89] Formatted | Microsoft Office User | 11/29/17 11:26:00 AM |
|---|---|---|

ont color: Text 1, English (UK)

| Page 4: [89] Formatted | Microsoft Office User | 11/29/17 11:26:00 AM |
|---|---|---|

ont color: Text 1, English (UK)

| Page 4: [89] Formatted | Microsoft Office User | 11/29/17 11:26:00 AM |
|---|---|---|

ont color: Text 1, English (UK)

| Page 4: [89] Formatted | Microsoft Office User | 11/29/17 11:26:00 AM |
|---|---|---|

ont color: Text 1, English (UK)

| Page 4: [89] Formatted | Microsoft Office User | 11/29/17 11:26:00 AM |
|---|---|---|

ont color: Text 1, English (UK)

| Page 4: [89] Formatted | Microsoft Office User | 11/29/17 11:26:00 AM |
|---|---|---|

ont color: Text 1, English (UK)

| Page 4: [89] Formatted | Microsoft Office User | 11/29/17 11:26:00 AM |
|---|---|---|

ont color: Text 1, English (UK)

| Page 4: [89] Formatted | Microsoft Office User | 11/29/17 11:26:00 AM |
|---|---|---|

ont color: Text 1, English (UK)

| Page 4: [89] Formatted | Microsoft Office User | 11/29/17 11:26:00 AM |
|---|---|---|

ont color: Text 1, English (UK)

| Page 4: [89] Formatted | Microsoft Office User | 11/29/17 11:26:00 AM |
|---|---|---|

ont color: Text 1, English (UK)

| Page 4: [89] Formatted | Microsoft Office User | 11/29/17 11:26:00 AM |
|---|---|---|

ont color: Text 1, English (UK)

| Page 4: [89] Formatted | Microsoft Office User | 11/29/17 11:26:00 AM |
|---|---|---|

ont color: Text 1, English (UK)

| Page 4: [89] Formatted | Microsoft Office User | 11/29/17 11:26:00 AM |
|---|---|---|

ont color: Text 1, English (UK)

| Page 4: [89] Formatted | Microsoft Office User | 11/29/17 11:26:00 AM |
|---|---|---|

ont color: Text 1, English (UK)

| Page 4: [89] Formatted | Microsoft Office User | 11/29/17 11:26:00 AM |
|---|---|---|

ont color: Text 1, English (UK)

| Page 4: [89] Formatted | Microsoft Office User | 11/29/17 11:26:00 AM |
|---|---|---|

ont color: Text 1, English (UK)

| Page 4: [89] Formatted | Microsoft Office User | 11/29/17 11:26:00 AM |
|---|---|---|

ont color: Text 1, English (UK)

| age 4: [89] Formatted | Microsoft Office User | 11/29/17 11:26:00 AM |

ont color: Text 1, English (UK)

| age 4: [89] Formatted | Microsoft Office User | 11/29/17 11:26:00 AM |

ont color: Text 1, English (UK)

| age 4: [89] Formatted | Microsoft Office User | 11/29/17 11:26:00 AM |

ont color: Text 1, English (UK)

| age 4: [89] Formatted | Microsoft Office User | 11/29/17 11:26:00 AM |

ont color: Text 1, English (UK)

| age 4: [89] Formatted | Microsoft Office User | 11/29/17 11:26:00 AM |

ont color: Text 1, English (UK)

| age 4: [89] Formatted | Microsoft Office User | 11/29/17 11:26:00 AM |

ont color: Text 1, English (UK)

| age 4: [89] Formatted | Microsoft Office User | 11/29/17 11:26:00 AM |

ont color: Text 1, English (UK)

| age 4: [89] Formatted | Microsoft Office User | 11/29/17 11:26:00 AM |

ont color: Text 1, English (UK)

| age 4: [89] Formatted | Microsoft Office User | 11/29/17 11:26:00 AM |

ont color: Text 1, English (UK)

| age 4: [90] Formatted | Microsoft Office User | 11/29/17 11:26:00 AM |

ont color: Text 1, English (UK)

| age 4: [90] Formatted | Microsoft Office User | 11/29/17 11:26:00 AM |

ont color: Text 1, English (UK)

| age 4: [90] Formatted | Microsoft Office User | 11/29/17 11:26:00 AM |

ont color: Text 1, English (UK)

| age 4: [90] Formatted | Microsoft Office User | 11/29/17 11:26:00 AM |

ont color: Text 1, English (UK)

| age 4: [90] Formatted | Microsoft Office User | 11/29/17 11:26:00 AM |

ont color: Text 1, English (UK)

| age 4: [90] Formatted | Microsoft Office User | 11/29/17 11:26:00 AM |

ont color: Text 1, English (UK)

| age 4: [90] Formatted | Microsoft Office User | 11/29/17 11:26:00 AM |

ont color: Text 1, English (UK)

| age 4: [90] Formatted | Microsoft Office User | 11/29/17 11:26:00 AM |

ont color: Text 1, English (UK)

| age 4: [90] Formatted | Microsoft Office User | 11/29/17 11:26:00 AM |

ont color: Text 1, English (UK)

| age 4: [90] Formatted | Microsoft Office User | 11/29/17 11:26:00 AM |

ont color: Text 1, English (UK)

| Page 4: [90] Formatted | Microsoft Office User | 11/29/17 11:26:00 AM |
|---|---|---|

Font color: Text 1, English (UK)

| Page 4: [90] Formatted | Microsoft Office User | 11/29/17 11:26:00 AM |
|---|---|---|

Font color: Text 1, English (UK)

| Page 4: [90] Formatted | Microsoft Office User | 11/29/17 11:26:00 AM |
|---|---|---|

Font color: Text 1, English (UK)

| Page 4: [90] Formatted | Microsoft Office User | 11/29/17 11:26:00 AM |
|---|---|---|

Font color: Text 1, English (UK)

| Page 4: [90] Formatted | Microsoft Office User | 11/29/17 11:26:00 AM |
|---|---|---|

Font color: Text 1, English (UK)

| Page 4: [90] Formatted | Microsoft Office User | 11/29/17 11:26:00 AM |
|---|---|---|

Font color: Text 1, English (UK)

| Page 4: [90] Formatted | Microsoft Office User | 11/29/17 11:26:00 AM |
|---|---|---|

Font color: Text 1, English (UK)

| Page 4: [90] Formatted | Microsoft Office User | 11/29/17 11:26:00 AM |
|---|---|---|

Font color: Text 1, English (UK)

| Page 4: [90] Formatted | Microsoft Office User | 11/29/17 11:26:00 AM |
|---|---|---|

Font color: Text 1, English (UK)

| Page 4: [90] Formatted | Microsoft Office User | 11/29/17 11:26:00 AM |
|---|---|---|

Font color: Text 1, English (UK)

| Page 4: [90] Formatted | Microsoft Office User | 11/29/17 11:26:00 AM |
|---|---|---|

Font color: Text 1, English (UK)

| Page 4: [90] Formatted | Microsoft Office User | 11/29/17 11:26:00 AM |
|---|---|---|

Font color: Text 1, English (UK)

| Page 4: [90] Formatted | Microsoft Office User | 11/29/17 11:26:00 AM |
|---|---|---|

Font color: Text 1, English (UK)

| Page 4: [90] Formatted | Microsoft Office User | 11/29/17 11:26:00 AM |
|---|---|---|

Font color: Text 1, English (UK)

| Page 4: [90] Formatted | Microsoft Office User | 11/29/17 11:26:00 AM |
|---|---|---|

Font color: Text 1, English (UK)

| Page 4: [90] Formatted | Microsoft Office User | 11/29/17 11:26:00 AM |
|---|---|---|

Font color: Text 1, English (UK)

| Page 4: [90] Formatted | Microsoft Office User | 11/29/17 11:26:00 AM |
|---|---|---|

Font color: Text 1, English (UK)

| Page 4: [90] Formatted | Microsoft Office User | 11/29/17 11:26:00 AM |
|---|---|---|

Font color: Text 1, English (UK)

| Page 4: [90] Formatted | Microsoft Office User | 11/29/17 11:26:00 AM |
|---|---|---|

Font color: Text 1, English (UK)

| age 4: [90] Formatted | Microsoft Office User | 11/29/17 11:26:00 AM |
| --- | --- | --- |
| ont color: Text 1, English (UK) | | |
| age 4: [90] Formatted | Microsoft Office User | 11/29/17 11:26:00 AM |
| ont color: Text 1, English (UK) | | |
| age 4: [91] Deleted | Microsoft Office User | 11/29/17 11:26:00 AM |

—————————————————————Section Break (Next Page)———————————————————————

ynamics

| age 4: [92] Formatted | Microsoft Office User | 11/29/17 11:26:00 AM |
|---|---|---|
| ont color: Text 1, English (UK) | | |
| age 4: [93] Formatted | Microsoft Office User | 11/29/17 11:26:00 AM |
| ont color: Text 1, English (UK) | | |
| age 4: [93] Formatted | Microsoft Office User | 11/29/17 11:26:00 AM |
| ont color: Text 1, English (UK) | | |
| age 4: [93] Formatted | Microsoft Office User | 11/29/17 11:26:00 AM |
| ont color: Text 1, English (UK) | | |
| age 4: [93] Formatted | Microsoft Office User | 11/29/17 11:26:00 AM |
| ont color: Text 1, English (UK) | | |
| age 4: [93] Formatted | Microsoft Office User | 11/29/17 11:26:00 AM |
| ont color: Text 1, English (UK) | | |
| age 4: [93] Formatted | Microsoft Office User | 11/29/17 11:26:00 AM |
| ont color: Text 1, English (UK) | | |
| age 4: [93] Formatted | Microsoft Office User | 11/29/17 11:26:00 AM |
| ont color: Text 1, English (UK) | | |
| age 4: [93] Formatted | Microsoft Office User | 11/29/17 11:26:00 AM |
| ont color: Text 1, English (UK) | | |
| age 4: [93] Formatted | Microsoft Office User | 11/29/17 11:26:00 AM |
| ont color: Text 1, English (UK) | | |
| age 4: [93] Formatted | Microsoft Office User | 11/29/17 11:26:00 AM |
| ont color: Text 1, English (UK) | | |
| age 4: [93] Formatted | Microsoft Office User | 11/29/17 11:26:00 AM |
| ont color: Text 1, English (UK) | | |
| age 4: [93] Formatted | Microsoft Office User | 11/29/17 11:26:00 AM |
| ont color: Text 1, English (UK) | | |
| age 4: [93] Formatted | Microsoft Office User | 11/29/17 11:26:00 AM |
| ont color: Text 1, English (UK) | | |
| age 4: [93] Formatted | Microsoft Office User | 11/29/17 11:26:00 AM |
| ont color: Text 1, English (UK) | | |
| age 4: [93] Formatted | Microsoft Office User | 11/29/17 11:26:00 AM |
| ont color: Text 1, English (UK) | | |
| age 4: [93] Formatted | Microsoft Office User | 11/29/17 11:26:00 AM |
| ont color: Text 1, English (UK) | | |

| age 4: [93] Formatted | Microsoft Office User | 11/29/17 11:26:00 AM |
| --- | --- | --- |

ont color: Text 1, English (UK)

| age 4: [93] Formatted | Microsoft Office User | 11/29/17 11:26:00 AM |
| --- | --- | --- |

ont color: Text 1, English (UK)

| age 4: [93] Formatted | Microsoft Office User | 11/29/17 11:26:00 AM |
| --- | --- | --- |

ont color: Text 1, English (UK)

| age 4: [93] Formatted | Microsoft Office User | 11/29/17 11:26:00 AM |
| --- | --- | --- |

ont color: Text 1, English (UK)

| age 4: [93] Formatted | Microsoft Office User | 11/29/17 11:26:00 AM |
| --- | --- | --- |

ont color: Text 1, English (UK)

| age 4: [93] Formatted | Microsoft Office User | 11/29/17 11:26:00 AM |
| --- | --- | --- |

ont color: Text 1, English (UK)

| age 4: [93] Formatted | Microsoft Office User | 11/29/17 11:26:00 AM |
| --- | --- | --- |

ont color: Text 1, English (UK)

| age 4: [93] Formatted | Microsoft Office User | 11/29/17 11:26:00 AM |
| --- | --- | --- |

ont color: Text 1, English (UK)

| age 4: [93] Formatted | Microsoft Office User | 11/29/17 11:26:00 AM |
| --- | --- | --- |

ont color: Text 1, English (UK)

| age 4: [93] Formatted | Microsoft Office User | 11/29/17 11:26:00 AM |
| --- | --- | --- |

ont color: Text 1, English (UK)

| age 4: [93] Formatted | Microsoft Office User | 11/29/17 11:26:00 AM |
| --- | --- | --- |

ont color: Text 1, English (UK)

| age 4: [93] Formatted | Microsoft Office User | 11/29/17 11:26:00 AM |
| --- | --- | --- |

ont color: Text 1, English (UK)

| age 4: [93] Formatted | Microsoft Office User | 11/29/17 11:26:00 AM |
| --- | --- | --- |

ont color: Text 1, English (UK)

| age 4: [93] Formatted | Microsoft Office User | 11/29/17 11:26:00 AM |
| --- | --- | --- |

ont color: Text 1, English (UK)

| age 4: [93] Formatted | Microsoft Office User | 11/29/17 11:26:00 AM |
| --- | --- | --- |

ont color: Text 1, English (UK)

| age 4: [93] Formatted | Microsoft Office User | 11/29/17 11:26:00 AM |
| --- | --- | --- |

ont color: Text 1, English (UK)

| age 4: [93] Formatted | Microsoft Office User | 11/29/17 11:26:00 AM |
| --- | --- | --- |

ont color: Text 1, English (UK)

| age 4: [93] Formatted | Microsoft Office User | 11/29/17 11:26:00 AM |
| --- | --- | --- |

ont color: Text 1, English (UK)

| age 4: [93] Formatted | **Microsoft Office User** | 11/29/17 11:26:00 AM |
|---|---|---|

ont color: Text 1, English (UK)

| age 4: [93] Formatted | **Microsoft Office User** | 11/29/17 11:26:00 AM |
|---|---|---|

ont color: Text 1, English (UK)

| age 4: [93] Formatted | **Microsoft Office User** | 11/29/17 11:26:00 AM |
|---|---|---|

ont color: Text 1, English (UK)

| age 4: [93] Formatted | **Microsoft Office User** | 11/29/17 11:26:00 AM |
|---|---|---|

ont color: Text 1, English (UK)

| age 4: [93] Formatted | **Microsoft Office User** | 11/29/17 11:26:00 AM |
|---|---|---|

ont color: Text 1, English (UK)

| age 4: [93] Formatted | **Microsoft Office User** | 11/29/17 11:26:00 AM |
|---|---|---|

ont color: Text 1, English (UK)

| age 4: [93] Formatted | **Microsoft Office User** | 11/29/17 11:26:00 AM |
|---|---|---|

ont color: Text 1, English (UK)

| age 4: [93] Formatted | **Microsoft Office User** | 11/29/17 11:26:00 AM |
|---|---|---|

ont color: Text 1, English (UK)

| age 4: [93] Formatted | **Microsoft Office User** | 11/29/17 11:26:00 AM |
|---|---|---|

ont color: Text 1, English (UK)

| age 4: [93] Formatted | **Microsoft Office User** | 11/29/17 11:26:00 AM |
|---|---|---|

ont color: Text 1, English (UK)

| age 4: [93] Formatted | **Microsoft Office User** | 11/29/17 11:26:00 AM |
|---|---|---|

ont color: Text 1, English (UK)

| age 4: [93] Formatted | **Microsoft Office User** | 11/29/17 11:26:00 AM |
|---|---|---|

ont color: Text 1, English (UK)

| age 4: [93] Formatted | **Microsoft Office User** | 11/29/17 11:26:00 AM |
|---|---|---|

ont color: Text 1, English (UK)

| age 4: [93] Formatted | **Microsoft Office User** | 11/29/17 11:26:00 AM |
|---|---|---|

ont color: Text 1, English (UK)

| age 4: [93] Formatted | **Microsoft Office User** | 11/29/17 11:26:00 AM |
|---|---|---|

ont color: Text 1, English (UK)

| age 4: [93] Formatted | **Microsoft Office User** | 11/29/17 11:26:00 AM |
|---|---|---|

ont color: Text 1, English (UK)

| age 4: [93] Formatted | **Microsoft Office User** | 11/29/17 11:26:00 AM |
|---|---|---|

ont color: Text 1, English (UK)

| age 4: [93] Formatted | **Microsoft Office User** | 11/29/17 11:26:00 AM |
|---|---|---|

ont color: Text 1, English (UK)

| Page 4: [93] Formatted | Microsoft Office User | 11/29/17 11:26:00 AM |
|---|---|---|

Font color: Text 1, English (UK)

| Page 4: [93] Formatted | Microsoft Office User | 11/29/17 11:26:00 AM |
|---|---|---|

Font color: Text 1, English (UK)

| Page 4: [93] Formatted | Microsoft Office User | 11/29/17 11:26:00 AM |
|---|---|---|

Font color: Text 1, English (UK)

| Page 4: [93] Formatted | Microsoft Office User | 11/29/17 11:26:00 AM |
|---|---|---|

Font color: Text 1, English (UK)

| Page 4: [94] Formatted | Microsoft Office User | 11/29/17 11:26:00 AM |
|---|---|---|

Font color: Text 1, English (UK)

| Page 4: [95] Formatted | Microsoft Office User | 11/29/17 11:26:00 AM |
|---|---|---|

Normal, Left, Indent: First line:  0", Right:  0", Space Before:  0 pt, Line spacing:  1.5 lines

| Page 4: [96] Formatted | Microsoft Office User | 11/29/17 11:26:00 AM |
|---|---|---|

Font:10 pt, Font color: Text 1, English (UK), Subscript, Not Raised by / Lowered by

| Page 4: [97] Formatted | Microsoft Office User | 11/29/17 11:26:00 AM |
|---|---|---|

Font color: Text 1, English (UK)

| Page 4: [97] Formatted | Microsoft Office User | 11/29/17 11:26:00 AM |
|---|---|---|

Font color: Text 1, English (UK)

| Page 4: [97] Formatted | Microsoft Office User | 11/29/17 11:26:00 AM |
|---|---|---|

Font color: Text 1, English (UK)

| Page 4: [98] Deleted | Microsoft Office User | 11/29/17 11:26:00 AM |
|---|---|---|

| Page 4: [99] Formatted | Microsoft Office User | 11/29/17 11:26:00 AM |
|---|---|---|

Font:10 pt, Font color: Text 1, English (UK)

| Page 4: [99] Formatted | Microsoft Office User | 11/29/17 11:26:00 AM |
|---|---|---|

Font:10 pt, Font color: Text 1, English (UK)

| Page 4: [100] Formatted | Microsoft Office User | 11/29/17 11:26:00 AM |
|---|---|---|

Font color: Text 1, English (UK)

| Page 4: [101] Formatted | Microsoft Office User | 11/29/17 11:26:00 AM |
|---|---|---|

Font color: Text 1, English (UK)

| Page 4: [102] Deleted | Microsoft Office User | 11/29/17 11:26:00 AM |
|---|---|---|

Previous validation studies have shown quantitative comparisons of these two types of data including quantitative comparisons in the

UTLS (i.e., Livesey et al., 2008; George et al., 2009; Liu et al., 2010a, 2010b; Kroon et al., 2011;

| Page 4: [103] Formatted | Microsoft Office User | 11/29/17 11:26:00 AM |
|---|---|---|

Font color: Text 1, English (UK)

| Page 4: [104] Formatted | Microsoft Office User | 11/29/17 11:26:00 AM |
|---|---|---|

Font color: Text 1, English (UK)

| Page 4: [105] Formatted | Microsoft Office User | 11/29/17 11:26:00 AM |
|---|---|---|

Normal, Left, Indent: First line: 0", Right: 0", Line spacing: 1.5 lines

| Page 4: [106] Formatted | Microsoft Office User | 11/29/17 11:26:00 AM |
|---|---|---|

English (UK)

| Page 4: [107] Formatted | Microsoft Office User | 11/29/17 11:26:00 AM |
|---|---|---|

English (UK)

| Page 4: [108] Formatted | Microsoft Office User | 11/29/17 11:26:00 AM |
|---|---|---|

English (UK)

| Page 4: [109] Formatted | Microsoft Office User | 11/29/17 11:26:00 AM |
|---|---|---|

English (UK)

| Page 4: [110] Formatted | Microsoft Office User | 11/29/17 11:26:00 AM |
|---|---|---|

English (UK)

| Page 4: [111] Deleted | Microsoft Office User | 11/29/17 11:26:00 AM |
|---|---|---|

distributions and variabilities. The comparisons will also help to assess

| Page 4: [112] Formatted | Microsoft Office User | 11/29/17 11:26:00 AM |
|---|---|---|

English (UK)

| Page 4: [113] Deleted | Microsoft Office User | 11/29/17 11:26:00 AM |
|---|---|---|

information of the nadir viewing datasets for ASM UTLS studies, and whether they complement the information provided by the

much widely used limb sensors, especially in representing the UTLS chemical tracers

| Page 4: [114] Formatted | Microsoft Office User | 11/29/17 11:26:00 AM |
|---|---|---|

English (UK)

| Page 4: [114] Formatted | Microsoft Office User | 11/29/17 11:26:00 AM |
|---|---|---|

English (UK)

| Page 4: [115] Formatted | Microsoft Office User | 11/29/17 11:26:00 AM |
|---|---|---|

English (UK), Not Expanded by / Condensed by

| Page 4: [115] Formatted | Microsoft Office User | 11/29/17 11:26:00 AM |
|---|---|---|

English (UK), Not Expanded by / Condensed by

| Page 4: [115] Formatted | Microsoft Office User | 11/29/17 11:26:00 AM |
|---|---|---|

English (UK), Not Expanded by / Condensed by

| Page 4: [115] Formatted | Microsoft Office User | 11/29/17 11:26:00 AM |
|---|---|---|

English (UK), Not Expanded by / Condensed by

| Page 4: [116] Formatted | Microsoft Office User | 11/29/17 11:26:00 AM |
|---|---|---|

English (UK), Not Expanded by / Condensed by

| Page 4: [116] Formatted | Microsoft Office User | 11/29/17 11:26:00 AM |
|---|---|---|

English (UK), Not Expanded by / Condensed by

| Page 4: [116] Formatted | Microsoft Office User | 11/29/17 11:26:00 AM |
|---|---|---|

English (UK), Not Expanded by / Condensed by

| Page 4: [117] Formatted | Microsoft Office User | 11/29/17 11:26:00 AM |

English (UK)

| Page 4: [117] Formatted | Microsoft Office User | 11/29/17 11:26:00 AM |

English (UK)

| Page 4: [118] Formatted | Microsoft Office User | 11/29/17 11:26:00 AM |

English (UK)

| Page 4: [118] Formatted | Microsoft Office User | 11/29/17 11:26:00 AM |

English (UK)

| Page 4: [118] Formatted | Microsoft Office User | 11/29/17 11:26:00 AM |

English (UK)

| Page 4: [119] Formatted | Microsoft Office User | 11/29/17 11:26:00 AM |

English (UK)

| Page 4: [120] Formatted | Microsoft Office User | 11/29/17 11:26:00 AM |

Heading 1, Left,  No bullets or numbering, Tabs:Not at  0.61"

| Page 4: [121] Deleted | Microsoft Office User | 11/29/17 11:26:00 AM |

15 present also aim to provide a perspective of whether the high density in horizontal sampling supplement the relatively weak vertical

information content when used to inform the dynamical variability of the upper troposphere.

| Page 4: [122] Formatted | Microsoft Office User | 11/29/17 11:26:00 AM |

Not Expanded by / Condensed by

| Page 4: [123] Formatted | Microsoft Office User | 11/29/17 11:26:00 AM |

Heading 1, Left, Indent: Left:  0"

| Page 4: [124] Formatted | Microsoft Office User | 11/29/17 11:26:00 AM |

Normal, Indent: Left:  0", Line spacing:  1.5 lines, Tabs:Not at  0.47"

| Page 4: [125] Formatted | Microsoft Office User | 11/29/17 11:26:00 AM |

Font color: Text 1, English (UK)

| Page 4: [125] Formatted | Microsoft Office User | 11/29/17 11:26:00 AM |

Font color: Text 1, English (UK)

| Page 4: [126] Formatted | Microsoft Office User | 11/29/17 11:26:00 AM |

Font color: Text 1, English (UK)

| Page 4: [127] Deleted | Microsoft Office User | 11/29/17 11:26:00 AM |

0

| Page 4: [128] Formatted | Microsoft Office User | 11/29/17 11:26:00 AM |

Font:10 pt, Font color: Text 1, English (UK), Not Expanded by / Condensed by

| Page 4: [128] Formatted | Microsoft Office User | 11/29/17 11:26:00 AM |

Font:10 pt, Font color: Text 1, English (UK), Not Expanded by / Condensed by

| age 4: [129] Formatted | Microsoft Office User | 11/29/17 11:26:00 AM |
|---|---|---|

ont color: Text 1, English (UK)

| age 5: [130] Formatted | Microsoft Office User | 11/29/17 11:26:00 AM |
|---|---|---|

ont color: Text 1, English (UK), Not Expanded by / Condensed by

| age 5: [130] Formatted | Microsoft Office User | 11/29/17 11:26:00 AM |
|---|---|---|

ont color: Text 1, English (UK), Not Expanded by / Condensed by

| age 5: [130] Formatted | Microsoft Office User | 11/29/17 11:26:00 AM |
|---|---|---|

ont color: Text 1, English (UK), Not Expanded by / Condensed by

| age 5: [130] Formatted | Microsoft Office User | 11/29/17 11:26:00 AM |
|---|---|---|

ont color: Text 1, English (UK), Not Expanded by / Condensed by

| age 5: [130] Formatted | Microsoft Office User | 11/29/17 11:26:00 AM |
|---|---|---|

ont color: Text 1, English (UK), Not Expanded by / Condensed by

| age 5: [130] Formatted | Microsoft Office User | 11/29/17 11:26:00 AM |
|---|---|---|

ont color: Text 1, English (UK), Not Expanded by / Condensed by

| age 5: [130] Formatted | Microsoft Office User | 11/29/17 11:26:00 AM |
|---|---|---|

ont color: Text 1, English (UK), Not Expanded by / Condensed by

| age 5: [130] Formatted | Microsoft Office User | 11/29/17 11:26:00 AM |
|---|---|---|

ont color: Text 1, English (UK), Not Expanded by / Condensed by

| age 5: [130] Formatted | Microsoft Office User | 11/29/17 11:26:00 AM |
|---|---|---|

ont color: Text 1, English (UK), Not Expanded by / Condensed by

| age 5: [130] Formatted | Microsoft Office User | 11/29/17 11:26:00 AM |
|---|---|---|

ont color: Text 1, English (UK), Not Expanded by / Condensed by

| age 5: [130] Formatted | Microsoft Office User | 11/29/17 11:26:00 AM |
|---|---|---|

ont color: Text 1, English (UK), Not Expanded by / Condensed by

| age 5: [130] Formatted | Microsoft Office User | 11/29/17 11:26:00 AM |
|---|---|---|

ont color: Text 1, English (UK), Not Expanded by / Condensed by

| age 5: [130] Formatted | Microsoft Office User | 11/29/17 11:26:00 AM |
|---|---|---|

ont color: Text 1, English (UK), Not Expanded by / Condensed by

| age 5: [130] Formatted | Microsoft Office User | 11/29/17 11:26:00 AM |
|---|---|---|

ont color: Text 1, English (UK), Not Expanded by / Condensed by

| age 5: [131] Formatted | Microsoft Office User | 11/29/17 11:26:00 AM |
|---|---|---|

ont color: Text 1, English (UK), Not Expanded by / Condensed by

| age 5: [131] Formatted | Microsoft Office User | 11/29/17 11:26:00 AM |
|---|---|---|

ont color: Text 1, English (UK), Not Expanded by / Condensed by

| age 5: [131] Formatted | Microsoft Office User | 11/29/17 11:26:00 AM |

ont color: Text 1, English (UK), Not Expanded by / Condensed by

| age 5: [131] Formatted | Microsoft Office User | 11/29/17 11:26:00 AM |

ont color: Text 1, English (UK), Not Expanded by / Condensed by

| age 5: [131] Formatted | Microsoft Office User | 11/29/17 11:26:00 AM |

ont color: Text 1, English (UK), Not Expanded by / Condensed by

| age 5: [131] Formatted | Microsoft Office User | 11/29/17 11:26:00 AM |

ont color: Text 1, English (UK), Not Expanded by / Condensed by

| age 5: [131] Formatted | Microsoft Office User | 11/29/17 11:26:00 AM |

ont color: Text 1, English (UK), Not Expanded by / Condensed by

| age 5: [131] Formatted | Microsoft Office User | 11/29/17 11:26:00 AM |

ont color: Text 1, English (UK), Not Expanded by / Condensed by

| age 5: [131] Formatted | Microsoft Office User | 11/29/17 11:26:00 AM |

ont color: Text 1, English (UK), Not Expanded by / Condensed by

| age 5: [131] Formatted | Microsoft Office User | 11/29/17 11:26:00 AM |

ont color: Text 1, English (UK), Not Expanded by / Condensed by

| age 5: [131] Formatted | Microsoft Office User | 11/29/17 11:26:00 AM |

ont color: Text 1, English (UK), Not Expanded by / Condensed by

| age 5: [131] Formatted | Microsoft Office User | 11/29/17 11:26:00 AM |

ont color: Text 1, English (UK), Not Expanded by / Condensed by

| age 5: [132] Formatted | Microsoft Office User | 11/29/17 11:26:00 AM |

ont color: Text 1, English (UK), Not Expanded by / Condensed by

| age 5: [132] Formatted | Microsoft Office User | 11/29/17 11:26:00 AM |

ont color: Text 1, English (UK), Not Expanded by / Condensed by

| age 5: [132] Formatted | Microsoft Office User | 11/29/17 11:26:00 AM |

ont color: Text 1, English (UK), Not Expanded by / Condensed by

| age 5: [132] Formatted | Microsoft Office User | 11/29/17 11:26:00 AM |

ont color: Text 1, English (UK), Not Expanded by / Condensed by

| age 5: [132] Formatted | Microsoft Office User | 11/29/17 11:26:00 AM |

ont color: Text 1, English (UK), Not Expanded by / Condensed by

| age 5: [132] Formatted | Microsoft Office User | 11/29/17 11:26:00 AM |

ont color: Text 1, English (UK), Not Expanded by / Condensed by

| age 5: [132] Formatted | Microsoft Office User | 11/29/17 11:26:00 AM |

ont color: Text 1, English (UK), Not Expanded by / Condensed by

| age 5: [132] Formatted | Microsoft Office User | 11/29/17 11:26:00 AM |

ont color: Text 1, English (UK), Not Expanded by / Condensed by

| age 5: [132] Formatted | Microsoft Office User | 11/29/17 11:26:00 AM |

ont color: Text 1, English (UK), Not Expanded by / Condensed by

| age 5: [132] Formatted | Microsoft Office User | 11/29/17 11:26:00 AM |
| --- | --- | --- |

ont color: Text 1, English (UK), Not Expanded by / Condensed by

| age 5: [132] Formatted | Microsoft Office User | 11/29/17 11:26:00 AM |
| --- | --- | --- |

ont color: Text 1, English (UK), Not Expanded by / Condensed by

| age 5: [132] Formatted | Microsoft Office User | 11/29/17 11:26:00 AM |
| --- | --- | --- |

ont color: Text 1, English (UK), Not Expanded by / Condensed by

| age 5: [132] Formatted | Microsoft Office User | 11/29/17 11:26:00 AM |
| --- | --- | --- |

ont color: Text 1, English (UK), Not Expanded by / Condensed by

| age 5: [132] Formatted | Microsoft Office User | 11/29/17 11:26:00 AM |
| --- | --- | --- |

ont color: Text 1, English (UK), Not Expanded by / Condensed by

| age 5: [132] Formatted | Microsoft Office User | 11/29/17 11:26:00 AM |
| --- | --- | --- |

ont color: Text 1, English (UK), Not Expanded by / Condensed by

| age 5: [132] Formatted | Microsoft Office User | 11/29/17 11:26:00 AM |
| --- | --- | --- |

ont color: Text 1, English (UK), Not Expanded by / Condensed by

| age 5: [133] Formatted | Microsoft Office User | 11/29/17 11:26:00 AM |
| --- | --- | --- |

ont color: Text 1, English (UK)

| age 5: [133] Formatted | Microsoft Office User | 11/29/17 11:26:00 AM |
| --- | --- | --- |

ont color: Text 1, English (UK)

| age 5: [134] Deleted | Microsoft Office User | 11/29/17 11:26:00 AM |
| --- | --- | --- |

| age 5: [135] Formatted | Microsoft Office User | 11/29/17 11:26:00 AM |
| --- | --- | --- |

ont color: Text 1, English (UK)

| age 5: [135] Formatted | Microsoft Office User | 11/29/17 11:26:00 AM |
| --- | --- | --- |

ont color: Text 1, English (UK)

| age 5: [135] Formatted | Microsoft Office User | 11/29/17 11:26:00 AM |
| --- | --- | --- |

ont color: Text 1, English (UK)

| age 6: [136] Deleted | Microsoft Office User | 11/29/17 11:26:00 AM |
| --- | --- | --- |

Nadir-viewing observations of CO are obtained from IASI (level 2 data) aboard EUMETSAT's Metop satellite. IASI measures the „thermal infrared" (TIR) spectrum emitted by the Earth-atmosphere system with twice daily near-global coverage (with 4 simultaneous pixels of 12 km diameter every 50 km), but limited vertical resolution (Clerbaux et al., 2009). The CO tropospheric abundance product is derived from the spectra using FORLI retrieval algorithm using a single a priori profile and

30   covariance matrix (Hurtmans et al., 2012; George et al., 2015). The IASI CO retrieval, however, contains 0.8 to 2.4 (1.5 to 2.0 at mid-latitudes) „independent pieces of information" (or degrees of freedom signal DOFS; George et al. 2009). This information content allows IASI CO retrieval to capture upper tropospheric variability at mid-latitude and tropical   latitudes,

———Section Break (Next Page)———

which is supported by in situ measurements from MOZAIC project (correlations ~ 0.7; Wachter et al., 2012), satellite observations from the MOPITT instrument (George et al., 2015). IASI data have been shown to reproduce monthly mean large-scale features in the UTLS over the ASM region from GEOS-Chem (a chemical transport model coupled to meteorological analysis from the Goddard Earth Observing System GEOS-5; Barret et al., 2016).

5 Nadir-viewing observations of $O_3$ are obtained from OMI, an $O_3$ sounder aboard the Aura satellite that provides daily global coverage at 13 km x 24 km footprint (Levelt et al., 2006). OMI $O_3$ products include retrievals of both total $O_3$ columns and vertical profiles. In this study, we use the $O_3$ profile product by Liu et al. (2010b) and Huang et al. (2016). $O_3$ profiles are retrieved at 24 vertical layers covering the surface to ~60 km using the optimal estimation technique constrained by a monthly and zonal mean $O_3$ profile climatology (McPeters et al., 2007). NCEP reanalysis tropopause pressure is used to separate the

10 stratosphere from the troposphere. To speed up the processing, the product is produced at a nadir spatial resolution of 52 km x 48 km by combining 4 pixels along the track. The retrievals have ~6.0-7.0 degrees of freedom (5.0-6.7 in the stratosphere; Liu et al., 2005; Liu et al., 2010b; Liu et al., 2010a). Although the vertical resolution of OMI $O_3$ retrievals in the troposphere is about 10 km (Liu et al., 2010b), the profile product has been shown to have significant and useful information in the UTLS transition region due to the large $O_3$ gradient across the tropopause and good stratospheric information content (Pittman et al.,

15 2009; Liu et al., 2010a; Liu et al., 2010b; Bak et al., 2013). In this work, we use a level-3 product gridded to 1° longitude x 1° latitude horizontal resolution. OMI has known cross-track dependent biases (Liu et al., 2010a; Liu et al., 2010b). Thus, we smoothed $O_3$ profiles by adjacent data which view zenith angles (VZA) is greater than 58°.

Note that the CO and $O_3$ data examined in this work are from different years. The CO data are from June-July-August (JJA) of 2012. 2012 was a standard year for IASI with no change in the L2 temperature data (which might induce jumps in the data

20 series). The $O_3$ are from JJA 2008, the season before the OMI instrument "row anomaly" in January 2009 which impacted the $O_3$ data quality since then (Huang et al., 2016; Huang, submitted). Since the $O_3$ and CO are examined separately and the work focuses on daily to sub-seasonal scale space-time variabilities, the choice of different years does not impact the analyses and the conclusions.

| Page 6: [137] Formatted | Microsoft Office User | 11/29/17 11:26:00 AM |
|---|---|---|

Normal, Left, Indent: First line: 0", Right: 0", Space Before: 0 pt, Line spacing: 1.5 lines

| Page 6: [138] Formatted | Microsoft Office User | 11/29/17 11:26:00 AM |
|---|---|---|

Font color: Text 1, English (UK)

| Page 6: [139] Deleted | Microsoft Office User | 11/29/17 11:26:00 AM |
|---|---|---|

| Page 6: [140] Formatted | Microsoft Office User | 11/29/17 11:26:00 AM |
|---|---|---|

Font:10 pt, Font color: Text 1, English (UK)

| Page 6: [140] Formatted | Microsoft Office User | 11/29/17 11:26:00 AM |
|---|---|---|

Font:10 pt, Font color: Text 1, English (UK)

| Page 6: [141] Formatted | Microsoft Office User | 11/29/17 11:26:00 AM |
|---|---|---|

ont color: Text 1, English (UK)

| age 6: [142] Formatted | Microsoft Office User | 11/29/17 11:26:00 AM |
|---|---|---|

ont color: Text 1, English (UK)

| age 6: [143] Formatted | Microsoft Office User | 11/29/17 11:26:00 AM |
|---|---|---|

ont color: Text 1, English (UK)

| age 6: [144] Formatted | Microsoft Office User | 11/29/17 11:26:00 AM |
|---|---|---|

ont color: Text 1, English (UK)

| age 6: [145] Formatted | Microsoft Office User | 11/29/17 11:26:00 AM |
|---|---|---|

ont color: Text 1, English (UK)

| age 6: [146] Formatted | Microsoft Office User | 11/29/17 11:26:00 AM |
|---|---|---|

ont color: Text 1, English (UK)

| age 6: [147] Formatted | Microsoft Office User | 11/29/17 11:26:00 AM |
|---|---|---|

ont color: Text 1, English (UK)

| age 6: [147] Formatted | Microsoft Office User | 11/29/17 11:26:00 AM |
|---|---|---|

ont color: Text 1, English (UK)

| age 6: [147] Formatted | Microsoft Office User | 11/29/17 11:26:00 AM |
|---|---|---|

ont color: Text 1, English (UK)

| age 6: [148] Formatted | Microsoft Office User | 11/29/17 11:26:00 AM |
|---|---|---|

ont color: Text 1, English (UK)

| age 6: [148] Formatted | Microsoft Office User | 11/29/17 11:26:00 AM |
|---|---|---|

ont color: Text 1, English (UK)

| age 6: [148] Formatted | Microsoft Office User | 11/29/17 11:26:00 AM |
|---|---|---|

ont color: Text 1, English (UK)

| age 6: [148] Formatted | Microsoft Office User | 11/29/17 11:26:00 AM |
|---|---|---|

ont color: Text 1, English (UK)

| age 6: [148] Formatted | Microsoft Office User | 11/29/17 11:26:00 AM |
|---|---|---|

ont color: Text 1, English (UK)

| age 6: [148] Formatted | Microsoft Office User | 11/29/17 11:26:00 AM |
|---|---|---|

ont color: Text 1, English (UK)

| age 6: [149] Formatted | Microsoft Office User | 11/29/17 11:26:00 AM |
|---|---|---|

ont color: Text 1, English (UK)

| age 6: [149] Formatted | Microsoft Office User | 11/29/17 11:26:00 AM |
|---|---|---|

ont color: Text 1, English (UK)

| age 6: [149] Formatted | Microsoft Office User | 11/29/17 11:26:00 AM |
|---|---|---|

ont color: Text 1, English (UK)

| age 6: [149] Formatted | Microsoft Office User | 11/29/17 11:26:00 AM |
|---|---|---|

ont color: Text 1, English (UK)

| age 6: [149] Formatted | Microsoft Office User | 11/29/17 11:26:00 AM |
|---|---|---|

ont color: Text 1, English (UK)

| Page 6: [149] Formatted | Microsoft Office User | 11/29/17 11:26:00 AM |
| --- | --- | --- |

Font color: Text 1, English (UK)

| Page 6: [149] Formatted | Microsoft Office User | 11/29/17 11:26:00 AM |
| --- | --- | --- |

Font color: Text 1, English (UK)

| Page 6: [149] Formatted | Microsoft Office User | 11/29/17 11:26:00 AM |
| --- | --- | --- |

Font color: Text 1, English (UK)

| Page 6: [149] Formatted | Microsoft Office User | 11/29/17 11:26:00 AM |
| --- | --- | --- |

Font color: Text 1, English (UK)

| Page 6: [149] Formatted | Microsoft Office User | 11/29/17 11:26:00 AM |
| --- | --- | --- |

Font color: Text 1, English (UK)

| Page 6: [149] Formatted | Microsoft Office User | 11/29/17 11:26:00 AM |
| --- | --- | --- |

Font color: Text 1, English (UK)

| Page 6: [149] Formatted | Microsoft Office User | 11/29/17 11:26:00 AM |
| --- | --- | --- |

Font color: Text 1, English (UK)

| Page 6: [149] Formatted | Microsoft Office User | 11/29/17 11:26:00 AM |
| --- | --- | --- |

Font color: Text 1, English (UK)

| Page 6: [149] Formatted | Microsoft Office User | 11/29/17 11:26:00 AM |
| --- | --- | --- |

Font color: Text 1, English (UK)

| Page 6: [149] Formatted | Microsoft Office User | 11/29/17 11:26:00 AM |
| --- | --- | --- |

Font color: Text 1, English (UK)

| Page 6: [149] Formatted | Microsoft Office User | 11/29/17 11:26:00 AM |
| --- | --- | --- |

Font color: Text 1, English (UK)

| Page 6: [149] Formatted | Microsoft Office User | 11/29/17 11:26:00 AM |
| --- | --- | --- |

Font color: Text 1, English (UK)

| Page 6: [149] Formatted | Microsoft Office User | 11/29/17 11:26:00 AM |
| --- | --- | --- |

Font color: Text 1, English (UK)

| Page 6: [150] Deleted | Microsoft Office User | 11/29/17 11:26:00 AM |
| --- | --- | --- |

me scales. Figure 1b shows a vertical information distribution comparison between IASI and MLS CO retrievals. For the interest of

his analyses, the IASI

| Page 6: [151] Formatted | Microsoft Office User | 11/29/17 11:26:00 AM |
| --- | --- | --- |

Font color: Text 1, English (UK)

| Page 6: [152] Formatted | Microsoft Office User | 11/29/17 11:26:00 AM |
| --- | --- | --- |

Normal, Left, Right:  0", Space Before:  0 pt

| Page 6: [153] Formatted | Microsoft Office User | 11/29/17 11:26:00 AM |
| --- | --- | --- |

Font color: Text 1, English (UK)

| Page 6: [154] Formatted | Microsoft Office User | 11/29/17 11:26:00 AM |
| --- | --- | --- |

Font color: Text 1, English (UK)

| Page 6: [155] Formatted | Microsoft Office User | 11/29/17 11:26:00 AM |
|---|---|---|

Font color: Text 1, English (UK)

| Page 6: [156] Formatted | Microsoft Office User | 11/29/17 11:26:00 AM |
|---|---|---|

Font color: Text 1, English (UK)

| Page 6: [157] Formatted | Microsoft Office User | 11/29/17 11:26:00 AM |
|---|---|---|

Font color: Text 1, English (UK)

| Page 6: [157] Formatted | Microsoft Office User | 11/29/17 11:26:00 AM |
|---|---|---|

Font color: Text 1, English (UK)

| Page 6: [158] Deleted | Microsoft Office User | 11/29/17 11:26:00 AM |
|---|---|---|

n the ASM region (30°E-140°E longitudes and 15°N-35°N latitudes) for a single day 1 June 2012. MLS

| Page 6: [159] Formatted | Microsoft Office User | 11/29/17 11:26:00 AM |
|---|---|---|

Font color: Text 1, English (UK)

| Page 6: [159] Formatted | Microsoft Office User | 11/29/17 11:26:00 AM |
|---|---|---|

Font color: Text 1, English (UK)

| Page 6: [160] Formatted | Microsoft Office User | 11/29/17 11:26:00 AM |
|---|---|---|

Font color: Text 1, English (UK)

| Page 6: [160] Formatted | Microsoft Office User | 11/29/17 11:26:00 AM |
|---|---|---|

Font color: Text 1, English (UK)

| Page 6: [161] Deleted | Microsoft Office User | 11/29/17 11:26:00 AM |
|---|---|---|

roduct (see MLS version 4.2x level 2 data quality description document, https://mls.jpl.nasa.gov/data/v4-

_data_quality_document.pdf).

| Page 6: [162] Formatted | Microsoft Office User | 11/29/17 11:26:00 AM |
|---|---|---|

ont:10 pt, Font color: Text 1, English (UK)

| Page 6: [162] Formatted | Microsoft Office User | 11/29/17 11:26:00 AM |
|---|---|---|

ont:10 pt, Font color: Text 1, English (UK)

| Page 6: [163] Formatted | Microsoft Office User | 11/29/17 11:26:00 AM |
|---|---|---|

ont color: Text 1, English (UK), Not Expanded by / Condensed by

| Page 6: [163] Formatted | Microsoft Office User | 11/29/17 11:26:00 AM |
|---|---|---|

ont color: Text 1, English (UK), Not Expanded by / Condensed by

| Page 6: [164] Deleted | Microsoft Office User | 11/29/17 11:26:00 AM |
|---|---|---|

Section Break (Next Page)

| age 6: [165] Formatted | Microsoft Office User | 11/29/17 11:26:00 AM |
| --- | --- | --- |

ont color: Text 1, English (UK)

| age 6: [166] Formatted | Microsoft Office User | 11/29/17 11:26:00 AM |
| --- | --- | --- |

ont color: Text 1, English (UK)

| age 6: [166] Formatted | Microsoft Office User | 11/29/17 11:26:00 AM |
| --- | --- | --- |

ont color: Text 1, English (UK)

| age 6: [167] Formatted | Microsoft Office User | 11/29/17 11:26:00 AM |
| --- | --- | --- |

ont color: Text 1, English (UK)

| age 6: [168] Deleted | Microsoft Office User | 11/29/17 11:26:00 AM |
| --- | --- | --- |

6 km (full-width at half maximum). This figure highlights the differences specific to questions of interest in this work.

| age 6: [169] Formatted | Microsoft Office User | 11/29/17 11:26:00 AM |
| --- | --- | --- |

ont color: Text 1, English (UK)

| age 6: [169] Formatted | Microsoft Office User | 11/29/17 11:26:00 AM |
| --- | --- | --- |

ont color: Text 1, English (UK)

| age 6: [170] Formatted | Microsoft Office User | 11/29/17 11:26:00 AM |
| --- | --- | --- |

ont color: Text 1, English (UK)

| age 6: [170] Formatted | Microsoft Office User | 11/29/17 11:26:00 AM |
| --- | --- | --- |

ont color: Text 1, English (UK)

| age 6: [170] Formatted | Microsoft Office User | 11/29/17 11:26:00 AM |
| --- | --- | --- |

ont color: Text 1, English (UK)

| age 6: [170] Formatted | Microsoft Office User | 11/29/17 11:26:00 AM |
| --- | --- | --- |

ont color: Text 1, English (UK)

| age 6: [170] Formatted | Microsoft Office User | 11/29/17 11:26:00 AM |
| --- | --- | --- |

ont color: Text 1, English (UK)

| age 6: [171] Deleted | Microsoft Office User | 11/29/17 11:26:00 AM |
| --- | --- | --- |

| age 6: [172] Formatted | Microsoft Office User | 11/29/17 11:26:00 AM |
| --- | --- | --- |

ndent: Left:  0", Space Before:  0 pt, Tabs:Not at  0.47"

| age 6: [173] Formatted | Microsoft Office User | 11/29/17 11:26:00 AM |
| --- | --- | --- |

Not Expanded by / Condensed by

| age 6: [173] Formatted | Microsoft Office User | 11/29/17 11:26:00 AM |
| --- | --- | --- |

Not Expanded by / Condensed by

| age 6: [174] Formatted | Microsoft Office User | 11/29/17 11:26:00 AM |
| --- | --- | --- |

Normal, Left, Right:  0", Space Before:  0 pt

| age 6: [175] Formatted | Microsoft Office User | 11/29/17 11:26:00 AM |
|---|---|---|

ont color: Text 1, English (UK)

| age 6: [175] Formatted | Microsoft Office User | 11/29/17 11:26:00 AM |
|---|---|---|

ont color: Text 1, English (UK)

| age 6: [175] Formatted | Microsoft Office User | 11/29/17 11:26:00 AM |
|---|---|---|

ont color: Text 1, English (UK)

| age 6: [175] Formatted | Microsoft Office User | 11/29/17 11:26:00 AM |
|---|---|---|

ont color: Text 1, English (UK)

| age 6: [175] Formatted | Microsoft Office User | 11/29/17 11:26:00 AM |
|---|---|---|

ont color: Text 1, English (UK)

| age 6: [175] Formatted | Microsoft Office User | 11/29/17 11:26:00 AM |
|---|---|---|

ont color: Text 1, English (UK)

| age 6: [175] Formatted | Microsoft Office User | 11/29/17 11:26:00 AM |
|---|---|---|

ont color: Text 1, English (UK)

| age 6: [175] Formatted | Microsoft Office User | 11/29/17 11:26:00 AM |
|---|---|---|

ont color: Text 1, English (UK)

| age 6: [175] Formatted | Microsoft Office User | 11/29/17 11:26:00 AM |
|---|---|---|

ont color: Text 1, English (UK)

| age 6: [175] Formatted | Microsoft Office User | 11/29/17 11:26:00 AM |
|---|---|---|

ont color: Text 1, English (UK)

| age 6: [175] Formatted | Microsoft Office User | 11/29/17 11:26:00 AM |
|---|---|---|

ont color: Text 1, English (UK)

| age 6: [175] Formatted | Microsoft Office User | 11/29/17 11:26:00 AM |
|---|---|---|

ont color: Text 1, English (UK)

| age 6: [175] Formatted | Microsoft Office User | 11/29/17 11:26:00 AM |
|---|---|---|

ont color: Text 1, English (UK)

| age 6: [175] Formatted | Microsoft Office User | 11/29/17 11:26:00 AM |
|---|---|---|

ont color: Text 1, English (UK)

| age 6: [175] Formatted | Microsoft Office User | 11/29/17 11:26:00 AM |
|---|---|---|

ont color: Text 1, English (UK)

| age 6: [175] Formatted | Microsoft Office User | 11/29/17 11:26:00 AM |
|---|---|---|

ont color: Text 1, English (UK)

| age 6: [175] Formatted | Microsoft Office User | 11/29/17 11:26:00 AM |
|---|---|---|

ont color: Text 1, English (UK)

| age 6: [175] Formatted | Microsoft Office User | 11/29/17 11:26:00 AM |
|---|---|---|

ont color: Text 1, English (UK)

| age 6: [175] Formatted | Microsoft Office User | 11/29/17 11:26:00 AM |
|---|---|---|

ont color: Text 1, English (UK)

| age 6: [175] Formatted | Microsoft Office User | 11/29/17 11:26:00 AM |
|---|---|---|

ont color: Text 1, English (UK)

| age 6: [175] Formatted | Microsoft Office User | 11/29/17 11:26:00 AM |
|---|---|---|

ont color: Text 1, English (UK)

| age 1: [176] Deleted | Microsoft Office User | 11/29/17 11:26:00 AM |
|---|---|---|
| age 7: [177] Formatted | Microsoft Office User | 11/29/17 11:26:00 AM |
|---|---|---|

ont color: Text 1, English (UK)

| age 7: [178] Formatted | Microsoft Office User | 11/29/17 11:26:00 AM |
|---|---|---|

ont color: Text 1, English (UK), Not Expanded by / Condensed by

| age 7: [179] Formatted | Microsoft Office User | 11/29/17 11:26:00 AM |
|---|---|---|

ont color: Text 1, English (UK)

| age 7: [180] Formatted | Microsoft Office User | 11/29/17 11:26:00 AM |
|---|---|---|

ont color: Text 1, English (UK), Not Expanded by / Condensed by

| age 7: [181] Formatted | Microsoft Office User | 11/29/17 11:26:00 AM |
|---|---|---|

ont color: Text 1, English (UK)

| age 7: [182] Formatted | Microsoft Office User | 11/29/17 11:26:00 AM |
|---|---|---|

ont color: Text 1, English (UK), Not Expanded by / Condensed by

| age 7: [183] Formatted | Microsoft Office User | 11/29/17 11:26:00 AM |
|---|---|---|

ont color: Text 1, English (UK)

| age 7: [184] Formatted | Microsoft Office User | 11/29/17 11:26:00 AM |
|---|---|---|

ont color: Text 1, English (UK), Not Expanded by / Condensed by

| age 7: [185] Formatted | Microsoft Office User | 11/29/17 11:26:00 AM |
|---|---|---|

ont color: Text 1, English (UK)

| age 7: [186] Formatted | Microsoft Office User | 11/29/17 11:26:00 AM |
|---|---|---|

ont color: Text 1, English (UK), Not Expanded by / Condensed by

| age 7: [187] Formatted | Microsoft Office User | 11/29/17 11:26:00 AM |
|---|---|---|

ont color: Text 1, English (UK)

| age 7: [188] Formatted | Microsoft Office User | 11/29/17 11:26:00 AM |
|---|---|---|

ont color: Text 1, English (UK), Not Expanded by / Condensed by

| age 7: [189] Formatted | Microsoft Office User | 11/29/17 11:26:00 AM |
|---|---|---|

ont color: Text 1, English (UK)

| age 7: [190] Formatted | Microsoft Office User | 11/29/17 11:26:00 AM |
|---|---|---|

ont color: Text 1, English (UK), Not Expanded by / Condensed by

| age 7: [191] Formatted | Microsoft Office User | 11/29/17 11:26:00 AM |
|---|---|---|

ont color: Text 1, English (UK)

| age 7: [192] Formatted | Microsoft Office User | 11/29/17 11:26:00 AM |
|---|---|---|

ont color: Text 1, English (UK), Not Expanded by / Condensed by

| age 7: [193] Formatted | Microsoft Office User | 11/29/17 11:26:00 AM |
|---|---|---|

ont color: Text 1, English (UK)

| age 7: [194] Formatted | Microsoft Office User | 11/29/17 11:26:00 AM |
|---|---|---|

ont color: Text 1, English (UK), Not Expanded by / Condensed by

| age 7: [195] Formatted | Microsoft Office User | 11/29/17 11:26:00 AM |
|---|---|---|

ont color: Text 1, English (UK)

| age 7: [196] Formatted | Microsoft Office User | 11/29/17 11:26:00 AM |
|---|---|---|

ont color: Text 1, English (UK), Not Expanded by / Condensed by

| age 7: [197] Formatted | Microsoft Office User | 11/29/17 11:26:00 AM |
|---|---|---|

ont color: Text 1, English (UK)

| age 7: [198] Formatted | Microsoft Office User | 11/29/17 11:26:00 AM |
|---|---|---|

ont color: Text 1, English (UK), Not Expanded by / Condensed by

| age 7: [199] Formatted | Microsoft Office User | 11/29/17 11:26:00 AM |
|---|---|---|

ont color: Text 1, English (UK)

| age 7: [200] Formatted | Microsoft Office User | 11/29/17 11:26:00 AM |
|---|---|---|

ont color: Text 1, English (UK), Not Expanded by / Condensed by

| age 7: [201] Formatted | Microsoft Office User | 11/29/17 11:26:00 AM |
|---|---|---|

ont color: Text 1, English (UK)

| age 7: [202] Formatted | Microsoft Office User | 11/29/17 11:26:00 AM |
|---|---|---|

ont color: Text 1, English (UK), Not Expanded by / Condensed by

| age 7: [203] Formatted | Microsoft Office User | 11/29/17 11:26:00 AM |
|---|---|---|

ont color: Text 1, English (UK)

| age 7: [204] Formatted | Microsoft Office User | 11/29/17 11:26:00 AM |
|---|---|---|

ont color: Text 1, English (UK), Not Expanded by / Condensed by

| age 7: [205] Formatted | Microsoft Office User | 11/29/17 11:26:00 AM |
|---|---|---|

ont color: Text 1, English (UK)

| age 7: [206] Formatted | Microsoft Office User | 11/29/17 11:26:00 AM |
|---|---|---|

ont color: Text 1, English (UK), Not Expanded by / Condensed by

| age 7: [207] Formatted | Microsoft Office User | 11/29/17 11:26:00 AM |
|---|---|---|

ont color: Text 1, English (UK)

| age 7: [208] Formatted | Microsoft Office User | 11/29/17 11:26:00 AM |
|---|---|---|

ont color: Text 1, English (UK), Not Expanded by / Condensed by

| Page 7: [209] Formatted | Microsoft Office User | 11/29/17 11:26:00 AM |
| --- | --- | --- |

ont color: Text 1, English (UK)

| Page 7: [210] Formatted | Microsoft Office User | 11/29/17 11:26:00 AM |
| --- | --- | --- |

ont color: Text 1, English (UK), Not Expanded by / Condensed by

| Page 7: [211] Formatted | Microsoft Office User | 11/29/17 11:26:00 AM |
| --- | --- | --- |

ont color: Text 1, English (UK)

| Page 7: [212] Formatted | Microsoft Office User | 11/29/17 11:26:00 AM |
| --- | --- | --- |

ont color: Text 1, English (UK), Not Expanded by / Condensed by

| Page 7: [213] Formatted | Microsoft Office User | 11/29/17 11:26:00 AM |
| --- | --- | --- |

ont color: Text 1, English (UK)

| Page 7: [214] Formatted | Microsoft Office User | 11/29/17 11:26:00 AM |
| --- | --- | --- |

ont color: Text 1, English (UK), Not Expanded by / Condensed by

| Page 7: [215] Formatted | Microsoft Office User | 11/29/17 11:26:00 AM |
| --- | --- | --- |

ont color: Text 1, English (UK)

| Page 7: [216] Formatted | Microsoft Office User | 11/29/17 11:26:00 AM |
| --- | --- | --- |

ont color: Text 1, English (UK), Not Expanded by / Condensed by

| Page 7: [217] Formatted | Microsoft Office User | 11/29/17 11:26:00 AM |
| --- | --- | --- |

ont color: Text 1, English (UK)

| Page 7: [218] Formatted | Microsoft Office User | 11/29/17 11:26:00 AM |
| --- | --- | --- |

ont color: Text 1, English (UK), Not Expanded by / Condensed by

| Page 7: [219] Formatted | Microsoft Office User | 11/29/17 11:26:00 AM |
| --- | --- | --- |

ont color: Text 1, English (UK)

| Page 7: [220] Formatted | Microsoft Office User | 11/29/17 11:26:00 AM |
| --- | --- | --- |

ont color: Text 1, English (UK), Not Expanded by / Condensed by

| Page 7: [221] Formatted | Microsoft Office User | 11/29/17 11:26:00 AM |
| --- | --- | --- |

ont color: Text 1, English (UK)

| Page 7: [222] Formatted | Microsoft Office User | 11/29/17 11:26:00 AM |
| --- | --- | --- |

ont color: Text 1, English (UK)

| Page 8: [223] Formatted | Microsoft Office User | 11/29/17 11:26:00 AM |
| --- | --- | --- |

Heading 1, Space Before:  0 pt,  No bullets or numbering, Tabs:Not at  0.61"

| Page 8: [224] Deleted | Microsoft Office User | 11/29/17 11:26:00 AM |
| --- | --- | --- |

10     Service/NOAA/U.S. Department of Commerce, 2000).

| Page 8: [224] Deleted | Microsoft Office User | 11/29/17 11:26:00 AM |
| --- | --- | --- |

10     Service/NOAA/U.S. Department of Commerce, 2000).

| Page 8: [224] Deleted | Microsoft Office User | 11/29/17 11:26:00 AM |
| --- | --- | --- |

10    Service/NOAA/U.S. Department of Commerce, 2000).

| Page 8: [225] Formatted | Microsoft Office User | 11/29/17 11:26:00 AM |

Normal, Left, Indent: First line:  0", Right:  0", Line spacing:  1.5 lines

| Page 8: [226] Deleted | Microsoft Office User | 11/29/17 11:26:00 AM |

Previous studies have shown that, during

| Page 8: [227] Formatted | Microsoft Office User | 11/29/17 11:26:00 AM |

Font color: Text 1, English (UK)

| Page 8: [228] Deleted | Microsoft Office User | 11/29/17 11:26:00 AM |

monsoon season (June-July-August, JJA), the seasonal mean ASM anticyclone has a pronounced

| Page 8: [229] Formatted | Microsoft Office User | 11/29/17 11:26:00 AM |

Font color: Text 1, English (UK)

| Page 8: [230] Deleted | Microsoft Office User | 11/29/17 11:26:00 AM |

signature with enhanced (diminished) mixing ratios of tropospheric (stratospheric) tracers such as CO ($O_3$) (Randel and Park,

2006; Park et al., 2007; Park et al., 2008; Randel et al., 2010;    Pan et al., 2016). To help identify

5    tracers"

| Page 8: [231] Formatted | Microsoft Office User | 11/29/17 11:26:00 AM |

Font color: Text 1, English (UK)

| Page 8: [232] Deleted | Microsoft Office User | 11/29/17 11:26:00 AM |

in relation to the anticyclone from the selected satellite

| Page 8: [233] Formatted | Microsoft Office User | 11/29/17 11:26:00 AM |

Font color: Text 1, English (UK)

| Page 8: [234] Deleted | Microsoft Office User | 11/29/17 11:26:00 AM |

sources, we first assess the ability of
the nadir-viewing dataset

| Page 8: [235] Formatted | Microsoft Office User | 11/29/17 11:26:00 AM |

Font color: Text 1, English (UK)

| Page 8: [235] Formatted | Microsoft Office User | 11/29/17 11:26:00 AM |

Font color: Text 1, English (UK)

| Page 8: [236] Formatted | Microsoft Office User | 11/29/17 11:26:00 AM |

Font color: Text 1, English (UK)

| Page 8: [237] Formatted | Microsoft Office User | 11/29/17 11:26:00 AM |

Font color: Text 1, English (UK)

| Page 8: [237] Formatted | Microsoft Office User | 11/29/17 11:26:00 AM |

Font color: Text 1, English (UK)

| age 8: [238] Formatted | Microsoft Office User | 11/29/17 11:26:00 AM |
|---|---|---|

ont color: Text 1, English (UK)

| age 8: [238] Formatted | Microsoft Office User | 11/29/17 11:26:00 AM |
|---|---|---|

ont color: Text 1, English (UK)

| age 8: [239] Formatted | Microsoft Office User | 11/29/17 11:26:00 AM |
|---|---|---|

ont color: Text 1, English (UK), Not Expanded by / Condensed by

| age 8: [239] Formatted | Microsoft Office User | 11/29/17 11:26:00 AM |
|---|---|---|

ont color: Text 1, English (UK), Not Expanded by / Condensed by

| age 8: [239] Formatted | Microsoft Office User | 11/29/17 11:26:00 AM |
|---|---|---|

ont color: Text 1, English (UK), Not Expanded by / Condensed by

| age 8: [239] Formatted | Microsoft Office User | 11/29/17 11:26:00 AM |
|---|---|---|

ont color: Text 1, English (UK), Not Expanded by / Condensed by

| age 8: [239] Formatted | Microsoft Office User | 11/29/17 11:26:00 AM |
|---|---|---|

ont color: Text 1, English (UK), Not Expanded by / Condensed by

| age 8: [239] Formatted | Microsoft Office User | 11/29/17 11:26:00 AM |
|---|---|---|

ont color: Text 1, English (UK), Not Expanded by / Condensed by

| age 8: [239] Formatted | Microsoft Office User | 11/29/17 11:26:00 AM |
|---|---|---|

ont color: Text 1, English (UK), Not Expanded by / Condensed by

| age 8: [239] Formatted | Microsoft Office User | 11/29/17 11:26:00 AM |
|---|---|---|

ont color: Text 1, English (UK), Not Expanded by / Condensed by

| age 8: [239] Formatted | Microsoft Office User | 11/29/17 11:26:00 AM |
|---|---|---|

ont color: Text 1, English (UK), Not Expanded by / Condensed by

| age 8: [239] Formatted | Microsoft Office User | 11/29/17 11:26:00 AM |
|---|---|---|

ont color: Text 1, English (UK), Not Expanded by / Condensed by

| age 8: [239] Formatted | Microsoft Office User | 11/29/17 11:26:00 AM |
|---|---|---|

ont color: Text 1, English (UK), Not Expanded by / Condensed by

| age 8: [239] Formatted | Microsoft Office User | 11/29/17 11:26:00 AM |
|---|---|---|

ont color: Text 1, English (UK), Not Expanded by / Condensed by

| age 8: [239] Formatted | Microsoft Office User | 11/29/17 11:26:00 AM |
|---|---|---|

ont color: Text 1, English (UK), Not Expanded by / Condensed by

| age 8: [239] Formatted | Microsoft Office User | 11/29/17 11:26:00 AM |
|---|---|---|

ont color: Text 1, English (UK), Not Expanded by / Condensed by

| age 8: [239] Formatted | Microsoft Office User | 11/29/17 11:26:00 AM |
|---|---|---|

ont color: Text 1, English (UK), Not Expanded by / Condensed by

| age 8: [239] Formatted | Microsoft Office User | 11/29/17 11:26:00 AM |
|---|---|---|

ont color: Text 1, English (UK), Not Expanded by / Condensed by

| age 8: [240] Formatted | Microsoft Office User | 11/29/17 11:26:00 AM |
|---|---|---|

ont color: Text 1, English (UK)

| age 8: [240] Formatted | Microsoft Office User | 11/29/17 11:26:00 AM |
|---|---|---|

ont color: Text 1, English (UK)

| age 8: [240] Formatted | Microsoft Office User | 11/29/17 11:26:00 AM |
|---|---|---|

ont color: Text 1, English (UK)

| age 8: [241] Formatted | Microsoft Office User | 11/29/17 11:26:00 AM |
|---|---|---|

ont color: Text 1, English (UK), Not Expanded by / Condensed by

| age 8: [241] Formatted | Microsoft Office User | 11/29/17 11:26:00 AM |
|---|---|---|

ont color: Text 1, English (UK), Not Expanded by / Condensed by

| age 8: [241] Formatted | Microsoft Office User | 11/29/17 11:26:00 AM |
|---|---|---|

ont color: Text 1, English (UK), Not Expanded by / Condensed by

| age 8: [241] Formatted | Microsoft Office User | 11/29/17 11:26:00 AM |
|---|---|---|

ont color: Text 1, English (UK), Not Expanded by / Condensed by

| age 8: [242] Deleted | Microsoft Office User | 11/29/17 11:26:00 AM |
|---|---|---|

0

| age 8: [243] Formatted | Microsoft Office User | 11/29/17 11:26:00 AM |
|---|---|---|

ont:10 pt, Font color: Text 1, English (UK)

| age 8: [243] Formatted | Microsoft Office User | 11/29/17 11:26:00 AM |
|---|---|---|

ont:10 pt, Font color: Text 1, English (UK)

| age 8: [244] Formatted | Microsoft Office User | 11/29/17 11:26:00 AM |
|---|---|---|

ont color: Text 1, English (UK)

| age 8: [245] Formatted | Microsoft Office User | 11/29/17 11:26:00 AM |
|---|---|---|

ont color: Text 1, English (UK)

| age 8: [246] Formatted | Microsoft Office User | 11/29/17 11:26:00 AM |
|---|---|---|

ont color: Text 1, English (UK)

| age 8: [247] Formatted | Microsoft Office User | 11/29/17 11:26:00 AM |
|---|---|---|

ont color: Text 1, English (UK)

| age 8: [247] Formatted | Microsoft Office User | 11/29/17 11:26:00 AM |
|---|---|---|

ont color: Text 1, English (UK)

| Page 8: [247] Formatted | Microsoft Office User | 11/29/17 11:26:00 AM |

Font color: Text 1, English (UK)

| Page 8: [247] Formatted | Microsoft Office User | 11/29/17 11:26:00 AM |

Font color: Text 1, English (UK)

| Page 8: [248] Formatted | Microsoft Office User | 11/29/17 11:26:00 AM |

Font color: Text 1, English (UK), Not Expanded by / Condensed by

| Page 8: [248] Formatted | Microsoft Office User | 11/29/17 11:26:00 AM |

Font color: Text 1, English (UK), Not Expanded by / Condensed by

| Page 8: [248] Formatted | Microsoft Office User | 11/29/17 11:26:00 AM |

Font color: Text 1, English (UK), Not Expanded by / Condensed by

| Page 8: [248] Formatted | Microsoft Office User | 11/29/17 11:26:00 AM |

Font color: Text 1, English (UK), Not Expanded by / Condensed by

| Page 8: [248] Formatted | Microsoft Office User | 11/29/17 11:26:00 AM |

Font color: Text 1, English (UK), Not Expanded by / Condensed by

| Page 8: [248] Formatted | Microsoft Office User | 11/29/17 11:26:00 AM |

Font color: Text 1, English (UK), Not Expanded by / Condensed by

| Page 8: [248] Formatted | Microsoft Office User | 11/29/17 11:26:00 AM |

Font color: Text 1, English (UK), Not Expanded by / Condensed by

| Page 8: [248] Formatted | Microsoft Office User | 11/29/17 11:26:00 AM |

Font color: Text 1, English (UK), Not Expanded by / Condensed by

| Page 8: [248] Formatted | Microsoft Office User | 11/29/17 11:26:00 AM |

Font color: Text 1, English (UK), Not Expanded by / Condensed by

| Page 8: [248] Formatted | Microsoft Office User | 11/29/17 11:26:00 AM |

Font color: Text 1, English (UK), Not Expanded by / Condensed by

| Page 8: [248] Formatted | Microsoft Office User | 11/29/17 11:26:00 AM |

Font color: Text 1, English (UK), Not Expanded by / Condensed by

| Page 8: [249] Deleted | Microsoft Office User | 11/29/17 11:26:00 AM |

from 147 hPa. For $O_3$, we use OMI data layer 18 (approximately 100 hPa) and MLS 100 hPa product. A

| Page 8: [250] Formatted | Microsoft Office User | 11/29/17 11:26:00 AM |

Font color: Text 1, English (UK)

| Page 8: [251] Formatted | Microsoft Office User | 11/29/17 11:26:00 AM |

Font color: Text 1, English (UK)

| Page 8: [252] Formatted | Microsoft Office User | 11/29/17 11:26:00 AM |

Normal, Line spacing: 1.5 lines

| Page 8: [253] Deleted | Microsoft Office User | 11/29/17 11:26:00 AM |

25    Figure 2 shows seasonal averages for (a) MLS CO at 147 hPa, (b) IASI CO at 12-16 km, (c) MLS $O_3$ at 100 hPa and (d) OMI $O_3$ in layer 18 during the active period of the ASM (JJA). To

| Page 8: [254] Moved from page 0 (Move #1) | Microsoft Office User | 11/29/17 11:26:00 AM |

igure

| Page 8: [255] Formatted | Microsoft Office User | 11/29/17 11:26:00 AM |
|---|---|---|

ont color: Text 1, English (UK)

| Page 8: [256] Formatted | Microsoft Office User | 11/29/17 11:26:00 AM |
|---|---|---|

ont:10 pt, Font color: Text 1, English (UK)

| Page 8: [257] Formatted | Microsoft Office User | 11/29/17 11:26:00 AM |
|---|---|---|

Normal, Space Before:  0 pt, Line spacing:  1.5 lines

| Page 8: [258] Formatted | Microsoft Office User | 11/29/17 11:26:00 AM |
|---|---|---|

ont color: Text 1, English (UK)

| Page 8: [259] Formatted | Microsoft Office User | 11/29/17 11:26:00 AM |
|---|---|---|

Normal, Left, Indent: First line:  0", Right:  0", Space Before:  0 pt, Line spacing:  1.5 lines

| Page 8: [260] Formatted | Microsoft Office User | 11/29/17 11:26:00 AM |
|---|---|---|

ont color: Text 1, English (UK), Not Expanded by / Condensed by

| Page 8: [260] Formatted | Microsoft Office User | 11/29/17 11:26:00 AM |
|---|---|---|

ont color: Text 1, English (UK), Not Expanded by / Condensed by

| Page 8: [260] Formatted | Microsoft Office User | 11/29/17 11:26:00 AM |
|---|---|---|

ont color: Text 1, English (UK), Not Expanded by / Condensed by

| Page 8: [260] Formatted | Microsoft Office User | 11/29/17 11:26:00 AM |
|---|---|---|

ont color: Text 1, English (UK), Not Expanded by / Condensed by

| Page 8: [260] Formatted | Microsoft Office User | 11/29/17 11:26:00 AM |
|---|---|---|

ont color: Text 1, English (UK), Not Expanded by / Condensed by

| Page 8: [260] Formatted | Microsoft Office User | 11/29/17 11:26:00 AM |
|---|---|---|

ont color: Text 1, English (UK), Not Expanded by / Condensed by

| Page 8: [260] Formatted | Microsoft Office User | 11/29/17 11:26:00 AM |
|---|---|---|

ont color: Text 1, English (UK), Not Expanded by / Condensed by

| Page 8: [260] Formatted | Microsoft Office User | 11/29/17 11:26:00 AM |
|---|---|---|

ont color: Text 1, English (UK), Not Expanded by / Condensed by

| Page 8: [260] Formatted | Microsoft Office User | 11/29/17 11:26:00 AM |
|---|---|---|

ont color: Text 1, English (UK), Not Expanded by / Condensed by

| Page 8: [260] Formatted | Microsoft Office User | 11/29/17 11:26:00 AM |
|---|---|---|

ont color: Text 1, English (UK), Not Expanded by / Condensed by

| Page 8: [260] Formatted | Microsoft Office User | 11/29/17 11:26:00 AM |
|---|---|---|

ont color: Text 1, English (UK), Not Expanded by / Condensed by

| Page 8: [260] Formatted | Microsoft Office User | 11/29/17 11:26:00 AM |
|---|---|---|

ont color: Text 1, English (UK), Not Expanded by / Condensed by

| Page 8: [260] Formatted | Microsoft Office User | 11/29/17 11:26:00 AM |
|---|---|---|

ont color: Text 1, English (UK), Not Expanded by / Condensed by

| Page 8: [260] Formatted | Microsoft Office User | 11/29/17 11:26:00 AM |
|---|---|---|

ont color: Text 1, English (UK), Not Expanded by / Condensed by

| Page 8: [260] Formatted | Microsoft Office User | 11/29/17 11:26:00 AM |
|---|---|---|

ont color: Text 1, English (UK), Not Expanded by / Condensed by

| Page 8: [261] Deleted | Microsoft Office User | 11/29/17 11:26:00 AM |
|---|---|---|

Fig. 2 also includes selected GPH contours and wind vectors at 150 hPa (panels a and b) and 100 hPa (panels c and d) from the same

eriod. The

| Page 8: [262] Formatted | Microsoft Office User | 11/29/17 11:26:00 AM |
|---|---|---|

ont color: Text 1, English (UK)

| Page 8: [263] Deleted | Microsoft Office User | 11/29/17 11:26:00 AM |
|---|---|---|

of the seasonally averaged ASM anticyclone is evident for all quantities. Both MLS and IASI CO data show enhancements

within the anticyclone, although IASI has lower values than MLS. Since the largest retrieval

30      sensitivity of IASI CO is in the middle troposphere (~ 500 hPa), the clear CO enhancement in the 12-16 km layer in the

Section Break (Next Page)

nticyclone

| age 8: [264] Formatted | Microsoft Office User | 11/29/17 11:26:00 AM |
|---|---|---|
| ont color: Text 1, English (UK) | | |
| age 8: [264] Formatted | Microsoft Office User | 11/29/17 11:26:00 AM |
| ont color: Text 1, English (UK) | | |
| age 8: [264] Formatted | Microsoft Office User | 11/29/17 11:26:00 AM |
| ont color: Text 1, English (UK) | | |
| age 8: [264] Formatted | Microsoft Office User | 11/29/17 11:26:00 AM |
| ont color: Text 1, English (UK) | | |
| age 8: [264] Formatted | Microsoft Office User | 11/29/17 11:26:00 AM |
| ont color: Text 1, English (UK) | | |
| age 8: [264] Formatted | Microsoft Office User | 11/29/17 11:26:00 AM |
| ont color: Text 1, English (UK) | | |
| age 8: [264] Formatted | Microsoft Office User | 11/29/17 11:26:00 AM |
| ont color: Text 1, English (UK) | | |
| age 8: [264] Formatted | Microsoft Office User | 11/29/17 11:26:00 AM |
| ont color: Text 1, English (UK) | | |
| age 8: [264] Formatted | Microsoft Office User | 11/29/17 11:26:00 AM |
| ont color: Text 1, English (UK) | | |
| age 8: [264] Formatted | Microsoft Office User | 11/29/17 11:26:00 AM |
| ont color: Text 1, English (UK) | | |
| age 8: [264] Formatted | Microsoft Office User | 11/29/17 11:26:00 AM |
| ont color: Text 1, English (UK) | | |
| age 8: [264] Formatted | Microsoft Office User | 11/29/17 11:26:00 AM |
| ont color: Text 1, English (UK) | | |
| age 8: [264] Formatted | Microsoft Office User | 11/29/17 11:26:00 AM |
| ont color: Text 1, English (UK) | | |
| age 8: [264] Formatted | Microsoft Office User | 11/29/17 11:26:00 AM |
| ont color: Text 1, English (UK) | | |
| age 8: [264] Formatted | Microsoft Office User | 11/29/17 11:26:00 AM |
| ont color: Text 1, English (UK) | | |
| age 8: [264] Formatted | Microsoft Office User | 11/29/17 11:26:00 AM |
| ont color: Text 1, English (UK) | | |
| age 8: [264] Formatted | Microsoft Office User | 11/29/17 11:26:00 AM |
| ont color: Text 1, English (UK) | | |

| age 8: [264] Formatted | Microsoft Office User | 11/29/17 11:26:00 AM |
|---|---|---|
| ont color: Text 1, English (UK) | | |
| age 8: [264] Formatted | Microsoft Office User | 11/29/17 11:26:00 AM |
| ont color: Text 1, English (UK) | | |
| age 8: [264] Formatted | Microsoft Office User | 11/29/17 11:26:00 AM |
| ont color: Text 1, English (UK) | | |
| age 8: [264] Formatted | Microsoft Office User | 11/29/17 11:26:00 AM |
| ont color: Text 1, English (UK) | | |
| age 8: [264] Formatted | Microsoft Office User | 11/29/17 11:26:00 AM |
| ont color: Text 1, English (UK) | | |
| age 8: [264] Formatted | Microsoft Office User | 11/29/17 11:26:00 AM |
| ont color: Text 1, English (UK) | | |
| age 8: [264] Formatted | Microsoft Office User | 11/29/17 11:26:00 AM |
| ont color: Text 1, English (UK) | | |
| age 8: [264] Formatted | Microsoft Office User | 11/29/17 11:26:00 AM |
| ont color: Text 1, English (UK) | | |
| age 8: [264] Formatted | Microsoft Office User | 11/29/17 11:26:00 AM |
| ont color: Text 1, English (UK) | | |
| age 8: [264] Formatted | Microsoft Office User | 11/29/17 11:26:00 AM |
| ont color: Text 1, English (UK) | | |
| age 8: [264] Formatted | Microsoft Office User | 11/29/17 11:26:00 AM |
| ont color: Text 1, English (UK) | | |
| age 8: [264] Formatted | Microsoft Office User | 11/29/17 11:26:00 AM |
| ont color: Text 1, English (UK) | | |
| age 8: [264] Formatted | Microsoft Office User | 11/29/17 11:26:00 AM |
| ont color: Text 1, English (UK) | | |
| age 8: [264] Formatted | Microsoft Office User | 11/29/17 11:26:00 AM |
| ont color: Text 1, English (UK) | | |
| age 8: [264] Formatted | Microsoft Office User | 11/29/17 11:26:00 AM |
| ont color: Text 1, English (UK) | | |
| age 8: [264] Formatted | Microsoft Office User | 11/29/17 11:26:00 AM |
| ont color: Text 1, English (UK) | | |
| age 8: [265] Deleted | Microsoft Office User | 11/29/17 11:26:00 AM |

imilarly, MLS and OMI data are

| Page 8: [266] Formatted | Microsoft Office User | 11/29/17 11:26:00 AM |

Font color: Text 1, English (UK)

| Page 8: [266] Formatted | Microsoft Office User | 11/29/17 11:26:00 AM |

Font color: Text 1, English (UK)

| Page 8: [267] Deleted | Microsoft Office User | 11/29/17 11:26:00 AM |

onsistent in their representations of $O_3$ distributions at 100 hPa, showing the lower $O_3$ mixing ratios within the anticyclone, despite the

| Page 8: [268] Formatted | Microsoft Office User | 11/29/17 11:26:00 AM |

Font color: Text 1, English (UK)

| Page 8: [269] Deleted | Microsoft Office User | 11/29/17 11:26:00 AM |

different sampling geometry and vertical resolutions. Note that

this chemical structure

| Page 8: [270] Formatted | Microsoft Office User | 11/29/17 11:26:00 AM |

Font color: Text 1, English (UK)

| Page 9: [271] Deleted | Microsoft Office User | 11/29/17 11:26:00 AM |

10 To further quantify the consistency of limb and nadir sounding data in representing the variability, we compare co-located bin averages from IASI CO to MLS (Fig. 3a) as well as OMI $O_3$ to MLS (Fig. 3b) in the monsoon region (0-50°N, 0-150°E) during JJA. These scatterplots show that variations from nadir viewing instruments are generally consistent with those from MLS. IASI and MLS CO are well correlated (r = 0.72; Fig. 3a) although the IASI layer averages have a smaller range of variability than MLS (represented by the small slope of the linear fit, 0.40), likely influenced by using a single a priori profile in CO

15 retrieval (George et al., 2015) and a weaker detection sensitivity in the upper troposphere. The $O_3$ correlation between OMI and MLS at 100 hPa is stronger (r = 0.93); here too, the nadir instrument has somewhat weaker variability (slope = 0.85).

| Page 9: [272] Formatted | Microsoft Office User | 11/29/17 11:26:00 AM |

Normal, Left, Right: 0", Space Before: 0 pt, Line spacing: 1.5 lines

| Page 9: [273] Formatted | Microsoft Office User | 11/29/17 11:26:00 AM |

Font color: Text 1, English (UK)

| Page 9: [273] Formatted | Microsoft Office User | 11/29/17 11:26:00 AM |

Font color: Text 1, English (UK)

| Page 9: [273] Formatted | Microsoft Office User | 11/29/17 11:26:00 AM |

Font color: Text 1, English (UK)

| Page 9: [273] Formatted | Microsoft Office User | 11/29/17 11:26:00 AM |

Font color: Text 1, English (UK)

| Page 9: [273] Formatted | Microsoft Office User | 11/29/17 11:26:00 AM |

Font color: Text 1, English (UK)

| Page 9: [273] Formatted | Microsoft Office User | 11/29/17 11:26:00 AM |

Font color: Text 1, English (UK)

| Page 9: [274] Deleted | Microsoft Office User | 11/29/17 11:26:00 AM |

how that the nadir viewing data has sufficient sensitivity to capture the UTLS chemical impact by ASM anticyclone. The IASI CO and

)MI O$_3$ products,

| Page 9: [275] Formatted | Microsoft Office User | 11/29/17 11:26:00 AM |
|---|---|---|

ont color: Text 1, English (UK), Not Expanded by / Condensed by

| Page 9: [275] Formatted | Microsoft Office User | 11/29/17 11:26:00 AM |
|---|---|---|

ont color: Text 1, English (UK), Not Expanded by / Condensed by

| Page 9: [275] Formatted | Microsoft Office User | 11/29/17 11:26:00 AM |
|---|---|---|

ont color: Text 1, English (UK), Not Expanded by / Condensed by

| Page 9: [275] Formatted | Microsoft Office User | 11/29/17 11:26:00 AM |
|---|---|---|

ont color: Text 1, English (UK), Not Expanded by / Condensed by

| Page 9: [275] Formatted | Microsoft Office User | 11/29/17 11:26:00 AM |
|---|---|---|

ont color: Text 1, English (UK), Not Expanded by / Condensed by

| Page 9: [275] Formatted | Microsoft Office User | 11/29/17 11:26:00 AM |
|---|---|---|

ont color: Text 1, English (UK), Not Expanded by / Condensed by

| Page 9: [275] Formatted | Microsoft Office User | 11/29/17 11:26:00 AM |
|---|---|---|

ont color: Text 1, English (UK), Not Expanded by / Condensed by

| Page 9: [275] Formatted | Microsoft Office User | 11/29/17 11:26:00 AM |
|---|---|---|

ont color: Text 1, English (UK), Not Expanded by / Condensed by

| Page 9: [275] Formatted | Microsoft Office User | 11/29/17 11:26:00 AM |
|---|---|---|

ont color: Text 1, English (UK), Not Expanded by / Condensed by

| Page 9: [275] Formatted | Microsoft Office User | 11/29/17 11:26:00 AM |
|---|---|---|

ont color: Text 1, English (UK), Not Expanded by / Condensed by

| Page 9: [275] Formatted | Microsoft Office User | 11/29/17 11:26:00 AM |
|---|---|---|

ont color: Text 1, English (UK), Not Expanded by / Condensed by

| Page 9: [275] Formatted | Microsoft Office User | 11/29/17 11:26:00 AM |
|---|---|---|

ont color: Text 1, English (UK), Not Expanded by / Condensed by

| Page 9: [275] Formatted | Microsoft Office User | 11/29/17 11:26:00 AM |
|---|---|---|

ont color: Text 1, English (UK), Not Expanded by / Condensed by

| Page 9: [275] Formatted | Microsoft Office User | 11/29/17 11:26:00 AM |
|---|---|---|

ont color: Text 1, English (UK), Not Expanded by / Condensed by

| Page 9: [275] Formatted | Microsoft Office User | 11/29/17 11:26:00 AM |
|---|---|---|

ont color: Text 1, English (UK), Not Expanded by / Condensed by

| Page 9: [276] Formatted | Microsoft Office User | 11/29/17 11:26:00 AM |
|---|---|---|

leading 1, Left,  No bullets or numbering, Tabs:Not at  0.61"

| Page 9: [277] Deleted | Microsoft Office User | 11/29/17 11:26:00 AM |
|---|---|---|

20 Park et al., 2007; Barret et al., 2008; Livesey et al., 2008; Liu et al., 2013; Huang et al., 2016; Yan et al., 2016). Encouraged by

the consistency shown by these correlations, we now proceed to analyze the sub-seasonal scale variability.

| Page 9: [277] Deleted | Microsoft Office User | 11/29/17 11:26:00 AM |
|---|---|---|

20 Park et al., 2007; Barret et al., 2008; Livesey et al., 2008; Liu et al., 2013; Huang et al., 2016; Yan et al., 2016). Encouraged by the consistency shown by these correlations, we now proceed to analyze the sub-seasonal scale variability.

| age 9: [280] Formatted | Microsoft Office User | 11/29/17 11:26:00 AM |
| --- | --- | --- |
| ont color: Text 1, English (UK) | | |

| age 9: [280] Formatted | Microsoft Office User | 11/29/17 11:26:00 AM |
| --- | --- | --- |
| ont color: Text 1, English (UK) | | |

| age 9: [280] Formatted | Microsoft Office User | 11/29/17 11:26:00 AM |
| --- | --- | --- |
| ont color: Text 1, English (UK) | | |

| age 9: [280] Formatted | Microsoft Office User | 11/29/17 11:26:00 AM |
| --- | --- | --- |
| ont color: Text 1, English (UK) | | |

| age 9: [280] Formatted | Microsoft Office User | 11/29/17 11:26:00 AM |
| --- | --- | --- |
| ont color: Text 1, English (UK) | | |

| age 9: [280] Formatted | Microsoft Office User | 11/29/17 11:26:00 AM |
| --- | --- | --- |
| ont color: Text 1, English (UK) | | |

| age 9: [280] Formatted | Microsoft Office User | 11/29/17 11:26:00 AM |
| --- | --- | --- |
| ont color: Text 1, English (UK) | | |

| age 9: [280] Formatted | Microsoft Office User | 11/29/17 11:26:00 AM |
| --- | --- | --- |
| ont color: Text 1, English (UK) | | |

| age 9: [280] Formatted | Microsoft Office User | 11/29/17 11:26:00 AM |
| --- | --- | --- |
| ont color: Text 1, English (UK) | | |

| age 9: [280] Formatted | Microsoft Office User | 11/29/17 11:26:00 AM |
| --- | --- | --- |
| ont color: Text 1, English (UK) | | |

| age 9: [280] Formatted | Microsoft Office User | 11/29/17 11:26:00 AM |
| --- | --- | --- |
| ont color: Text 1, English (UK) | | |

| age 9: [280] Formatted | Microsoft Office User | 11/29/17 11:26:00 AM |
| --- | --- | --- |
| ont color: Text 1, English (UK) | | |

| age 9: [280] Formatted | Microsoft Office User | 11/29/17 11:26:00 AM |
| --- | --- | --- |
| ont color: Text 1, English (UK) | | |

| age 9: [280] Formatted | Microsoft Office User | 11/29/17 11:26:00 AM |
| --- | --- | --- |
| ont color: Text 1, English (UK) | | |

| age 9: [280] Formatted | Microsoft Office User | 11/29/17 11:26:00 AM |
| --- | --- | --- |
| ont color: Text 1, English (UK) | | |

| age 9: [280] Formatted | Microsoft Office User | 11/29/17 11:26:00 AM |
| --- | --- | --- |
| ont color: Text 1, English (UK) | | |

| age 9: [280] Formatted | Microsoft Office User | 11/29/17 11:26:00 AM |
| --- | --- | --- |
| ont color: Text 1, English (UK) | | |

| age 9: [280] Formatted | Microsoft Office User | 11/29/17 11:26:00 AM |
| --- | --- | --- |
| ont color: Text 1, English (UK) | | |

| age 9: [280] Formatted | Microsoft Office User | 11/29/17 11:26:00 AM |
|---|---|---|

ont color: Text 1, English (UK)

| age 9: [280] Formatted | Microsoft Office User | 11/29/17 11:26:00 AM |
|---|---|---|

ont color: Text 1, English (UK)

| age 9: [280] Formatted | Microsoft Office User | 11/29/17 11:26:00 AM |
|---|---|---|

ont color: Text 1, English (UK)

| age 9: [280] Formatted | Microsoft Office User | 11/29/17 11:26:00 AM |
|---|---|---|

ont color: Text 1, English (UK)

| age 9: [280] Formatted | Microsoft Office User | 11/29/17 11:26:00 AM |
|---|---|---|

ont color: Text 1, English (UK)

| age 9: [281] Formatted | Microsoft Office User | 11/29/17 11:26:00 AM |
|---|---|---|

ont color: Text 1, English (UK)

| age 9: [281] Formatted | Microsoft Office User | 11/29/17 11:26:00 AM |
|---|---|---|

ont color: Text 1, English (UK)

| age 9: [281] Formatted | Microsoft Office User | 11/29/17 11:26:00 AM |
|---|---|---|

ont color: Text 1, English (UK)

| age 9: [281] Formatted | Microsoft Office User | 11/29/17 11:26:00 AM |
|---|---|---|

ont color: Text 1, English (UK)

| age 9: [281] Formatted | Microsoft Office User | 11/29/17 11:26:00 AM |
|---|---|---|

ont color: Text 1, English (UK)

| age 9: [281] Formatted | Microsoft Office User | 11/29/17 11:26:00 AM |
|---|---|---|

ont color: Text 1, English (UK)

| age 9: [281] Formatted | Microsoft Office User | 11/29/17 11:26:00 AM |
|---|---|---|

ont color: Text 1, English (UK)

| age 9: [281] Formatted | Microsoft Office User | 11/29/17 11:26:00 AM |
|---|---|---|

ont color: Text 1, English (UK)

| age 9: [281] Formatted | Microsoft Office User | 11/29/17 11:26:00 AM |
|---|---|---|

ont color: Text 1, English (UK)

| age 9: [281] Formatted | Microsoft Office User | 11/29/17 11:26:00 AM |
|---|---|---|

ont color: Text 1, English (UK)

| age 9: [281] Formatted | Microsoft Office User | 11/29/17 11:26:00 AM |
|---|---|---|

ont color: Text 1, English (UK)

| age 9: [281] Formatted | Microsoft Office User | 11/29/17 11:26:00 AM |
|---|---|---|

ont color: Text 1, English (UK)

| age 9: [281] Formatted | Microsoft Office User | 11/29/17 11:26:00 AM |
|---|---|---|

ont color: Text 1, English (UK)

| age 9: [281] Formatted | Microsoft Office User | 11/29/17 11:26:00 AM |
|---|---|---|

ont color: Text 1, English (UK)

| age 9: [281] Formatted | Microsoft Office User | 11/29/17 11:26:00 AM |
|---|---|---|

ont color: Text 1, English (UK)

| age 9: [281] Formatted | Microsoft Office User | 11/29/17 11:26:00 AM |
|---|---|---|

ont color: Text 1, English (UK)

| age 9: [281] Formatted | Microsoft Office User | 11/29/17 11:26:00 AM |
|---|---|---|

ont color: Text 1, English (UK)

| age 9: [281] Formatted | Microsoft Office User | 11/29/17 11:26:00 AM |
|---|---|---|

ont color: Text 1, English (UK)

| age 9: [281] Formatted | Microsoft Office User | 11/29/17 11:26:00 AM |
|---|---|---|

ont color: Text 1, English (UK)

| age 9: [281] Formatted | Microsoft Office User | 11/29/17 11:26:00 AM |
|---|---|---|

ont color: Text 1, English (UK)

| age 9: [281] Formatted | Microsoft Office User | 11/29/17 11:26:00 AM |
|---|---|---|

ont color: Text 1, English (UK)

| age 9: [281] Formatted | Microsoft Office User | 11/29/17 11:26:00 AM |
|---|---|---|

ont color: Text 1, English (UK)

| age 9: [281] Formatted | Microsoft Office User | 11/29/17 11:26:00 AM |
|---|---|---|

ont color: Text 1, English (UK)

| age 9: [281] Formatted | Microsoft Office User | 11/29/17 11:26:00 AM |
|---|---|---|

ont color: Text 1, English (UK)

| age 9: [281] Formatted | Microsoft Office User | 11/29/17 11:26:00 AM |
|---|---|---|

ont color: Text 1, English (UK)

| age 9: [281] Formatted | Microsoft Office User | 11/29/17 11:26:00 AM |
|---|---|---|

ont color: Text 1, English (UK)

| age 9: [282] Deleted | Microsoft Office User | 11/29/17 11:26:00 AM |
|---|---|---|

| age 9: [283] Formatted | Microsoft Office User | 11/29/17 11:26:00 AM |
|---|---|---|

ont:10 pt, Font color: Text 1, English (UK)

| age 9: [283] Formatted | Microsoft Office User | 11/29/17 11:26:00 AM |
|---|---|---|

ont:10 pt, Font color: Text 1, English (UK)

| age 9: [284] Deleted | Microsoft Office User | 11/29/17 11:26:00 AM |
|---|---|---|

Popovic and Plumb, 2001; Garny and Randel, 2013; Pan et al., 2016).

| age 9: [285] Formatted | Microsoft Office User | 11/29/17 11:26:00 AM |
|---|---|---|

ont color: Text 1, English (UK)

Because of the large differences in horizontal sampling density, the limb and nadir data are mapped differently. The daily representation from MLS data requires interpolation to increase the density in coverage, while the IASI and OMI data densities

30      are reduced by binned averages. We explored three interpolation algorithms for mapping MLS CO and $O_3$ to approximately 5° ×5° latitude-longitude grid. All three methods (cosine smoothing, natural neighbor, inverse distance) are similar, conceptually, in filling the empty cell with weighted mean of nearby observations, but the weightings are determined differently. The

————Section Break (Next Page)————

differences produced by these methods do not impact the conclusions of this study. Figure 4 shows maps of retrieved data only (Fig. 4a) and the interpolated data (Fig. 4b), using one day of MLS CO at 147 hPa. Both maps exhibit the co-locations of enhanced CO and high GPH, showing the consistency of the CO enhancement with the large scale dynamical fields that highlight the location of the anticyclone. The small scale structure produced by the mapping should not be over interpreted.

5 **4.1. UT CO analysis using MLS and IASI data**

| age 9: [287] Formatted | Microsoft Office User | 11/29/17 11:26:00 AM |
|---|---|---|

ont color: Text 1, English (UK), Not Expanded by / Condensed by

| age 9: [287] Formatted | Microsoft Office User | 11/29/17 11:26:00 AM |
|---|---|---|

ont color: Text 1, English (UK), Not Expanded by / Condensed by

| age 9: [287] Formatted | Microsoft Office User | 11/29/17 11:26:00 AM |
|---|---|---|

ont color: Text 1, English (UK), Not Expanded by / Condensed by

| age 9: [287] Formatted | Microsoft Office User | 11/29/17 11:26:00 AM |
|---|---|---|

ont color: Text 1, English (UK), Not Expanded by / Condensed by

| age 9: [287] Formatted | Microsoft Office User | 11/29/17 11:26:00 AM |
|---|---|---|

ont color: Text 1, English (UK), Not Expanded by / Condensed by

| age 9: [287] Formatted | Microsoft Office User | 11/29/17 11:26:00 AM |
|---|---|---|

ont color: Text 1, English (UK), Not Expanded by / Condensed by

| age 9: [287] Formatted | Microsoft Office User | 11/29/17 11:26:00 AM |
|---|---|---|

ont color: Text 1, English (UK), Not Expanded by / Condensed by

| age 9: [287] Formatted | Microsoft Office User | 11/29/17 11:26:00 AM |
|---|---|---|

ont color: Text 1, English (UK), Not Expanded by / Condensed by

| age 9: [287] Formatted | Microsoft Office User | 11/29/17 11:26:00 AM |
|---|---|---|

ont color: Text 1, English (UK), Not Expanded by / Condensed by

| age 9: [287] Formatted | Microsoft Office User | 11/29/17 11:26:00 AM |
|---|---|---|

ont color: Text 1, English (UK), Not Expanded by / Condensed by

| age 9: [287] Formatted | Microsoft Office User | 11/29/17 11:26:00 AM |
|---|---|---|

ont color: Text 1, English (UK), Not Expanded by / Condensed by

| age 9: [287] Formatted | Microsoft Office User | 11/29/17 11:26:00 AM |
|---|---|---|

ont color: Text 1, English (UK), Not Expanded by / Condensed by

| age 9: [287] Formatted | Microsoft Office User | 11/29/17 11:26:00 AM |
|---|---|---|

ont color: Text 1, English (UK), Not Expanded by / Condensed by

| age 9: [287] Formatted | Microsoft Office User | 11/29/17 11:26:00 AM |
|---|---|---|

ont color: Text 1, English (UK), Not Expanded by / Condensed by

| age 9: [287] Formatted | Microsoft Office User | 11/29/17 11:26:00 AM |
|---|---|---|

ont color: Text 1, English (UK), Not Expanded by / Condensed by

| age 9: [287] Formatted | Microsoft Office User | 11/29/17 11:26:00 AM |

ont color: Text 1, English (UK), Not Expanded by / Condensed by

| age 9: [287] Formatted | Microsoft Office User | 11/29/17 11:26:00 AM |

ont color: Text 1, English (UK), Not Expanded by / Condensed by

| age 9: [287] Formatted | Microsoft Office User | 11/29/17 11:26:00 AM |

ont color: Text 1, English (UK), Not Expanded by / Condensed by

| age 9: [287] Formatted | Microsoft Office User | 11/29/17 11:26:00 AM |

ont color: Text 1, English (UK), Not Expanded by / Condensed by

| age 9: [287] Formatted | Microsoft Office User | 11/29/17 11:26:00 AM |

ont color: Text 1, English (UK), Not Expanded by / Condensed by

| age 9: [287] Formatted | Microsoft Office User | 11/29/17 11:26:00 AM |

ont color: Text 1, English (UK), Not Expanded by / Condensed by

| age 9: [287] Formatted | Microsoft Office User | 11/29/17 11:26:00 AM |

ont color: Text 1, English (UK), Not Expanded by / Condensed by

| age 9: [287] Formatted | Microsoft Office User | 11/29/17 11:26:00 AM |

ont color: Text 1, English (UK), Not Expanded by / Condensed by

| age 9: [287] Formatted | Microsoft Office User | 11/29/17 11:26:00 AM |

ont color: Text 1, English (UK), Not Expanded by / Condensed by

| age 9: [287] Formatted | Microsoft Office User | 11/29/17 11:26:00 AM |

ont color: Text 1, English (UK), Not Expanded by / Condensed by

| age 9: [287] Formatted | Microsoft Office User | 11/29/17 11:26:00 AM |

ont color: Text 1, English (UK), Not Expanded by / Condensed by

| age 9: [287] Formatted | Microsoft Office User | 11/29/17 11:26:00 AM |

ont color: Text 1, English (UK), Not Expanded by / Condensed by

| age 9: [287] Formatted | Microsoft Office User | 11/29/17 11:26:00 AM |

ont color: Text 1, English (UK), Not Expanded by / Condensed by

| age 9: [287] Formatted | Microsoft Office User | 11/29/17 11:26:00 AM |

ont color: Text 1, English (UK), Not Expanded by / Condensed by

| age 9: [287] Formatted | Microsoft Office User | 11/29/17 11:26:00 AM |

ont color: Text 1, English (UK), Not Expanded by / Condensed by

| age 9: [287] Formatted | Microsoft Office User | 11/29/17 11:26:00 AM |

ont color: Text 1, English (UK), Not Expanded by / Condensed by

| age 9: [287] Formatted | Microsoft Office User | 11/29/17 11:26:00 AM |

ont color: Text 1, English (UK), Not Expanded by / Condensed by

| age 9: [287] Formatted | Microsoft Office User | 11/29/17 11:26:00 AM |

ont color: Text 1, English (UK), Not Expanded by / Condensed by

**age 9: [287] Formatted** | **Microsoft Office User** | **11/29/17 11:26:00 AM**

ont color: Text 1, English (UK), Not Expanded by / Condensed by

**age 9: [287] Formatted** | **Microsoft Office User** | **11/29/17 11:26:00 AM**

ont color: Text 1, English (UK), Not Expanded by / Condensed by

**age 9: [287] Formatted** | **Microsoft Office User** | **11/29/17 11:26:00 AM**

ont color: Text 1, English (UK), Not Expanded by / Condensed by

**age 9: [287] Formatted** | **Microsoft Office User** | **11/29/17 11:26:00 AM**

ont color: Text 1, English (UK), Not Expanded by / Condensed by

**age 9: [287] Formatted** | **Microsoft Office User** | **11/29/17 11:26:00 AM**

ont color: Text 1, English (UK), Not Expanded by / Condensed by

**age 9: [287] Formatted** | **Microsoft Office User** | **11/29/17 11:26:00 AM**

ont color: Text 1, English (UK), Not Expanded by / Condensed by

**age 9: [287] Formatted** | **Microsoft Office User** | **11/29/17 11:26:00 AM**

ont color: Text 1, English (UK), Not Expanded by / Condensed by

**age 9: [287] Formatted** | **Microsoft Office User** | **11/29/17 11:26:00 AM**

ont color: Text 1, English (UK), Not Expanded by / Condensed by

**age 9: [287] Formatted** | **Microsoft Office User** | **11/29/17 11:26:00 AM**

ont color: Text 1, English (UK), Not Expanded by / Condensed by

**age 9: [287] Formatted** | **Microsoft Office User** | **11/29/17 11:26:00 AM**

ont color: Text 1, English (UK), Not Expanded by / Condensed by

**age 9: [287] Formatted** | **Microsoft Office User** | **11/29/17 11:26:00 AM**

ont color: Text 1, English (UK), Not Expanded by / Condensed by

**age 9: [287] Formatted** | **Microsoft Office User** | **11/29/17 11:26:00 AM**

ont color: Text 1, English (UK), Not Expanded by / Condensed by

**age 9: [287] Formatted** | **Microsoft Office User** | **11/29/17 11:26:00 AM**

ont color: Text 1, English (UK), Not Expanded by / Condensed by

**age 9: [287] Formatted** | **Microsoft Office User** | **11/29/17 11:26:00 AM**

ont color: Text 1, English (UK), Not Expanded by / Condensed by

**age 9: [287] Formatted** | **Microsoft Office User** | **11/29/17 11:26:00 AM**

ont color: Text 1, English (UK), Not Expanded by / Condensed by

**age 9: [287] Formatted** | **Microsoft Office User** | **11/29/17 11:26:00 AM**

ont color: Text 1, English (UK), Not Expanded by / Condensed by

**age 9: [287] Formatted** | **Microsoft Office User** | **11/29/17 11:26:00 AM**

ont color: Text 1, English (UK), Not Expanded by / Condensed by

**age 9: [287] Formatted** | **Microsoft Office User** | **11/29/17 11:26:00 AM**

ont color: Text 1, English (UK), Not Expanded by / Condensed by

**age 9: [287] Formatted** | **Microsoft Office User** | **11/29/17 11:26:00 AM**

ont color: Text 1, English (UK), Not Expanded by / Condensed by

| age 9: [287] Formatted | Microsoft Office User | 11/29/17 11:26:00 AM |

ont color: Text 1, English (UK), Not Expanded by / Condensed by

| age 9: [287] Formatted | Microsoft Office User | 11/29/17 11:26:00 AM |

ont color: Text 1, English (UK), Not Expanded by / Condensed by

| age 9: [287] Formatted | Microsoft Office User | 11/29/17 11:26:00 AM |

ont color: Text 1, English (UK), Not Expanded by / Condensed by

| age 9: [287] Formatted | Microsoft Office User | 11/29/17 11:26:00 AM |

ont color: Text 1, English (UK), Not Expanded by / Condensed by

| age 9: [287] Formatted | Microsoft Office User | 11/29/17 11:26:00 AM |

ont color: Text 1, English (UK), Not Expanded by / Condensed by

| age 9: [287] Formatted | Microsoft Office User | 11/29/17 11:26:00 AM |

ont color: Text 1, English (UK), Not Expanded by / Condensed by

| age 9: [287] Formatted | Microsoft Office User | 11/29/17 11:26:00 AM |

ont color: Text 1, English (UK), Not Expanded by / Condensed by

| age 9: [287] Formatted | Microsoft Office User | 11/29/17 11:26:00 AM |

ont color: Text 1, English (UK), Not Expanded by / Condensed by

| age 9: [287] Formatted | Microsoft Office User | 11/29/17 11:26:00 AM |

ont color: Text 1, English (UK), Not Expanded by / Condensed by

| age 9: [287] Formatted | Microsoft Office User | 11/29/17 11:26:00 AM |

ont color: Text 1, English (UK), Not Expanded by / Condensed by

| age 9: [287] Formatted | Microsoft Office User | 11/29/17 11:26:00 AM |

ont color: Text 1, English (UK), Not Expanded by / Condensed by

| age 9: [287] Formatted | Microsoft Office User | 11/29/17 11:26:00 AM |

ont color: Text 1, English (UK), Not Expanded by / Condensed by

| age 9: [287] Formatted | Microsoft Office User | 11/29/17 11:26:00 AM |

ont color: Text 1, English (UK), Not Expanded by / Condensed by

| age 9: [287] Formatted | Microsoft Office User | 11/29/17 11:26:00 AM |

ont color: Text 1, English (UK), Not Expanded by / Condensed by

| age 9: [287] Formatted | Microsoft Office User | 11/29/17 11:26:00 AM |

ont color: Text 1, English (UK), Not Expanded by / Condensed by

| age 9: [287] Formatted | Microsoft Office User | 11/29/17 11:26:00 AM |

ont color: Text 1, English (UK), Not Expanded by / Condensed by

| age 9: [287] Formatted | Microsoft Office User | 11/29/17 11:26:00 AM |

ont color: Text 1, English (UK), Not Expanded by / Condensed by

| age 9: [287] Formatted | Microsoft Office User | 11/29/17 11:26:00 AM |

ont color: Text 1, English (UK), Not Expanded by / Condensed by

| age 9: [287] Formatted | Microsoft Office User | 11/29/17 11:26:00 AM |

ont color: Text 1, English (UK), Not Expanded by / Condensed by

| age 9: [287] Formatted | Microsoft Office User | 11/29/17 11:26:00 AM |

ont color: Text 1, English (UK), Not Expanded by / Condensed by

| age 9: [287] Formatted | Microsoft Office User | 11/29/17 11:26:00 AM |

ont color: Text 1, English (UK), Not Expanded by / Condensed by

| age 9: [287] Formatted | Microsoft Office User | 11/29/17 11:26:00 AM |

ont color: Text 1, English (UK), Not Expanded by / Condensed by

| age 9: [287] Formatted | Microsoft Office User | 11/29/17 11:26:00 AM |

ont color: Text 1, English (UK), Not Expanded by / Condensed by

| age 9: [287] Formatted | Microsoft Office User | 11/29/17 11:26:00 AM |

ont color: Text 1, English (UK), Not Expanded by / Condensed by

| age 9: [287] Formatted | Microsoft Office User | 11/29/17 11:26:00 AM |

ont color: Text 1, English (UK), Not Expanded by / Condensed by

| age 9: [287] Formatted | Microsoft Office User | 11/29/17 11:26:00 AM |

ont color: Text 1, English (UK), Not Expanded by / Condensed by

| age 9: [287] Formatted | Microsoft Office User | 11/29/17 11:26:00 AM |

ont color: Text 1, English (UK), Not Expanded by / Condensed by

| age 9: [287] Formatted | Microsoft Office User | 11/29/17 11:26:00 AM |

ont color: Text 1, English (UK), Not Expanded by / Condensed by

| age 9: [287] Formatted | Microsoft Office User | 11/29/17 11:26:00 AM |

ont color: Text 1, English (UK), Not Expanded by / Condensed by

| age 9: [287] Formatted | Microsoft Office User | 11/29/17 11:26:00 AM |

ont color: Text 1, English (UK), Not Expanded by / Condensed by

| age 9: [287] Formatted | Microsoft Office User | 11/29/17 11:26:00 AM |

ont color: Text 1, English (UK), Not Expanded by / Condensed by

| age 9: [287] Formatted | Microsoft Office User | 11/29/17 11:26:00 AM |

ont color: Text 1, English (UK), Not Expanded by / Condensed by

| age 9: [287] Formatted | Microsoft Office User | 11/29/17 11:26:00 AM |

ont color: Text 1, English (UK), Not Expanded by / Condensed by

| age 9: [287] Formatted | Microsoft Office User | 11/29/17 11:26:00 AM |

ont color: Text 1, English (UK), Not Expanded by / Condensed by

| age 9: [287] Formatted | Microsoft Office User | 11/29/17 11:26:00 AM |

ont color: Text 1, English (UK), Not Expanded by / Condensed by

| age 9: [287] Formatted | Microsoft Office User | 11/29/17 11:26:00 AM |

ont color: Text 1, English (UK), Not Expanded by / Condensed by

| Page 9: [287] Formatted | Microsoft Office User | 11/29/17 11:26:00 AM |

Font color: Text 1, English (UK), Not Expanded by / Condensed by

| Page 9: [287] Formatted | Microsoft Office User | 11/29/17 11:26:00 AM |

Font color: Text 1, English (UK), Not Expanded by / Condensed by

| Page 9: [287] Formatted | Microsoft Office User | 11/29/17 11:26:00 AM |

Font color: Text 1, English (UK), Not Expanded by / Condensed by

| Page 9: [287] Formatted | Microsoft Office User | 11/29/17 11:26:00 AM |

Font color: Text 1, English (UK), Not Expanded by / Condensed by

| Page 9: [287] Formatted | Microsoft Office User | 11/29/17 11:26:00 AM |

Font color: Text 1, English (UK), Not Expanded by / Condensed by

| Page 9: [287] Formatted | Microsoft Office User | 11/29/17 11:26:00 AM |

Font color: Text 1, English (UK), Not Expanded by / Condensed by

| Page 9: [287] Formatted | Microsoft Office User | 11/29/17 11:26:00 AM |

Font color: Text 1, English (UK), Not Expanded by / Condensed by

| Page 9: [287] Formatted | Microsoft Office User | 11/29/17 11:26:00 AM |

Font color: Text 1, English (UK), Not Expanded by / Condensed by

| Page 9: [287] Formatted | Microsoft Office User | 11/29/17 11:26:00 AM |

Font color: Text 1, English (UK), Not Expanded by / Condensed by

| Page 9: [287] Formatted | Microsoft Office User | 11/29/17 11:26:00 AM |

Font color: Text 1, English (UK), Not Expanded by / Condensed by

| Page 9: [287] Formatted | Microsoft Office User | 11/29/17 11:26:00 AM |

Font color: Text 1, English (UK), Not Expanded by / Condensed by

| Page 9: [287] Formatted | Microsoft Office User | 11/29/17 11:26:00 AM |

Font color: Text 1, English (UK), Not Expanded by / Condensed by

| Page 9: [287] Formatted | Microsoft Office User | 11/29/17 11:26:00 AM |

Font color: Text 1, English (UK), Not Expanded by / Condensed by

| Page 9: [287] Formatted | Microsoft Office User | 11/29/17 11:26:00 AM |

Font color: Text 1, English (UK), Not Expanded by / Condensed by

| Page 9: [287] Formatted | Microsoft Office User | 11/29/17 11:26:00 AM |

Font color: Text 1, English (UK), Not Expanded by / Condensed by

| Page 9: [287] Formatted | Microsoft Office User | 11/29/17 11:26:00 AM |

Font color: Text 1, English (UK), Not Expanded by / Condensed by

| Page 9: [287] Formatted | Microsoft Office User | 11/29/17 11:26:00 AM |

Font color: Text 1, English (UK), Not Expanded by / Condensed by

| Page 9: [288] Deleted | Microsoft Office User | 11/29/17 11:26:00 AM |

Pan et al., 2016; Zhang et al., 2002). In this sequence, the anticyclone was initially in the Tibetan mode (Aug 18, center of the

anticyclone was located near the  southern

| Page 10: [289] Deleted | Microsoft Office User | 11/29/17 11:26:00 AM |

10  edge of the Tibetan Plateau) and migrated westward toward the Iranian mode (Aug 20, center of the anticyclone located west of

70° E). It then continued to elongate and migrate further west (Aug 24), eventually the center splits and the anticyclone was in

a double center phase (Aug 26). The spatial distribution of the CO enhancement during this period indicates a similar evolution,

following a region of relatively high CO as it migrates together with the center of the anticyclone. Although its horizontal

resolution is limited, MLS successfully captures the day to day co-variability of CO with the dynamical fields. This migration

15 of the anticyclone center was also found in terms of potential vorticity (Garny and Randel, 2013) and model simulated CO (Pan

et al., 2016).

The IASI 12-16 km CO maps during the same period are shown in Fig. 6. The sequence of maps shows that the daily UT CO

enhancement over the ASM region are well detected by IASI retrievals, and the spatial distributions are generally co-located

with the anticyclone as represented by the 150 hPa GPH field. Compared to the MLS, the spatial distribution shows many

20 additional finer scale structures (note the color scales in Figs. 5 and 6 are different). On the Aug 18 and 20, IASI 12-16 km CO

are much more extended in longitudinal range compared to the MLS, co-located and mimic the east-west extent of the

anticyclone. The distributions Aug 24 and 26, in contrast, show clear double centered structure in CO enhancement that are not

as well co-located with the 150 hPa GPH distribution. In addition, the IASI CO field shows evidence of eastward shedding over

the western Pacific, which subsequently migrates southward following the anticyclonic flow. Additional CO enhancement

5        features appear on the south of the anticyclone migrating toward southwest following the cross–equator flow. It is not clear if

his feature originated from the shedding of anticyclone trapped UT air or if it is from an additional boundary layer source.

| Page 11: [290] Formatted | Microsoft Office User | 11/29/17 11:26:00 AM |

ont color: Text 1, English (UK), Not Expanded by / Condensed by

| Page 11: [290] Formatted | Microsoft Office User | 11/29/17 11:26:00 AM |

ont color: Text 1, English (UK), Not Expanded by / Condensed by

| Page 11: [290] Formatted | Microsoft Office User | 11/29/17 11:26:00 AM |

ont color: Text 1, English (UK), Not Expanded by / Condensed by

| Page 11: [290] Formatted | Microsoft Office User | 11/29/17 11:26:00 AM |

ont color: Text 1, English (UK), Not Expanded by / Condensed by

| Page 11: [290] Formatted | Microsoft Office User | 11/29/17 11:26:00 AM |

ont color: Text 1, English (UK), Not Expanded by / Condensed by

| Page 11: [290] Formatted | Microsoft Office User | 11/29/17 11:26:00 AM |

ont color: Text 1, English (UK), Not Expanded by / Condensed by

| Page 11: [290] Formatted | Microsoft Office User | 11/29/17 11:26:00 AM |

ont color: Text 1, English (UK), Not Expanded by / Condensed by

| Page 11: [290] Formatted | Microsoft Office User | 11/29/17 11:26:00 AM |

ont color: Text 1, English (UK), Not Expanded by / Condensed by

| Page 11: [290] Formatted | Microsoft Office User | 11/29/17 11:26:00 AM |

ont color: Text 1, English (UK), Not Expanded by / Condensed by

| age 11: [290] Formatted | Microsoft Office User | 11/29/17 11:26:00 AM |
|---|---|---|

ont color: Text 1, English (UK), Not Expanded by / Condensed by

| age 11: [290] Formatted | Microsoft Office User | 11/29/17 11:26:00 AM |
|---|---|---|

ont color: Text 1, English (UK), Not Expanded by / Condensed by

| age 11: [290] Formatted | Microsoft Office User | 11/29/17 11:26:00 AM |
|---|---|---|

ont color: Text 1, English (UK), Not Expanded by / Condensed by

| age 11: [290] Formatted | Microsoft Office User | 11/29/17 11:26:00 AM |
|---|---|---|

ont color: Text 1, English (UK), Not Expanded by / Condensed by

| age 11: [290] Formatted | Microsoft Office User | 11/29/17 11:26:00 AM |
|---|---|---|

ont color: Text 1, English (UK), Not Expanded by / Condensed by

| age 11: [290] Formatted | Microsoft Office User | 11/29/17 11:26:00 AM |
|---|---|---|

ont color: Text 1, English (UK), Not Expanded by / Condensed by

| age 11: [290] Formatted | Microsoft Office User | 11/29/17 11:26:00 AM |
|---|---|---|

ont color: Text 1, English (UK), Not Expanded by / Condensed by

| age 11: [290] Formatted | Microsoft Office User | 11/29/17 11:26:00 AM |
|---|---|---|

ont color: Text 1, English (UK), Not Expanded by / Condensed by

| age 11: [290] Formatted | Microsoft Office User | 11/29/17 11:26:00 AM |
|---|---|---|

ont color: Text 1, English (UK), Not Expanded by / Condensed by

| age 11: [290] Formatted | Microsoft Office User | 11/29/17 11:26:00 AM |
|---|---|---|

ont color: Text 1, English (UK), Not Expanded by / Condensed by

| age 11: [290] Formatted | Microsoft Office User | 11/29/17 11:26:00 AM |
|---|---|---|

ont color: Text 1, English (UK), Not Expanded by / Condensed by

| age 11: [290] Formatted | Microsoft Office User | 11/29/17 11:26:00 AM |
|---|---|---|

ont color: Text 1, English (UK), Not Expanded by / Condensed by

| age 11: [290] Formatted | Microsoft Office User | 11/29/17 11:26:00 AM |
|---|---|---|

ont color: Text 1, English (UK), Not Expanded by / Condensed by

| age 11: [290] Formatted | Microsoft Office User | 11/29/17 11:26:00 AM |
|---|---|---|

ont color: Text 1, English (UK), Not Expanded by / Condensed by

| age 11: [290] Formatted | Microsoft Office User | 11/29/17 11:26:00 AM |
|---|---|---|

ont color: Text 1, English (UK), Not Expanded by / Condensed by

| age 11: [290] Formatted | Microsoft Office User | 11/29/17 11:26:00 AM |
|---|---|---|

ont color: Text 1, English (UK), Not Expanded by / Condensed by

| age 11: [290] Formatted | Microsoft Office User | 11/29/17 11:26:00 AM |
|---|---|---|

ont color: Text 1, English (UK), Not Expanded by / Condensed by

| age 11: [290] Formatted | Microsoft Office User | 11/29/17 11:26:00 AM |
|---|---|---|

ont color: Text 1, English (UK), Not Expanded by / Condensed by

| Page 11: [290] Formatted | Microsoft Office User | 11/29/17 11:26:00 AM |
|---|---|---|

Font color: Text 1, English (UK), Not Expanded by / Condensed by

| Page 11: [290] Formatted | Microsoft Office User | 11/29/17 11:26:00 AM |
|---|---|---|

Font color: Text 1, English (UK), Not Expanded by / Condensed by

| Page 11: [290] Formatted | Microsoft Office User | 11/29/17 11:26:00 AM |
|---|---|---|

Font color: Text 1, English (UK), Not Expanded by / Condensed by

| Page 11: [290] Formatted | Microsoft Office User | 11/29/17 11:26:00 AM |
|---|---|---|

Font color: Text 1, English (UK), Not Expanded by / Condensed by

| Page 11: [290] Formatted | Microsoft Office User | 11/29/17 11:26:00 AM |
|---|---|---|

Font color: Text 1, English (UK), Not Expanded by / Condensed by

| Page 11: [290] Formatted | Microsoft Office User | 11/29/17 11:26:00 AM |
|---|---|---|

Font color: Text 1, English (UK), Not Expanded by / Condensed by

| Page 11: [290] Formatted | Microsoft Office User | 11/29/17 11:26:00 AM |
|---|---|---|

Font color: Text 1, English (UK), Not Expanded by / Condensed by

| Page 11: [290] Formatted | Microsoft Office User | 11/29/17 11:26:00 AM |
|---|---|---|

Font color: Text 1, English (UK), Not Expanded by / Condensed by

| Page 11: [290] Formatted | Microsoft Office User | 11/29/17 11:26:00 AM |
|---|---|---|

Font color: Text 1, English (UK), Not Expanded by / Condensed by

| Page 11: [290] Formatted | Microsoft Office User | 11/29/17 11:26:00 AM |
|---|---|---|

Font color: Text 1, English (UK), Not Expanded by / Condensed by

| Page 11: [290] Formatted | Microsoft Office User | 11/29/17 11:26:00 AM |
|---|---|---|

Font color: Text 1, English (UK), Not Expanded by / Condensed by

| Page 11: [290] Formatted | Microsoft Office User | 11/29/17 11:26:00 AM |
|---|---|---|

Font color: Text 1, English (UK), Not Expanded by / Condensed by

| Page 11: [291] Formatted | Microsoft Office User | 11/29/17 11:26:00 AM |
|---|---|---|

Font color: Text 1, English (UK)

| Page 11: [291] Formatted | Microsoft Office User | 11/29/17 11:26:00 AM |
|---|---|---|

Font color: Text 1, English (UK)

| Page 11: [292] Formatted | Microsoft Office User | 11/29/17 11:26:00 AM |
|---|---|---|

Font color: Text 1, English (UK)

| Page 11: [292] Formatted | Microsoft Office User | 11/29/17 11:26:00 AM |
|---|---|---|

Font color: Text 1, English (UK)

| Page 11: [292] Formatted | Microsoft Office User | 11/29/17 11:26:00 AM |
|---|---|---|

Font color: Text 1, English (UK)

| Page 11: [293] Formatted | Microsoft Office User | 11/29/17 11:26:00 AM |
|---|---|---|

Font:10 pt, Font color: Text 1, English (UK), Not Expanded by / Condensed by

| Page 11: [293] Formatted | Microsoft Office User | 11/29/17 11:26:00 AM |
|---|---|---|

Font:10 pt, Font color: Text 1, English (UK), Not Expanded by / Condensed by

Figure

6 indicates that, despite the limited vertical resolution, IASI contains sufficient information to detect UT CO enhancement. To further examine the consistency of the IASI CO profile vertical structure with the circulation in the monsoon region, Fig. 7 shows a pressure-latitude cross-section of IASI CO for a single day during the monsoon season at 90° E (where CO is persistently high). Enhanced CO exists throughout the middle and upper troposphere between 20°N and 30°N, with a

5 plume of CO that extends from the surface to the tropopause near 20° N. While the limited vertical resolution of IASI no doubt smoothes the distribution of CO, it is important that IASI is able to detect the signature of upward transport of CO associated with the region of strong vertical winds over Northern India and the southern flank of the Tibetan Plateau. Overall, the vertical structure of the CO enhancement is consistent with the flow field, i.e., the UT CO enhancement is co-located with deep ascending branch of the monsoon Hadley cell (Wang, 2006). For more discussion on the climatological flow structure in the

10 meridional plane, see analyses in Zhang et al. (2002).

To characterize the sub-seasonal variability represented by MLS and IASI CO, Fig. 8 displays

n the ASM region (15-35° N, 0-150°E) during JJA 2012. We examine the zonal propagation of anomalies and compare the migration of dynamical and chemical features from different datasets throughout the season.
Ⓒ Author(s) 2017. CC BY 3.0 License.

anomalies (Fig. 8a) migrate across the region approximately every 10-20

5 days. MLS CO anomalies also show westward migration but with amplitudes and timing that differ from GPH. However, the westward migration is less regular and limited to east of 30°E for IASI CO anomalies, although they do exhibit the persistent positive anomaly between 60-90°E that is common to all three quantities. Interestingly, IASI CO anomalies show eastward migrating features between 120° and 150° E, as shown in Fig. 6. This type

Font color: Text 1, English (UK)

hedding event was seen in the model analysis, following the western Pacific mode of the anticyclone (Pan et al., 2016). We speculate

| age 13: [301] Formatted | Microsoft Office User | 11/29/17 11:26:00 AM |
|---|---|---|
| ont color: Text 1, English (UK) | | |

| age 13: [302] Formatted | Microsoft Office User | 11/29/17 11:26:00 AM |
|---|---|---|
| ont color: Text 1, English (UK) | | |

| age 13: [303] Formatted | Microsoft Office User | 11/29/17 11:26:00 AM |
|---|---|---|
| ont color: Text 1, English (UK) | | |

| age 13: [304] Deleted | Microsoft Office User | 11/29/17 11:26:00 AM |
|---|---|---|
| trongly impacted by the sub-seasonal scale east-west oscillation of the ASM anticyclone. MLS CO and IASI CO agree on regional | | |

| age 13: [305] Formatted | Microsoft Office User | 11/29/17 11:26:00 AM |
|---|---|---|
| ont color: Text 1, English (UK) | | |

| age 13: [306] Formatted | Microsoft Office User | 11/29/17 11:26:00 AM |
|---|---|---|
| leading 1, Indent: Left:  0", Space Before:  0 pt | | |

| age 13: [307] Deleted | Microsoft Office User | 11/29/17 11:26:00 AM |
|---|---|---|

30 mean fluctuations despite the fact that the two sensors are influenced by different over vertical columns. The sub-seasonal variability exhibited in the CO data during the Asian Summer Monsoon is broadly consistent with atmospheric model output (Pan et al., 2016). MLS and IASI data have different advantages. MLS data are better for examining features with a shallow vertical extent such as the ASM anticyclone provided those features have a large enough horizontal scale. IASI CO data have a higher horizontal resolution and are able to detect the impacts of vertical transport in the troposphere. The spatial distribution

———————————————————Section Break (Next Page)———————————————————

of IASI CO in UTLS shows additional finer scale structures than MLS and the eastward shedding over the western Pacific is evident in IASI CO field.

.2.

| age 13: [308] Formatted | Microsoft Office User | 11/29/17 11:26:00 AM |
|---|---|---|
| ont color: Text 1 | | |

| age 13: [308] Formatted | Microsoft Office User | 11/29/17 11:26:00 AM |
|---|---|---|
| ont color: Text 1 | | |

| age 13: [308] Formatted | Microsoft Office User | 11/29/17 11:26:00 AM |
|---|---|---|
| ont color: Text 1 | | |

| age 13: [309] Formatted | Microsoft Office User | 11/29/17 11:26:00 AM |
|---|---|---|
| ont color: Text 1 | | |

| age 13: [310] Formatted | Microsoft Office User | 11/29/17 11:26:00 AM |
|---|---|---|
| Jormal, Left, Right:  0", Space Before:  0 pt, Line spacing:  1.5 lines | | |

| age 13: [311] Formatted | Microsoft Office User | 11/29/17 11:26:00 AM |
|---|---|---|
| ont color: Text 1, English (UK) | | |

| age 13: [312] Formatted | Microsoft Office User | 11/29/17 11:26:00 AM |
|---|---|---|
| ont color: Text 1, English (UK) | | |

| age 13: [313] Formatted | Microsoft Office User | 11/29/17 11:26:00 AM |
|---|---|---|
| ont:10 pt, Font color: Text 1, English (UK), Not Raised by / Lowered by | | |

| age 13: [313] Formatted | Microsoft Office User | 11/29/17 11:26:00 AM |
|---|---|---|
| ont:10 pt, Font color: Text 1, English (UK), Not Raised by / Lowered by | | |

| age 13: [314] Deleted | Microsoft Office User | 11/29/17 11:26:00 AM |
|---|---|---|

| age 13: [315] Formatted | Microsoft Office User | 11/29/17 11:26:00 AM |
|---|---|---|
| ont:10 pt, Font color: Text 1, English (UK) | | |

| age 13: [315] Formatted | Microsoft Office User | 11/29/17 11:26:00 AM |
|---|---|---|
| ont:10 pt, Font color: Text 1, English (UK) | | |

| age 13: [315] Formatted | Microsoft Office User | 11/29/17 11:26:00 AM |
|---|---|---|
| ont:10 pt, Font color: Text 1, English (UK) | | |

| age 13: [315] Formatted | Microsoft Office User | 11/29/17 11:26:00 AM |
|---|---|---|
| ont:10 pt, Font color: Text 1, English (UK) | | |

| age 13: [316] Formatted | Microsoft Office User | 11/29/17 11:26:00 AM |
|---|---|---|
| ont color: Text 1, English (UK) | | |

| age 13: [316] Formatted | Microsoft Office User | 11/29/17 11:26:00 AM |
| --- | --- | --- |
| ont color: Text 1, English (UK) | | |
| age 13: [316] Formatted | Microsoft Office User | 11/29/17 11:26:00 AM |
| ont color: Text 1, English (UK) | | |
| age 13: [316] Formatted | Microsoft Office User | 11/29/17 11:26:00 AM |
| ont color: Text 1, English (UK) | | |
| age 13: [316] Formatted | Microsoft Office User | 11/29/17 11:26:00 AM |
| ont color: Text 1, English (UK) | | |
| age 13: [316] Formatted | Microsoft Office User | 11/29/17 11:26:00 AM |
| ont color: Text 1, English (UK) | | |
| age 13: [316] Formatted | Microsoft Office User | 11/29/17 11:26:00 AM |
| ont color: Text 1, English (UK) | | |
| age 13: [316] Formatted | Microsoft Office User | 11/29/17 11:26:00 AM |
| ont color: Text 1, English (UK) | | |
| age 13: [316] Formatted | Microsoft Office User | 11/29/17 11:26:00 AM |
| ont color: Text 1, English (UK) | | |
| age 13: [316] Formatted | Microsoft Office User | 11/29/17 11:26:00 AM |
| ont color: Text 1, English (UK) | | |
| age 13: [316] Formatted | Microsoft Office User | 11/29/17 11:26:00 AM |
| ont color: Text 1, English (UK) | | |
| age 13: [316] Formatted | Microsoft Office User | 11/29/17 11:26:00 AM |
| ont color: Text 1, English (UK) | | |
| age 13: [317] Formatted | Microsoft Office User | 11/29/17 11:26:00 AM |
| ont color: Text 1, English (UK), Not Expanded by / Condensed by | | |
| age 13: [317] Formatted | Microsoft Office User | 11/29/17 11:26:00 AM |
| ont color: Text 1, English (UK), Not Expanded by / Condensed by | | |
| age 13: [317] Formatted | Microsoft Office User | 11/29/17 11:26:00 AM |
| ont color: Text 1, English (UK), Not Expanded by / Condensed by | | |
| age 13: [317] Formatted | Microsoft Office User | 11/29/17 11:26:00 AM |
| ont color: Text 1, English (UK), Not Expanded by / Condensed by | | |
| age 13: [317] Formatted | Microsoft Office User | 11/29/17 11:26:00 AM |
| ont color: Text 1, English (UK), Not Expanded by / Condensed by | | |
| age 13: [317] Formatted | Microsoft Office User | 11/29/17 11:26:00 AM |
| ont color: Text 1, English (UK), Not Expanded by / Condensed by | | |
| age 13: [317] Formatted | Microsoft Office User | 11/29/17 11:26:00 AM |
| ont color: Text 1, English (UK), Not Expanded by / Condensed by | | |
| age 13: [317] Formatted | Microsoft Office User | 11/29/17 11:26:00 AM |
| ont color: Text 1, English (UK), Not Expanded by / Condensed by | | |

**age 13: [317] Formatted**          **Microsoft Office User**          **11/29/17 11:26:00 AM**

ont color: Text 1, English (UK), Not Expanded by / Condensed by

**age 13: [318] Formatted**          **Microsoft Office User**          **11/29/17 11:26:00 AM**

ont color: Text 1, English (UK), Not Expanded by / Condensed by

**age 13: [318] Formatted**          **Microsoft Office User**          **11/29/17 11:26:00 AM**

ont color: Text 1, English (UK), Not Expanded by / Condensed by

**age 13: [318] Formatted**          **Microsoft Office User**          **11/29/17 11:26:00 AM**

ont color: Text 1, English (UK), Not Expanded by / Condensed by

**age 13: [318] Formatted**          **Microsoft Office User**          **11/29/17 11:26:00 AM**

ont color: Text 1, English (UK), Not Expanded by / Condensed by

**age 13: [318] Formatted**          **Microsoft Office User**          **11/29/17 11:26:00 AM**

ont color: Text 1, English (UK), Not Expanded by / Condensed by

**age 13: [318] Formatted**          **Microsoft Office User**          **11/29/17 11:26:00 AM**

ont color: Text 1, English (UK), Not Expanded by / Condensed by

**age 13: [318] Formatted**          **Microsoft Office User**          **11/29/17 11:26:00 AM**

ont color: Text 1, English (UK), Not Expanded by / Condensed by

**age 13: [318] Formatted**          **Microsoft Office User**          **11/29/17 11:26:00 AM**

ont color: Text 1, English (UK), Not Expanded by / Condensed by

**age 13: [318] Formatted**          **Microsoft Office User**          **11/29/17 11:26:00 AM**

ont color: Text 1, English (UK), Not Expanded by / Condensed by

**age 13: [318] Formatted**          **Microsoft Office User**          **11/29/17 11:26:00 AM**

ont color: Text 1, English (UK), Not Expanded by / Condensed by

**age 13: [318] Formatted**          **Microsoft Office User**          **11/29/17 11:26:00 AM**

ont color: Text 1, English (UK), Not Expanded by / Condensed by

**age 13: [318] Formatted**          **Microsoft Office User**          **11/29/17 11:26:00 AM**

ont color: Text 1, English (UK), Not Expanded by / Condensed by

**age 13: [318] Formatted**          **Microsoft Office User**          **11/29/17 11:26:00 AM**

ont color: Text 1, English (UK), Not Expanded by / Condensed by

**age 13: [318] Formatted**          **Microsoft Office User**          **11/29/17 11:26:00 AM**

ont color: Text 1, English (UK), Not Expanded by / Condensed by

**age 13: [318] Formatted**          **Microsoft Office User**          **11/29/17 11:26:00 AM**

ont color: Text 1, English (UK), Not Expanded by / Condensed by

| age 13: [318] Formatted | Microsoft Office User | 11/29/17 11:26:00 AM |

ont color: Text 1, English (UK), Not Expanded by / Condensed by

| age 13: [318] Formatted | Microsoft Office User | 11/29/17 11:26:00 AM |

ont color: Text 1, English (UK), Not Expanded by / Condensed by

| age 13: [318] Formatted | Microsoft Office User | 11/29/17 11:26:00 AM |

ont color: Text 1, English (UK), Not Expanded by / Condensed by

| age 13: [318] Formatted | Microsoft Office User | 11/29/17 11:26:00 AM |

ont color: Text 1, English (UK), Not Expanded by / Condensed by

| age 13: [318] Formatted | Microsoft Office User | 11/29/17 11:26:00 AM |

ont color: Text 1, English (UK), Not Expanded by / Condensed by

| age 13: [318] Formatted | Microsoft Office User | 11/29/17 11:26:00 AM |

ont color: Text 1, English (UK), Not Expanded by / Condensed by

| age 13: [318] Formatted | Microsoft Office User | 11/29/17 11:26:00 AM |

ont color: Text 1, English (UK), Not Expanded by / Condensed by

| age 13: [318] Formatted | Microsoft Office User | 11/29/17 11:26:00 AM |

ont color: Text 1, English (UK), Not Expanded by / Condensed by

| age 13: [318] Formatted | Microsoft Office User | 11/29/17 11:26:00 AM |

ont color: Text 1, English (UK), Not Expanded by / Condensed by

| age 13: [318] Formatted | Microsoft Office User | 11/29/17 11:26:00 AM |

ont color: Text 1, English (UK), Not Expanded by / Condensed by

| age 13: [318] Formatted | Microsoft Office User | 11/29/17 11:26:00 AM |

ont color: Text 1, English (UK), Not Expanded by / Condensed by

| age 13: [318] Formatted | Microsoft Office User | 11/29/17 11:26:00 AM |

ont color: Text 1, English (UK), Not Expanded by / Condensed by

| age 13: [318] Formatted | Microsoft Office User | 11/29/17 11:26:00 AM |

ont color: Text 1, English (UK), Not Expanded by / Condensed by

| age 13: [318] Formatted | Microsoft Office User | 11/29/17 11:26:00 AM |

ont color: Text 1, English (UK), Not Expanded by / Condensed by

| age 13: [318] Formatted | Microsoft Office User | 11/29/17 11:26:00 AM |

ont color: Text 1, English (UK), Not Expanded by / Condensed by

| age 13: [318] Formatted | Microsoft Office User | 11/29/17 11:26:00 AM |

ont color: Text 1, English (UK), Not Expanded by / Condensed by

| age 13: [318] Formatted | Microsoft Office User | 11/29/17 11:26:00 AM |

ont color: Text 1, English (UK), Not Expanded by / Condensed by

| age 13: [318] Formatted | Microsoft Office User | 11/29/17 11:26:00 AM |

ont color: Text 1, English (UK), Not Expanded by / Condensed by

| age 13: [318] Formatted | Microsoft Office User | 11/29/17 11:26:00 AM |
|---|---|---|

ont color: Text 1, English (UK), Not Expanded by / Condensed by

| age 13: [318] Formatted | Microsoft Office User | 11/29/17 11:26:00 AM |
|---|---|---|

ont color: Text 1, English (UK), Not Expanded by / Condensed by

| age 13: [318] Formatted | Microsoft Office User | 11/29/17 11:26:00 AM |
|---|---|---|

ont color: Text 1, English (UK), Not Expanded by / Condensed by

| age 13: [318] Formatted | Microsoft Office User | 11/29/17 11:26:00 AM |
|---|---|---|

ont color: Text 1, English (UK), Not Expanded by / Condensed by

| age 13: [318] Formatted | Microsoft Office User | 11/29/17 11:26:00 AM |
|---|---|---|

ont color: Text 1, English (UK), Not Expanded by / Condensed by

| age 13: [318] Formatted | Microsoft Office User | 11/29/17 11:26:00 AM |
|---|---|---|

ont color: Text 1, English (UK), Not Expanded by / Condensed by

| age 13: [318] Formatted | Microsoft Office User | 11/29/17 11:26:00 AM |
|---|---|---|

ont color: Text 1, English (UK), Not Expanded by / Condensed by

| age 13: [318] Formatted | Microsoft Office User | 11/29/17 11:26:00 AM |
|---|---|---|

ont color: Text 1, English (UK), Not Expanded by / Condensed by

| age 13: [318] Formatted | Microsoft Office User | 11/29/17 11:26:00 AM |
|---|---|---|

ont color: Text 1, English (UK), Not Expanded by / Condensed by

| age 13: [318] Formatted | Microsoft Office User | 11/29/17 11:26:00 AM |
|---|---|---|

ont color: Text 1, English (UK), Not Expanded by / Condensed by

| age 13: [318] Formatted | Microsoft Office User | 11/29/17 11:26:00 AM |
|---|---|---|

ont color: Text 1, English (UK), Not Expanded by / Condensed by

| age 13: [318] Formatted | Microsoft Office User | 11/29/17 11:26:00 AM |
|---|---|---|

ont color: Text 1, English (UK), Not Expanded by / Condensed by

| age 13: [319] Formatted | Microsoft Office User | 11/29/17 11:26:00 AM |
|---|---|---|

ont color: Text 1, English (UK)

| age 13: [319] Formatted | Microsoft Office User | 11/29/17 11:26:00 AM |
|---|---|---|

ont color: Text 1, English (UK)

| age 13: [319] Formatted | Microsoft Office User | 11/29/17 11:26:00 AM |
|---|---|---|

ont color: Text 1, English (UK)

| age 13: [319] Formatted | Microsoft Office User | 11/29/17 11:26:00 AM |
|---|---|---|

ont color: Text 1, English (UK)

| age 13: [320] Formatted | Microsoft Office User | 11/29/17 11:26:00 AM |
|---|---|---|

ont color: Text 1, English (UK), Not Expanded by / Condensed by

| age 13: [320] Formatted | Microsoft Office User | 11/29/17 11:26:00 AM |
|---|---|---|

ont color: Text 1, English (UK), Not Expanded by / Condensed by

| Page 13: [320] Formatted | Microsoft Office User | 11/29/17 11:26:00 AM |

Font color: Text 1, English (UK), Not Expanded by / Condensed by

| Page 13: [321] Formatted | Microsoft Office User | 11/29/17 11:26:00 AM |

Font color: Text 1, English (UK), Not Expanded by / Condensed by

| Page 13: [321] Formatted | Microsoft Office User | 11/29/17 11:26:00 AM |

Font color: Text 1, English (UK), Not Expanded by / Condensed by

| Page 13: [321] Formatted | Microsoft Office User | 11/29/17 11:26:00 AM |

Font color: Text 1, English (UK), Not Expanded by / Condensed by

| Page 13: [322] Formatted | Microsoft Office User | 11/29/17 11:26:00 AM |

Font color: Text 1, English (UK)

| Page 13: [323] Formatted | Microsoft Office User | 11/29/17 11:26:00 AM |

Font color: Text 1, English (UK), Not Expanded by / Condensed by

| Page 13: [323] Formatted | Microsoft Office User | 11/29/17 11:26:00 AM |

Font color: Text 1, English (UK), Not Expanded by / Condensed by

| Page 13: [323] Formatted | Microsoft Office User | 11/29/17 11:26:00 AM |

Font color: Text 1, English (UK), Not Expanded by / Condensed by

| Page 13: [323] Formatted | Microsoft Office User | 11/29/17 11:26:00 AM |

Font color: Text 1, English (UK), Not Expanded by / Condensed by

| Page 13: [323] Formatted | Microsoft Office User | 11/29/17 11:26:00 AM |

Font color: Text 1, English (UK), Not Expanded by / Condensed by

| Page 13: [323] Formatted | Microsoft Office User | 11/29/17 11:26:00 AM |

Font color: Text 1, English (UK), Not Expanded by / Condensed by

| Page 13: [324] Deleted | Microsoft Office User | 11/29/17 11:26:00 AM |

0

| Page 13: [325] Formatted | Microsoft Office User | 11/29/17 11:26:00 AM |

Font:10 pt, Font color: Text 1, English (UK)

| Page 13: [325] Formatted | Microsoft Office User | 11/29/17 11:26:00 AM |

Font:10 pt, Font color: Text 1, English (UK)

| Page 13: [326] Formatted | Microsoft Office User | 11/29/17 11:26:00 AM |

Font color: Text 1, English (UK)

| Page 13: [326] Formatted | Microsoft Office User | 11/29/17 11:26:00 AM |

Font color: Text 1, English (UK)

| Page 13: [326] Formatted | Microsoft Office User | 11/29/17 11:26:00 AM |

Font color: Text 1, English (UK)

| Page 13: [326] Formatted | Microsoft Office User | 11/29/17 11:26:00 AM |

Font color: Text 1, English (UK)

| Page 13: [326] Formatted | Microsoft Office User | 11/29/17 11:26:00 AM |

ont color: Text 1, English (UK)

| age 13: [326] Formatted | Microsoft Office User | 11/29/17 11:26:00 AM |
|---|---|---|

ont color: Text 1, English (UK)

| age 13: [326] Formatted | Microsoft Office User | 11/29/17 11:26:00 AM |
|---|---|---|

ont color: Text 1, English (UK)

| age 13: [326] Formatted | Microsoft Office User | 11/29/17 11:26:00 AM |
|---|---|---|

ont color: Text 1, English (UK)

| age 13: [326] Formatted | Microsoft Office User | 11/29/17 11:26:00 AM |
|---|---|---|

ont color: Text 1, English (UK)

| age 13: [326] Formatted | Microsoft Office User | 11/29/17 11:26:00 AM |
|---|---|---|

ont color: Text 1, English (UK)

| age 13: [326] Formatted | Microsoft Office User | 11/29/17 11:26:00 AM |
|---|---|---|

ont color: Text 1, English (UK)

| age 13: [326] Formatted | Microsoft Office User | 11/29/17 11:26:00 AM |
|---|---|---|

ont color: Text 1, English (UK)

| age 13: [326] Formatted | Microsoft Office User | 11/29/17 11:26:00 AM |
|---|---|---|

ont color: Text 1, English (UK)

| age 13: [326] Formatted | Microsoft Office User | 11/29/17 11:26:00 AM |
|---|---|---|

ont color: Text 1, English (UK)

| age 13: [326] Formatted | Microsoft Office User | 11/29/17 11:26:00 AM |
|---|---|---|

ont color: Text 1, English (UK)

| age 13: [326] Formatted | Microsoft Office User | 11/29/17 11:26:00 AM |
|---|---|---|

ont color: Text 1, English (UK)

| age 13: [326] Formatted | Microsoft Office User | 11/29/17 11:26:00 AM |
|---|---|---|

ont color: Text 1, English (UK)

| age 13: [326] Formatted | Microsoft Office User | 11/29/17 11:26:00 AM |
|---|---|---|

ont color: Text 1, English (UK)

| age 13: [327] Formatted | Microsoft Office User | 11/29/17 11:26:00 AM |
|---|---|---|

ont color: Text 1, English (UK), Not Expanded by / Condensed by

| age 13: [327] Formatted | Microsoft Office User | 11/29/17 11:26:00 AM |
|---|---|---|

ont color: Text 1, English (UK), Not Expanded by / Condensed by

| age 13: [327] Formatted | Microsoft Office User | 11/29/17 11:26:00 AM |
|---|---|---|

ont color: Text 1, English (UK), Not Expanded by / Condensed by

| age 13: [327] Formatted | Microsoft Office User | 11/29/17 11:26:00 AM |
|---|---|---|

ont color: Text 1, English (UK), Not Expanded by / Condensed by

| Page 13: [327] Formatted | Microsoft Office User | 11/29/17 11:26:00 AM |
|---|---|---|

Font color: Text 1, English (UK), Not Expanded by / Condensed by

| Page 13: [327] Formatted | Microsoft Office User | 11/29/17 11:26:00 AM |
|---|---|---|

Font color: Text 1, English (UK), Not Expanded by / Condensed by

| Page 13: [327] Formatted | Microsoft Office User | 11/29/17 11:26:00 AM |
|---|---|---|

Font color: Text 1, English (UK), Not Expanded by / Condensed by

| Page 13: [327] Formatted | Microsoft Office User | 11/29/17 11:26:00 AM |
|---|---|---|

Font color: Text 1, English (UK), Not Expanded by / Condensed by

| Page 13: [327] Formatted | Microsoft Office User | 11/29/17 11:26:00 AM |
|---|---|---|

Font color: Text 1, English (UK), Not Expanded by / Condensed by

| Page 13: [328] Formatted | Microsoft Office User | 11/29/17 11:26:00 AM |
|---|---|---|

Font color: Text 1, English (UK)

| Page 13: [328] Formatted | Microsoft Office User | 11/29/17 11:26:00 AM |
|---|---|---|

Font color: Text 1, English (UK)

| Page 13: [328] Formatted | Microsoft Office User | 11/29/17 11:26:00 AM |
|---|---|---|

Font color: Text 1, English (UK)

| Page 13: [328] Formatted | Microsoft Office User | 11/29/17 11:26:00 AM |
|---|---|---|

Font color: Text 1, English (UK)

| Page 13: [328] Formatted | Microsoft Office User | 11/29/17 11:26:00 AM |
|---|---|---|

Font color: Text 1, English (UK)

| Page 13: [328] Formatted | Microsoft Office User | 11/29/17 11:26:00 AM |
|---|---|---|

Font color: Text 1, English (UK)

| Page 13: [328] Formatted | Microsoft Office User | 11/29/17 11:26:00 AM |
|---|---|---|

Font color: Text 1, English (UK)

| Page 13: [328] Formatted | Microsoft Office User | 11/29/17 11:26:00 AM |
|---|---|---|

Font color: Text 1, English (UK)

| Page 13: [328] Formatted | Microsoft Office User | 11/29/17 11:26:00 AM |
|---|---|---|

Font color: Text 1, English (UK)

| Page 13: [329] Formatted | Microsoft Office User | 11/29/17 11:26:00 AM |
|---|---|---|

Font color: Text 1, English (UK)

| Page 14: [330] Deleted | Microsoft Office User | 11/29/17 11:26:00 AM |
|---|---|---|

15 anticyclone as well as four distinct distributions that are indicative of the east-west oscillation of the ASM anticyclone: the eastern phase with the $O_3$ minimum centered between 60° and 90° E on July 16 (Fig. 10a), the western phase on Jul 19 (Fig. 10b), the elongated phase on Jul 20 (Fig. 10c), and a double-centered phase on Jul 23 (Fig. 10d). The distribution of $O_3$ from OMI also shows similar distribution of low $O_3$ inside the anticyclone but the $O_3$ inside the anticyclone is significantly lower than the MLS. Quantitatively, OMI data show a sharper transition of $O_3$ field across the edge of the anticyclone (as indicted by

20    the 105 hPa tropopause contour) and lower $O_3$ in the anticyclone. The differences between MLS and OMI are more pronounced at low latitudes (i.e., south of 40° N), where OMI data show finer scale structures.

[remaining 95,074 characters of this post omitted]

---

## Author Response (AR2)

Laura Pan on behalf of all co-authors

We thank the reviewer for many suggestions. Please see the point-by-point responses below.

**Point-by-point responses**

In general, the authors have done a good job of responding to referee comments. The manuscript has been substantially revised; in fact, in some places it has been almost entirely re-written. The writing and structural flow of the manuscript have been improved. In the process of editing to address reviewer comments, the authors have articulated a new emphasis on "process-based retrieval evaluation", whereby, unlike in more traditional validation approaches, satellite measurements are assessed qualitatively through their dynamical consistency with meteorological fields (e.g., winds, GPH). The authors stress the value of the nadir sounders' dense horizontal sampling, but in my view what is probably the most important result from this paper --- the observational evidence supporting the conceptual model of the preferred ASM vertical transport pathway --- was enabled more by the ability of IASI to discriminate variability in the upper troposphere from that in the lower/middle troposphere than by its horizontal resolution.

Because of the extensive nature of the revisions, I have more comments on this draft than is typical for a re-review. However, the vast majority of them should be straightforward to address.

General comment:

I still object to the way the vertical information content of the nadir sounder measurements is characterized. IASI data may be reported on 19 surfaces, but as George et al. [2009] point out, the number of retrieval altitude levels is not representative of the vertical resolution. The statement is made (P5, L16-19): "Averaging kernels for layers centred at 13.5, 14.5, and 15.5 km are highlighted in the figure, because these layers are relevant to the analysis, which is focused on the 150 hPa pressure level. This level is contributed mostly by IASI CO product layer centred at 14.5 km and with a small fraction from the 13.5 km and 15.5 km layers." I do not see the basis for this statement in Figure 1b. Layers from ~8 to ~18 km seem to contribute almost equally to the value reported at 14.5 km. That IASI cannot discriminate between 150 and 200 hPa is also clearly demonstrated in Figure 4 and accompanying discussion. Although the authors do acknowledge that IASI has maximum sensitivity in the middle troposphere (P5, L23-25), the language they employ could be misleading to many readers, especially those who do not go through the data description subsection carefully. Figure 1b shows that it is simply NOT possible to "focus on the 150 hPa level" with IASI CO data. Thus, the authors may choose to analyze the IASI measurements reported on the 14.5 km level in the data file, but those values represent an average over a broad region of the upper troposphere and cannot be labelled "IASI CO at 150 hPa", as is done in numerous places throughout the manuscript (e.g., P6, L19; P7, L10, L13, L21, L26, L29, L32, L33; P9, L29; P10, L5, L19; P11, L28; P14, L11, L12, L15; P21, L3; P22, L2; P23, L4; P27, L1; P29, L1). A similar comment applies to IASI data "at 500 hPa" (P9, L29; P10, L7). I think it is fair to say that IASI can distinguish between the lower/middle and the upper

troposphere, and that is how the data analyzed here should be referred to, not by association with a specific pressure level.

The same comment can be made for the characterization of OMI data. It is stated (P4, L35) that the "layer 18" OMI product is "comparable to the 100 hPa level", and that the analysis focuses on "100 hPa OMI O3" data (e.g., P12, L4, L14, L33; P14, L26, L28; P32, L2, etc.). Given that the vertical resolution of OMI profiles near 100 hPa is 11-12 km [Liu et al., 2010b], such a characterization is not justified.

Although this is a significant issue, it is fundamentally a question of semantics that is easily rectified by explicitly noting that the IASI and OMI data represent averages over relatively thick (10-12 km) layers in the upper and lower/middle troposphere and appropriately labelling them as such in a consistent manner throughout the manuscript, rather than referring to them as 500, 150, or 100 hPa values.

We agree with the reviewer that this is an issue of semantics. When we refer to the IASI 150 hPa, it is the designation as a retrieval product. Same as the OMI layer 18 be comparable to 100 hPa. The specific information contributing to the product is discussed in the section on averaging kernels, in the comparisons and so on. Smoothing error is an issue for all satellite data. MLS 147 hPa CO retrieval for example, is also contributed by a thick layer of several kilometers. To further minimize any possible confusion about this, we have added specific "disclaimer" sentences at the beginning.

Note that the averaging kernels are results of model analyses and they have limitations in what they can show themselves. The importance of this work is that we show by using process understanding, we can demonstrate that the IASI retrieval over the region indeed has ~ 2 pieces of independent information, which is reflected in the independent upper tropospheric and lower/middle tropospheric variability. This approach we are demonstrating is complementing the ability of the averaging kernels. Due to the strong tradition in the satellite retrieval field, the averaging kernels may have been held to the position beyond its actual information content. We think the value of this work is to demonstrate how much more we can learn if we let our understanding of the process help us in an open-minded fashion.

Specific substantive comments:

* P3, L23: MLS O3 is not mentioned in this sentence, but the Livesey et al. [2008] validation paper cited for MLS CO also covered UTLS O3.

Modified.

* P4, L1-2: In the MLS Data Quality Document cited here, the accuracies of the v4 CO and O3 data are given as the RSS or the sum of the ppbv/ppmv and percentage uncertainties, respectively. That is, the systematic uncertainty for 147 hPa CO should be quoted as the RSS of 26 ppbv and 30%, and for 100 hPa O3 it should be quoted as the sum of 5 ppbv + 7%. The multiplicative terms are probably not negligible even for O3, and certainly not for CO in the ASM anticyclone.

In addition, it would be good to add "single-profile" in front of "precision" in these lines, since the precision is substantially improved when profiles are averaged together.

Modified.

* P6, L26-31: The issues with IASI data over elevated terrain, discussed later in the manuscript, should probably be mentioned here as well as a possible factor in explaining some of the IASI/MLS differences. In addition, it might be good to clarify that the "missing data in IASI" (L27) arise because of cloud contamination.

Modified.

* P7, L33-34: "… the IASI 150 hPa product is significantly contributed by the atmosphere at lower UT levels, including the 200 hPa range". While this is no doubt true, based on Figure 1b it seems possible that some of the features seen in the IASI upper tropospheric seasonal map could have come from even lower in the atmosphere. It would be good to show the IASI JJA map for the lower/middle troposphere in Figure 4 as well. This is done for a daily map in Figure 7, but since the presence of a systematic vertical tilt in the chemical structure of the anticyclone is being argued on the basis of these seasonal maps, a plot representing the lower layer average might be illuminating.

We do not consider the average of lower to middle troposphere here is helpful. The upper level seasonal average is dominated by the dynamics of anticyclone which is not the case in the lower level. We understand the desire of the reviewer to further quantify the smoothing error, however we do not think there is enough information to do so here. The important point in this work is that the IASI retrieval in the region can discriminate the upper tropospheric variability from that in the middle and lower troposphere.

* P8, L30: It is stated (here and in the caption to Figure 6) that daily means are calculated over the longitude range 0-220E. But the study domain was previously characterized as extending over 0-180E (P5, L6), and the Hovmoller plots only cover that hemisphere as well, so it raises the question of why the daily means are calculated over a broader region.

This is because the study region is dominated by the three Highs. When calculating anomaly, it is necessary to include part of the background outside the highs for the positive anomaly to be identified. We added a note in the text.

* P9, L12-14: A reference (Pan et al. [2016]?) would be appropriate for this mention of the modeling analysis.

Added.

* P9, L32-33: "… effective test whether the retrieval sensitivity is sufficient to resolve independent CO variability". This statement should be qualified. The comparison represents an effective test of whether IASI's sensitivity is sufficient to resolve independent upper and lower/middle tropospheric variability.

Modified.

* P11, L9-11: The weaker CO enhancement over the Tibetan Plateau is attributed to the impact of the elevated terrain on the retrieval. However, the enhancement in Fig. 9a is weak only over part of the Plateau area -- it is relatively strong around 29-30N.

The weakening here is mostly toward the altitude of enhancement. The IASI enhancement in this case ended at lower altitude and is not appearing at the 150 hPa level. A sentence is added to explain this.

* P11, L13: "the enhanced layers are centered near 150 hPa and vertically extended between 100 and 200 hPa". Again, IASI cannot really discriminate between these levels. Given the "smoothing error" in the retrieval, it is not possible to quantify the true vertical extent of this feature.

Modified.

* P11, L12-15: I am a little confused by the discussion of CO enhancement over the Western Pacific High being associated with strong easterlies in the case of Fig. 9b (though not in the case of Fig. 9d). Here and previously (P8), this feature is attributed to eastward eddy shedding, which seems inconsistent with the presence of strong easterlies. Some elaboration of this point would be helpful.

This is purely an observation, which is somewhat interesting when you look at the cross section and them together. They are on two different sides of the Tibetan High so not all eastward. We deleted the sentence since it causes unnecessary confusions here.

* P12, L8: The previous work of Park et al. [2007, Fig. 9] is described as analyzing MLS 100 hPa CO, but it would be more appropriate here (in a paragraph about O3) to say that Park et al. analyzed the MLS 100 hPa CO-O3 relationship.

Added $O_3$.

* P12, L10: In addition to its relatively coarse vertical resolution, it might be appropriate to mention the potential impacts of clouds and terrain on the OMI data, as discussed on P13.

This is a general statement and we prefer not to go into the details here.

* P12, L23-24: The weaker signature of in-mixing of stratospheric air in the OMI data is attributed to averaging of variable fine-scale structure in the seasonal map. But it seems to me that it is probably more related to the coarse vertical resolution of the OMI measurements, which effectively smears out the signature of this relatively shallow layer.

Modified.

* P14, L9-10: The weaker correlation between IASI CO and GPH is largely attributed to the effects of elevated terrain, specifically the Tibetan Plateau, on the retrieval. But I do not believe that the relative contribution of all possible factors was quantified in this work. Couldn't the coarser vertical resolution and cloud effects (missing data) also have played a role?

In this case we only point out the obvious that the terrain effect jumps out as the leading factor. Detailed attribution of all contributors is beyond what we can address in this work.

* P14, L12-13: "the IASI 150 hPa data include contributions from the level below". Again, this wording gives the impression that a much narrower layer is influencing the IASI measurements at 150 hPa than is actually the case.

Again, the main point is that IASI can sense the independent variability in the upper troposphere. However much the lower levels contribute, it did not change this qualitatively. We do not have information to say more about it quantitatively.

* P14, L32: It is stated that the analysis shows that IASI data have sufficient information content "to resolve upper tropospheric CO variability". I think it would be clearer and more accurate to employ wording similar to that in the abstract (P1, L24-25). I suggest something along the lines of: "to discriminate upper tropospheric CO variability from that in the lower to middle troposphere".

Modified.

* P15, L5-6: It is stated that "Although the retrieval has fewer degrees of freedom for each profile …". I think it would be clearer and more accurate to say "Although the retrieval has limited vertical resolution and is degraded over elevated terrain, …".

We have detailed these issues many times in the paper. At this point, this is a general comment contrasting the information of each profile compounded in the large number of profiles.

* P26, Figure 7: Are GFS winds at 150 hPa overlaid on both the MLS and IASI upper tropospheric CO maps? I expected the wind vectors to be the same in both panels, but close examination reveals small differences. Are the meteorological fields also gridded differently (as the satellite data are)?

The small differences are due to the 18Z versus daily mean GFS. We have updated the figure to all use daily mean.

* P29, Figure 10: Similarly, I expected the 150 hPa GPH fields (colored contours) to be identical between Figures 6 and 10, but there are small differences between them.

Same as the above. Fig. 6 has been updated to using daily average.

Minor points of clarification and suggested wording / figure changes (leaving most typos and grammar points to the journal copy-editors):

We have included all wording suggestions.

* P2, L1: I think that "demonstrate the value of" would work better here than "advocate for the use of"

* P2, L10: "composition … displays"

* P3, L18: "variabilities" --> "variability"

* P3, L21: "weak" does not seem like the right word for "resolution"; I suggest "poor" or "coarse" here instead

* P3, L27: "for using … diagnosis" --> "to use … diagnostic"

*P4, L17: delete "works of"

* P5, L9: "no IASI retrieval product once the cloud is greater than 25% in the pixel" is somewhat awkward. I think that something like "no IASI products are available if the cloud fraction in the pixel exceeds 25%" would sound better.

* P6, L20: "rational" --> "rationale"

* P7, L35: delete "are"

* P9, L23: "is almost always referred" --> "almost always refers"

* P10, L14: I think "degraded" would work better here than "weakened"

* P10, L21: "maps" --> "values"

* P10, L34: "over the Iranian mode" --> "over the Iranian Plateau"

* P11, L9-11: "weakening" --> "degrading"

* P12, L15-16: "anticyclonic flow over the ASM" would be better as "anticyclonic flow over the ASM region" or "anticyclonic flow associated with the ASM"

* P12, L17: "interception" should be "intersection". The same comment applies to P13, L4 and P30, L5

* P13, L5: The text states that Gaussian smoothing is applied to the 1 x 1 degree maps (i.e., OMI), but the figure caption states that smoothing is applied to all maps

* P13, L14: Just to be clear, it would be good to add "in OMI O3" after "structure"

* P13, L16-17: "the weaker vertical resolution for this potentially shallow layer in OMI may contribute" --> "the coarser vertical resolution of OMI for resolving this shallow layer may contribute"

* P13, L32: "surface elevation on retrieval" --> "surface elevation on the OMI retrieval"

* P14, L4: "limb data" --> "higher vertical resolution limb data"

* P14, L9: "the season studied" --> "the 2008 JJA season studied"

* P14, L28-29: "model-based conceptual model" --> "conceptual model"

* P20, Figure 1a: the continent outlines are hard to see. Perhaps it would help to thicken the lines, or use a different color.

* P20, Figure 1b: It would be good to add a vertical line marking the zero line.

* P26, Figure 7: please either re-draw this figure so that panel (a) is at the top, as readers expect and is the case in all other multi-panel figures in this manuscript, or simply label the bottom panel as (c) and alter the references to this figure in the text accordingly.

* P26, L4: "terrains are" --> "terrain is"

* P28, Figure 9 caption: The various overlays, which presumably represent the westerly and easterly jets, tropopause height, theta surfaces, and wind vectors, all need to be defined.

Added. Thank you.

* P32, Figure 13 caption: "tropopause pressure contour" --> "tropopause pressure contour (black)". Also, although the Tibetan Plateau is clearly marked on other figures, it is pretty hard to see here. Perhaps a different color could be used (or perhaps pink could be used for the tropopause pressure, as in Figure 11, and then black could be used for the TP).

Modified.

Laura Pan on behalf of all co-authors

We thank the reviewer for many suggestions. Please see the point-by-point responses below.

I carefully read and revised the new version of the study and I apologize with the authors for the long time I eventually needed. The revised version of the manuscript by Luo et al. was greatly improved in its overall quality and clarity. The questions posed by the authors are very interesting and now more clearly expressed and investigated. However, in my opinion, the authors failed to overcome the main shortages that were pointed out during the ACP discussion phase, i.e. the lack of robustness in the adopted IASI CO profiles and how they are interpreted in the paper. I share with the authors the conviction that many interesting results are shown but these results carry the lack of possibility for the reader to discern robust patterns versus artificious (i.e., induced by the retrieval with poor sensitivity) patterns in the nadir CO (and to a lesser extent in the O3) data. I feel the authors should give more importance to having a consistent and quantified quality of the adopted data in support of their aims. In their current analysis, high and low quality results are presented under uncontrolled circumstances. I cannot therefore recommend the manuscript for publication unless these significant flaws are overcome. I report here below a number of comments that I hope the authors will consider to improve their study.

GENERAL COMMENTS

As mentioned above, in my opinion, there are major flaws in the CO data quality and the way the data are interpreted that make the whole study unreliable. These are expected because CO data are used in the analysis outside the validated range, at a single layer with no consideration of the vertical resolution and with faulty data not removed from the analysis. This was pointed out in the ACPD phase by the reviewers citing relevant literature and persists. Please see the details in the previous report. I find it interesting and see no restrictions in trying to extract more information by comparing the CO data to known processes with your aim of a processed based evaluation. But then I think the data quality needs to be kept completely under control through error and information content analysis on each individual profile considered, and through sensitivity tests showing under which conditions the profile is correctly reproducing the CO in the sounded airmasses (see e.g. Fig 10 of George et al 2015 where a bump of high CO over a thick middle tropospheric layer is completely missed in the retrieved profile). In their current version, the reader cannot understand what parts of the maps are presenting real CO details and what parts are showing retrieval artefacts. This is clearly evident in the comparison of 3 month averages of Figure 3, which removing variability show clear biases e.g. over Africa, south of the anticyclone, over the Tibetan plateau and possibly elsewhere. This is possibly an

effect of vertical displacement of the airmasses which are not correctly picked by the retrieved profiles, leading to 50-100% differences with MLS. There is no possibility to correctly place the CO at a km scale, so at times the CO will be at the right altitude and will be retrieved, other times not. The issue affects also the vertical cross sections shown in Figure 9 where clear agreement of the CO fields with dynamical fields are seen in some parts, unexplained changes are seen in other parts. How much, where and when the reader can trust the data is not evaluated in the study and this severely affects the analysis and conclusions.

I think the general idea of a processed-based evaluation of the nadir data is interesting and important. The results in the study are however presented as they are unexpectedly better than what the averaging kernels are showing. I feel the authors should reconcile the limited information content available with the results that are shown, the latter being in fair agreement with the 2 independent points expected in the troposphere. This information is barely given and then disregarded in the analysis (e.g. in Figure 9 reporting cross sections): the key robust message in my view should be how well the CO vertical profiles are interpolating on 1-km layers the two available independent points, and how well this can describe the real atmosphere. Unless this is made clear and brought forward in the analysis, the reader is wrongly induced to think there is far more information than what is available in the data. If the same figures were to be replicated using raster plots with 2 single cells as deep as the information content is (about 10 km each), then the image would change but the advances of the dense nadir sounding may still be visible and correct. Again, sensitivity tests would clearly show how the profiles respond to the real atmosphere giving the needed support to this study.

I appreciated the effort of the author to adopt the same period of time for CO and O3. However, I missed an even brief comparison of the results of the two targets. There are clear similarities in the deficiency of CO and O3 nadir data as compared to the MLS data in the region south of the anticyclone, and likely above the Tibetan plateau. MLS in these very large regions show even 100% more CO and O3 than the nadir data. These differences are shown both on seasonal averages and on daily data, and are therefore robust and need to be addressed. Also, if the nadir data over the Tibetan plateau is unreliable (as it is for CO and at least partly for O3), then it should be removed from the analysis or a detailed error map shown in couple with the absolute values.

Response to the General Comments:

We appreciate the reviewer's concern for using satellite data carefully and the caution for unclear data quality due to limited validation. This work is very much motivated by the fact that validation opportunities are limited. To push this limitation, we explore satellite data creatively using known physical processes. We disagree with the reviewer on the statement that the data are flawed and the study is unreliable.

Although MLS is a better understood and more validated dataset, it is still incorrect to conclude that the IASI data are wrong in places the two datasets disagree. The conclusions of this study do not depend on every IASI or MLS data point on the map giving correct CO value, but rather, on how consistently both datasets show the impact of Asian monsoon dynamics. The analysis has clearly shown that both nadir data sets are useful for investigating the sub-seasonal variability of the Asian monsoon dynamics and transport. A key point is that IASI can sense the independent variability in the upper troposphere, albeit the retrievals are weighted within a broad atmospheric layer.

All satellite data are known to have "smoothing errors", which do not make the data flawed. You can find from Fig. 1b that the UTLS MLS CO retrievals are also associated with a thick layer of ~5 kilometers. The importance of this work is that we show by using process understanding that the IASI retrieval over the region indeed has important independent information in the upper tropospheric and lower/middle tropospheric. This result is not surprising, and it is consistent with and complementary to the information content analysis based on the averaging kernels and DOFS. To show this, we have included one more figure, Fig. 1c, to show that the DOFS calculated for the most important part of the study domain is distributed near 1.9-2.0, which supports the independent variability for UT and Mid-Troposphere. We think the value of this work is to demonstrate how much more we can learn if we let our understanding of the process help interpretation of the separate data sets.

As we have mentioned in the 'overview of the revisions' file in the last round of the review, CO is a tropospheric tracer and the UT CO variability is associated with both the convective uplifting of the boundary layer air and the horizontal transport driven by the anticyclone dynamics. However, $O_3$ is a more complicated tracer at UT. We focus on the impact of the anticyclone and tropopause structure on the $O_3$ mixing ratio at 100 hPa level and how MLS and OMI data show it in this study.

In any case, we view our work as a first look at these interesting questions, and as motivation for future studies. More quantitative validation and detailed attribution of all aspects of the differences between limb-viewing and nadir-viewing data are beyond what we can address in this work.

DETAILS (P=page, L=line number)

Abstract: please revise as needed

P3 L 15-17: In order to support their aims, I think the authors should underline that the same (or nearby) airmasses are sounded but with different geometries that involve crossing other portions of the atmosphere. If it were not the case, then the comparison would make no sense.

To be more accurate, the air masses sensed by the two sensors have overlap, which is more like a loaf of French bread with a pencil sticking into it. We have modified the wording to

> .. the air masses they are sensing represent very different volume and spatial extent (described in section 2).

L18-19: it is not the retrieval which is being evaluated but the retrieved data, which include the whole chain from measurement, through retrieval, to data manipulation. I suggest also to specify the evaluation is left at a qualitative level, i.e. no quantitative information is given on the goodness of this evaluation.

Our analyses include both qualitative and quantitative comparisons of the MLS, IASI and OMI data sets. It is clear that we use the retrieved data in the study, and we believe it is appropriate to refer to the method as 'process-based retrieval evaluation'.

L19-20: I do not see any effort in the manuscript to relate the information content of the data to the dynamical/chemical patterns that are described. Therefore, I feel the sentence "From the remote sensing information content point of view" should be rephrased. Unless of course an analysis of the information content of the data was introduced in parallel as suggested in the first revision and still strongly recommended.

Revised although we did include a new figure on DOFS.

L21: to me the 10-14 km vertical resolution is poor for the aims of describing vertical resolved CO tropospheric structures, rather than "relatively weak".

Modified.

L26: why is there no attempt to improve the analysis by comparing results for CO and O3? It seems straightforward to see that shortages south of the anticyclone in the nadir-limb comparison in CO are also present in O3.

See response to general comments. The CO and $O_3$ are analyzed independently in this work. We chose two different tracers to examine two different aspects of the ASM dynamics in this study.

P4 L8-20: the reader should be informed that the IASI CO data is used outside the validated range (which extends up to 225 hPa in Wachter 2012 and was performed on 2 partial columns) and that the use of one point of the profile at 150 hPa is equivalent to take into account a partial column about 10 km high given the available vertical resolution. I think the study would be greatly improved if these essential details were more carefully explained and dealt with throughout the paper.

225 hPa was chose as an upper limit of IASI CO data in Wachter 2012 rather than the actual validated range of IASI CO data.

We have added 'Note that these layers are referring to the product identification. The physical information for these layer products is contributed from a broad layer, as suggested by the averaging kernels.' in the revised manuscript.

L14: I think it is necessary to add that given the 0.8-2.4 DOFS, only 1 or 2 independent partial columns can be generally extracted and the vertical profiles will be based on 1 or 2 points. The 1-km vertical grid is oversampling the available information, acting as interpolation based on the 2 independent points. This is a fundamental information for the reader and I think it needs to be correctly emphasized here and in the analyses. Again, I think the manuscript would be greatly improved by showing detailed calculation of the DOFS for the horizontal maps and vertical cross sections shown in the manuscript, possibly showing better consistency with MLS when the DOFS is higher. Having this information under control would support the strategy adopted by the authors.

We have included a new figure Fig. 1c, to show the distribution of DOFS. The distribution shows that most of the profiles in the interested region have ~ 2 pieces of information. Our analysis is highly consistent with what indicated by the DOFS.

P5 L15-20/Figure 1b: the averaging kernel at 14.5 km is completely overlapped with those from nearby altitudes. What do the authors mean when writing "This level is contributed mostly by IASI CO product layer centred at 14.5 km and with a small fraction from the 13.5 km and 15.5 km layers."? Where is this information taken from? The averaging kernels are so spread that the vertical resolution (e.g. half width half maximum of the averaging kernel) is of the order 10-14 km. In fact, the plot shows that the signal at 14.5 km is almost evenly contributed by averaging kernels from 10 to 16 km and with still relevant contributions from layers below down to at least 7 km altitude. I think the sentence needs to be rephrased and the correct emphasis to what the plot is showing given here and throughout the analysis (as it is mentioned a few lines below).

To make it more clear that the IASI CO product in the UT is contributed from a broad layer, and the 14.5 km or 150 hPa are only used for product designation, we have added a few sentences:

Note that these layers are referring to the product identification. The physical information for these layer products is contributed from a broad layer, as suggested by the averaging kernels. Fig. 1b clearly indicates that the 1-km layer products are not intended for representing independent information from each layer. To provide a perspective of retrieval information content in the study region,

we show the distribution of DOFS for IASI profiles in Figure 1c. The distribution shows that the majority of the profiles are estimated to have DOFS close to 2, which supports the aim of this study to evaluate the UT CO variability using IASI CO product.

As we have mentioned, smoothing error and limited vertical resolution is an issue for all satellite data. MLS 147 hPa CO retrieval for example, is also contributed by a thick layer of ~5 kilometers.

L28-31: I appreciated the clarity for the usability of O3 in this study.

Thanks.

P6 L6-16: Wachter et al. 2012 finds very large sensitivity variations in CO between day and night and over land/ocean, with a shift of the maximum sensitivity from the low to the middle troposphere. Maybe this is affecting part of the nadir data and could help reconciling them with MLS?

Thanks for the suggestion. We have compared IASI CO data between daytime and nighttime when we were preparing this study. The patterns are similar to each other, and both show UT variability consistent with dynamical fields.

L33-34: I would clearly state that this comparison is needed because the data is used beyond their validated range and with an approach that is not supported by the current literature.

We have clearly stated that we focus on variability. We have made quantitative comparisons beyond the past validation study.

P7 L5: typo: "is the strongest" ◊ rephrase?

Modified.

L9: it would be useful to specify how the "the vertical information content of IASI CO retrieval" led to use the 150 hPa level.

This is included in our previous responses.

L11: typo: "of THE single year 2008"

Unchanged. "a single year of 2008" is correct.

L16-19/Figure 3: Considering this is a three-month average over fairly large boxes, the agreement seems to me not so good, with a broad spread showing very large

differences in the dataset at uncontrolled times and locations: 50 ppbv of IASI CO can be associated to 30-100 ppbv of MLS CO, 50 ppbv in MLS CO can be seen as 50-100 ppbv in IASI. Indeed, the measurement and retrieval shortages in IASI have an impact on the spread and I think this should be quantified and its goodness tested. Would removing regions producing the significant difference seen in Figure 4 greatly improve the correlation shown in Figure 3 and its significance? If so, then it should be possible to isolate those regions, investigate the source of error and seek a method for keeping or rejecting nadir data.

First, it should be noted that Fig. 3 uses daily data and not seasonal averages, as clearly stated in the figure caption. The important point in this work is that the IASI retrieval in the region can discriminate the upper tropospheric variability from that in the middle and lower troposphere, and enable analysis of sub-seasonal variability of chemical distribution. We agree that improvement in the IASI retrieval is possible and necessary, but it is beyond this study.

L25-27/Figure 4: the comparison of seasonal averages should give the best results given that temporal and spatial sampling shortages are removed. The figure clearly shows that there are serious deficiencies that need to be investigated and isolated in order to support the analysis at shorter time scale. There are large differences which do not appear to have a simple and consistent interpretation, e.g. over Africa, south of the anticyclone, over the Tibetan plateau. Looking at a IASI CO map it is not possible to understand whether the data in one region is trustable or not. Since IASI CO at 150 hPa is picking signal from different layers, it is very likely that if high CO values are encountered at the wrong altitude the retrieved data may not show it. Showing maps of error and information content may help understanding why there is agreement in some regions and not in others.

The 150 hPa IASI CO data is the designation as the retrieval product, but represent a broad layer and clearly noted in the revised text. We have stated that the information is from a broad layer rather than a wrong altitude. Additional work on data validation and quality control will be useful, but that work is beyond what we focus on here.

L29-30: the blob of high CO at 120-150E in MLS at 215 hPa is very similar to that in IASI at 150 hPa. This strongly suggests IASI is picking signal from the wrong layer and there is no control on the vertical distribution as correctly shown by the averaging kernels.

This is because physical information of IASI CO is from a relative thicker layer rather than from wrong layer.

L35: typo: "likely involve"

Revised.

P8 L1-3: I agree. So, if there are regions where the data is so poor, then I suggest these data to be removed or marked somehow. But if this is the case for a certain region, how can one further trust the data at a more detailed level? I think these sentences in the text should be translated in careful selection of what is trustable and what is not.

This work is a first look at these data comparisons, and we believe it is useful to include all of the data without determining what is or is not 'trustable'.

L5-7: following the above comments, the comparison shows significant shortages in CO data that are not characterized nor explained. Further examples: the patch of high CO is at 60E in IASI while it is over India in MLS. How to explain the weak IASI CO at 30N/30E? MLS nicely fills the white contour. What about the high CO patch below 0N/30E in MLS? There is no sign of it in IASI CO.

We agree that the differences between datasets exist, but these are part of the interesting results! Further comparisons and validation work may resolve detailed differences and ascertain if retrieval improvement is possible.

P8L11-12/Figure 5: Why are the peaks in MLS CO always south of the peaks in GPH? Is it related to the tropopause interception of 100 and 105 hPa as shown for O3? A comparison of the results of the two targets may help.

Firstly, chemical distribution is not expected to be completely co-aligned with GPH fields in all situations (especially at low latitudes). Secondly, the constructed grid data of MLS is 5x5 degree which is much coarser than GFS analysis. We simply note this behavior and leave it as a topic for future study.

L29-30: possibly rephrase? E.g., a latitude band restricted to a longitude interval. The plots are for 0-180E and not for 0-220 as written.

We have added a note in the text.

Figure 6: the figure is difficult to read apart from the zero lines. Fewer color levels may help promptly identifying highs and lows. Or possibly the addition of a few contour lines? Would it be possible to extend the longitude range to 20W or so in order to capture the zero-anomaly contour line in both GPH and MLS CO?

We have revised Fig. 6 with fewer contours.

P9 L12: what model analysis are the authors referring to? Is this literature or performed within this study? Please clarify.

Reference added.

Figure 7: Indeed, IASI CO shows very interesting information but it is again difficult to understand what parts are robust and what not. For example, why would MLS pick signal along the 135E cross section at 150hPa that is not seen in IASI? The maximum contribution of an individual measurement in MLS comes from the tangent point (say 14 km altitude in this case) along the line of sight with a spread of a few hundred km, i.e. maximum +_2/3 degrees latitude. Moving 10 degrees latitude away from the tangent point, the line of sight crosses the atmosphere already 100 km higher. So, there is no possible contamination of nearby portion of the atmosphere at a distance of 10 degrees latitude and the MLS data must be trustable within their validated errors. Why is then IASI CO about 45 ppbv at 0-20 N along the 135E cross section where MLS is seeing 80 ppbv? This and other examples show IASI CO is picking signal but with no control on when it is placing it at the right location (see comments below for Figure 9). Even more, it seems various regions lead to persistent lower values as compared to MLS, pointing to physical conditions that affect the retrieval in the same way. Again, IASI CO above the Tibetan plateau seems completely wrong. I would recommend to remove known faulty data.

We agree that there are differences in horizontal distribution in UT between IASI CO and MLS CO, and we have noted that IASI CO is influenced by terrain and cloud. The different geometries also affect datasets. However, the key point is that IASI can sense independent variability in the UT, albeit within a broad vertical layer. There may be ways to improve data retrieval in the future, but that is beyond this study.

Figure 9: please add to the caption that black contour lines represent zonal winds.

Added. Thank you.

Figure 9b: I see oscillations along the longitudes with vertical plumes of low and high CO (60/100 ppbv) at 750/250 hPa between 10N to 35N. The wind field does not seem to predict those. The same in other regions and figures. On the contrary other portions seem to be in very good agreement. However, in my view, the wind field may be consistent with CO fields that are different from those shown and 50-100% changes could be easily accommodated. Therefore, it is very difficult to validate the use of nadir CO through this possible agreement with dynamical fields with no further quality control. Which parts can one trust and which not? I find it essential to support these interesting findings with a thorough analysis of the information content, error budgets and sensitivity tests on the ability of the profiles to reproduce the real data and not produce artefact. It is very well known that artefacts/oscillations can be produced by retrievals under low information content conditions, which in fact usually lead to data rejection.

The focus of our work demonstrates that IASI can sense independent CO variability in

the upper troposphere. While there is indeed some complex structure in the retrieved CO at lower levels, we have chosen not to focus on these details in our analyses.

P10 L5-8: to me the example show two independent fields can be obtained, not whether the retrieval is reproducing the two layers correctly. Nor whether the 150 hPa is unrelated to the 250 or 300 hPa level. What is the correct altitude of those high CO values?

When we refer to the IASI 150 hPa, it is the designation as a retrieval product. As responded above, we have noted in the text that the physical information for the layer products is contributed from a broad layer.

L13-17: because of the many flaws reported above, I think the comparison does not look convincing.

See previous response.

P11 L9-11: if the data is known to be of bad quality, why not removing it from the analysis?

We think it is very useful to show these limitations which help us better understand the factors limiting the preformation of our sensors.

P11 L16-19: this is likely to be the case but the study fails to have support from the missing data quality analysis: with what error? How much is real? It seems IASI CO can very likely pick strong localized CO enhancements but how it depicts them along the vertical profile is unknown. I recall George et al. 2015 Fig 10 example.

We think in this case the physical understanding of the circulation of the region is very informative.

L21-22: again the results showed IASI CO is able to pick variability, not that it is able to correctly reproduce it.

See response to General comments.

L26-27: I do not agree. The results presented in this manuscript are fairly in agreement with the limited information content calculated by the retrieval and are coupled with the several shortages that were discussed. I think this work deserves to be fully supported by a detailed quantification of the goodness of the data and retrieval sensitivity tests that show how much, when and where the profiles are correctly reproducing the sensed CO.

See response to General comments.

L28-32/Figure 10: this figure shows the limited agreement of the data with a not refined spatial and temporal reproduction by IASI CO of the real atmosphere. This does not seem to be dependent only on the Tibetan plateau faulty data. I would recommend to repeat the correlation removing the Tibetan plateau to test it.

We think it is more useful to examine what the dataset itself can show. The imperfect correlation informs the limitation of the data.

Figure 11: considering these are 3-month averages over such a broad region and that the target is the well defined O3, I think the comparison shows significant differences that need to be addressed. This is the case of Africa and the whole regions south of the anticyclone. And the Tibetan plateau as already discussed for CO. The strong disagreement south of the anticyclone was already found in CO and inter-comparison of the two targets would likely bring new insight in these shortages. Could MLS be picking stratospheric air (as seen by O3) which OMI is missing, and IASI retrieved CO be affected by this vertical displacement of stratospheric air along its line of sight? But why would MLS see more CO than IASI does?

In this study, we focus on showing what the three datasets tell us rather than how to improve them. Given the broad layer averages, we think the overall agreement in Fig. 11 is quite encouraging.

P12 L10 please revise the conclusions accordingly with the rest. This (10-14 km) is a very poor vertical resolution for resolving the troposphere in detail, I do not see how a different claim can be made.

See our response to the General comments.

L17: does this also explain why peak CO is displaced south of the highest GPH in Figure 7 and following?

Yes. They are very similar.

**The list of changes for the manuscript**

1. All the changes are shown in the marked-up manuscript. Please see it below.

2. We have added a new figure (Fig. 1c) and revised Figs. 6, 7, and 13.

[revised manuscript text omitted]

---

## Author Response (AR3)

Laura Pan on behalf of all co-authors

General response and summary of revisions

We appreciate the reviewer's concerns that referring the nadir sensors data as "level" data can be misleading for the readers. We have made changes to refer these data products as $CO/O_3$ mixing ratio in respective layers throughout the paper. We believe these changes have take care of the concerns.

The specific responses are given below each point (Reviewers comments in black. Our responses in blue).

I have appreciated the efforts of the authors in responding to the reviewers, which I think helped in better defining the aims of the authors and give a more balanced presentation. As previously stated, I share with them that they present interesting ideas and results. However, I was not fully convinced by their revision and think that 1) the way their results are presented can still be misleading, and 2) they did not perform enough (and easily achievable) analysis to support their claims. I am afraid in my opinion these two issues should still be taken in consideration to fully support publication of the manuscript. The authors stressed that they show IASI retrieved data carry sufficient information to have independent variability in the lower and upper troposphere. This is evident from both maps and averaging kernels, and the new panel in Figure 1 greatly improved this by constraining most of the profiles to 1.7-2 degrees of freedom. Since better constrains were given to the available information content, I think better care should still be given to the presentation.

We agree with the reviewer that the issue is mainly in the "presentation" and we have made changes to be clear.

As previously stated, the performed analysis does not leave room for understating whether this variability is correct and from what altitude it is coming from, and the qualitative comparisons performed in e.g. Fig. 3 and 4 is in my view not satisfying enough to exploit the data in detail, nor it is the comparison of Fig. 6 and 10 (Hovemoeller diagrams for MLS and IASI) where MLS show a much better agreement with GPH.

We have provided as much information as possible for where the information content:
- The averaging kernels in Fig. 1b provide the answer to this from signal point of view.
- The consistency in variability with the MLS and the dynamical field provide additional support on the UTLS information.

I would therefore expect to stress pros and cons in a more balanced way throughout the manuscript (see e.g. abstract, and the discussion in page 7 and related text – whereas I appreciated the much more balanced text in the conclusions). This includes discussion of the 10-14 km vertical resolution (still missing in the text and misleading in Fig 9) and removal of labelling 150 hPa (or 500 hPa)

layers in favor of "upper troposphere" and "lower-middle troposphere" layers. The wrong statement that the 150 hPa layer "is contributed mostly by IASI CO product layer centred at 14.5 km and with a small fraction from the 13.5 km and 15.5 km layers" (P5 L19-20) was not removed and is not supported by the averaging kernels shown in Fig. 1.

We have made revisions along this direction throughout the paper. Specifically
- OMI product layer 18 is a layer centered near 100 hPa and with broad 10-km vertical resolution.
- IASI CO UT layer used product layer 12-16 km, which includes information as shown by the averaging kernels.
- The IASI CO are presented as UT layer mixing ratio and MT layer mixing ratio.
- The statement that the 150 hPa layer "is contributed mostly by IASI CO product layer centred at 14.5 km and with a small fraction from the 13.5 km and 15.5 km layers" (P5 L19-20) has been removed. Although these statements are not wrong, we agree that they can be misleading for the casual readers. We use a different wording to explain that the IASI UT layer is derived from these product layers.

Of course, also MLS have shortages, but these are known, kept under control by using validated data and, moreover, a 5 km (3 km for O3) vertical resolution and related smoothing errors are very different from IASI CO 10-14 km (10-12 km OMI O3) vertical resolution. I feel the authors' results and ideas would be much better supported by a more balanced presentation. In my views, given the aims of the authors to show an open-minded and novel approach to explore the information content of nadir data, it remains a clear shortage not supporting their claims with available knowledge: It would be very straightforward to perform a few simulations of the retrieval under the various CO and O3 scenarios encountered in the ASM area, or to use a chemical transport model to show how CO is vertically distributed and how different layers may impact IASI "lower-middle troposphere" and "upper troposphere" layers.

These retrieval simulations are not within the scope of the paper. We will pass these comments to the retrieval team and hope this will motivate their additional work along this direction.

Because of the ambiguity in assigning the retrieved CO to the correct altitude, I think this further analysis is essential and not simply further work for the future. I therefore remain convinced that the promising ideas and results presented require further refinement to be fully supported and recommendable for publication.

We have renamed and included additional emphasis that the data are layer product. These re-assignment in the paper should have taken care of the above concerns.

**The list of changes for the manuscript**

All the changes are shown in the marked-up manuscript. Please see it below.

[revised manuscript text omitted]